# Reinforcement Learning for LLM Post-Training: A Survey

## Abstract

Large language models (LLMs) trained via pretraining and supervised fine-tuning (SFT) can still produce harmful and misaligned outputs, or struggle in domains like math and coding. Reinforcement learning (RL)-based post-training methods, including Reinforcement Learning from Human Feedback (RLHF) methods like Direct Preference Optimization (DPO) and Reinforcement Learning with Verifiable Rewards (RLVR) approaches like PPO and GRPO, have made remarkable gains to alleviate these issues. Yet, no existing work offers a technically detailed comparison of the various methods driving this progress. In order to fill this gap, we present a timely survey that connects foundational components with latest advancements. We adopt and extend the unified policy-gradient framework of Shao et al. (2024) as an organizing lens spanning pretraining, SFT, RLHF, RLVR, and more recent techniques. The main contributions of our survey are as follows: (1) a self-contained introduction to MLE, RLHF, and RLVR foundations and the unified policy gradient framework; (2) detailed technical analysis of PPO- and GRPO-based methods alongside offline and iterative DPO approaches, decomposed along prompt sampling, response sampling, and gradient coefficient axes; (3) standardized notation enabling direct cross-method comparison; and (4) comprehensive comparison of implementation details and experimental setup of each method in Appendix. We aim to serve as a technically grounded reference for researchers and practitioners working on LLM post-training.

## Contents

## 1    Introduction

We first briefly outline the core stages of LLM training, i.e., pretraining, SFT, RLHF, and RLVR. Next, we proceed to discuss the structure and contributions of our survey, guiding readers through the technical foundations of RL and its role in LLM post-training.

### 1.1    Pretraining and SFT

The rapid ascent of LLMs has been propelled by scaling decoder-only Transformers (Vaswani et al., 2017) trained with self-supervised next-token prediction (Radford et al., 2019) on trillions of tokens, yielding models with broad world knowledge and emergent capabilities. However, a pretrained model is optimized to continue its training distribution, not to follow user instructions. Performing SFT on curated instruction–response pairs (Brown et al., 2020; Ouyang et al., 2022) closes this gap, teaching the model to produce helpful responses to human queries.

### 1.2    RLHF

SFT does not, however, guarantee alignment with human values: models can still generate outputs that are unhelpful, dishonest, or unsafe. RLHF (Christiano et al., 2017; Stiennon et al., 2020; Ouyang et al., 2022; Bai et al., 2022a) addresses this by training a reward model on pairwise human preferences and optimizing

the LLM policy, typically via PPO (Schulman et al., 2017b), to maximize the reward subject to a KL penalty that anchors the policy to a reference model. RLHF induces frontier systems including GPT-4 (OpenAI et al., 2024), Claude (Anthropic, 2024), and Gemini (Gemini, 2025). The RLHF/PPO pipeline is resource-intensive, requiring four models to be held in memory simultaneously, i.e., the policy, reference, reward, and value models along with on-policy rollouts at every gradient step. DPO addresses this by establishing a direct mapping between the reward model and the optimal policy, enabling joint optimization through offline pairwise preference data.

## 1.3 RLVR

RLHF and DPO rely on reward signals derived from subjective human judgments. For domains where correctness is objectively verifiable, i.e., mathematics and code generation, a stronger training signal is available: whether the final answer matches the ground truth or the generated code passes its test suite. RLVR exploits this signal to cultivate reasoning capabilities. DeepSeek-R1 (Guo et al., 2025) showed that outcome-based RL, powered by GRPO (Shao et al., 2024), can elicit sophisticated chain-of-thought (CoT) reasoning (Wei et al., 2023) directly from a base model without any supervised reasoning data.

## 1.4 Contributions

Several recent surveys discuss reinforcement learning, preference learning, or reasoning-oriented LLMs, but they differ from our work in both scope and technical lens. General RL surveys such as Ghasemi et al. (2025) cover classical and deep RL algorithms, but do not address LLM-specific post-training issues such as token-level policy optimization, reference-model regularization, reward modeling, verifiable rewards, or response length effects. Preference-learning and DPO surveys, including Gao et al. (2024) and Liu et al. (2025b), provide useful taxonomies for alignment and DPO-style methods, but they do not systematically connect DPO with PPO-based RLHF, GRPO-based RLVR, SFT, and pretraining under a common update rule. Surveys on RL for large reasoning models, such as Zhang et al. (2025c), are closer in topic and recency, but are primarily landscape-oriented, emphasizing reasoning capabilities, resources, applications, infrastructure, and scaling challenges rather than the fine-grained mechanics of post-training objectives.

Taken together, these limitations leave a need for a unified, mechanics-focused account of how modern LLM post-training objectives relate to one another. This survey fills that gap with the following contributions:

- **Self-contained RL and LLM post-training foundations.** Section 2 provides a self-contained treatment of all reinforcement learning foundations and key LLM post-training algorithms, beginning from MLE and REINFORCE through RLHF, PPO, and DPO to RLVR and GRPO. This covers every concept required to understand the methods surveyed in later sections without consulting any external reference. For each algorithm, we derive the policy-gradient objective and identify the *gradient coefficient* that encapsulates its core design decisions, thereby expressing it within the unified analytical lens of Shao et al. (2024) that we adopt and extend throughout the survey.

- **Unified policy gradient framework with systematic decomposition.** We adopt and extend the unified policy gradient framework by Shao et al. (2024) that includes PPO-based RLHF, RLVR, and DPO-based alignment. This framework decomposes RLVR along three orthogonal design axes: prompt sampling (Section 3), response sampling (Section 4), and gradient coefficient (Section 5). Section 6 extends the framework to on-policy RLHF and iterative DPO methods, including RLAIF and Nash learning. Section 7 covers offline DPO-based methods, which share the same policy gradient foundation but exhibit greater diversity in preference signal design, and is organized along response generation, reward modeling, regularization, and SFT integration.

- **Standardized notation enabling direct technical comparison.** We introduce a unified notation applied consistently across all reviewed papers, expressing every method's update rule in terms of the same gradient coefficient decomposition. This common formalism makes design choices directly comparable across otherwise disparate methods, and is supported by detailed per-paper comparison tables that catalog not only base models, training datasets, and benchmarks, but also fine-grained

algorithmic attributes including importance sampling ratio, clipping strategy, reward signal, baseline, advantage normalization, length normalization, partition function, KL regularization, and entropy regularization thereby enabling systematic cross-method comparison under the unified framework.

## 1.5 Survey Scope and Literature Selection

**Scope and criteria**    We include papers that introduce, analyze, or substantially refine LLM post-training objectives, training algorithms, reward mechanisms, or sampling strategies. Following the unified policy-gradient lens in Section 2, we characterize methods by their data or prompt source, response-sampling strategy, gradient coefficient, and stabilization mechanism. We prioritize methodological contributions over works focused mainly on empirical evaluation. Accordingly, we generally exclude papers centered on benchmark construction, dataset curation, inference-time prompting, or performance studies without a corresponding post-training innovation, while citing them when needed for context.

**Sources and time range**    The survey draws on arXiv preprints, peer-reviewed papers from major machine learning and NLP venues such as NeurIPS, ICML, ICLR, ACL, and EMNLP, and industry technical reports accompanying frontier model releases, including DeepSeek-R1 (Guo et al., 2025), InstructGPT (Ouyang et al., 2022), and others. The collection is methodology-driven rather than strictly chronological. Most surveyed methods appeared between 2023 and April 2026, our cut-off date, with earlier foundational works such as REINFORCE (Williams, 1992), RLHF (Christiano et al., 2017), and PPO (Schulman et al., 2017b) included for completeness.

**Search procedure**    Candidate papers were identified through arXiv, Google Scholar, and Semantic Scholar searches using algorithmic terms such as RLHF, DPO, PPO, GRPO, RLVR, policy gradient, actor-critic, and preference optimization; component terms such as reward model, verifiable reward, advantage estimation, importance sampling, and KL regularization; and application terms such as post-training, alignment, reasoning, and chain-of-thought. We further expanded the set through forward and backward citation tracing.

**Paper Timeline**    Figure 1 gives a chronological map of the representative methods surveyed in this paper. Methods are ordered by their earliest public appearance, typically the first arXiv posting or technical-report release, and coloured by the penultimate taxonomy node corresponding to their main methodological contribution. It highlights the shift from PPO-based RLHF and offline preference optimization toward the rapid growth of RLVR methods for reasoning. For methods spanning multiple mechanisms, we assign the colour according to the headline contribution and discuss secondary connections in the relevant sections.

## 1.6 Paper Organization

As shown in Figure 2, the post-training of LLM will be reviewed from six interconnected modules: Prompt (human-generated or synthetic), Response (on-policy, off-policy, offline, and SFT), Reward Models (rule-based, AI/human feedback, and outcome vs. process rewards), Reward signal formulations (pointwise, pairwise, listwise, negative, and token-level), Reinforcement Learning algorithms (REINFORCE, PPO, GRPO, DPO), and Regularization techniques (divergence and entropy). Arrows indicate the training loop: prompts are fed to the LLM, generated responses are scored by reward models to produce reward signals, and the RL algorithm updates the LLM with regularization constraints. Section 2 adopts and extends a unified post-training framework (UPT) by Shao et al. (2024), deriving pretraining through MLE, REINFORCE, actor-critic, TRPO, PPO, DPO, and GRPO for RLHF and RLVR. Section 3 covers prompt sampling, including generation (human-generated and synthetic) and selection (static curriculum, adaptive difficulty, and reward-based filtering). Section 4 addresses response sampling, covering generation methods (on-policy, off-policy, asynchronous, and tree-structured rollouts) and selection strategies. Section 5 dissects the gradient coefficient across importance sampling ratio, advantage shaping, normalization, length normalization, and regularization. Section 6 examines on-policy based RLHF, RLAIF and iterative DPO and Nash learning methods. Section 7 examines offline-policy learning, covering response generation, reward modeling, regularization, optimization iterations, and SFT merging. Section 8 outlines future directions, and Section 9 concludes the survey. Detailed characterizations of all RLVR and RLHF papers appear in the appendices  A and  B.

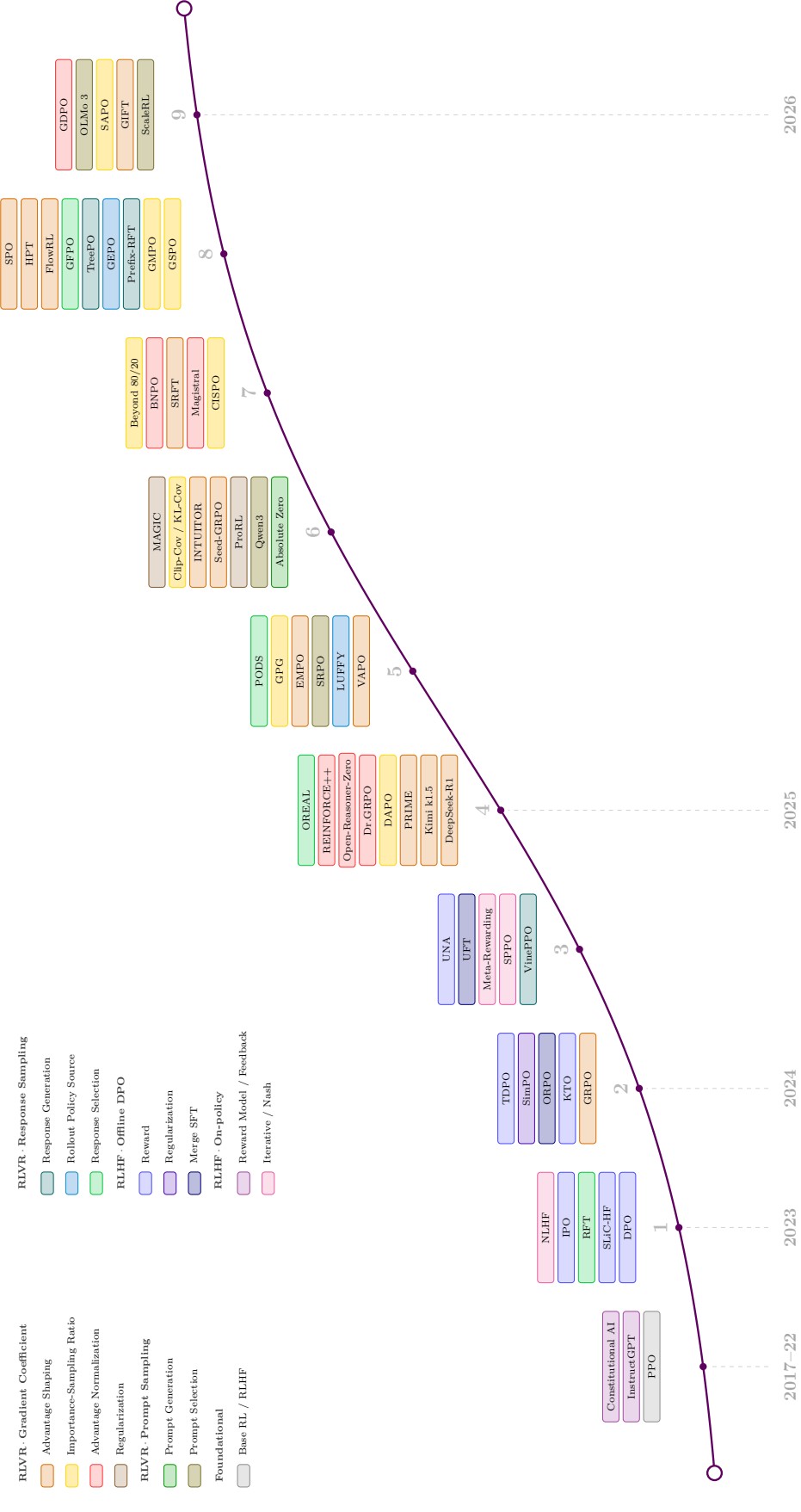

Figure 1: Timeline of representative RL-based LLM post-training methods covered in this survey, tracing the trajectory from preference-based RLHF/DPO to RLVR reasoning models (milestones ordered by arXiv first-posting). Each method is coloured by the *penultimate* taxonomy node of its primary contribution; hue families denote the parent branch (warm: RLVR gradient-coefficient axes; greens: RLVR prompt sampling; teal/cyan: RLVR response sampling; blue/indigo: offline DPO; violet/magenta: on-policy RLHF). Methods that contribute to several nodes are coloured by their headline mechanism.

## 2 Evolution of Language Model Training Paradigms

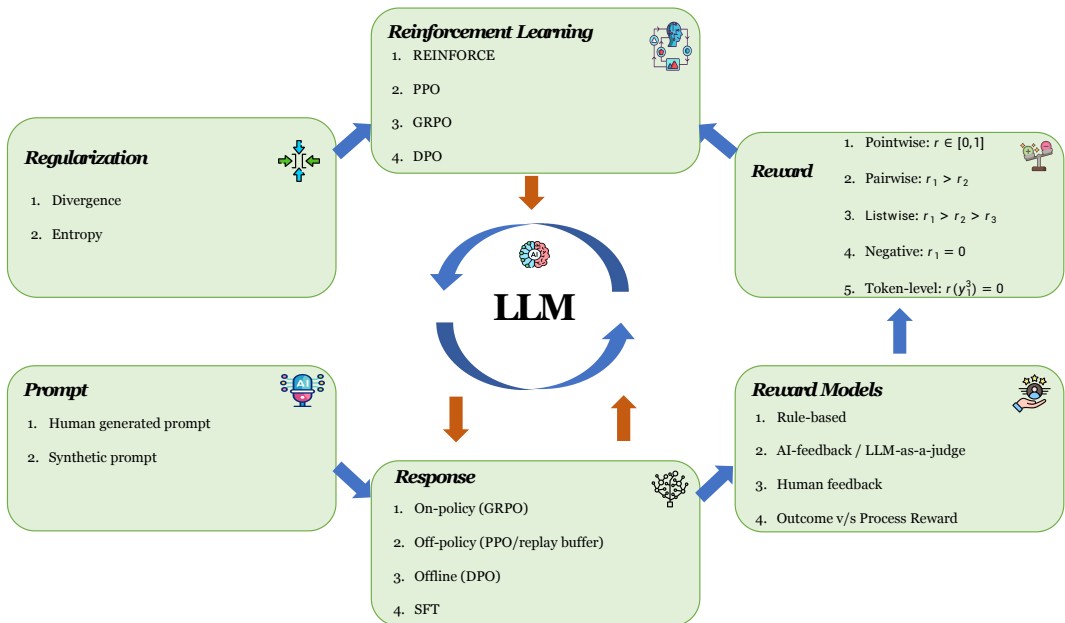

Figure 2: Overview of the key components in the reinforcement learning-based post-training pipeline for LLMs.

In this section, we adopt and extend the unified post-training framework based on a single policy gradient estimator by Shao et al. (2024). Building on this framework, we sequentially introduce key training paradigms for LLMs: LLM pretraining through MLE, SFT, and RLHF/RLVR through REINFORCE, AC, TRPO and PPO and GRPO across 1. Data/prompt, 2. LLM policy, 3. Response(s), 4. Reward / advantage, 5. Gradient coefficient and 6. Update as shown in Figure 2. For each paper, it is further analyzed from 1. importance sampling, 2. clipping, 3. reward, 4. baseline, 5. advantage normalization, 6. length normalization, 7. KL divergence, 8. Entropy and 9. partition function as shown in Figure 3. More detailed paper-by-paper comparisons of the different methods are provided in the following sections, while direct metric-by-metric comparisons can be found in Appendix A and B.

### 2.1 Unified Post-Training (UPT) via a Unified Policy Gradient Estimator

Post-training methods, ranging from supervised fine-tuning to reinforcement learning, can all be understood through a unified objective function gradient structure (Shao et al., 2024). On-policy RL methods optimize a reverse-KL-regularized reward objective in Eq. 2.1.1.

$$J(\theta) = \mathbb{E}_{x \sim \mathcal{D}}\big[\mathbb{E}_{y \sim \pi_\theta(\cdot|x)}[r(x,y)] - \beta \, \mathrm{KL}(\pi_\theta(\cdot|x) \, \| \, \pi_{\mathrm{ref}}(\cdot|x))\big], \quad \beta \geq 0 \qquad (2.1.1)$$

Expanding the KL into the expectation, the per-prompt objective is Eq. 2.1.2.

$$J(\theta) = \mathbb{E}_{x \sim \mathcal{D}}\left[\sum_y \pi_\theta(y|x)\left(r(x,y) - \beta \log \frac{\pi_\theta(y|x)}{\pi_{\mathrm{ref}}(y|x)}\right)\right] \qquad (2.1.2)$$

For a given $x$, $\pi_\theta$ appears twice in the inner sum: as the expectation weight and inside the log-ratio. Letting $f(y,\theta) = r(x,y) - \beta \log \frac{\pi_\theta(y|x)}{\pi_{\mathrm{ref}}(y|x)}$, the product rule gives $\nabla_\theta[\sum_y \pi_\theta f] = \underbrace{\sum_y (\nabla_\theta \pi_\theta) f}_{\text{Term 1}} + \underbrace{\sum_y \pi_\theta \nabla_\theta f}_{\text{Term 2}}$. Term 2

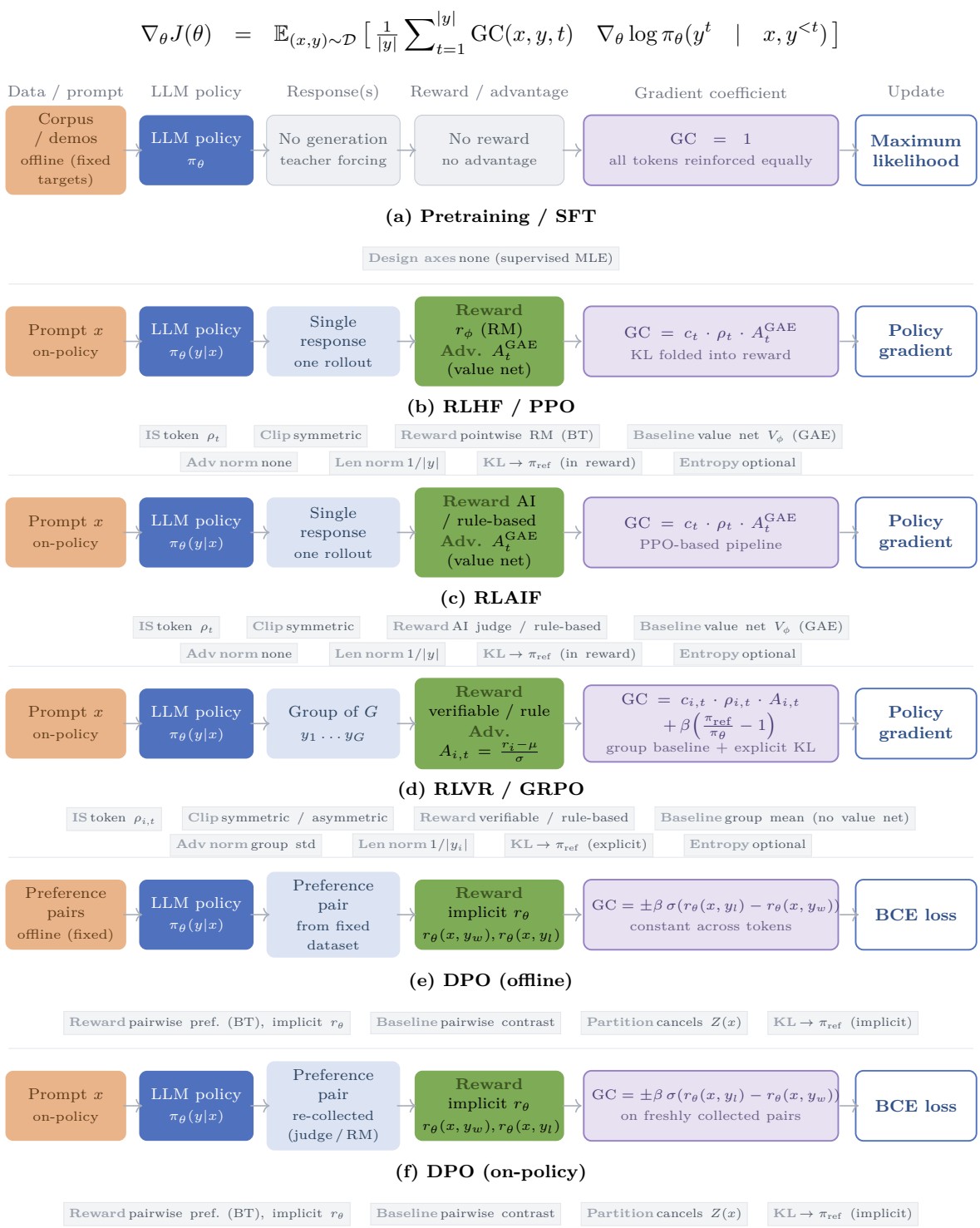

Figure 3: Overview of the nine per-paper analysis axes used to characterize each method—importance sampling, clipping, reward, baseline, advantage normalization, length normalization, KL divergence, entropy, and partition function—and the six visualization categories: pretraining/SFT, RLHF/PPO, RLAIF, RLVR/GRPO, DPO (offline), and DPO (on-policy).

differentiates the log-ratio: $\nabla_\theta f = -\beta \nabla_\theta \log \pi_\theta$, so Term 2 $= -\beta \sum_y \pi_\theta(y|x) \frac{\nabla_\theta \pi_\theta(y|x)}{\pi_\theta(y|x)} = -\beta \sum_y \nabla_\theta \pi_\theta(y|x) = -\beta \nabla_\theta \sum_y \pi_\theta(y|x) = -\beta \nabla_\theta 1 = 0$, since probabilities sum to one. Only Term 1 survives; applying the log-derivative trick $\nabla_\theta \pi_\theta = \pi_\theta \nabla_\theta \log \pi_\theta$ and restoring the outer expectation over $x$ in Eq. 2.1.3.

$$\nabla_\theta J(\theta) = \mathbb{E}_{x \sim \mathcal{D},\, y \sim \pi_\theta(\cdot|x)} \left[ \underbrace{\left( r(x,y) - \beta \log \frac{\pi_\theta(y|x)}{\pi_{\text{ref}}(y|x)} \right)}_{\text{gradient coefficient}} \nabla_\theta \log \pi_\theta(y|x) \right] \tag{2.1.3}$$

### 2.1.1 The Unified Gradient Coefficient

More generally, the gradient of any post-training method can be written in token-level form (Shao et al., 2024) in Eq. 2.1.1.1.

$$\nabla_\theta J(\theta) = \mathbb{E}_{(x,y) \sim \mathcal{D}} \left[ \frac{1}{|y|} \sum_{t=1}^{|y|} \text{GC}(x,y,t) \, \nabla_\theta \log \pi_\theta(y^t|x, y^{<t}) \right] \tag{2.1.1.1}$$

where $\mathcal{D}$ is the data source and GC is the gradient coefficient that determines the magnitude and sign of reinforcement for each token. Practical post-training methods are recovered by varying three interchangeable components: (i) the data source $\mathcal{D}$, which can be offline (from a fixed dataset), off-policy (from a different behavior policy $\pi_b$) or on-policy (from the previous steps $\pi_{\theta_{\text{old}}}$ or from the current policy $\pi_\theta$); (ii) the gradient coefficient GC; and (iii) a stabilization mechanism such as PPO clipping or a KL penalty. We now derive the objective and gradient for each representative method.

### 2.2 LLM Pretraining: MLE

Pretraining is the most compute-intensive stage of building an LLM: a randomly initialized transformer is trained on trillions of tokens via self-supervised next-token prediction. Given a pretraining corpus $\mathcal{D}_{\text{pre}} = \{x^{(i)}\}$ with $x = (x^1, \ldots, x^{|x|})$, the model $\pi_\theta$ maximizes the average log-likelihood in Eq. 2.2.1.

$$J_{\text{pre}}(\theta) = \mathbb{E}_{x \sim \mathcal{D}_{\text{pre}}} \left[ \frac{1}{|x|} \sum_{t=1}^{|x|} \log \pi_\theta(x^t|x^{<t}) \right] \tag{2.2.1}$$

Maximizing Eq. 2.2.1 is equivalent to minimizing the forward KL divergence between the data distribution and the model in Eq. 2.2.2.

$$\min_\theta D_{\text{KL}}(\pi_{\text{pre}}(x) \| \pi_\theta(x)) = \min_\theta \mathbb{E}_{x \sim D_{\text{pre}}} \left[ \log \frac{\pi_{\text{pre}}(x)}{\pi_\theta(x)} \right]$$

$$= \underbrace{\mathbb{E}_{x \sim D_{\text{pre}}}[\log \pi_{\text{pre}}(x)]}_{-H(\pi_{\text{pre}}),\, \text{const. w.r.t. } \theta} - \mathbb{E}_{x \sim D_{\text{pre}}} \left[ \sum_{t=1}^{|x|} \log \pi_\theta(x^t|x^{<t}) \right] \tag{2.2.2}$$

Since $H(\pi_{\text{pre}})$ is independent of $\theta$, minimizing $D_{\text{KL}}$ recovers the MLE objective in Eq. 2.2.1 (up to a scaling constant). Differentiating yields the log-likelihood gradient in Eq. 2.2.3, which matches the unified form in Eq. 2.1.1.1.

$$\nabla_\theta J_{\text{pre}}(\theta) = \mathbb{E}_{x \sim \mathcal{D}_{\text{pre}}} \left[ \frac{1}{|x|} \sum_{t=1}^{|x|} \nabla_\theta \log \pi_\theta(x^t|x^{<t}) \right] \tag{2.2.3}$$

The constant gradient coefficient $\text{GC}_{\text{pre}}(x,t) = 1$ means every token is reinforced equally, regardless of quality or downstream relevance. This uniform, task-agnostic objective makes pretraining scalable, i.e., requiring no reward model, annotations, or prompt-response structure. However, the resulting base model cannot distinguish preferred from dispreferred outputs. Subsequent post-training methods (SFT, RFT, Online RFT, etc.) refine it toward aligned and capable behavior.

9

### 2.2.1 SFT

SFT maximizes the log-likelihood on a curated demonstration dataset $\mathcal{D}_{\mathrm{sft}}$ composed of prompt $x$ and golden response $y$ in Eq. 2.2.1.1. Differentiating with respect to $\theta$ derives Eq. 2.2.1.2, yielding a constant gradient coefficient $\mathrm{GC}_{\mathrm{SFT}}(x, y, t) = 1$, with the curated dataset serving as an implicit human-selection reward.

$$J_{\mathrm{SFT}}(\theta) = \mathbb{E}_{(x,y)\sim\mathcal{D}_{\mathrm{sft}}} \left[ \frac{1}{|y|} \sum_{t=1}^{|y|} \log \pi_\theta(y^t | x, y^{<t}) \right] \qquad (2.2.1.1)$$

$$\nabla_\theta J_{\mathrm{SFT}}(\theta) = \mathbb{E}_{(x,y)\sim\mathcal{D}_{\mathrm{sft}}} \left[ \frac{1}{|y|} \sum_{t=1}^{|y|} \nabla_\theta \log \pi_\theta(y^t | x, y^{<t}) \right] \qquad (2.2.1.2)$$

### 2.2.2 RFT

RFT (Yuan et al., 2023a; Dong et al., 2023) samples multiple responses from the SFT model $\pi_{\mathrm{sft}}(y|x)$ and trains only on those with correct answers in Eq. 2.2.2.1 where $\mathbb{I}(y = y^*) = 1$ if the answer is correct and 0 otherwise.

$$J_{\mathrm{RFT}}(\theta) = \mathbb{E}_{x\sim\mathcal{D}_{\mathrm{sft}},\, y\sim\pi_{\mathrm{sft}}(\cdot|x)} \left[ \frac{1}{|y|} \sum_{t=1}^{|y|} \mathbb{I}(y = y^*) \log \pi_\theta(y^t | x, y^{<t}) \right] \qquad (2.2.2.1)$$

Differentiating with respect to $\theta$ derives Eq. 2.2.2.2 giving gradient coefficient $\mathrm{GC}_{\mathrm{RFT}}(x, y, t) = \mathbb{I}(y = y^*)$: uniform reinforcement of correct responses, zero for incorrect ones. RFT is an offline method since outputs are sampled once from $\pi_{\mathrm{sft}}$.

$$\nabla_\theta J_{\mathrm{RFT}}(\theta) = \mathbb{E}_{x\sim\mathcal{D}_{\mathrm{sft}},\, y\sim\pi_{\mathrm{sft}}(\cdot|x)} \left[ \frac{1}{|y|} \sum_{t=1}^{|y|} \mathbb{I}(y = y^*)\, \nabla_\theta \log \pi_\theta(y^t | x, y^{<t}) \right] \qquad (2.2.2.2)$$

### 2.2.3 Online RFT

The only difference between online RFT and RFT is that responses are sampled from the current policy $\pi_\theta$ rather than the fixed $\pi_{\mathrm{sft}}$ as shown in Eq. 2.2.3.1, and differentiating with respect to $\pi_\theta$ obtains Eq. 2.2.3.2.

$$J_{\mathrm{OnRFT}}(\theta) = \mathbb{E}_{x\sim\mathcal{D}_{\mathrm{sft}},\, y\sim\pi_\theta(\cdot|x)} \left[ \frac{1}{|y|} \sum_{t=1}^{|y|} \mathbb{I}(y = y^*) \log \pi_\theta(y^t | x, y^{<t}) \right] \qquad (2.2.3.1)$$

$$\nabla_\theta J_{\mathrm{OnRFT}}(\theta) = \mathbb{E}_{x\sim\mathcal{D}_{\mathrm{sft}},\, y\sim\pi_\theta(\cdot|x)} \left[ \frac{1}{|y|} \sum_{t=1}^{|y|} \mathbb{I}(y = y^*)\, \nabla_\theta \log \pi_\theta(y^t | x, y^{<t}) \right] \qquad (2.2.3.2)$$

The gradient coefficient remains $\mathbb{I}(y = y^*)$, but the on-policy data source allows the model to explore beyond the initial SFT distribution, yielding continued improvement in later training stages where the actor has diverged significantly from the SFT model.

## 2.3 From REINFORCE to Early RL for Language Models

Before the modern RLHF paradigm, several works explored reinforcement learning to directly optimize sequence-level metrics for text generation. In a general Markov decision process (MDP), an agent in state $s_t$ takes action $a_t$, receives reward $r_t$, and transitions to $s_{t+1}$, producing a trajectory $\tau = (s_1, a_1, s_2, a_2, \ldots, s_T, a_T)$. The agent accumulates the *discounted return* $G_t$ with discount factor $\gamma \in [0, 1]$, and the objective is to maximize the expected return $J(\theta)$ in Eq. 2.3.1.

$$G_t = \sum_{t'=t}^{T} \gamma^{t'-t} r_{t'}, \qquad J(\theta) = \mathbb{E}_{\tau \sim \pi_\theta}\left[\sum_{t=1}^{T} \gamma^{t-1} r_t\right] = \mathbb{E}_{\tau \sim \pi_\theta}[G_t] \tag{2.3.1}$$

The form of $r_t$ distinguishes two reward paradigms:

- **Outcome Reward Model (ORM):** Only the terminal state receives reward. With $\gamma = 1$, the return simplifies to $G_t = r(x, y)$, as defined in Equation (2.3.2).

$$r_t = \begin{cases} r(x,y) & t = T \\ 0 & t < T \end{cases}, \qquad G_t = r(x,y) \tag{2.3.2}$$

- **Process Reward Model (PRM)** (Lightman et al., 2023) assigns rewards sparsely at reasoning step boundaries. Let $S_{terminal} \subseteq \{1, \ldots, |y|\}$ be the set of step-ending token positions, as defined in Equation (2.3.3).

$$r_t = \begin{cases} r^p(x, y^{\leq t}) & \text{if } t \in S_{terminal} \\ 0 & \text{otherwise} \end{cases}, \qquad G_t = \sum_{t' \in S_{terminal}, \, t' \geq t} r^p(x, y^{\leq t'}) \tag{2.3.3}$$

### 2.3.1 REINFORCE

The REINFORCE algorithm (Williams, 1992) derives an unbiased gradient estimator for the expected return. The trajectory probability factorizes as $\pi_\theta(\tau) = \prod_{t=1}^{T} \pi_\theta(a_t|s_t)\, p(s_{t+1}|s_t, a_t)$. Starting from the definition of $J(\theta) = J_{\text{REINFORCE}}(\theta) = \mathbb{E}_{\tau \sim \pi_\theta}[G_t] = \mathbb{E}_{\tau \sim \pi_\theta}\left[\sum_{t=1}^{T} \gamma^{t-1} r_t\right]$ in Eq. 2.3.1, the derivation relies on two identities. Firstly, the *log-derivative trick*: $\nabla_\theta \pi_\theta(x) = \pi_\theta(x)\, \nabla_\theta \log \pi_\theta(x)$. Secondly, since the transition dynamics $p(s_{t+1}|s_t, a_t)$ do not depend on $\theta$, the log-trajectory decomposes as $\nabla_\theta \log \pi_\theta(\tau) = \sum_{t'=1}^{T} \nabla_\theta \log \pi_\theta(a_{t'}|s_{t'})$. Applying these identities, the policy gradient derivation proceeds as Eq. 2.3.1.1

$$\nabla_\theta J_{\text{REINFORCE}}(\theta) = \nabla_\theta \sum_\tau \pi_\theta(\tau)\ \sum_{t=1}^{T} \gamma^{t-1} r_t$$

$$= \sum_\tau (\nabla_\theta \pi_\theta(\tau)) \sum_{t=1}^{T} \gamma^{t-1} r_t \qquad \text{(linearity of } \sum \text{)}$$

$$= \sum_\tau \pi_\theta(\tau)\, \nabla_\theta \log \pi_\theta(\tau)\ \sum_{t=1}^{T} \gamma^{t-1} r_t \qquad \text{(log-derivative trick)}$$

$$= \mathbb{E}_{\tau \sim \pi_\theta}\left[\sum_{t=1}^{T} \gamma^{t-1} r_t\, \nabla_\theta \log \pi_\theta(\tau)\right] \qquad (\sum_\tau \pi_\theta[\cdot] = \mathbb{E}_{\pi_\theta}[\cdot])$$

$$= \mathbb{E}_{\tau \sim \pi_\theta}\left[\sum_{t=1}^{T} \gamma^{t-1} r_t \sum_{t'=1}^{T} \nabla_\theta \log \pi_\theta(a_{t'}|s_{t'})\right] \qquad \text{(log-trajectory decomposition)}$$

$$= \mathbb{E}_{\tau \sim \pi_\theta}\left[\sum_{t'=1}^{T} \nabla_\theta \log \pi_\theta(a_{t'}|s_{t'}) \sum_{t=1}^{T} \gamma^{t-1} r_t\right] \qquad \text{(swap finite sums)}$$

$$= \mathbb{E}_{\tau \sim \pi_\theta}\left[\sum_{t'=1}^{T} \nabla_\theta \log \pi_\theta(a_{t'}|s_{t'}) \sum_{t=1}^{t'} \gamma^{t-1} r_t\right] \qquad \text{(causality)}$$

$$= \mathbb{E}_{\tau \sim \pi_\theta}\left[\sum_{t=1}^{T} G_t\, \nabla_\theta \log \pi_\theta(a_t|s_t)\right] \qquad \text{(causality, } t' \to t) \tag{2.3.1.1}$$

where the last step applies *causality*: $r_t$ for $t < t'$ is determined entirely by the trajectory up to step $t$ and cannot depend on the future action $a_{t'}$; moreover, $\mathbb{E}_{a_{t'} \sim \pi_\theta(\cdot|s_{t'})}[\nabla_\theta \log \pi_\theta(a_{t'}|s_{t'})] = \nabla_\theta \sum_{a_{t'}} \pi_\theta(a_{t'}|s_{t'}) = \nabla_\theta 1 = 0$, so past-reward terms vanish in expectation. Dropping these zero-expectation terms and relabeling $t' \to t$, each $\nabla_\theta \log \pi_\theta(a_t|s_t)$ pairs only with its future discounted return $G_t = \sum_{t'=t}^{T} \gamma^{t'-t} r_{t'}$.

Substituting the LLM instantiation $s_t = (x, y^{<t})$, $a_t = y^t$ with outcome reward model and $\gamma = 1$ derives Eq. 2.3.1.2 giving gradient coefficient $\text{GC}_{\text{REINFORCE}}(x, y, t) = r(x, y)$.

$$\nabla_\theta J_{\text{REINFORCE}}(\theta) = \mathbb{E}_{x \sim \mathcal{D}, \, y \sim \pi_\theta(\cdot|x)} \left[ \frac{1}{|y|} \sum_{t=1}^{|y|} r(x, y) \nabla_\theta \log \pi_\theta(y^t|x, y^{<t}) \right] \tag{2.3.1.2}$$

### 2.3.2  Early Applications to Sequence Models

Ranzato et al. (2016) introduced MIXER wherein they applied RL to sequence generation models. They combined cross-entropy pretraining with REINFORCE fine-tuning, using task-level evaluation metrics (e.g., BLEU) as rewards and a learned linear regression baseline over the RNN hidden states to reduce variance without introducing bias. This work established that sequence-level RL objectives could surpass token-level cross-entropy training on translation and summarization, although the approach was validated only on small recurrent models.

### 2.4  RLHF: AC, TRPO and PPO

A central challenge in policy-gradient methods is preventing each update from catastrophically degrading pretrained capabilities. Actor-critic methods (AC) address this by replacing the raw return with a learned advantage, reducing gradient variance. TRPO (Schulman et al., 2017a) constrains updates to a KL trust region, guaranteeing monotonic improvement but requiring expensive second-order computations. PPO (Schulman et al., 2017b) approximates the same constraint with a clipped first-order surrogate, making it the dominant RLHF algorithm at scale.

### 2.4.1  AC methods

AC methods (Konda & Tsitsiklis, 1999) replace the high-variance Monte Carlo return $G_t$ in REINFORCE with a learned advantage. The advantage $A^\pi(s_t, a_t) = Q^\pi(s_t, a_t) - V^\pi(s_t) = Q^\pi(s_t, a_t) - \mathbb{E}_{a_t \sim \pi}[Q^\pi(s_t, a_t)]$ measures how much better action $a_t$ is compared to the average action under $\pi$ in state $s_t$. The policy gradient theorem (Sutton et al., 1999) then writes $J_{\text{AC}}(\theta) = \mathbb{E}_{\tau \sim \pi_\theta}[A^\pi(s_t, a_t)]$ and $\nabla_\theta J_{\text{AC}}(\theta)$ in Eq. 2.4.1.1.

$$\nabla_\theta J_{\text{AC}}(\theta) = \mathbb{E}_{\tau \sim \pi_\theta} \left[ \sum_{t=1}^{T} A^\pi(s_t, a_t) \nabla_\theta \log \pi_\theta(a_t|s_t) \right] \tag{2.4.1.1}$$

Starting from REINFORCE, $\nabla_\theta J_{\text{REINFORCE}}(\theta) = \mathbb{E}_{\tau \sim \pi_\theta} \left[ \sum_{t=1}^{T} G_t \nabla_\theta \log \pi_\theta(a_t|s_t) \right]$. Subtracting any state-dependent baseline $b(s_t)$ from the coefficient preserves unbiasedness, because $\mathbb{E}_{a_t \sim \pi_\theta}[b(s_t)\nabla_\theta \log \pi_\theta(a_t|s_t)] = b(s_t) \nabla_\theta \sum_{a_t} \pi_\theta(a_t|s_t) = b(s_t) \nabla_\theta 1 = 0$. Choosing $b(s_t) = V^\pi(s_t)$ is optimal in that it reduces the variance of the gradient estimator, and the resulting coefficient is exactly the advantage $A^\pi(s_t, a_t) = Q^\pi(s_t, a_t) - V^\pi(s_t)$. Because the advantage is centered, i.e., $\mathbb{E}_{a_t \sim \pi}[A^\pi(s_t, a_t)] = 0$, its magnitude is much smaller than the raw return $G_t$, substantially reducing gradient variance without introducing any bias.

Actor-critic methods require **two separately trained models**: (i) the *actor* (policy model) $\pi_\theta$, updated via the policy gradient in Eq. 2.4.1.1, and (ii) the *critic* (value model) $V_\phi \approx V^\pi$, trained by regressing on target values $L_{\text{critic}}(\phi) = \mathbb{E}_{\tau \sim \pi_\theta} \left[ \sum_{t=1}^{T} (V_\phi(s_t) - \hat{V}_t^\pi)^2 \right]$. The target $\hat{V}_t^\pi$ is either the Monte Carlo return $G_t$ (unbiased, high variance) or the one-step temporal-difference (TD) target $r_t + \gamma V_{\phi^-}(s_{t+1})$ with stop-gradient parameters $\phi^-$ (biased, lower variance). The one-step TD residual $\delta_t = r_t + \gamma V_{\phi^-}(s_{t+1}) - V_\phi(s_t)$ measures the prediction error of the value function at each step and serves as the building block for advantage estimation. Generalized Advantage Estimation (GAE) (Schulman et al., 2018) interpolates between the high-bias one-step TD estimate ($\lambda = 0$) and the high-variance Monte Carlo estimate ($\lambda = 1$) via an exponentially weighted sum of multi-step TD residuals in Eq. 2.4.1.2.

$$A_t^{\mathrm{GAE}(\gamma,\lambda)} = \sum_{l=0}^{T-t} (\gamma\lambda)^l \, \delta_{t+l} \qquad (2.4.1.2)$$

At each iteration, both actor and critic models are updated jointly: the critic minimizes $L_{\mathrm{critic}}(\phi)$ (or its TD variant) while the actor ascends along $\nabla_\theta J_{\mathrm{AC}}(\theta)$ using the GAE advantage estimates in Eq. 2.4.1.2. Substituting the LLM instantiation with $s_t = (x, y^{<t})$, $a_t = y^t$, $\gamma = 1$ yields Eq. 2.4.1.3 giving gradient coefficient $\mathrm{GC}_{\mathrm{AC}}(x,y,t) = A^\pi(x, y^{\leq t})$.

$$\nabla_\theta J_{\mathrm{AC}}(\theta) = \mathbb{E}_{x\sim\mathcal{D},\, y\sim\pi_\theta(\cdot|x)}\left[\sum_{t=1}^{|y|} A^\pi(x, y^{\leq t}) \, \nabla_\theta \log \pi_\theta(y^t|x, y^{<t})\right] \qquad (2.4.1.3)$$

### 2.4.2 TRPO and PPO

The actor-critic gradient places no constraint on how far the policy moves per step; an unconstrained update can invalidate the advantage estimates and catastrophically destroy pretrained capabilities. TRPO (Schulman et al., 2017a) constrains each update to a KL trust region (Eq. 2.4.2.1), originally enforced via the Fisher information matrix, natural gradient, conjugate gradient, and backtracking line search. A later variant relaxes this hard constraint into a KL penalty (Eq. 2.4.2.2) to reduce computational cost.

$$J_{\mathrm{TRPO}}(\theta) = \mathbb{E}_{s_t,a_t\sim\pi_{\theta_{\mathrm{old}}}}\left[\frac{\pi_\theta(a_t|s_t)}{\pi_{\theta_{\mathrm{old}}}(a_t|s_t)}\, A^{\pi_{\theta_{\mathrm{old}}}}(s_t, a_t)\right],\ \mathbb{E}_{s_t\sim\pi_{\theta_{\mathrm{old}}}}[\mathrm{KL}(\pi_{\theta_{\mathrm{old}}}(\cdot|s_t)\,\|\,\pi_\theta(\cdot|s_t))] \leq \varepsilon_{\mathrm{KL}} \qquad (2.4.2.1)$$

$$\begin{aligned} J_{\mathrm{TRPO\text{-}pen}}(\theta) &= \mathbb{E}_{s_t,a_t\sim\pi_{\theta_{\mathrm{old}}}}\left[\frac{\pi_\theta(a_t|s_t)}{\pi_{\theta_{\mathrm{old}}}(a_t|s_t)}\, A^{\pi_{\theta_{\mathrm{old}}}}(s_t, a_t)\right] - \beta\, \mathbb{E}_{s_t\sim\pi_{\theta_{\mathrm{old}}}}[\mathrm{KL}(\pi_{\theta_{\mathrm{old}}}(\cdot|s_t)\,\|\,\pi_\theta(\cdot|s_t))] \\ &= \mathbb{E}_{s_t,a_t\sim\pi_{\theta_{\mathrm{old}}}}\left[\frac{\pi_\theta(a_t|s_t)}{\pi_{\theta_{\mathrm{old}}}(a_t|s_t)}\, A^{\pi_{\theta_{\mathrm{old}}}}(s_t, a_t) - \beta\log\left(\frac{\pi_{\theta_{\mathrm{old}}}(\cdot|s_t)}{\pi_\theta(\cdot|s_t)}\right)\right] \end{aligned} \qquad (2.4.2.2)$$

PPO (Schulman et al., 2017b) replaces the KL penalty with a clipped surrogate. Rollouts from $\pi_{\theta_{\mathrm{old}}}$ are reused over multiple epochs, with $\rho_t = \frac{\pi_\theta(a_t|s_t)}{\pi_{\theta_{\mathrm{old}}}(a_t|s_t)}$ correcting for the distribution mismatch in Eq. 2.4.2.3 for objective and Eq. 2.4.2.4 for gradient where $A_t = A_t^{\mathrm{GAE}(\gamma,\lambda)}$ in Eq. 2.4.1.2.

$$J_{\mathrm{PPO}}(\theta) = \mathbb{E}_{s_t,a_t\sim\pi_{\theta_{\mathrm{old}}}}\left[\sum_{t=1}^{T}\min(\rho_t\, A^{\pi_{\theta_{\mathrm{old}}}}(s_t, a_t),\ \mathrm{clip}(\rho_t, 1-\varepsilon, 1+\varepsilon)\, A^{\pi_{\theta_{\mathrm{old}}}}(s_t, a_t))\right] \qquad (2.4.2.3)$$

$$\nabla_\theta J_{\mathrm{PPO}}(\theta) = \mathbb{E}_{s_t,a_t\sim\pi_{\theta_{\mathrm{old}}}}\left[\sum_{t=1}^{T} c_t\, \rho_t\, A^{\pi_{\theta_{\mathrm{old}}}}(s_t, a_t)\, \nabla_\theta \log \pi_\theta(a_t|s_t)\right] \qquad (2.4.2.4)$$

The clipping indicator $c_t$ equals 0 when $\rho_t$ exceeds $[1-\varepsilon, 1+\varepsilon]$ in the direction favored by $A_t^{\mathrm{GAE}}$, zeroing the gradient, and 1 otherwise as shown in Eq. 2.4.2.5.

$$c_t = 1 - \mathbb{I}\big[\rho_t > 1+\varepsilon,\ A_t^{\mathrm{GAE}} > 0\big] - \mathbb{I}\big[\rho_t < 1-\varepsilon,\ A_t^{\mathrm{GAE}} < 0\big] \qquad (2.4.2.5)$$

Substituting the LLM setting $s_t = (x, y^{<t})$, $a_t = y^t$, $\gamma = 1$ and averaging over response length yields $\mathrm{GC}_{\mathrm{PPO}} = c_t\, \rho_t\, A_t^{\mathrm{GAE}}$ with $\rho_t = \frac{\pi_\theta(y^t|x, y^{<t})}{\pi_{\theta_{\mathrm{old}}}(y^t|x, y^{<t})}$ in Eq. 2.4.2.6 and Eq. 2.4.2.7.

$$J_{\mathrm{PPO}}(\theta) = \mathbb{E}_{x\sim\mathcal{D}_{\mathrm{ppo}},\, y\sim\pi_{\theta_{\mathrm{old}}}(\cdot|x)}\left[\frac{1}{|y|}\sum_{t=1}^{|y|}\min\big(\rho_t\, A_t^{\mathrm{GAE}},\ \mathrm{clip}(\rho_t, 1-\varepsilon, 1+\varepsilon)\, A_t^{\mathrm{GAE}}\big)\right] \qquad (2.4.2.6)$$

$$\nabla_\theta J_{\text{PPO}}(\theta) = \mathbb{E}_{x \sim \mathcal{D}_{\text{ppo}}, \, y \sim \pi_{\theta_{\text{old}}}(\cdot|x)} \left[ \frac{1}{|y|} \sum_{t=1}^{|y|} c_t \, \rho_t \, A_t^{\text{GAE}} \, \nabla_\theta \log \pi_\theta(y^t|x, y^{<t}) \right] \quad (2.4.2.7)$$

In standard RLHF, the per-token KL penalty is folded directly into the reward signal *before* advantage estimation: $r_t = r_\phi(x, y^{\leq t}) - \beta \log \frac{\pi_\theta(y^t|x,y^{<t})}{\pi_{\text{ref}}(y^t|x,y^{<t})}$, so the GAE advantage $A_t^{\text{GAE}}$ already incorporates the KL regularization.

### 2.4.3 RLHF

**RLHF - OpenAI**   Building on the PPO framework derived above, the RLHF pipeline (Christiano et al., 2017; Stiennon et al., 2020; Ouyang et al., 2022; OpenAI et al., 2024) replaces automatic proxy metrics such as BLEU (Papineni et al., 2002), ROUGE (Lin, 2004), or BERTScore (Zhang et al., 2020) with learned human preferences through a three-stage workflow: (1) *SFT* on human demonstrations initializes the reference policy $\pi_{\text{ref}}$ (Eq. 2.2.1.1); (2) *Reward model training* fits a pointwise reward $r_\phi(x, y)$ to human pairwise preferences via the Bradley–Terry (BT) model (Bradley & Terry, 1952); (3) *PPO* optimizes the policy against $r_\phi$ under a KL constraint to $\pi_{\text{ref}}$. The process of SFT is consistent with Section 2.2.1.

*Reward modeling.* For each prompt $x$, $G$ candidate responses (typically $G \in [4, 9]$) are sampled from the SFT policy and presented to human labelers, who rank them from best to worst. These listwise rankings are reduced to $\binom{G}{2}$ pairwise comparisons, i.e., prompt $x$, preferred response $y_w$ and dispreferred response $y_l$. Under the BT model, the probability that response $y_w$ is preferred over $y_l$ given prompt $x$ is defined as $P(y_w \succ y_l \mid x) = \sigma[r_\phi(x, y_w) - r_\phi(x, y_l)]$. Based on this formulation, we train the reward model by minimizing the binary cross-entropy loss in Eq. 2.4.3.1.

$$L_{\text{RM}}(r_\phi) = -\mathbb{E}_{(x, y_w, y_l) \sim \mathcal{D}_{\text{RM}}} \left[ \log \left( \sigma \left( r_\phi(x, y_w) - r_\phi(x, y_l) \right) \right) \right] \quad (2.4.3.1)$$

Training all $\binom{G}{2}$ comparisons jointly as a single batch rather than naively shuffling is essential to prevent overfitting from correlated candidate responses generated by the same prompt.

*Policy optimization.* The policy is optimized using the KL-regularized objective in Eq. 2.4.2.3. Two structural differences distinguish this instantiation from generic PPO. First, generic PPO (inheriting from TRPO) constrains the *forward* KL $D_{\text{KL}}(\pi_{\theta_{\text{old}}} \| \pi_\theta)$ between successive iterates, whereas RLHF penalizes the *reverse* KL $D_{\text{KL}}(\pi_\theta \| \pi_{\text{ref}})$. The reverse KL penalty $D_{\text{KL}}(\pi_\theta \| \pi_{\text{ref}}) = \sum_y \pi_\theta(y) \log \frac{\pi_\theta(y)}{\pi_{\text{ref}}(y)}$ is *zero-forcing*: it diverges to $+\infty$ whenever $\pi_\theta$ places mass outside the support of $\pi_{\text{ref}}$, preventing the RL optimizer from reward-hacking via capability-destroying or degenerate outputs that the reference model considers essentially impossible. Second, the anchor $\pi_{\text{ref}}$ is frozen after SFT, serving as a fixed trust anchor against reward hacking, whereas $\pi_{\theta_{\text{old}}}$ in generic PPO is refreshed every iteration.

In practice, larger reward models (up to 175B) achieved lower validation loss but were unstable and expensive, leading the authors to adopt a single 6B model across all policy sizes. A remaining limitation is that all preference pairs are weighted equally regardless of score margins, motivating later listwise or strength-aware methods (Liu et al., 2025c). Empirically, RLHF improved human preference win-rates on helpfulness, honesty, and harmlessness, reduced hallucination and toxicity (Lin et al., 2022).

**RLHF - Anthropic**   Anthropic's RLHF study (Bai et al., 2022a) explores how data-collection strategy and model scale affect alignment across models ranging from 13M to 52B parameters. They selected crowdworkers for writing quality rather than label agreement, accepting lower researcher-crowdworker agreement ($\sim$63%) in favor of data diversity, and separated "helpful" and "harmless" objectives into distinct datasets, collecting the latter via adversarial red-teaming where crowdworkers choose the *more harmful* model response to probe vulnerabilities. Although these objectives are strongly anti-correlated, i.e., training a PM on one alone yields worse-than-chance accuracy on the other, a single PM trained on combined data learns both effectively, with robustness to data mixture increasing with scale.

A central finding challenges the "alignment tax": RLHF degrades smaller models on standard NLP benchmarks but yields an *alignment bonus* for 13B and 52B models on nearly all zero-shot and few-shot evaluations,

without mixing in pretraining gradients. Alignment is also compatible with specialized skills; RLHF improves Python-finetuned code models on HumanEval, and mixing summarization with HH preference data incurs no degradation in either task. PM accuracy scales roughly log-linearly with model and dataset size. To probe robustness, the authors split preference data 50:50 into separate *train* and *test* PMs, then train RL policies against the train PM while evaluating on the test PM. The two scores agree early but diverge at high reward, i.e., the train PM over-credits the policy, indicating reward over-optimization, though larger PMs are substantially more robust to this effect. The paper also identifies an approximately linear relationship between $\sqrt{D_{\mathrm{KL}}(\pi\|\pi_0)}$ and reward, with learning curves running roughly parallel across policy sizes, suggesting most of RLHF training remains in a perturbative regime around the initial policy. Finally, they propose *iterated online RLHF*, retraining PMs and policies on a roughly weekly cadence with fresh human feedback from deployed models, which yields substantially higher crowdworker Elo ratings.

### 2.4.4 DPO

RLHF with PPO uses a two-stage pipeline, i.e., reward model training followed by RL policy optimization, incurring substantial overhead from multiple models, dual data collection, and overfitting monitoring. To alleviate these complexities, DPO was proposed (Rafailov et al., 2023). Unlike REINFORCE and PPO, DPO was developed directly for the LLM setting and is natively formulated in terms of prompts $x$ and complete responses $y$, bypassing the state-action formalism in MDP. DPO departs from the KL-penalty objective in Eq. 2.4.2.2 by operating entirely *offline*: the preference pairs $(y_w, y_l)$ are sampled from a static dataset typically generated by a separate model or collected from human annotators rather than rolled out from the current policy $\pi_\theta$. Because the training data distribution is fixed and independent of $\pi_\theta$, no importance-sampling ratio $\rho_t$ is required. The optimal policy $\pi^*(y|x)$ is the maximizer of the KL-regularized reward objective in Eq. 2.4.2.2.

Given a reward model $r_\theta(x, y)$, the optimal policy admits a closed-form solution (Eq. 2.4.4.1), where $Z(x)$ is a normalization constant depending only on the prompt, $\pi_{\mathrm{ref}}$ is the reference policy, and $\beta$ controls deviation from it. Rearranging yields the reward expressed in terms of the policy.

$$r_\theta(x, y) = \beta \log\left(\frac{\pi_\theta(y|x)}{\pi_{\mathrm{ref}}(y|x)}\right) + \beta \log Z(x), \qquad Z(x) = \sum_y \pi_{\mathrm{ref}}(y|x) e^{\frac{1}{\beta} r_\theta(x,y)} \tag{2.4.4.1}$$

This formulation enables joint optimization of the reward and policy. However, $Z(x)$ is intractable due to summation over all outputs. DPO removes this term by considering reward differences between preferred and dispreferred responses. Substituting into the BT model yields the pairwise preference probability in Eq. 2.4.4.2, which is then used as a cross-entropy objective to obtain the final DPO loss in Eq. 2.4.4.3.

$$P_\theta(y_w > y_l|x) = \sigma(r_\theta(x, y_w) - r_\theta(x, y_l)) = \sigma\left[\beta \log\left(\frac{\pi_\theta(y_w|x)}{\pi_{\mathrm{ref}}(y_w|x)}\right) - \beta \log\left(\frac{\pi_\theta(y_l|x)}{\pi_{\mathrm{ref}}(y_l|x)}\right)\right] \tag{2.4.4.2}$$

$$\begin{aligned} J_{\mathrm{DPO}}(\pi_\theta) &= \mathbb{E}_{(x, y_w, y_l) \sim \mathcal{D}_{\mathrm{DPO}}} \log P_\theta(y_w > y_l|x) \\ &= \mathbb{E}_{(x, y_w, y_l) \sim \mathcal{D}_{\mathrm{DPO}}} \log\left\{\sigma\left[\beta \log\left(\frac{\pi_\theta(y_w|x)}{\pi_{\mathrm{ref}}(y_w|x)}\right) - \beta \log\left(\frac{\pi_\theta(y_l|x)}{\pi_{\mathrm{ref}}(y_l|x)}\right)\right]\right\} \end{aligned} \tag{2.4.4.3}$$

The gradient of this loss, shown in Eq. 2.4.4.4, increases the likelihood difference between the preferred and rejected responses, with a weighting term that emphasizes hard-to-separate pairs. Expanding to token level and negating (to match the unified maximization form in Eq. 2.1.1.1), the gradient coefficient for each token of the preferred and dispreferred responses is $\mathrm{GC}_{\mathrm{DPO}}^w(x, y_w, y_l) = \beta \sigma(r_\theta(x, y_l) - r_\theta(x, y_w))$ and $\mathrm{GC}_{\mathrm{DPO}}^l(x, y_w, y_l) = -\beta \sigma(r_\theta(x, y_l) - r_\theta(x, y_w))$. The positive coefficient for $y_w$ increases its likelihood while the negative coefficient for $y_l$ decreases it. The gradient coefficient is constant across all tokens within each response, reflecting DPO's response-level formulation.

$$\nabla_\theta J_{\text{DPO}}(\pi_\theta) = \mathbb{E}_{(x,y_w,y_l) \sim \mathcal{D}_{\text{DPO}}}[\beta\sigma(r_\theta(x,y_l) - r_\theta(x,y_w))(\nabla_\theta \log \pi_\theta(y_w|x) - \nabla_\theta \log \pi_\theta(y_l|x))] \quad (2.4.4.4)$$

The authors further showed that reward functions differing only by a prompt-dependent term $f(x)$ are equivalent, implying that $\beta \log \frac{\pi_\theta(y|x)}{\pi_{\text{ref}}(y|x)}$ suffices to recover the same optimal policy. Consequently, DPO directly learns the aligned policy without explicitly training a reward model, reducing RLHF to a simple classification loss. The framework is also extended to noisy labels by replacing the cross-entropy label weights in Eq. 2.4.4.3 with $\epsilon$ and $1 - \epsilon$.

## 2.5 RLVR: GRPO

When ground-truth verification is available (e.g., mathematical correctness, code execution), RLHF can bypass learned reward models entirely and use verifiable rewards directly.

### 2.5.1 GRPO and RLOO

Like DPO, GRPO (Shao et al., 2024) was designed directly for the LLM post-training setting and is natively formulated in terms of prompts and responses, without passing through the generic state–action MDP formalism used by REINFORCE and PPO. GRPO eliminates the value model by estimating the baseline from group scores. For each prompt $x$, a group of $G$ responses $\{y_1, \ldots, y_G\}$ is sampled from $\pi_{\theta_{\text{old}}}$. One of the main architectural distinctions between PPO and GRPO lies in how the KL divergence term is handled. In PPO-based RLHF, the KL penalty is absorbed into the per-token reward $r_t$ before computing GAE advantages (Eq. 2.4.1.2), so the clipped surrogate operates on KL-contaminated advantages; in GRPO, the clipped surrogate acts only on the raw reward-based advantage while the KL term is applied separately and unclipped. This decoupling keeps the advantage $A_{i,t}$ computed purely from raw reward scores via group normalization that replaces the learned value function. Like PPO, GRPO uses a clipped importance-sampling surrogate for the advantage term in Eq. 2.5.1.1

$$J_{\text{GRPO}}(\theta) = \mathbb{E}_{x \sim \mathcal{D}_{\text{GRPO}},\ \{y_i\}_{i=1}^{G} \sim \pi_{\theta_{\text{old}}}(\cdot|x)} \left[ \frac{1}{G} \sum_{i=1}^{G} \frac{1}{|y_i|} \sum_{t=1}^{|y_i|} \Big( \min(\rho_{i,t} A_{i,t},\ \text{clip}(\rho_{i,t}, 1-\varepsilon, 1+\varepsilon)\, A_{i,t}) - \beta\, D_{\text{KL}}^{(i,t)} \Big) \right]$$

$$(2.5.1.1)$$

where $\rho_{i,t} = \frac{\pi_\theta(y_i^t|x,y_i^{<t})}{\pi_{\theta_{\text{old}}}(y_i^t|x,y_i^{<t})}$ is the per-token importance ratio, and $D_{\text{KL}}^{(i,t)} = \frac{\pi_{\text{ref}}(y_i^t|x,y_i^{<t})}{\pi_\theta(y_i^t|x,y_i^{<t})} - \log \frac{\pi_{\text{ref}}(y_i^t|x,y_i^{<t})}{\pi_\theta(y_i^t|x,y_i^{<t})} - 1$ is the Schulman KL estimator (Schulman, 2020) for $\text{KL}(\pi_\theta\|\pi_{\text{ref}})$, guaranteed to be non-negative. Differentiating (with the same clipping indicator $c_{i,t}$ as in Eq. (2.4.2.5), applied to the advantage term) yields Eq. 2.5.1.2.

$$\nabla_\theta J_{\text{GRPO}}(\theta) = \mathbb{E}_{x \sim \mathcal{D}_{\text{GRPO}},\ \{y_i\}_{i=1}^{G} \sim \pi_{\theta_{\text{old}}}(\cdot|x)} \left[ \frac{1}{G} \sum_{i=1}^{G} \frac{1}{|y_i|} \sum_{t=1}^{|y_i|} \Big( c_{i,t}\, \rho_{i,t}\, A_{i,t} \right.$$

$$\left. + \beta \Big( \frac{\pi_{\text{ref}}(y_i^t|x,y_i^{<t})}{\pi_\theta(y_i^t|x,y_i^{<t})} - 1 \Big) \Big) \nabla_\theta \log \pi_\theta(y_i^t|x,y_i^{<t}) \right] \quad (2.5.1.2)$$

This yields the gradient coefficient $\text{GC}_{\text{GRPO}}(x, y_i, t) = c_{i,t}\, \rho_{i,t}\, A_{i,t} + \beta\Big( \frac{\pi_{\text{ref}}(y_i^t|x,y_i^{<t})}{\pi_\theta(y_i^t|x,y_i^{<t})} - 1 \Big)$, combining the clipped advantage term with the KL regularization.

The crucial distinction from PPO lies in the advantage computation. Rather than learning a value function, GRPO estimates the advantage via group normalization. Let $\mu(x)$ and $\sigma(x)$ denote the mean and standard deviation of the group rewards in Eq. 2.5.1.3 and the GRPO advantage for response $y_i$ is then computed using Eq. 2.5.1.4.

$$\mu(x) = \frac{1}{G}\sum_{j=1}^{G} r(x, y_j), \qquad \sigma(x) = \sqrt{\frac{1}{G}\sum_{j=1}^{G}(r(x, y_j) - \mu(x))^2} \tag{2.5.1.3}$$

$$A_{\text{GRPO}}(x, y_i) = \frac{r(x, y_i) - \mu(x)}{\sigma(x)} \tag{2.5.1.4}$$

Notably, $A_{\text{GRPO}}(x, y_i)$ is computed at the *response level*, i.e., depending only on the total reward $r(x, y_i)$ for the entire response, yet is assigned as the gradient coefficient for *every token t* uniformly: $A_{i,t} = A_{\text{GRPO}}(x, y_i)$ for all $t = 1, \ldots, |y_i|$. This contrasts with PPO's token-level GAE advantage $A_t^{\text{GAE}}$, which varies across positions within a response.

Another important distinction from PPO concerns the number of optimization steps per rollout. PPO reuses the same batch of rollouts from $\pi_{\theta_{\text{old}}}$ over multiple gradient steps, updating $\theta$ several times before refreshing $\pi_{\theta_{\text{old}}}$ with a new round of generation. GRPO, by contrast, performs only a single gradient update per rollout batch (Shao et al., 2024), refreshing $\pi_{\theta_{\text{old}}}$ after every update. However, both PPO and GRPO are defined as an on-policy training strategy in this survey.

REINFORCE leave-one-out (RLOO) (Williams, 1992; Kool et al., 2019; Ahmadian et al., 2024) predates GRPO and differs only in how the baseline is defined. In Eq. 2.5.1.5, the mean is computed by excluding the reward of the current response, and the standard deviation is fixed to 1.

$$\mu(x, y_i) = \frac{1}{G-1}\sum_{j=1, j\neq i}^{G} r(x, y_j), \qquad A_{\text{RLOO}}(x, y_i) = r(x, y_i) - \mu(x, y_i) \tag{2.5.1.5}$$

### 2.5.2 DeepSeek R1

DeepSeek-R1 (Guo et al., 2025) demonstrates that reasoning capabilities in LLMs can be incentivized through pure reinforcement learning, without relying on human-annotated reasoning trajectories. Using GRPO (Eq. 2.5.1.1) as the RL algorithm on top of the DeepSeek-V3 base model, the authors first train *DeepSeek-R1-Zero*, which bypasses SFT entirely and uses only rule-based rewards combining accuracy verification (e.g., answer matching, code execution) and format adherence (`<think>`...`</think>` tags). Notably, neural reward models are deliberately avoided to prevent reward hacking during large-scale RL. During training, sophisticated reasoning behaviors, i.e., self-verification, reflection, and dynamic strategy exploration emerge organically without explicit instruction, including a striking "aha moment" where the model spontaneously learns to pause and re-evaluate its reasoning (e.g., generating "Wait, let me reconsider...").

To address practical issues in R1-Zero such as poor readability and language mixing, the full *DeepSeek-R1* adopts a multi-stage pipeline that alternates between SFT and RL:

1. **Cold-start SFT.** Starting from the *base model*, SFT on curated long CoT examples instills a clean `<think>`...`</think>` reasoning format before RL begins.

2. **Reasoning-focused RL.** Starting from the *Stage-1 checkpoint*, GRPO trains with *rule-based rewards only* (accuracy, format, language consistency).

3. **Rejection sampling + SFT.** The *Stage-2 model* generates candidates; the best are selected via rejection sampling and mixed with non-reasoning data ($\sim$800K samples). SFT is then applied to the *Stage-2 checkpoint*, consolidating reasoning while restoring general-purpose capabilities.

4. **All-scenario RL.** Starting from the *Stage-3 checkpoint*, a second GRPO phase adds *model-based preference rewards* for helpfulness and safety alongside rule-based rewards for reasoning, polishing the model across all task types.

The two RL stages differ in both scope and reward design: Stage 2 targets reasoning tasks exclusively with rule-based rewards to safely bootstrap fragile reasoning skills, whereas Stage 4 broadens to all scenarios and introduces neural network-based preference rewards, which is feasible only after Stages 2–3 have solidified the model's reasoning foundation.

### 2.5.3 GRPO and RLVR Enhancement

Various RLVR methods have been proposed as modifications to the GRPO framework. A known degenerate case arises when all responses $y$ within a group are either entirely correct or entirely incorrect, yielding zero advantages and rendering the update trivial. Dynamic sampling (Yu et al., 2025) in Section 5.4.2 has been introduced to address this issue by retaining only non-trivial prompts during training.

Regarding the importance sampling ratio, both PPO and GRPO compute $\rho_{i,t}$ at the token level. Subsequent research has explored alternatives, including sequence-level ratios $\rho_i$ (Zheng et al., 2025) in Section 5.1.2, or removing the importance ratio entirely in Section 5.4.3. For the clipping mechanism, PPO and GRPO both employ symmetric clipping, whereas several later works have proposed asymmetric clipping variants, i.e., clip higher (Yu et al., 2025) in Section 5.4.2.

On the reward side, different objectives can be pursued through the choice of reward signal: outcome rewards in Section 5.2.1, process rewards in Section 5.2.4, and length penalties in Section 5.2.2 have each been investigated to serve distinct training goals. For the baseline, one may either train a separate critic model as in PPO in Section 2.4.2, or compute a baseline from group-level in Section 2.5.1 or batch-level statistics (e.g., mean and standard deviation) in Section 5.4.3, or combine both approaches in Section 5.3.3.

Finally, concerning response length normalization, PPO and GRPO normalize by the individual response length $\frac{1}{|y_i|}$. Alternative approaches include group-level length normalization $\frac{1}{\sum_{i=1}^{G} |y_i|}$ in Section 5.4.2, or omitting length normalization altogether.

### 2.6 Hybrid Post-Training (HPT)

Although the unified view shows that RL and SFT optimize the same objective, they exhibit opposite failure modes: on-policy RL collapses when the model cannot produce any correct rollouts, while SFT suppresses exploration once the model has surpassed its reference demonstrations. The optimal signal therefore depends on model capability, i.e., RL benefits stronger base models far more than weaker ones, whereas SFT remains helpful for both (Lv et al., 2025), motivating a mechanism that selects the more informative gradient estimator per prompt.

HPT instantiates this unified view by dynamically switching between an on-policy RL loss (Dr. GRPO) in Section 5.4.3 and an SFT loss based on per-prompt rollout performance. For each prompt $x$, the model draws $G$ on-policy responses and computes the group mean $\mu(x) = \frac{1}{G} \sum_{i=1}^{G} r(x, y_i)$ where $r(x, y_i) \in \{0, 1\}$ is a rule-based reward verifier. A gate threshold $\kappa$ selects the training signal: when $\mu(x) > \kappa$ the model already shows competence, so RL fosters further exploration; otherwise SFT provides direct guidance as shown in Eq. 2.6.1. The gate defaults to $\kappa = 0$ for Qwen models and $\kappa = 2/8$ for LLaMA.

$$(w_{\text{RL}},\, w_{\text{SFT}}) = \begin{cases} (1,\, 0) & \text{if } \mu(x) > \kappa \\ (0,\, 1) & \text{if } \mu(x) \leq \kappa \end{cases} \qquad L(\theta) = w_{\text{RL}}\, L_{\text{RL}}(\theta) + w_{\text{SFT}}\, L_{\text{SFT}}(\theta) \qquad (2.6.1)$$

### 2.7 The Art of Scaling RL: ScaleRL

While RL compute budgets grow rapidly, the field still lacked a principled basis for predicting how performance scales with compute. Khatri et al. (2026) address this gap by modeling reward $r_C$ as a sigmoidal function of compute $C$ (Eq. 2.7.1), where $A \in [0, 1]$ is the *asymptotic performance ceiling*, $B > 0$ controls compute efficiency, and $C_{\text{mid}}$ is the compute at which half the total gain is achieved. This separates two frequently conflated objectives, i.e., raising the ceiling $A$ vs. accelerating convergence $B$ and enables reliable extrapolation from early, cheap runs to large-scale performance.

$$r_C = r_0 + (A - r_0)\frac{1}{1 + \left(\frac{C_{\mathrm{mid}}}{C}\right)^B} \tag{2.7.1}$$

Guided by a 400,000+ GPU-hour empirical study (Khatri et al., 2026), the authors derive **ScaleRL**: an asynchronous RL recipe built on GRPO (Shao et al., 2024) that integrates existing techniques into a single scalable combination. Concretely, it adopts the CISPO (MiniMax-M1 Team, 2025) in Section 5.1.1.

The framework shows that RL compute can be scaled along four orthogonal axes, each trading off $A$ against $B$ in a predictable way. Model scale is the most impactful: the 17B×16 MoE matches and surpasses the 8B dense model using only one-sixth of the RL compute, with stable sigmoidal scaling throughout. Longer generation budgets and larger batch size raise $A$ at the cost of lower initial $B$. Joint training on math and code preserves the sigmoidal structure per domain, confirming the framework generalizes beyond single-task settings.

## 2.8  Method Selection and Paper Summaries

Before turning to the per-paper summaries, we provide a practical procedure for selecting a post-training strategy given a base model and a target task (Figure 4). The procedure escalates from the cheapest intervention to the most powerful: when zero-shot performance already suffices, prompt engineering avoids training altogether; otherwise, a capability gap is first closed with SFT on gold demonstrations. When preference alignment is needed, the choice turns on exploration and reward source: DPO learns directly from offline preference pairs when such data already cover the target distribution, whereas RLVR is favoured when reward is machine-verifiable, as in mathematics or code. When reward is not verifiable, the remaining on-policy, subjective-reward methods—RLHF with a learned human reward model, RLAIF with an LLM-as-judge reward, and iterative DPO with on-policy preference pairs—are compared. Within these branches, algorithm selection should be guided by the reliability, granularity, and cost of the reward signal. For RLHF, KL-regularized PPO remains a strong default when online rollouts and reward-model scoring are affordable, while DPO-style objectives are preferable when high-quality offline preference pairs are available; iterative DPO provides an intermediate option when periodic on-policy sampling is useful but full RLHF is unnecessarily expensive. For RLVR, the main challenge is often sparse reward and high-variance credit assignment rather than reward subjectivity, making critic-free group-relative methods such as GRPO and DAPO with dynamic sampling and length normalization natural starting points for mathematics and code tasks, while PPO may be useful for long horizon problems. These families are complementary rather than mutually exclusive and are frequently composed in sequence, so the tree should be read as a first-order guide rather than a strict prescription. Detailed RLVR method selection is organized around prompt sampling (Section 3), response sampling (Section 4), and gradient coefficients (Section 5); RLHF and iterative DPO are discussed in Section 6, and offline DPO in Section 7. Across these methods, we analyze nine recurring design dimensions: importance sampling, clipping, reward, baseline, advantage normalization, length normalization, KL divergence, entropy, and the partition function.

Algorithm 1 for RLVR and Algorithm 2 for RLHF share a unified policy-gradient form: at each token position the gradient of the log-policy is scaled by a *gradient coefficient* (GC) that encapsulates every method-specific design choice. Algorithm 1 (RLVR) instantiates this as a three-stage loop: (i) prompt sampling with optional difficulty or reward-based filtering (Section 3); (ii) generating a group of candidate responses scored by verifiable rewards, filtered to a subset via positive/negative splits or advantage thresholds (Section 4); and (iii) token-level GC assembly from an importance-sampling ratio (with optional clipping), an advantage term (reward minus baseline, with normalization and length adjustment), and a KL or entropy regularization (Section 5). For a comprehensive taxonomy of RLVR, we refer the reader to Figure 5.

Algorithm 2 (RLHF/DPO) reuses the same gradient form but branches from different directions: on-policy methods (Section 6) generate fresh rollouts scored by a learned or AI reward model, while offline methods (Section 7) draw pairwise, single, or listwise responses from a fixed dataset, shape the GC through divergence, entropy, or length regularisation, and optionally apply a post-training SFT merge. For a comprehensive taxonomy of RLHF and DPO, we refer the reader to Figure 19.

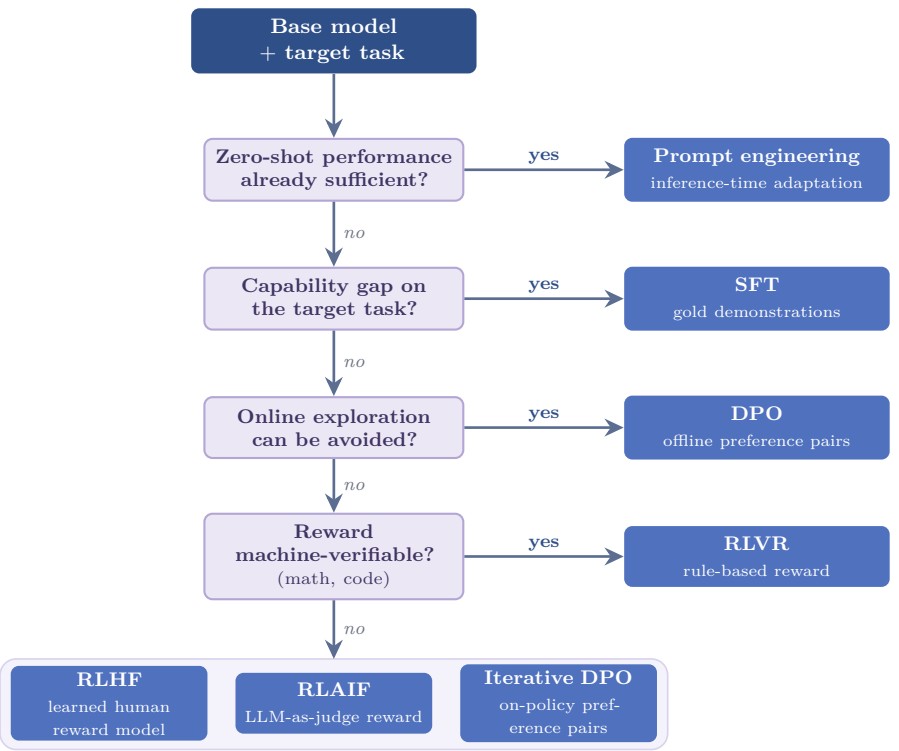

Figure 4: Method-selection decision tree for LLM post-training. From a base model and target task, one follows the "no" spine through increasingly powerful interventions—prompt engineering, SFT, DPO, and RLVR—branching right ("yes") as soon as a condition is met. When reward is not machine-verifiable, the on-policy, subjective-reward methods (RLHF, RLAIF, and iterative DPO) are compared. The tree is a first-order practical guide; the method families are complementary and often combined.

Accompanying per-paper summary tables catalogue these selections along two axes. *Methodology* tables (Tables 2-6 for RLVR; Tables 12-13 for RLHF methods) record design dimensions such as importance sampling, clipping, reward signal, baseline, normalization, KL penalty, and entropy bonus. *Experimental* tables (Tables 7–11 and Tables 14–15) summarise each paper's base model, training data, benchmarks, and compared methods. Each row is a self-contained snapshot; cross-referencing the two axes reveals not only *what* choices were made but *under which conditions* they were validated, with novel elements highlighted to make each paper's contribution easy to identify.

---

**Algorithm 1** RLVR: Key Design Choices

---

**Unified Policy Gradient:** $\nabla_\theta J = \mathbb{E}_{(x,y)\sim\mathcal{D}}\left[\frac{1}{|y|}\sum_{t=1}^{|y|}\mathrm{GC}(x,y,t)\,\nabla_\theta\log\pi_\theta(y^t\mid x,y^{<t})\right]$, $\mathrm{GC}(x,y_i,t) =$

$\underbrace{c_{i,t}\,\rho_{i,t}}_{5.1}\cdot\underbrace{A_{i,t}^{\mathrm{norm}}}_{5.2-5.4}+\underbrace{\beta\left(\frac{\pi_{\mathrm{ref}}}{\pi_\theta}-1\right)}_{5.5}$

1: **Initialise:** $\pi_\theta,\ \pi_{\mathrm{ref}},\ \pi_{\theta_{\mathrm{old}}}\leftarrow\pi_\theta$, group size $G$
2: **for** $iter = 1, 2, \ldots, N$ **do**
3:     **3 Prompt Sampling** sample $x\sim\mathcal{D}$
       **3.1** *Generation:* human-curated | synthetic self-play
       **3.2** *Selection:* static curriculum | adaptive difficulty | reward-based filter
4:     **4 Response Sampling**
       **4.1** *Generation:* draw $\mathcal{G}:=\{y_1,\ldots,y_G\}$, compute $r_i = r(x,y_i)$; on-policy | off-policy (replay / distillation) | tree-structured
       **4.2** *Selection:* filter $\mathcal{G}$ to $\tilde{\mathcal{G}}$; positive/negative split | reward-based filter | advantage-based filter
5:     **for** $y_i\in\tilde{\mathcal{G}};\ t = 1,\ldots,|y_i|$ **do**
6:        **5 Gradient Coefficient**
          **5.1** *IS Ratio $\rho_{i,t}$:* token-level $\rho_{i,t}$ w/ clip $c_{i,t}$ | sequence-level $\rho_i$ w/ clip $c_i$ | none
          **5.2** *Advantage $A_i = r_i - b$:* *Reward:* ORM | PRM | hybrid; *Baseline b:* group mean | leave-one-out | value fn $V_\phi$
          **5.3–5.4** *Norm:* *Adv:* group std | batch std | none; *Len:* response | group | none
          **5.5** *Regularisation:* KL penalty | entropy bonus | none
7:        **Accumulate:** $g \mathrel{+}= \mathrm{GC}(x,y_i^t)\cdot\nabla_\theta\log\pi_\theta(y_i^t\mid x,y_i^{<t})$
8:     $\theta\leftarrow\theta+\eta\,g;\quad\pi_{\theta_{\mathrm{old}}}\leftarrow\pi_\theta$
9: **return** $\pi_\theta$

---

## 3 RLVR: Prompt Sampling

This section surveys how training prompts are *generated*, ranging from human-curated datasets to fully synthetic self-play and *selected*, through static curricula, adaptive difficulty scheduling, and reward-based filtering. Together, these two dimensions determine the quality, diversity, and difficulty of the experiences the policy model encounters.

### 3.1 Prompt Generation

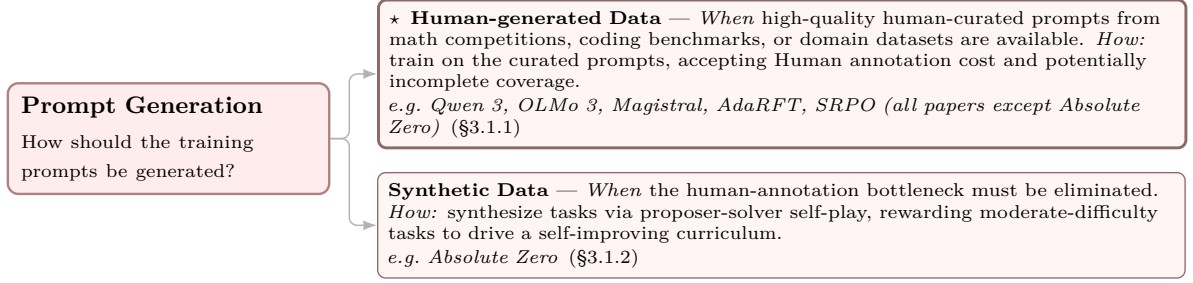

Figure 6: **Prompt generation.** The origin of RLVR training tasks—human-curated prompts versus autonomous proposer–solver self-play synthesis—trading annotation cost and difficulty coverage against a self-improving curriculum (§3.1; ⋆ default).

Prompt generation determines the origin of training tasks (Figure 6), with two paradigms: human-curated data that offers high quality but requires annotation effort, and synthetic self-play that automates task creation to eliminate the human-data bottleneck entirely.

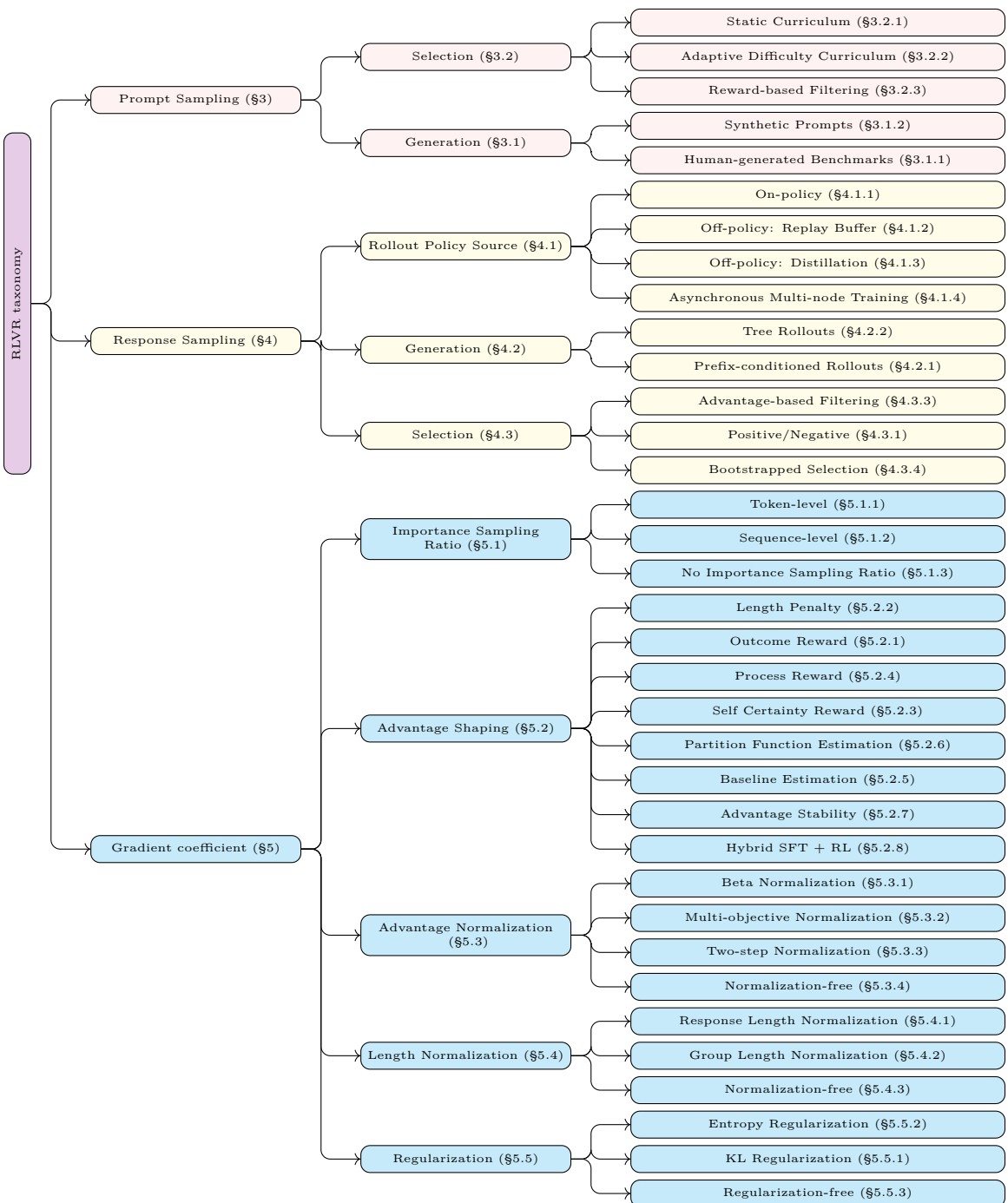

Figure 5: Comprehensive RLVR taxonomy.

### 3.1.1 Human-generated Data

Human-curated prompts from math competitions, coding benchmarks, and domain-specific datasets provide high-quality training signal but require annotation effort and may not cover the full difficulty spectrum. Except Section 3.1.2, all the papers are based on human-generated data.

### 3.1.2 Synthetic Data

Synthetic data methods automatically generate training prompts without human annotation.

**Absolute Zero** The Absolute Zero paradigm (Zhao et al., 2025a) (App. Tables 2 and 7) eliminates dependence on human-curated data by having a single policy $\pi_\theta$ simultaneously propose and solve tasks through self-play. The policy acts in two roles-proposer $\pi_\theta^{\text{propose}}$ and solver $\pi_\theta^{\text{solve}}$-interacting with a code executor that serves as both task validator and reward verifier. Tasks are defined over program triplets $(p, i, o)$ where $o = p(i)$, yielding three reasoning modes: *deduction* (predict $o$ given $p, i$), *abduction* (infer $i$ given $p, o$), and *induction* (synthesize $p$ from $i, o$). To estimate the *learnability* of a proposed task, i.e., whether it lies in the solver's zone of proximal development where $G$ Monte Carlo rollouts $\{y_i\}_{i=1}^{G} \sim \pi_\theta^{\text{solve}}(\cdot \mid x)$ are sampled for a task query $x$ at non-zero temperature, and the average solver success rate $\mu_{\text{solve}}$ is computed. This difficulty estimate drives the proposer reward: tasks that are trivially easy ($\mu_{\text{solve}} = 1$) or completely unsolvable ($\mu_{\text{solve}} = 0$) yield zero reward, while tasks of moderate difficulty, where the solver occasionally succeeds, yield the highest reward, encouraging a self-improving curriculum. The proposer and solver rewards are defined in Eq. 3.1.2.1.

$$r_{\text{solve}}(x, y) = \mathbb{I}(y = y^*), \quad \mu_{\text{solve}}(x) = \frac{1}{G} \sum_{i=1}^{G} r_{\text{solve}}(x, y_i)$$

$$r_{\text{propose}} = \begin{cases} 0 & \text{if } \mu_{\text{solve}}(x) = 0 \\ 1 - \mu_{\text{solve}}(x) & \text{otherwise} \end{cases} \tag{3.1.2.1}$$

Three types of tasks are explored: **deduction, abduction, and induction**. Both roles, i.e., proposer and solver are optimized jointly via **Task-Relative REINFORCE++ (TRR++)**, a GRPO-based method (Eq. 2.5.1.1) that removes KL divergence and adds entropy regularization. Importantly, the $G$ rollouts above serve only the *reward estimation* (propose phase); they are *not* the rollouts used for the RL gradient update. For the update step, TRR++ departs from standard GRPO by generating a single response per prompt ($G = 1$) and computing the advantage baseline across the global batch grouped by (task, role), replacing the per-prompt advantage with a task-role-grouped advantage in Eq. 3.1.2.2

$$A_{\text{task,role}}(x, y) = \frac{r_{\text{task,role}}(x, y) - \mu_{\text{task,role}}}{\sigma_{\text{task,role}}} \tag{3.1.2.2}$$

where task $\in \{\text{ind}, \text{ded}, \text{abd}\}$, role $\in \{\text{propose}, \text{solve}\}$, $\mu_{\text{task,role}}$ and $\sigma_{\text{task,role}}$ are the mean and standard deviation of outcome rewards $r_{\text{task,role}}(x, y)$ across all samples in the batch sharing the same (task, role) pair, yielding six separate baselines: an interpolation between per-question baselines (as in GRPO) and a global baseline. The gradient coefficient is $\text{GC}_{\text{TRR++}}(x, y, t) = c_{i,t} \, \rho_{i,t} \, A_{\text{task,role}}$, following the GRPO gradient coefficient (Eq. 2.5.1.2).

## 3.2 Prompt Selection

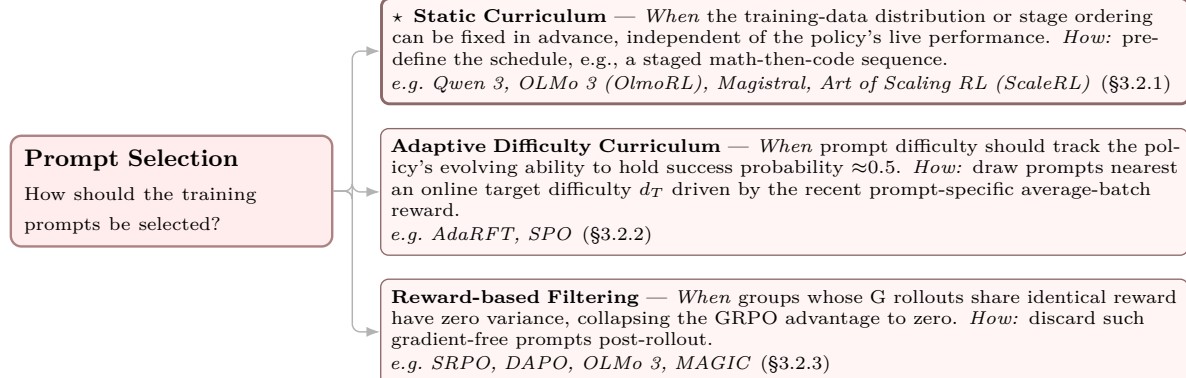

Figure 7: **Prompt selection.** How the prompts trained on at each step are chosen—a fixed curriculum set before training, an adaptive difficulty curriculum tracking live reward, or post-rollout filtering of zero-variance gradient-free groups (§3.2; ⋆ default).

Prompt selection determines *which* prompts the policy trains on at each step (Figure 7), spanning static curricula, adaptive difficulty scheduling, and reward-based filtering that discards prompts yielding no gradient signal.

### 3.2.1 Static Curriculum

Static curricula pre-define the training data distribution or stage ordering before training begins. Aside from the following works, the Art of Scaling RL (Khatri et al., 2026) in Section 2.7 similarly employs a staged math-then-code data sequence.

**Qwen 3**   The Qwen 3 (Yang et al., 2025) (App. Tables 2 and 7) series introduces native thinking-mode with adaptive thinking-budget, allowing users to adjust inference-time compute without switching architectures. The series spans six dense models (0.6B through 32B) and two MoE variants, with flagship Qwen3-235B-A22B activating 22B of 235B parameters per forward pass. Language coverage expands from 29 to 119 languages. Post-training proceeds in four stages for flagship models (with a separate distillation path for lightweight models). Stage 1: Long-CoT Cold Start instills reasoning via filtered math, code, and STEM data. Stage 2: Reasoning RL employs GRPO (Shao et al., 2024) (Eq. 2.5.1.1) with outcome-based verifier reward $r(x, y)$. Entropy regularization is used to control the model's entropy to increase steadily or remain stable. The training uses 3,995 query-verifier pairs, large batch sizes, a high number of rollouts per query, and off-policy training. The AIME'24 score increases from 70.1 to 85.1 over 170 RL training steps. Stage 3: Thinking Mode Fusion via SFT enables seamless switching between thinking and non-thinking modes. Stage 4: General RL combines rule-based rewards with model-based scoring. Strong-to-weak distillation (off-policy followed by on-policy) transfers capabilities to smaller models (0.6B–14B and 30B-A3B), requiring roughly $\frac{1}{10}$ of the GPU hours of RL for comparable gains.

**OLMo 3**   OLMo 3 (Olmo Team et al., 2025) (App. Tables 2 and 7) advances open-source AI by releasing a state-of-the-art language model with complete transparency, providing the full lifecycle including every training stage, checkpoint, data point, and dependency. The training pipeline proceeds through Pretraining, Mid-training, Long context, Thinking SFT, Thinking DPO, and finally RLVR, powered by **OlmoRL**, i.e., a GRPO-based reinforcement learning framework that integrates advances from DAPO (Yu et al., 2025) and Dr. GRPO (Liu et al., 2025d).

OlmoRL introduces six key improvements over vanilla GRPO: (1) zero-gradient filtering removes groups where all rewards are identical to prevent zero-advantage batches, akin to dynamic sampling in DAPO (Yu et al., 2025); (2) active sampling dynamically replenishes filtered slots to maintain a consistent batch size

of non-zero-gradient completions; (3) token-level group length normalization divides the loss by the total number of tokens across the batch rather than per-sample, following DAPO, to eliminate length bias; (4) no KL loss allows less-restricted policy updates without over-optimization or destabilization; (5) asymmetric clipping (clip-higher) sets $\varepsilon_{\text{high}} > \varepsilon_{\text{low}}$ to permit larger updates on high-advantage tokens; and (6) truncated importance sampling caps the log-probability ratio $\rho$ between the inference and training engines to correct for off-policy drift. No standard-deviation normalization is applied to the group advantage, following Dr.GRPO, to avoid amplifying advantages on low-variance (too-easy or too-hard) questions.

**Magistral**  Magistral (Mistral-AI et al., 2025) (App. Tables 2 and 7) is built on GRPO with $G$ generations per prompt $x$ from $\pi_{\theta_{\text{old}}}$ and computes the baseline through the group mean $\mu(x)$ (Eq. 2.5.1.3). With the input outcome reward $r(x, y_i)$, Magistral modifies GRPO (Eq. 2.5.1.1) by: (i) removing the KL divergence term; (ii) using asymmetric clipping with distinct $\varepsilon_{\text{low}}$ and $\varepsilon_{\text{high}}$; (iii) replacing per-response length normalization with group length normalization $\frac{1}{\sum_{i=1}^{G} |y_i|}$; (iv) using batch std for advantage normalization, i.e., first $A_i = r(x, y_i) - \mu(x)$ (group mean only, no group std), then $A_{i,t}^{\text{norm}} = \frac{A_i - \hat{A}^\mu}{\hat{A}^\sigma}$ across the minibatch; and (v) dynamic sampling, which filters out prompts whose responses all receive the same reward (zero advantage), reducing gradient noise. Lastly, entropy regularization is not used.

Reward shaping focuses on formatting (proper use of think tags), correctness (using SymPy for math and C++20 compilers for code), and language consistency via a classifier ensuring the internal monologue matches the user's prompt language. Training data is restricted to problems with verifiable solutions. The distributed, asynchronous RL pipeline has three components: Trainers for weight updates, Generators for rollouts, and Verifiers for reward calculation. Generators operate continuously, receiving weight updates while still generating tokens.

### 3.2.2   Adaptive Difficulty Curriculum

Adaptive difficulty curricula continuously adjust prompt selection based on the current policy's live performance, avoiding rigid fixed schedules. Beyond AdaRFT, SPO (Xu & Ding, 2026) in Section 5.2.5 incorporates difficulty-weighted prompt sampling via a persistent per-prompt Beta value tracker.

**Adaptive Curriculum Reinforcement Finetuning (AdaRFT)**   Shi et al. (2025) (App. Tables 2 and 7) proposes AdaRFT, an adaptive curriculum for reinforcement fine-tuning that overcomes the rigidity of static data filtering (pre-selecting a fixed subset of prompts once, typically by reference difficulty) and fixed difficulty schedules (moving the target difficulty by a pre-defined rule regardless of learning progress). Each prompt $x$ is sent to a reference LLM to generate multiple responses and their average reward is computed as $\mu_{\text{ref}}(x, y)$. Then, the difficulty score is assigned to the prompt as $d = 100 \times (1 - \mu_{\text{ref}}(x, y))$.

AdaRFT maintains a global target difficulty $d_T$ (on the same 0–100 scale as $d$, but not tied to any single prompt) that is updated online based on the current policy's batch-average reward $\mu_\theta(x, y)$ (i.e., the same reward averaged over the selected batch, but under the current policy rather than the reference model) as in Eq. 3.2.2.1

$$d_T \leftarrow \text{clip}(d_T + \eta \cdot \tanh(\alpha \cdot (\mu_\theta(x, y) - p^*)), d_{min}, d_{max}) \tag{3.2.2.1}$$

where $\eta > 0$ is the update step size, $\alpha > 0$ is the sensitivity scaling factor that controls saturation of the reward gap, $p^* \in \mathbb{R}$ is the target reward threshold, and $d_{\min}, d_{\max}$ are the clipping bounds that keep $d_T$ within a valid operating range.

During training, AdaRFT forms a feedback loop: given a target difficulty $d_T$, it samples the $B$ prompts with smallest $|d_i - d_T|$ (closest to $d_T$), then applies any standard RL optimizer (e.g., PPO/GRPO/REINFORCE++) to update the policy model parameters on this batch and compute the resulting batch-average reward $\mu_\theta(x, y)$; finally, it updates $d_T$ by the rule above to keep the success rate near $p^*$ (harder when $\mu_\theta(x, y) > p^*$, easier when $\mu_\theta(x, y) < p^*$). The paper motivates $p^* \approx 0.5$ for binary rewards since the learning signal is strongest near a 50% success rate. AdaRFT is algorithm-agnostic and is instantiated with PPO in this work.

### 3.2.3 Reward-based Filtering

Reward-based filtering removes prompts whose rollouts yield no useful gradient signal, specifically those where all rollouts are correct and the group advantage collapses to zero. Aside from SRPO, dynamic sampling strategies, i.e., DAPO (Yu et al., 2025) in Section 5.4.2, OLMo 3 (Olmo Team et al., 2025) in Section 3.2.1, and MAGIC (He et al., 2025) in Section 5.5.2 also filter zero-advantage groups. In the Response Sampling section, GFPO (Shrivastava et al., 2026) in Section 4.3.2, PODS (Xu et al., 2025) in Section 4.3.3, and CPPO (Lin et al., 2025) in Section 4.3.3 apply analogous filtering at the *response* level.

**Two-Staged history-Resampling Policy Optimization (SRPO)**    Vanilla GRPO faces three bottlenecks in cross-domain settings: (i) a response-length conflict: math benefits from long CoT while coding favors short outputs; (ii) degenerate groups where all $G$ rollouts share the same reward, making $A_{\mathrm{GRPO}}(x, y_i)$ (Eq. (2.5.1.4)) collapse to zero; and (iii) premature saturation from insufficient data difficulty. SRPO (App. Tables 2 and 7) introduces two mechanisms on top of the GRPO objective (Eq. (2.5.1.1)). **Two-stage training**: Stage 1 trains on math-only data to develop extended CoT (reflective pauses, backtracking); Stage 2 adds coding data, building on the reasoning foundation. **History Resampling (HR)**: at each epoch boundary, prompts $x$ for which all $G$ rollouts were correct are filtered out, retaining only **mixed- or all-incorrect-outcome prompts**. Mixed-outcome prompts guarantee non-zero reward variance and thus non-trivial $A_{\mathrm{GRPO}}$ and all-incorrect prompts currently yield zero advantage but are kept because the updated $\pi_\theta$ may partially solve them in later epochs, similarly to curriculum learning. The objective follows the GRPO formulation (Eq. (2.5.1.1)), using the input outcome reward $r(x, y_i)$. Additionally, the advantage is zeroed for responses exceeding the maximum length. The gradient coefficient is $\mathrm{GC}_{\mathrm{SRPO}}(x, y_i, t) = c_{i,t}\, \rho_{i,t}\, A_{i,t}$.

## 4   RLVR: Response Sampling

This section surveys how RLVR training responses are constructed and filtered. **Rollout Policy Source** covers which policy generates the rollouts, ranging from on-policy sampling to off-policy replay, distillation, and asynchronous multi-node training. **Response Generation** covers prefix-conditioned and tree-structured rollout strategies. **Response Selection** surveys filtering methods that determine which responses enter the policy gradient update.

In Figure 8, five terms of response sampling, i.e., on-policy, off-policy, advantage filtering, prefix-conditioned rollouts and tree rollouts are illustrated with examples. On-policy and off-policy refer to whether responses are sampled from the current training model itself or from a separate model; advantage filtering keeps only the most informative ones, while prefix-conditioned rollouts generate multiple completions branching from a shared prefix, and tree rollouts extend this idea by building a branching tree of partial rollouts that are selectively expanded into full responses.

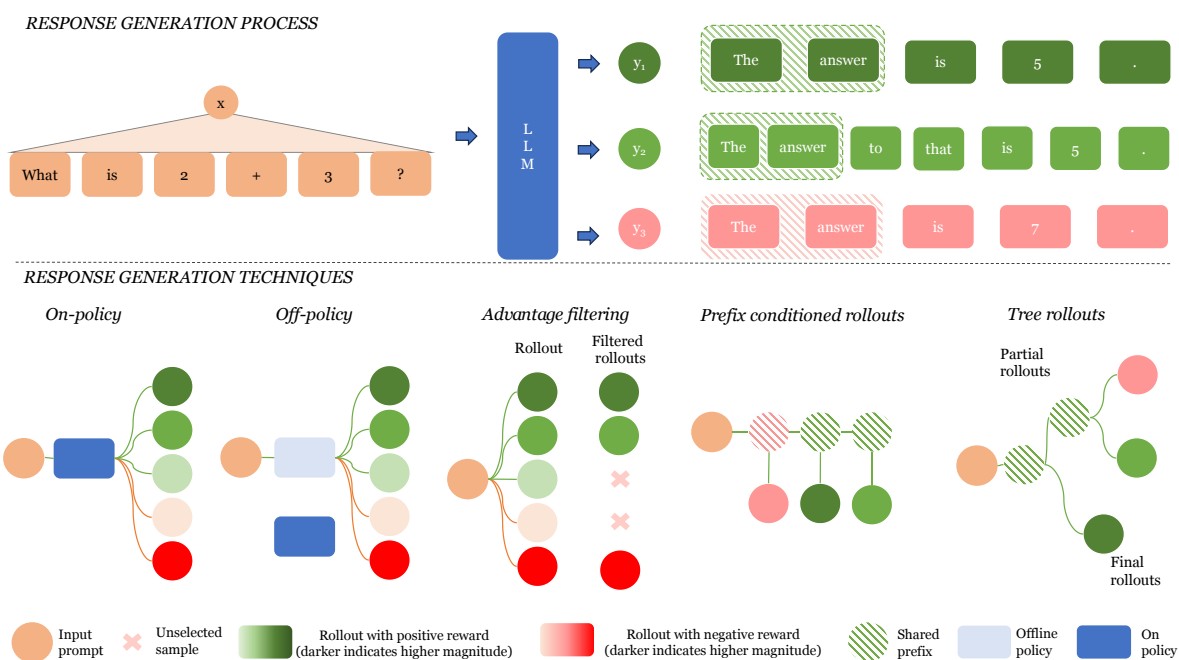

Figure 8: Response Sampling including On-policy, Off-policy, Advantage filtering, Prefix-conditioned rollouts and Tree rollouts.

## 4.1   Rollout Policy Source

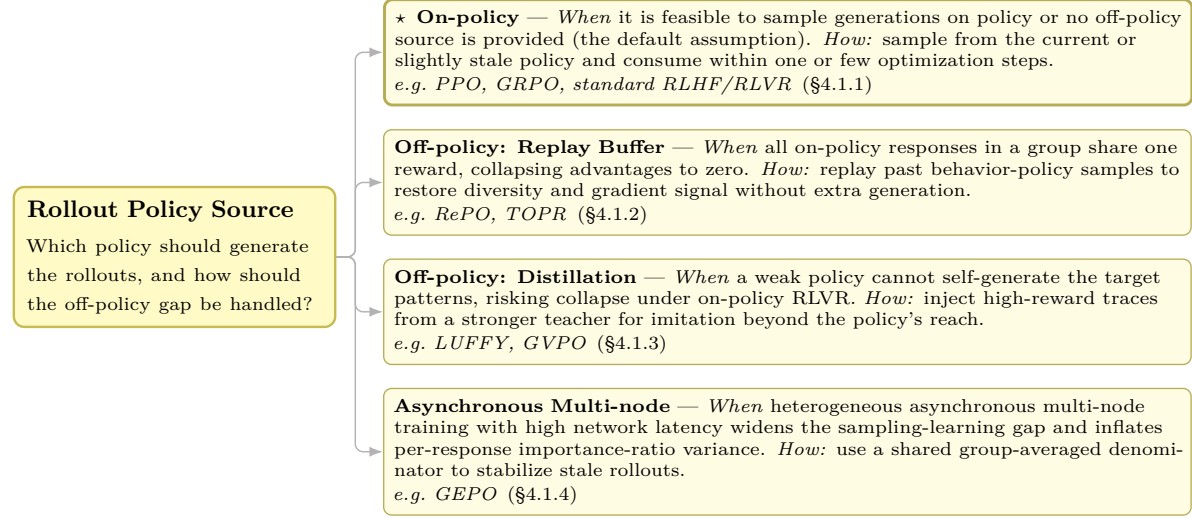

Figure 9: **Rollout policy source.** Which distribution generates training rollouts and how the on/off-policy mismatch is corrected—the current policy, a replay buffer of past behavior, a stronger teacher's traces, or infrastructure-stale asynchronous rollouts (§4.2; ⋆ default).

Standard GRPO (Shao et al., 2024) samples $G$ independent responses on-policy per prompt (Figure 9). This subsection surveys modifications to that baseline: on-policy rollouts (default); off-policy replay buffers; distillation from a stronger teacher; asynchronous multi-node training.

### 4.1.1 On-policy Rollouts

On-policy rollouts serve as the foundation for PPO (Schulman et al., 2017b), GRPO (Shao et al., 2024), and RLHF and RLVR approaches. In these methods, $G$ responses are sampled from the current policy $\pi_{\theta_{\text{old}}}$ and used to update the model for one or more optimization steps. Although the rollouts may originate from slightly earlier versions of the model, they are still generally treated as on-policy data. Unless explicitly stated otherwise, we assume methods follow the on-policy learning paradigm.

### 4.1.2 Off-policy: Replay Buffer

The following works augment GRPO (Shao et al., 2024) with off-policy responses to improve data efficiency and stabilize training when on-policy rollouts collapse.

**Replay-Enhanced Policy Optimization (RePO)**  GRPO's reliance on purely on-policy samples is both computationally expensive and fragile: when all sampled responses receive the same reward, advantages collapse to zero and provide no gradient signal. RePO (Li et al., 2025a) (App. Tables 3 and 8) augments on-policy training $G^{\text{on}}$ from $x \sim \mathcal{D}$, $\{y_i^{\text{on}}\}_{i=1}^{G^{\text{on}}} \sim \pi_{\theta_{\text{old}}}(\cdot|x)$ with $G^{\text{off}}$ off-policy responses retrieved from a replay buffer $\mathcal{B}$: $\{y_i^{\text{off}}, \pi_b(y_i^{\text{off}}|x)\}_{i=1}^{G^{\text{off}}} \sim \mathcal{B}(x,y)$, where $\pi_b$ is the behavior policy that originally generated each replay sample, broadening the sample set per prompt without additional rollouts. The combined objective is given in Eq. 4.1.2.1.

$$J_{\text{RePO}}(\theta; \mathcal{B}) = \underbrace{J_{\text{on}}(\theta)}_{\text{current samples}} + \underbrace{J_{\text{off}}(\theta; \mathcal{B})}_{\text{replay samples}} \tag{4.1.2.1}$$

Each term follows the GRPO objective (Eq. 2.5.1.1) with the KL penalty removed, using the outcome reward $r(x, y_i^\star)$ for $\star \in \{\text{on}, \text{off}\}$. The importance ratios are $\rho_{i,t}^{\text{on}} = \frac{\pi_\theta(y_i^{\text{on},t}|x, y_i^{\text{on},<t})}{\pi_{\theta_{\text{old}}}(y_i^{\text{on},t}|x, y_i^{\text{on},<t})}$ and $\rho_{i,t}^{\text{off}} = \frac{\pi_\theta(y_i^{\text{off},t}|x, y_i^{\text{off},<t})}{\pi_b(y_i^{\text{off},t}|x, y_i^{\text{off},<t})}$. RePO estimates advantages separately for on-policy and off-policy groups (split strategy): $A_i^\star = \frac{r(x, y_i^\star) - \mu(\mathcal{G}^\star)}{\sigma(\mathcal{G}^\star)}$, where $\mathcal{G}^\star = \{r(x, y_i^\star)\}_{i=1}^{G^\star}$ and $\mu(\cdot)$, $\sigma(\cdot)$ denote the group mean and standard deviation, respectively. The gradient coefficient is $\text{GC}_{\text{RePO}}(x, y_i^\star, t) = c_{i,t}\, \rho_{i,t}^\star\, A_{i,t}^\star$, identical to the GRPO gradient coefficient (Eq. 2.5.1.2) but without the KL regularization term. The off-policy update activates only after a threshold epoch. Lately, four retrieval strategies (Full-scope, Recency-based, Reward-oriented, Variance-driven) are evaluated for sampling from the replay buffer and Recency-based and Reward-oriented consistently outperform the others.

**Tapered Off-Policy REINFORCE (TOPR)**  In the off-policy setting, $\pi_\theta$ is trained for many steps on data from a frozen behavior policy $\pi_b$, leading to divergence. With negative rewards, naive off-policy REINFORCE collapses as $\pi_\theta$ pushes their probabilities to zero, making the objective $\log(\pi_\theta)$ unbounded and pushing model logits toward negative infinity, eventually causing degenerate behavior. TOPR (Roux et al., 2025) (App. Tables 3 and 8) resolves this through an asymmetric taper applied separately per reward sign, enabling KL-free, fully offline fine-tuning that remains stable even as $\pi_\theta$ diverges substantially from $\pi_b$. Concretely, TOPR samples $G$ responses per prompt from the frozen behavior policy $\pi_b$, labels each with a binary reward $r(x, y) \in \{-1, +1\}$, and optimizes $\pi_\theta$ over this off-policy dataset. The off-policy correction enters via the sequence-level importance ratio $\rho = \frac{\pi_\theta(y|x)}{\pi_b(y|x)} = \prod_{t=1}^{|y|} \frac{\pi_\theta(y^t|x, y^{<t})}{\pi_b(y^t|x, y^{<t})}$, which quantifies how much the current policy $\pi_\theta$ has drifted from the data-generating distribution $\pi_b$.

TOPR applies an asymmetric taper $\mathcal{T}$ to $\rho$ differently for each reward sign. For positive responses ($r(x, y) \geq 0$), $\mathcal{T}(\rho, 1, 1)$ uses the log-ratio surrogate $1 + \log \rho$, whose gradient with respect to $\theta$ reduces to $\nabla \log \pi_\theta$, i.e., a gradient coefficient of 1 independent of $\rho$, yielding a SFT-style update that remains effective even when $\pi_\theta$ has drifted far from $\pi_b$ and $\rho \ll 1$, the regime where standard importance-weighted updates would vanish. For negative responses ($r(x, y) < 0$), $\mathcal{T}(\rho, 0, 1)$ clips $\rho$ to $[0, 1]$, bounding the gradient magnitude and preventing the destructive blow-up that arises in importance-weighted updates when $\rho \gg 1$. The objective is Eq. 4.1.2.2, where $\mathcal{T}$ is defined in Eq. 4.1.2.3.

$$
\begin{aligned}
J_{\text{TOPR}}(\theta) = \mathbb{E}_{x \sim \mathcal{D},\, y \sim \pi_b(\cdot|x)} \Big[ &\mathbb{I}\{r(x, y) \geq 0\}\, \mathcal{T}(\rho,\, 1,\, 1)\, r(x, y) \\
&+ \mathbb{I}\{r(x, y) < 0\}\, \mathcal{T}(\rho,\, 0,\, 1)\, r(x, y) \Big]
\end{aligned}
\tag{4.1.2.2}
$$

$$\mathcal{T}(\rho, a, b) = \begin{cases} a\left(1 + \log\frac{\rho}{a}\right) & \text{if } \rho < a \\ b\left(1 + \log\frac{\rho}{b}\right) & \text{if } \rho > b \\ \rho & \text{if } \rho \in [a, b] \end{cases} \tag{4.1.2.3}$$

The resulting sequence-level gradient coefficient is $\text{GC}_{\text{TOPR}}(x, y) = \mathbb{I}\{r \geq 0\}\, r(x, y) + \mathbb{I}\{r < 0\}\, \text{clip}(\rho, 0, 1)\, r(x, y)$. Crucially, positive-reward responses carry a constant gradient weight of 1 regardless of how much $\pi_\theta$ has drifted from $\pi_b$, while negative-reward responses are progressively downweighted as $\pi_\theta$ moves away from the behavior policy.

### 4.1.3 Off-policy: Distillation

The following works augment on-policy rollouts with demonstrations from a stronger fixed teacher model. Aside from these works, Prefix-RFT (Huang et al., 2025) in Section 4.2.1 pursues a related goal, i.e., anchoring rollouts to expert prefixes.

**Learning to reason Under oFF-policY guidance (LUFFY)** On-policy RLVR is fundamentally limited by the model's initial capabilities: it cannot learn reasoning patterns that it cannot already self-generate, which often leads to training collapse in weaker models. To address this limitation, LUFFY (Yan et al., 2025) (App. Tables 3 and 8) incorporates off-policy reasoning traces from a stronger teacher (e.g., DeepSeek-R1), exposing the policy to strategies beyond its current reach. Concretely, LUFFY builds on GRPO and extends it to a mixed-policy setting. For each prompt $x$, it combines on-policy rollouts $\mathcal{G}_{\text{on}} = \{y_i \sim \pi_{\theta_{\text{old}}}(\cdot \mid x)\}_{i=1}^{G_{\text{on}}}$ with off-policy reasoning traces $\mathcal{G}_{\text{off}} = \{y_j \sim \pi_b(\cdot \mid x)\}_{j=1}^{G_{\text{off}}}$, dynamically balancing exploration and imitation. Since off-policy traces consistently yield high rewards, they receive high positive advantages when the model's own rollouts fail, enabling imitation; once the model succeeds, on-policy solutions dominate, preserving exploration. The advantage for each response, with the outcome reward $r(x, y)$, is computed via group mean over all responses to the same prompt in Eq. 4.1.3.1.

$$A_i = r(x, y_i) - \mu(x), \quad \mu(x) = \frac{1}{G}\sum_{i=1}^{G} r(x, y_i), \quad y_i \in \{\mathcal{G}_{\text{on}} \cup \mathcal{G}_{\text{off}}\} \tag{4.1.3.1}$$

The standard deviation normalization is removed following Dr. GRPO (Liu et al., 2025d). The mixed-policy objective applies a **policy shaping** function $f(u) = \frac{u}{u+\kappa}$ (with $\kappa = 0.1$) to the off-policy term and the standard clipped surrogate to the on-policy term, with group length normalization. In practice, the off-policy behavior policy $\pi_b$ is set to 1 (i.e., the teacher model's token probabilities are not computed), so the off-policy importance ratio simplifies to $\pi_\theta(y_i^t \mid x, y_i^{<t})$, and the clip operation is omitted for off-policy rollouts. The objective is given in Eq. 4.1.3.2.

$$J_{\text{LUFFY}}(\theta) = \mathbb{E}_{x \sim \mathcal{D},\, \{y_i\}_{i=1}^{G_{\text{on}}} \sim \pi_{\theta_{\text{old}}}(\cdot|x),\, \{y_j\}_{j=1}^{G_{\text{off}}} \sim \pi_b(\cdot|x)} \frac{1}{\sum_{y_j \in \mathcal{G}_{\text{off}}} |y_j| + \sum_{y_i \in \mathcal{G}_{\text{on}}} |y_i|}$$

$$\left[ \underbrace{\sum_{y_j \in \mathcal{G}_{\text{off}}} \sum_{t=1}^{|y_j|} f\big(\pi_\theta(y_j^t \mid x, y_j^{<t})\big) A_j}_{\text{off-policy with shaping}} + \underbrace{\sum_{y_i \in \mathcal{G}_{\text{on}}} \sum_{t=1}^{|y_i|} \min(\rho_{i,t} A_i,\, \text{clip}(\rho_{i,t}, 1-\varepsilon, 1+\varepsilon)\, A_i)}_{\text{on-policy with clipping}} \right] \tag{4.1.3.2}$$

The policy shaping function amplifies learning signals for low-probability but crucial tokens from off-policy traces, preventing entropy collapse. The gradient coefficient for the on-policy term is $\text{GC}_{\text{LUFFY,on}}(x, y_i, t) = c_{i,t}\, \rho_{i,t}\, A_i$, identical to the GRPO gradient coefficient (Eq. 2.5.1.2) without the KL regularization term. For the off-policy term, differentiating through the shaping function yields $\text{GC}_{\text{LUFFY,off}}(x, y_j, t) = f'\big(\pi_\theta(y_j^t \mid x, y_j^{<t})\big) A_j$, which up-weights low-probability tokens relative to the linear weighting in standard importance sampling.

29

**Group Variance Policy Optimization (GVPO)**  GRPO suffers from training instability due to importance sampling weights and gradient clipping, and is further limited by its inherently on-policy sampling scheme. GVPO (Zhang et al., 2025b) (App. Tables 3 and 8) incorporates the closed-form solution of the KL-constrained reward objective (Eq. (2.1.1)) directly into the gradient weights, thereby improving stability and relaxing the strict reliance on on-policy updates. The implicit reward is consistent with DPO, i.e., $r_\theta(x, y) = \beta \log \frac{\pi_\theta(y|x)}{\pi_{\theta_{\text{old}}}(y|x)} + \beta \log Z(x)$. Although the partition function $Z(x)$ is intractable, it cancels out when the per-group weights satisfy $\sum_{i=1}^{G} w_i = 0$. Let $\pi_b$ denote an arbitrary behavior policy, and GVPO's optimality guarantee holds for any $\pi_b$ whose support covers that of $\pi_{\text{ref}}$, enabling off-policy training without importance sampling. The resulting objective, with input reward $r(x, y_i)$, minimizes the mean squared error between the central distances of implicit and actual rewards, as shown in Eq. 4.1.3.3

$$J_{\text{GVPO}}(\theta) = \mathbb{E}_{x \sim \mathcal{D}, \ \{y_i\}_{i=1}^{G} \sim \pi_b(\cdot|x)} \left[ -\frac{1}{G} \sum_{i=1}^{G} \left[ (r_\theta(x, y_i) - \mu_\theta(x)) - (r(x, y_i) - \mu(x)) \right]^2 \right] \quad (4.1.3.3)$$

where $\mu_\theta(x) = \frac{1}{G} \sum_{i=1}^{G} r_\theta(x, y_i)$ is its group mean, $\mu(x) = \frac{1}{G} \sum_{i=1}^{G} r(x, y_i)$ is the group mean of actual rewards. The sampling distribution $\pi_b$ can differ from $\pi_\theta$, enabling off-policy training without importance sampling. The gradient coefficient is $\text{GC}_{\text{GVPO}}(x, y_i, t) = \beta[(r(x, y_i) - \mu(x)) - (r_\theta(x, y_i) - \mu_\theta(x))]$, constant across all tokens $t$ within a response. Unlike GRPO, GVPO uses no importance sampling ratios, no clipping, and no group standard deviation normalization in the advantage.

### 4.1.4 Asynchronous Multi-node Training

GEPO (Zhang et al., 2025a) targets distribution shift caused by stale rollout policies in heterogeneous asynchronous training by replacing the per-response IS ratio with a group-averaged denominator.

**Group Expectation Policy Optimization (GEPO)**  GEPO (Zhang et al., 2025a) (App. Tables 3 and 8) addresses training instability in *heterogeneous* asynchronous RL, i.e., environments where high network latency widens the gap between the sampling policy $\pi_{\theta_{\text{old}}}$ and the learning policy $\pi_\theta$, causing the sequence-level importance ratio $\rho_i = \frac{\pi_\theta(y_i|x)}{\pi_{\theta_{\text{old}}}(y_i|x)}$ to have high variance and destabilize training. The key insight is to replace the per-response denominator $\pi_{\theta_{\text{old}}}(y_i \mid x)$ with a single *group-level* estimate shared by all $G$ responses. For each prompt $x$ and group $\{y_1, \ldots, y_G\} \sim \pi_{\theta_{\text{old}}}(\cdot \mid x)$, let $\tilde{\pi}(y_i \mid x) = \frac{\pi_{\theta_{\text{old}}}(y_i|x)}{\sum_{j=1}^{G} \pi_{\theta_{\text{old}}}(y_j|x)}$ be the within-group normalized weight. The group-level denominator is the expectation of $\pi_{\theta_{\text{old}}}(y \mid x)$ *under $\tilde{\pi}$* in Eq. 4.1.4.1.

$$\mathbb{E}_{\tilde{\pi}}[\pi_{\theta_{\text{old}}}(y \mid x)] = \sum_{i=1}^{G} \tilde{\pi}(y_i \mid x) \, \pi_{\theta_{\text{old}}}(y_i \mid x) = \frac{\sum_{i=1}^{G} \pi_{\theta_{\text{old}}}(y_i \mid x)^2}{\sum_{i=1}^{G} \pi_{\theta_{\text{old}}}(y_i \mid x)} \quad (4.1.4.1)$$

GEPO replaces $\rho_i$ with a *sequence-level importance ratio* in Eq. 4.1.4.2.

$$\rho_i^{\text{GEPO}} = \frac{\pi_\theta(y_i \mid x)}{\mathbb{E}_{\tilde{\pi}}[\pi_{\theta_{\text{old}}}(y \mid x)]} \quad (4.1.4.2)$$

Substituting $\rho_i^{\text{GEPO}}$ for $\rho_{i,t}$ in the GRPO objective and gradient (Eqs. (2.5.1.1)–(2.5.1.2)) yields the GEPO update directly. The resulting gradient coefficient is $\text{GC}_{\text{GEPO}}(x, y_i, t) = c_{i,t} \, \rho_i^{\text{GEPO}} \, A_{i,t} + \beta \left( \frac{\pi_{\text{ref}}(y_i^t | x, y_i^{<t})}{\pi_\theta(y_i^t | x, y_i^{<t})} - 1 \right)$, identical to the GRPO gradient coefficient (Eq. 2.5.1.2) but with the group-level importance ratio $\rho_i^{\text{GEPO}}$ replacing the per-token ratio $\rho_{i,t}$.

## 4.2 Response Generation

Figure 10: **Response generation (rollout structure).** How the rollout group departs from flat on-policy sampling—expert-prefix conditioning that densifies sparse outcome rewards, or tree/Monte-Carlo/serial structures that extract finer-grained credit (§4.2; ⋆ default).

Response generation determines how the rollout group departs from flat, independent on-policy sampling (Figure 10). Prefix-conditioned rollouts anchor generation to an expert prefix to densify sparse outcome rewards, while tree-structured rollouts introduce non-flat tree, Monte-Carlo, or serial structures for finer-grained credit assignment.

### 4.2.1 Prefix-conditioned Rollouts

The following works address the cold-start problem by anchoring rollouts to expert prefixes, providing a denser reward signal for models that cannot solve problems from scratch. SRFT (Fu et al., 2025) in Section 5.2.8 and HPT (Lv et al., 2025) in Section 2.6 pursue the complementary goal of unifying SFT and RL in a single gradient objective.

**Branch Rollouts and Expert Anchors for Densified Rewards (BREAD)** The standard $\text{SFT} + \text{RL}$ pipeline can fail for small language models when (1) expert traces are too complex for the student to express, rendering SFT uninformative, or (2) a poorly initialized policy rarely produces correct traces, leaving RL with sparse rewards. BREAD (Zhang et al., 2025f) (App. Tables 3 and 8) is based on GRPO with $G$ generations per prompt $x \sim \mathcal{D}$, $\{y_i\}_{i=1}^{G} \sim \pi_{\theta_{\text{old}}}(\cdot | x, y^{\text{pre}})$, and computes the baseline through the group mean $\mu(x)$ and group standard deviation $\sigma(x)$ (Eq. 2.5.1.3). BREAD addresses both problems of GRPO by conditioning rollouts on a short expert prefix $y^{\text{pre}}$ rather than imitating full expert solutions. An **Episode Anchor Search (EAS)** binary-searches the expert trace for the shortest prefix such that rollouts $\{y_i\}_{i=1}^{G} \sim \pi_{\theta_{\text{old}}}(\cdot \mid x, y^{\text{pre}})$ achieve nontrivial success; when the policy already solves the prompt, no prefix is used. As training progresses the anchor shortens, yielding a self-paced curriculum that densifies rewards. The objective follows the GRPO formulation (Eq. 2.5.1.1), using the outcome reward $r(x, y_i)$, with the data source changed to prefix-conditioned rollouts $(x, y^{\text{pre}})$, where the prefix $y^{\text{pre}}$ is an expert anchor selected via EAS. The per-token importance ratio becomes $\rho_{i,t} = \frac{\pi_\theta(y_i^t | x, y^{\text{pre}}, y_i^{<t})}{\pi_{\theta_{\text{old}}}(y_i^t | x, y^{\text{pre}}, y_i^{<t})}$, and the advantage is computed via group normalization in Eq. 2.5.1.4. The gradient coefficient is $\text{GC}_{\text{BREAD}}(x, y_i, t) = c_{i,t} \, \rho_{i,t} \, A_{i,t} + \beta \left( \frac{\pi_{\text{ref}}(y_i^t | x, y^{\text{pre}}, y_i^{<t})}{\pi_\theta(y_i^t | x, y^{\text{pre}}, y_i^{<t})} - 1 \right)$, identical to the GRPO gradient coefficient (Eq. 2.5.1.2) but with prefix-conditioned rollouts.

**Prefix Reinforcement Fine-Tuning (Prefix-RFT)** Unlike BREAD, which conditions all $G$ rollouts on an EAS-selected expert prefix to densify rewards for cold-start models, Prefix-RFT (Huang et al., 2025) (App. Tables 3 and 8) targets the formal integration of SFT and RFT within a single training step: it mixes one hybrid trajectory (expert prefix concatenated with on-policy continuation) into the standard rollout group, and applies an entropy-based gradient mask on prefix tokens to selectively incorporate dense demonstration supervision without allowing it to dominate the RFT exploration signal. For each prompt $x$ with an offline demonstration $y^*$, the authors:

1. Sample $G - 1$ on-policy rollouts $y_i \sim \pi_{\theta_{\text{old}}}(\cdot \mid x)$.

2. Construct one hybrid trajectory by concatenating a prefix of length $L$ from $y^*$ with an on-policy continuation from $\pi_{\theta_{\text{old}}}(\cdot \mid x, y^{<t})$.

Following the design of Dr. GRPO, the authors remove standard deviation normalization, length normalization, and the KL penalty, and the advantage is defined as $A(x, y_i) = r(x, y_i) - \mu(x)$. A key innovation is an entropy-based mask $m_{i,t}$ that selectively gates gradients of prefix tokens according to policy entropy as defined in Eq. (4.2.1.1)

$$m_{i,t} = \mathbb{I}[t > L] + \mathbb{I}[t \leq L] \cdot \mathbb{I}\big[H(\pi_\theta(\cdot \mid x, y_i^{<t})) \geq \eta\big] \tag{4.2.1.1}$$

where $\eta$ is the $k$-th percentile entropy threshold: only the top-$k\%$ highest-entropy prefix tokens receive gradients, targeting uncertain junctures while avoiding sharp distribution shifts. The gradient coefficient is $\text{GC}_{\text{Prefix-RFT}}(x, y_i, t) = m_{i,t} c_{i,t} \rho_{i,t} A_{i,t}$, identical to the GRPO gradient coefficient (Eq. 2.5.1.2) but without the KL regularization term and with the additional entropy-based mask $m_{i,t}$. In addition, a cosine decay scheduler gradually reduces $L$ over training, transitioning from imitation-heavy to exploration-dominant.

### 4.2.2 Tree-structured Rollouts

The following works introduce non-flat rollout structures for finer-grained credit assignment.

**Tree-based Policy Optimization (TreePO)**  Standard RLVR approaches independently sample multiple trajectories per prompt, causing redundant KV cache computation across shared prefixes and only flat, coarse credit assignment. TreePO (Li et al., 2025b) (App. Tables 3 and 8) restructures rollouts as a shared-prefix tree to amortize computation over common prefixes and enable hierarchical advantage estimation at each branching depth. TreePO adopts DAPO's modifications: asymmetric clipping ($\varepsilon_{\text{low}} < \varepsilon_{\text{high}}$), group length normalization $\frac{1}{\sum_{i=1}^{G} |y_i|}$, and dynamic rejection sampling that filters prompts where all responses are correct or all incorrect.

TreePO introduces a tree-based sampling scheme where, for each prompt $x$, the policy $\pi_{\theta_{\text{old}}}$ expands an $N$-array tree up to depth $d_{\max}$, sampling $N$ continuations of at most $L_{\text{seg}}$ tokens at each node. Each response decomposes into segments, and sub-groups sharing longer prefixes nest within shallower ones in Eq. 4.2.2.1.

$$y_i = s_1 \oplus s_2 \oplus \cdots \oplus s_j, \quad j \leq d_{\max}, \qquad\qquad G_{|J|} \subseteq \cdots \subseteq G_2 \subseteq G_1 = G \tag{4.2.2.1}$$

The key innovation is the tree-based advantage estimation that replaces the standard GRPO group advantage (Eq. 2.5.1.4) with a hierarchical two-step normalization. First, sub-group advantages are computed at each tree depth $j$ by subtracting the sub-group mean. Then, these sub-group advantages are averaged across depths and normalized by their standard deviation, incorporating the global variance normalization strategy from REINFORCE++. The tree-based advantage is given in Eq. 4.2.2.2

$$A_{i,t} = \frac{\sum_{j=1}^{|J|} \hat{A}_{i,t,j}}{|J| \cdot \sigma\left(\{\hat{A}_{i,t,j}\}^{|J|-1}\right)}, \qquad \hat{A}_{i,t,j} = r(x, y_i) - \frac{1}{G_j} \sum_{j=1}^{G_j} r(x, y_{i,j}) \tag{4.2.2.2}$$

where $i$ indexes the response, $t$ indexes the token position, $j$ indexes the tree depth, and $G_j$ denotes the set of trajectories sharing the same predecessor node at depth $j$. Note that $\hat{A}_{i,t,j} = \hat{A}_{i,j}$ is constant across tokens $t$ within a response since it depends only on the outcome reward $r(x, y_i)$ and the $t$ subscript is inherited from the token-level policy gradient in which $A_{i,t}$ appears. Each response participates in multiple sub-groups at different tree depths (from the root group $G_1 = G$ down to the sibling group $G_{|J|}$ at the deepest shared branching point). This hierarchical advantage reveals nuanced segment-level differences among trajectories and provides more precise credit assignment than the flat group advantage. The gradient coefficient is $\text{GC}_{\text{TreePO}}(x, y_i, t) = c_{i,t} \rho_{i,t} A_{i,t}$, identical to the GRPO gradient coefficient (Eq. 2.5.1.2) but without the KL regularization term.

**VinePPO**  VinePPO (Kazemnejad et al., 2025) (App. Tables 3 and 8) is based on PPO with $G$ response generations per prompt $x \sim \mathcal{D}$, $\{y_i\}_{i=1}^{G} \sim \pi_{\theta_{\text{old}}}(\cdot|x)$, and estimates the baseline via a Monte Carlo (MC) value function $V_{\text{MC}}$ instead of a learned value network $V_\phi$. VinePPO addresses the credit assignment problem by exploiting the resettable structure of the language environment: any partial response $(x, y^{<t})$ can be re-fed to the policy to sample alternative continuations. For each intermediate position $t$ in a training trajectory, $G$ auxiliary continuations $y_1, \ldots, y_G \sim \pi_\theta(\cdot \mid x, y^{<t})$, each generated autoregressively from position $t$ to the terminal token, are sampled and their episode returns averaged to form the MC value estimate in Eq. 4.2.2.3, where $r(x, y^{<t}, y_i)$ denotes the outcome reward for the full trajectory composed of prefix $y^{<t}$ and continuation $y_i$.

$$V_{\text{MC}}(x, y^{<t}) = \frac{1}{G} \sum_{i=1}^{G} r(x, y^{<t}, y_i) \tag{4.2.2.3}$$

The advantage at token $y^t$ is then computed via Eq. 4.2.2.4 with $\gamma = 1$ since language generation is an undiscounted finite-horizon problem and intermediate rewards are zero ($r_t = 0$ for non-terminal tokens).

$$A_{\text{MC},t} = r_t + \gamma V_{\text{MC}}(x, y^{<t+1}) - V_{\text{MC}}(x, y^{<t}) = V_{\text{MC}}(x, y^{<t+1}) - V_{\text{MC}}(x, y^{<t}) \tag{4.2.2.4}$$

This replaces the GAE advantage in the PPO objective (Eq. (2.4.2.6)). The auxiliary continuations $y_i$ are used exclusively for value estimation and do not enter the policy gradient. For any $G \geq 1$ the resulting gradient is unbiased, and increasing $G$ reduces estimator variance at the cost of additional sampling. The advantages are normalized to zero mean and unit variance across the entire training batch (batch std). The gradient coefficient is $\text{GC}_{\text{VinePPO}}(x, y_i, t) = c_{i,t}\, \rho_{i,t}\, A_{\text{MC},t} + \beta\left(\frac{\pi_{\text{ref}}(y_i^t|x, y_i^{<t})}{\pi_\theta(y_i^t|x, y_i^{<t})} - 1\right)$, where $\rho_{i,t} = \frac{\pi_\theta(y_i^t|x, y_i^{<t})}{\pi_{\theta_{\text{old}}}(y_i^t|x, y_i^{<t})}$, identical to the PPO gradient coefficient (Eq. (2.4.2.7)) but with MC-based token-level advantages replacing GAE advantages and KL divergence following GRPO.

**Serial GRPO (S-GRPO)**  Standard GRPO's binary outcome rewards fail to regulate intermediate reasoning processes, causing reasoning models to generate unnecessarily long chains of thought, a phenomenon known as overthinking. Unlike standard GRPO which samples $G$ independent responses in parallel, S-GRPO (Dai et al., 2025b) (App. Tables 3 and 8) constructs a **serial group** via a two-phase rollout. In Phase 1, a single complete reasoning path $y_0$ is generated from $\pi_{\theta_{\text{old}}}$. In Phase 2, $m$ positions are randomly sampled along $y_0$; at each position the model is prompted to stop reasoning and produce an answer, yielding early-exit outputs $\{y_1, \ldots, y_m\}$ that truncate the same reasoning path at different depths. The serial group $\{y_0, y_1, \ldots, y_m\}$ thus contains $G = m + 1$ outputs from a single reasoning path. A **decaying reward** assigns exponentially decreasing credit to successive correct answers along the path, with the outcome reward $r(x, y_i)$ defined in Eq. 4.2.2.5

$$r(x, y_i) = \begin{cases} \frac{1}{2^{N_{\text{right}} - 1}} & \text{if } y_i \text{ is correct} \\ 0 & \text{otherwise} \end{cases} \tag{4.2.2.5}$$

where $N_{\text{right}}$ counts the number of correct answers up to and including position $i$ (ordered by exit depth). Earlier correct exits receive higher reward, incentivizing the model to produce sufficient reasoning as early as possible. The advantage removes standard-deviation normalization compared to GRPO, computed as $A_i = r(x, y_i) - \mu(x)$. The objective follows the GRPO formulation (Eq. 2.5.1.1) with the KL divergence term removed. The gradient coefficient is $\text{GC}_{\text{S-GRPO}}(x, y_i, t) = c_{i,t}\, \rho_{i,t}\, A_{i,t}$, identical to the GRPO gradient coefficient (Eq. 2.5.1.2) but without the KL regularization term.

### 4.3 Response Selection

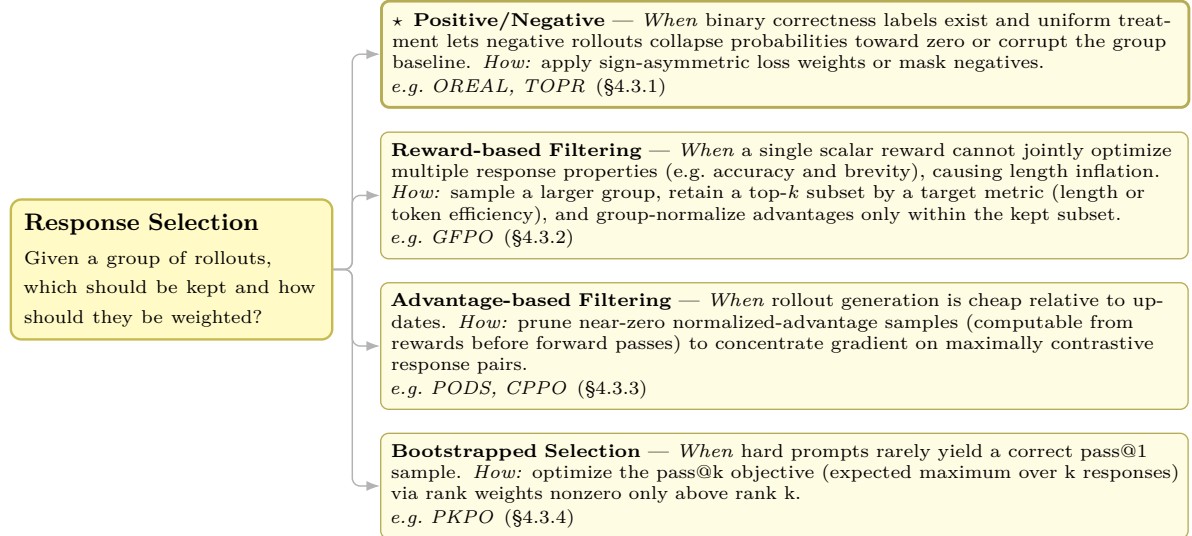

Figure 11: **Response selection.** How generated rollouts are weighted into the gradient update—sign-asymmetric treatment of correct versus incorrect responses, reward-based filtering to a top-$k$ subset, pruning of near-zero-advantage samples, or rank weighting that targets the pass@$k$ objective (§4.3; ⋆ default).

Response selection determines which rollouts enter the policy gradient update (Figure 11). The methods include positive/negative separation, reward-based filtering, advantage-based pruning, and bootstrapped pass@$k$ optimization.

#### 4.3.1 Positive/Negative

Positive/negative separation treats correct and incorrect rollouts asymmetrically, applying different loss weights or masking to prevent negative samples from corrupting the baseline. Other than the following work, OREAL (Lyu et al., 2025) in Section 5.1.1 and TOPR(Roux et al., 2025) in Section 4.1.2 have also applied asymmetric treatment to positive and negative responses.

#### 4.3.2 Reward-based Filtering

Reward-based filtering selects a subset of rollouts based on a target metric before computing group advantages. Besides GFPO, RFT (Yuan et al., 2023a) in Section 2.2.2 pioneered the idea of selecting only correct rollouts for supervised fine-tuning.

**Group Filtered Policy Optimization (GFPO)**   GRPO's single scalar reward makes it difficult to jointly optimize multiple response properties (e.g., accuracy and brevity), often leading to length inflation. GFPO (Shrivastava et al., 2026) (App. Tables 3 and 8) addresses length inflation by sampling a larger group of $G$ responses per prompt, filtering to a retained subset $S \subseteq \{1, \ldots, G\}$ of size $k$ based on a target metric (e.g., response length or token efficiency $\frac{r(x,y_i)}{|y_i|}$), and computing advantages only within $S$. With the outcome reward $r(x, y_i)$, a binary mask $m_i = \mathbb{I}[i \in S]$ zeroes out the advantage for rejected responses. The masked advantage is computed via group normalization within $S$, as given in Eq. 4.3.2.1.

$$A_{i,t} = \frac{r(x,y_i) - \mu_S}{\sigma_S} \, m_i, \qquad \mu_S = \frac{1}{k} \sum_{i \in S} r(x, y_i), \quad \sigma_S = \sqrt{\frac{1}{k} \sum_{i \in S} (r(x,y_i) - \mu_S)^2} \qquad (4.3.2.1)$$

The GFPO objective utilizes the modified advantage $A_{i,t}$, adopts group length normalization $\frac{1}{\sum_{i=1}^{G}|y_i|}$ (DAPO token-level loss aggregation only; symmetric clipping and KL divergence), and adds an entropy bonus $\lambda_{\text{ent}} H(\pi_\theta)$, as given in Eq. 4.3.2.2.

$$J_{\text{GFPO}}(\theta) = J_{\text{GRPO}}(\theta) + \lambda_{\text{ent}} H(\pi_\theta) \tag{4.3.2.2}$$

The gradient coefficient below is identical to GRPO
$\text{GC}_{\text{GFPO}}(x, y_i, t) = c_{i,t}\, \rho_{i,t}\, A_{i,t} + \beta\left(\frac{\pi_{\text{ref}}(y_i^t|x, y_i^{<t})}{\pi_\theta(y_i^t|x, y_i^{<t})} - 1\right) - \lambda_{\text{ent}} \log \pi_\theta(y_i^t|x, y_i^{<t})$. Adaptive Difficulty GFPO further adjusts $k$ per question based on real-time difficulty estimates via streaming reward quartiles, retaining more responses for harder problems and aggressively pruning easy ones.

### 4.3.3 Advantage-based filtering

The following works prune rollouts with near-zero normalized advantages, i.e., minimum variance before the expensive policy update, focusing gradient computation on the most contrastive response pairs.

**Policy Optimization with Down-Sampling (PODS)** RLVR training faces a compute asymmetry between cheap, parallelizable rollout generation and expensive policy updates. Not all rollouts contribute equally, and beyond a certain group size, additional rollouts reduce reward variance and introduce redundant, low-contrast signal. PODS (Xu et al., 2025) (App. Tables 3 and 8) decouples the two phases by generating a full group of $G$ rollouts per prompt $x$ from $\pi_{\theta_{\text{old}}}$ during inference, then training on only a strategically selected subset $S \subset \{1, \ldots, G\}$ of size $m < G$ during the policy update. The objective follows the GRPO formulation (Eq. 2.5.1.1) but restricts the sum to the selected subset $S$ (averaging over $m$ instead of $G$) and removes the KL penalty, with the outcome reward $r(x, y_i)$. The subset advantage is $A_{S,i} = \frac{r(x,y_i) - \mu_S}{\sigma_S}$, with $\mu_S$ and $\sigma_S$ computed within the selected subset $S$. The key selection criterion is **max-variance down-sampling**: choose $S$ to maximize $\text{Var}(\{r(x, y_i) \mid i \in S\})$, thereby preserving the strongest contrastive signals between successful and unsuccessful reasoning paths. The authors prove that the optimal subset always consists of the $k$ highest-reward and $(m-k)$ lowest-reward rollouts for some $k$:

$$S^* = \underset{k \in \{0, \ldots, m\}}{\text{argmax}} \ \text{Var}(\{r(x, y_1), \ldots, r(x, y_{m-k})\} \cup \{r(x, y_{G-k+1}), \ldots, r(x, y_G)\}) \tag{4.3.3.1}$$

where $\{y_1, \ldots, y_G\}$ are sorted so that $r(x, y_1) \leq \cdots \leq r(x, y_G)$. This reduces the combinatorial search to $O(G \log G)$ time (dominated by sorting), and in the common binary-reward setting simplifies to picking $m/2$ correct and $m/2$ incorrect rollouts. The gradient coefficient is $\text{GC}_{\text{PODS}}(x, y_i, t) = c_{i,t}\, \rho_{i,t}\, A_{S,i}$, identical to the GRPO gradient coefficient (Eq. 2.5.1.2) but without the KL regularization term.

**Completion Pruning Policy Optimization (CPPO)** CPPO (Lin et al., 2025) (App. Tables 3 and 8) reduces GRPO's training cost by pruning low-advantage completions before expensive forward passes, and reallocates the saved GPU capacity to additional prompts via dynamic completion allocation. GRPO's training cost scales with the group size $G$: computing the objective for each completion requires a forward pass through three models: $\pi_\theta$, $\pi_{\theta_{\text{old}}}$, and $\pi_{\text{ref}}$, yielding $3G$ forward passes per prompt. CPPO observes that the group-normalized advantage $A_i$ is computable *before* these forward passes as it depends only on rewards and completions with near-zero $|A_i|$ contribute negligibly to the policy gradient. CPPO performs *response-level* pruning: it modifies the GRPO objective (Eq. (2.5.1.1)) by replacing the full group average $\frac{1}{G}\sum_{i=1}^{G}$ with a restricted average $\frac{1}{k}\sum_{i \in \mathcal{I}}$, where $\mathcal{I} = \{i : |A_i| \text{ is among the top-}k \text{ values}\}$ and $k = \lfloor G(1-P) \rfloor$ for pruning rate $P$. Entire completions are either retained or discarded based on their sequence-level advantage $A_i$ and token-level terms within a retained completion are unchanged. A dynamic completion allocation strategy then fills freed GPU memory with pruned completions from additional prompts, maximizing device utilization and further reducing total training steps. The objective, with the outcome reward $r(x, y_i)$, follows the GRPO formulation (Eq. 2.5.1.1) but restricts the summation to a pruned subset $\mathcal{I} = \{i \in \{1, \ldots, G\} \mid |A_i| \text{ is among the top-}k \text{ values}\}$, where $k = \lfloor G \times (1-P) \rfloor$ for pruning rate $P \in (0, 1]$. The gradient coefficient is $\text{GC}_{\text{CPPO}}(x, y_i, t) = \mathbb{I}[i \in \mathcal{I}]\left(c_{i,t}\, \rho_{i,t}\, A_{i,t} + \beta\left(\frac{\pi_{\text{ref}}(y_i^t|x, y_i^{<t})}{\pi_\theta(y_i^t|x, y_i^{<t})} - 1\right)\right)$.

### 4.3.4 Bootstrapped selection

PKPO (Walder & Karkhanis, 2025) optimizes the pass@$k$ objective rather than pass@1, assigning rank-weighted gradient contributions only to responses ranked $k$-th or higher.

**Pass@$k$ Policy Optimization (PKPO)** Standard RLVR methods optimize pass@1, which prioritizes individual sample performance over the collective utility of the response group and limits exploration on harder problems where individually-sampled solutions are rarely correct. PKPO (Walder & Karkhanis, 2025) (App. Tables 3 and 8) addresses this by directly optimizing pass@$k$ which is the expected maximum reward across $k$ jointly sampled responses, preserving model diversity and unblocking learning on challenging task sets. For each prompt $x \sim \mathcal{D}$, $G$ on-policy responses are sampled $\{y_i\}_{i=1}^{G} \sim \pi_\theta(\cdot|x)$ with outcome rewards $r(x, y_i)$ and PKPO directly optimizes pass@$k$: the expected maximum reward over $k$ i.i.d. responses in Eq. 4.3.4.1.

$$J_{\mathrm{PKPO}}(\theta) = \mathbb{E}_{x \sim \mathcal{D},\ \{y_i\}_{i=1}^{k} \sim \pi_\theta(\cdot|x)} \left[ \max_{i \in \{1,\ldots,k\}} r(x, y_i) \right] \tag{4.3.4.1}$$

Sampling $G$ responses and averaging over all size-$k$ subsets yields a closed-form rank-weighted estimator (with $r_{(1)} \leq \cdots \leq r_{(G)}$ the sorted rewards) in Eq. 4.3.4.2.

$$\hat{J}_{\mathrm{PKPO}}(\theta) = \mathbb{E}_{x \sim \mathcal{D}} \left[ \sum_{i=k}^{G} \omega_i\, r_{(i)} \right], \quad \omega_i = \frac{\binom{i-1}{k-1}}{\binom{G}{k}} \tag{4.3.4.2}$$

Only responses ranked $k$-th or higher receive nonzero weight. Applying the leave-one-out principle, i.e., averaging each response's marginal contribution $r_{(i)} - \max_{j \in S} r_{(j)}$ across all size-$(k-1)$ subsets $S \subset \{1, \ldots, i-1\}$ gives the gradient coefficient in Eq. 4.3.4.3.

$$\mathrm{GC}_{\mathrm{PKPO}}(x, y_{(i)}) = \frac{1}{\binom{G}{k}} \sum_{\substack{S \subset \{1,\ldots,i-1\} \\ |S|=k-1}} \left( r_{(i)} - \max_{j \in S} r_{(j)} \right) \tag{4.3.4.3}$$

For binary rewards, incorrect responses receive zero gradient as the rewards are sorted in non-decreasing order and each correct response's gradient equals the fraction of size-$k$ subsets in which it is the sole correct one. PKPO supports annealing $k$ from large to 1 over training, shifting the objective from pass@k encouraging the model to explore diverse and low-probability solution strategies to pass@1 which demands high individual-sample accuracy and consolidates probability mass onto the most reliable solutions.

## 5 RLVR: Gradient Coefficient

This section surveys how recent RL fine-tuning methods modify the gradient coefficient along five orthogonal axes: (1) the importance sampling (IS) ratio, (2) advantage shaping, (3) advantage normalization, (4) length normalization, and (5) regularization. The two foundational baselines are PPO (Schulman et al., 2017b; Ouyang et al., 2022) and GRPO (Shao et al., 2024).

## 5.1 Importance Sampling Ratio

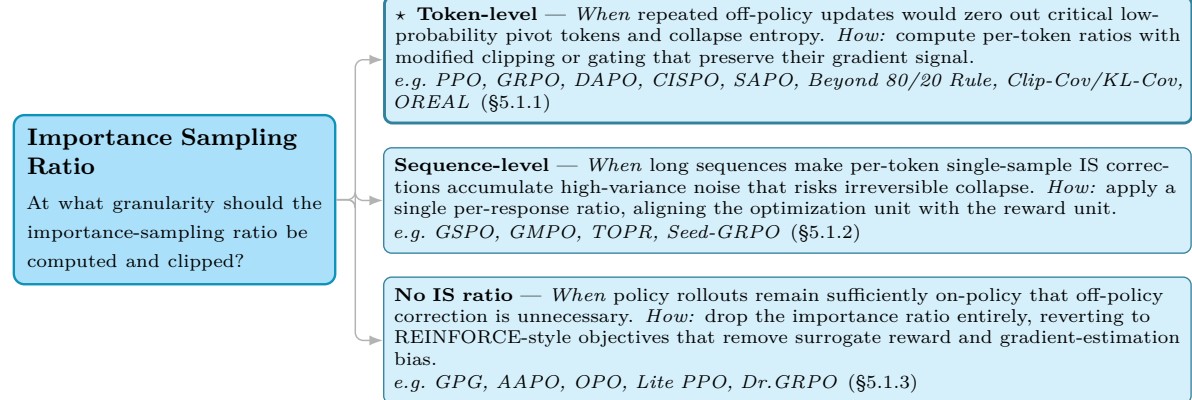

**Importance Sampling Ratio**

At what granularity should the importance-sampling ratio be computed and clipped?

★ **Token-level** — *When* repeated off-policy updates would zero out critical low-probability pivot tokens and collapse entropy. *How:* compute per-token ratios with modified clipping or gating that preserve their gradient signal.
*e.g. PPO, GRPO, DAPO, CISPO, SAPO, Beyond 80/20 Rule, Clip-Cov/KL-Cov, OREAL (§5.1.1)*

**Sequence-level** — *When* long sequences make per-token single-sample IS corrections accumulate high-variance noise that risks irreversible collapse. *How:* apply a single per-response ratio, aligning the optimization unit with the reward unit.
*e.g. GSPO, GMPO, TOPR, Seed-GRPO (§5.1.2)*

**No IS ratio** — *When* policy rollouts remain sufficiently on-policy that off-policy correction is unnecessary. *How:* drop the importance ratio entirely, reverting to REINFORCE-style objectives that remove surrogate reward and gradient-estimation bias.
*e.g. GPG, AAPO, OPO, Lite PPO, Dr.GRPO (§5.1.3)*

Figure 12: **Importance-sampling ratio.** The granularity at which the off-policy correction is computed and clipped—per token, a single per-response ratio, or no ratio at all under fresh on-policy rollouts—trading preserved signal on pivotal low-probability tokens against long-sequence variance (§5.1; ★ default).

The IS ratio $\rho_{i,t} = \frac{\pi_\theta(y_i^t|x,y_i^{<t})}{\pi_{\theta_{\text{old}}}(y_i^t|x,y_i^{<t})}$ corrects for distribution shift when reusing data from $\pi_{\theta_{\text{old}}}$ (Figure 12). This subsection surveys token-level IS and clipping, sequence-level IS and clipping, and complete elimination of the IS ratio. Among papers primarily discussed elsewhere, TOPR (Roux et al., 2025) in Section 4.1.2 also employs a sequence-level IS formulation. DAPO (Yu et al., 2025) in Section 5.4.2, Reinforce-Rej (Xiong et al., 2025b) in Section 5.4.2, and Lite PPO (Liu et al., 2026b) in Section 5.4.2 use asymmetric token-level clipping. ORZ (Hu et al., 2025b) in Section 5.2.5, VC-PPO (Yuan et al., 2025) in Section 5.2.5, VAPO (Yue et al., 2025) in Section 5.2.5, and SPO (Xu & Ding, 2026) in Section 5.2.5 retain token-level IS.

### 5.1.1 Token-level

Token-level IS methods compute a separate importance ratio per generated token. These methods share the goal of preserving gradient signal for critical reasoning tokens while preventing dramatic updates, each through a different modification of the standard clipping or gating mechanism. PPO (Schulman et al., 2017b; Ouyang et al., 2022) in Section 2.4.2 and GRPO (Shao et al., 2024) in Section 2.5.1 and DAPO (Yu et al., 2025) in Section 5.4.2 are the token-level IS baselines.

**Clipped Importance Sampling Policy Optimization (CISPO)**   In GRPO, tokens with importance ratios $\rho_{i,t}$ outside $[1\pm\varepsilon]$ are fully zeroed by clipping. In long-reasoning regimes with repeated off-policy updates, this disproportionately suppresses low-probability but crucial reasoning pivots (e.g., "Wait", "Recheck"), harming entropy and performance. CISPO (MiniMax-M1 Team, 2025) (App. Tables 4 and 9) addresses this by replacing GRPO's clipped surrogate $\min(\rho_{i,t}A_{i,t}, \text{clip}(\rho_{i,t}, 1-\varepsilon, 1+\varepsilon)A_{i,t})$ with a REINFORCE-style objective, where the importance ratio is clipped inside a stop-gradient: $\text{sg}(\hat{\rho}_{i,t}) = \text{sg}(\text{clip}(\rho_{i,t}, 1-\varepsilon_{\text{low}}, 1+\varepsilon_{\text{high}}))$ scales each token as $\text{sg}(\hat{\rho}_{i,t})A_{i,t}$, where $\text{sg}(\cdot)$ denotes stop-gradient. With the input reward $r(x,y_i)$ being the outcome-based score, the advantage is computed via group mean and std normalization (Eq. 2.5.1.4). In practice, $\varepsilon_{\text{low}}^{\text{IS}}$ is set to a large value (effectively removing the lower bound) and only $\varepsilon_{\text{high}}^{\text{IS}}$ is tuned, yielding asymmetric clipping. Group length normalization $\frac{1}{\sum_{i=1}^{G}|y_i|}$ is applied across the group. Because $\text{sg}(\hat{\rho}_{i,t})$ is treated as a constant during differentiation, the gradient reduces to $\text{sg}(\hat{\rho}_{i,t})A_{i,t}\nabla_\theta \log \pi_\theta$, yielding the gradient coefficient $\text{GC}_{\text{CISPO}}(x,y_i,t) = \text{sg}(\hat{\rho}_{i,t})A_{i,t}$ so that no token gradient is ever zeroed out and no KL penalty or entropy regularization.

**Soft Adaptive Policy Optimization (SAPO)**   Both GRPO and GSPO rely on binary hard clipping that zeros gradients at the token and sequence level respectively. SAPO (Gao et al., 2025) (App. Tables 4 and 9)

replaces the hard clipping function in the GRPO objective (Eq. 2.5.1.1) with a smooth, temperature-controlled soft gate $f_{i,t}^{\text{SAPO}}(\rho_{i,t})$, where $A_i = \frac{r(x,y_i)-\mu(x)}{\sigma(x)}$ is the group-normalized advantage (Eq. 2.5.1.4) and $r(x,y_i)$ is the outcome reward. The soft gating function is Eq. 5.1.1.1.

$$f_{i,t}^{\text{SAPO}}(\rho_{i,t}) = \frac{4}{\tau_i}\,\sigma(\tau_i\,(\rho_{i,t}-1))\,, \quad \tau_i = \begin{cases} \tau^+ & \text{if } A_i > 0 \\ \tau^- & \text{if } A_i \le 0 \end{cases} \tag{5.1.1.1}$$

The soft gate peaks at $\rho_{i,t}=1$ and decays smoothly as the ratio deviates, so near-on-policy tokens receive full gradients while off-policy tokens are progressively down-weighted rather than zeroed. Setting $\tau^- > \tau^+$ attenuates negative-advantage updates more aggressively, since they spread logit mass across many irrelevant tokens and are more prone to instability. The gradient coefficient is $\text{GC}_{\text{SAPO}}(x,y_i,t) = w_{i,t}(\theta)\rho_{i,t}(\theta)A_i$, where $w_{i,t}(\theta) = 4\,p_{i,t}(\theta)\,(1-p_{i,t}(\theta))$ and $p_{i,t}(\theta) = \sigma(\tau_i\,(\rho_{i,t}(\theta)-1))$, replacing the hard clipping indicator $c_{i,t}$ in the GRPO gradient coefficient (Eq. 2.5.1.2) with a smooth weight $w_{i,t}(\theta)$.

**Beyond 80/20 Rule**  Existing RLVR methods train uniformly on all tokens, neglecting that only a high-entropy minority of tokens serve as critical "forking" decision points while the majority of low-entropy tokens merely continue established reasoning paths. The key contribution is an entropy-based token filter that restricts policy gradient updates to the top-$p_\rho$ fraction of highest-entropy tokens per batch, based on the observation that high-entropy minority tokens act as critical reasoning forks while low-entropy tokens merely follow established paths (Wang et al., 2025a) (App. Tables 4 and 9).

Let $H_{i,t}$ denote the entropy of $\pi_\theta(\cdot|x,y_i^{<t})$ at position $t$ in response $y_i$, and $\tau_\rho$ the batch-level threshold selecting the top-$p_\rho$ fraction. The objective modifies the DAPO formulation by multiplying each per-token contribution by $\mathbb{I}[H_{i,t} \ge \tau_\rho]$ and adjusting the token-count normalization from $\sum_{i=1}^{G}|y_i|$ to $\sum_{i=1}^{G}\sum_{t=1}^{|y_i|}\mathbb{I}[H_{i,t} \ge \tau_\rho]$. The gradient coefficient is $\text{GC}_{\text{Beyond80/20Rule}}(x,y_i,t) = \mathbb{I}[H_{i,t} \ge \tau_\rho]c_{i,t}\,\rho_{i,t}A_i$, identical to the DAPO gradient coefficient but gated by the entropy indicator.

**Covariance-Based Entropy Regularization (Clip-Cov / KL-Cov)**  Policy entropy collapses in early RL training because a small fraction of tokens, i.e., those where high log-probability and high advantage coincide, drive disproportionately large entropy reductions, causing performance to plateau. While prior GRPO variants apply a uniform trust-region constraint to every token, Clip-Cov and KL-Cov identify and suppress only this responsible subset, leaving all remaining gradients untouched (Cui et al., 2025b) (App. Tables 4 and 9). A token-level centered cross-product over all $N_{\text{tok}}$ rollout tokens quantifies this in Eq. (5.1.1.2):

$$\text{Cov}(y_i^t) = \left(\log\pi_\theta(y_i^t|x,y_i^{<t}) - \frac{1}{N_{\text{tok}}}\sum_{j=1}^{G}\sum_{s=1}^{|y_j|}\log\pi_\theta(y_j^s|x,y_j^{<s})\right)\left(A_{i,t} - \frac{1}{N_{\text{tok}}}\sum_{j=1}^{G}\sum_{s=1}^{|y_j|}A_{j,s}\right) \tag{5.1.1.2}$$

where $N_{\text{tok}} = \sum_{j=1}^{G}|y_j|$ is the total token count, and advantages use GRPO group normalization $A_i = \frac{r(x,y_i)-\mu(x)}{\sigma(x)}$ (Eq. (2.5.1.4)). Two complementary strategies selectively suppress these high-covariance tokens, sharing the same objective structure (Eq. (5.1.1.3))

$$J(\theta) = \mathbb{E}_{x\sim\mathcal{D},\,y\sim\pi_{\theta_{\text{old}}}(\cdot|x)}\left[\frac{1}{|y|}\sum_{t=1}^{|y|}\ell_t\right] \tag{5.1.1.3}$$

**Clip-Cov** (hard suppression) randomly selects $\lfloor r\cdot N_{\text{tok}}\rfloor$ tokens whose covariance falls in $[\omega_{\text{low}},\omega_{\text{high}}]$ and zeros out their gradient (Eq. (5.1.1.4)–(5.1.1.5)):

$$I_{\text{clip}} \sim \text{Uniform}\big(\{(i,t) \mid \text{Cov}(y_i^t) \in [\omega_{\text{low}},\omega_{\text{high}}]\},\, \lfloor r\cdot N_{\text{tok}}\rfloor\big) \tag{5.1.1.4}$$

$$\ell_t^{\text{clip}} = \begin{cases} \rho_t\,A_t & \text{if } t \notin I_{\text{clip}} \\ 0 & \text{if } t \in I_{\text{clip}} \end{cases} \tag{5.1.1.5}$$

**KL-Cov** (soft suppression) selects the top-$k$ fraction of tokens ranked by covariance and, instead of removing them, adds a targeted forward KL penalty $D_{\mathrm{KL}}(\pi_{\theta_{\mathrm{old}}} \| \pi_\theta)$ to slow their update toward the rollout policy (Eq. (5.1.1.6)–(5.1.1.7)).

$$I_{\mathrm{KL}} = \{(i, t) \mid \mathrm{Rank}(\mathrm{Cov}(y_i^t)) \le k \cdot N_{\mathrm{tok}}\} \tag{5.1.1.6}$$

$$\ell_t^{\mathrm{KL}} = \begin{cases} \rho_t\, A_t & \text{if } t \notin I_{\mathrm{KL}} \\ \rho_t\, A_t - \beta\, D_{\mathrm{KL}}(\pi_{\theta_{\mathrm{old}}}(\cdot|x, y^{<t}) \| \pi_\theta(\cdot|x, y^{<t})) & \text{if } t \in I_{\mathrm{KL}} \end{cases} \tag{5.1.1.7}$$

KL-Cov does not use PPO-style clipping where the selective KL penalty serves as the trust-region constraint. The gradient coefficient is $\mathrm{GC}_{\text{KL-Cov}}(x, y_i, t) = \rho_t\, A_t$ for $t \notin I_{\mathrm{KL}}$, and $\rho_t\, A_t + \beta\left(\frac{\pi_{\theta_{\mathrm{old}}}(y_i^t|x, y_i^{<t})}{\pi_\theta(y_i^t|x, y_i^{<t})} - 1\right)$ for $t \in I_{\mathrm{KL}}$, penalizing divergence from the rollout policy $\pi_{\theta_{\mathrm{old}}}$ rather than a reference policy.

**Outcome REwArd-based reinforcement Learning (OREAL)**   Unlike prior methods that apply GRPO-style advantage normalization without theoretical grounding for binary sparse rewards in long reasoning chains, OREAL (Lyu et al., 2025) (App. Tables 4 and 9) provides a principled framework that proves behavior cloning on BoN-sampled positives suffices for KL-regularized optimality, and replaces heuristic credit assignment with a co-trained token-level reward model. OREAL introduces two key ideas: (1) behavior cloning on Best-of-$N$ positives with a KL constraint recovers the optimal policy and negatives are incorporated via a separate penalty term with hyperparameter $\eta_{\mathrm{neg}}$ and the reward shaping factor $(1 - \mu)$; (2) a co-trained token-level reward model $w(x, y^{\le t})$ produces per-token importance weights: $\omega_t^+ = \max(2\sigma(w) - 1, 0)$ for reinforcing key tokens in $y^+$, and $\omega_t^- = \max(1 - 2\sigma(w), 0)$ for penalizing error-causing tokens in $y^-$, providing credit assignment without a value network. Like GRPO, OREAL samples $G$ responses per prompt from $\pi_\theta$ using binary outcome reward $r(x, y) \in \{0, 1\}$, discards prompts where all responses are correct or all wrong, and selects one correct $y^+$ and one incorrect $y^-$ per prompt. The group accuracy rate $\mu = \frac{1}{G} \sum_{k=1}^G r_k$ serves as the baseline. To maintain gradient consistency with the Best-of-$N$ distribution, negative samples receive a shaped reward $r^*(x, y) = (1 - \mu)\, r(x, y)$ and correspondingly, the negative-sample loss applies a generalized preprocessing $F(1 - \mu)$ (e.g., $F(1 - \mu) = \frac{r_i - \mu}{\sigma(x)}$ in GRPO) as the advantage coefficient. The reward model $w$ is co-trained with the policy via cross-entropy on binary outcome rewards: $\mathcal{L}_{\mathrm{CE}} = -\mathbb{E}_{(x,y,r)}[r \log \hat{r}(x, y) + (1 - r) \log(1 - \hat{r}(x, y))]$, where $\hat{r}(x, y) = \sigma\left(\frac{1}{|y|} \sum_{t=1}^{|y|} w(x, y^{\le t})\right)$ is the predicted probability of correctness.

The OREAL objective (negating the loss) is in Eq. 5.1.1.8

$$J_{\mathrm{OREAL}}(\theta) = \mathbb{E}_{x \sim \mathcal{D},\, y \sim \pi_{\theta_{\mathrm{old}}}(\cdot|x)}\Bigg[\sum_{t=1}^{|y|}\Bigg(\omega_t^+\, \mathbb{I}[r(x, y) = 1]\, \log \pi_\theta(y^t|x, y^{<t})$$

$$- \eta_{\mathrm{neg}}\, \omega_t^-\, \mathbb{I}[r(x, y) = 0]\, \log \frac{\pi_\theta(y^t|x, y^{<t})}{\pi_{\mathrm{old}}(y^t|x, y^{<t})}\Bigg)\Bigg] - \beta\, \mathrm{KL}(\pi_\theta(\cdot|x) \| \pi_{\mathrm{old}}(\cdot|x)) \tag{5.1.1.8}$$

where $\eta_{\mathrm{neg}}$ is a hyperparameter that balances positive and negative terms. The gradient coefficient is $\mathrm{GC}_{\mathrm{OREAL}}(x, y, t) = \omega_t^+\, \mathbb{I}[r(x, y) = 1] - \eta_{\mathrm{neg}}\, \omega_t^-\, \mathbb{I}[r(x, y) = 0] + \beta\left(\frac{\pi_{\mathrm{old}}(y^t|x, y^{<t})}{\pi_\theta(y^t|x, y^{<t})} - 1\right)$, combining the token-weighted advantage terms with KL regularization.

### 5.1.2   Sequence-level

Sequence-level IS methods aggregate token-level probabilities into a single ratio per response in the following works. Apart from these works, TOPR (Roux et al., 2025) in Section 4.1.2 and Seed-GRPO (Chen et al., 2025) in Section 5.2.3 also employs a sequence-level IS ratio.

**Group Sequence Policy Optimization (GSPO)**   GSPO (Zheng et al., 2025) (App. Tables 4 and 9) is motivated by the ill-posedness of GRPO's token-level importance sampling: applying a single-sample IS correction per token accumulates high-variance training noise over long sequences, leading to catastrophic and often irreversible model collapse. The key innovation is replacing GRPO's token-level importance ratio $\rho_{i,t}$ with a *sequence-level* importance ratio $\rho_i$, aligning the optimization unit with the reward unit. The objective is in Eq. 5.1.2.1

$$J_{\text{GSPO}}(\theta) = \mathbb{E}_{x \sim \mathcal{D}, \{y_i\}_{i=1}^{G} \sim \pi_{\theta_{\text{old}}}(\cdot|x)} \left[ \frac{1}{G} \sum_{i=1}^{G} \min(\rho_i(\theta) A_i, \ \text{clip}(\rho_i(\theta), 1-\varepsilon, 1+\varepsilon) A_i) \right] \qquad (5.1.2.1)$$

where the advantage uses group normalization as in GRPO (Eq. (2.5.1.4)). The length-normalized sequence-level importance ratio is in Eq. 5.1.2.2.

$$\rho_i(\theta) = \left( \frac{\pi_\theta(y_i|x)}{\pi_{\theta_{\text{old}}}(y_i|x)} \right)^{\frac{1}{|y_i|}} = \exp \left( \frac{1}{|y_i|} \sum_{t=1}^{|y_i|} \log \frac{\pi_\theta(y_i^t|x, y_i^{<t})}{\pi_{\theta_{\text{old}}}(y_i^t|x, y_i^{<t})} \right) \qquad (5.1.2.2)$$

The $\frac{1}{|y_i|}$ exponent serves as length normalization, embedded within the importance ratio rather than as a separate factor as in GRPO. The gradient is in Eq. 5.1.2.3

$$\nabla_\theta J_{\text{GSPO}}(\theta) = \mathbb{E}_{x \sim \mathcal{D}, \{y_i\}_{i=1}^{G} \sim \pi_{\theta_{\text{old}}}(\cdot|x)} \left[ \frac{1}{G} \sum_{i=1}^{G} \frac{1}{|y_i|} c_i \, \rho_i(\theta) A_i \sum_{t=1}^{|y_i|} \nabla_\theta \log \pi_\theta(y_i^t|x, y_i^{<t}) \right] \qquad (5.1.2.3)$$

where $c_i$ is the sequence-level clipping indicator (same form as Eq. (2.4.2.5), with $\rho_i(\theta)$ and $A_i$ replacing per-token $\rho_{i,t}$ and $A_{i,t}$), so the importance ratio for the entire response is clipped at once rather than per token. The gradient coefficient is $\text{GC}_{\text{GSPO}}(x, y_i, t) = c_i \, \rho_i(\theta) A_i$, identical in structure to the GRPO gradient coefficient (Eq. 2.5.1.2) but with the sequence-level importance ratio and clipping indicator replacing their token-level counterparts, and without the KL regularization term. This uniform weighting also resolves a key source of MoE instability: expert token-routing can change after gradient updates, causing GRPO's per-token ratios $\rho_{i,t}$ to fluctuate as numerator and denominator are evaluated under different sub-networks, previously requiring *Routing Replay* at extra memory cost. GSPO obviates this because $\rho_i(\theta)$ aggregates over the full sequence and remains stable despite routing changes.

**Geometric-Mean Policy Optimization (GMPO)** GRPO's arithmetic-mean aggregation of token-level importance-weighted rewards is sensitive to extreme $\rho_{i,t}$, driving aggressive updates and entropy collapse. GMPO (Zhao et al., 2026) (App. Tables 4 and 9) replaces it with the **geometric mean**, which is inherently outlier-robust. The objective with token-level clipping is Eq. 5.1.2.4

$$J_{\text{GMPO}}(\theta) = \mathbb{E}_{x \sim \mathcal{D}, \{y_i\}_{i=1}^{G} \sim \pi_{\theta_{\text{old}}}(\cdot|x)} \left[ \frac{1}{G} \sum_{i=1}^{G} \left\{ \prod_{t=1}^{|y_i|} \Big| \min\Big( \rho_{i,t} A_{i,t}, \right. \right.$$

$$\left. \left. \text{clip}(\rho_{i,t}, \varepsilon_{\text{low}}, \varepsilon_{\text{high}}) A_{i,t} \Big) \Big| \right\}^{\frac{1}{|y_i|}} \cdot \text{sgn}(A_{i,t}) \right] \qquad (5.1.2.4)$$

where $\text{sgn}(A_{i,t})$ is the sign function ($+1$ if $A_{i,t} > 0$, $-1$ if $< 0$), restoring the correct optimization direction after the absolute value. Differentiating the unclipped objective derives Eq. 5.1.2.5

$$\nabla_\theta J_{\text{GMPO}}(\theta) = \mathbb{E}_{x \sim \mathcal{D}, \{y_i\}_{i=1}^{G} \sim \pi_{\theta_{\text{old}}}(\cdot|x)} \left[ \frac{1}{G} \sum_{i=1}^{G} \frac{1}{|y_i|} \left( \prod_{k=1}^{|y_i|} \rho_{i,k} \right)^{\frac{1}{|y_i|}} \sum_{t=1}^{|y_i|} A_{i,t} \nabla_\theta \log \pi_\theta(y_i^t|x, y_i^{<t}) \right] \qquad (5.1.2.5)$$

yielding $\text{GC}_{\text{GMPO}} = \left( \prod_{k=1}^{|y_i|} \rho_{i,k} \right)^{\frac{1}{|y_i|}} A_{i,t}$: every token shares the sequence-level geometric mean of ratios rather than its individual $\rho_{i,t}$, giving a more balanced update. Since the geometric mean never exceeds the arithmetic mean (AM–GM inequality), $|J_{\text{GMPO}}^*| \leq |J_{\text{GRPO}}^*|$, giving a narrower objective range and lower variance. Because the geometric mean already dampens outlier ratios, GMPO can afford a wider clipping range $(e^{-\varepsilon}, e^{\varepsilon})$ than GRPO's $(1 - \varepsilon, 1 + \varepsilon)$, encouraging greater exploration without sacrificing stability.

### 5.1.3 No Importance Sampling Ratio

GPG (Chu et al., 2026) discards the IS ratio entirely, reverting to on-policy REINFORCE-style objectives at the cost of requiring fresh rollouts. Besides GPG, AAPO (Xiong et al., 2025a) in Section 5.2.7, OPO (Hao et al., 2025) in Section 5.2.5 also discard the IS ratio. Lite PPO Liu et al. (2026b) in Section 5.4.2 and Dr.GRPO Liu et al. (2025d) in Section 5.4.3 retain PPO-style clipping but effectively operate without an IS correction.

**Group Policy Gradient (GPG)**  Unlike GRPO, which relies on a surrogate IS-ratio loss, PPO-style clipping, a reference model, and KL regularization, GPG (Chu et al., 2026) (App. Tables 4 and 9) strips the training pipeline to a minimal REINFORCE-style objective, i.e., directly optimizing the original RL objective without any surrogate approximation, auxiliary model, or distributional constraint. GPG addresses two sources of bias in GRPO: (i) *reward bias* from std-normalization in the advantage; and (ii) *gradient estimation bias* from groups with identical rewards contributing zero gradient while batch averaging still divides by the full batch size. GPG replaces GRPO's clipped importance-sampling surrogate with the direct REINFORCE loss, eliminating the importance ratio, clipping, KL penalty, and the reference model. The advantage uses the input reward $r(x, y_i)$ with group mean baseline $\mu(x)$ (Eq. 2.5.1.3) and sets $F_{\text{norm}} = 1$ (no $\sigma$-normalization), removing the reward bias introduced by GRPO's group $\sigma$ division. To address gradient estimation bias from groups with identical rewards, Accurate Gradient Estimation (AGE) rescales the loss by a batch-dependent factor $\alpha_{\text{GPG}} = \frac{B}{B-M}$ where $M$ is the number of zero-gradient groups in a batch of $B$ prompts. When $\alpha_{\text{GPG}}$ exceeds $\alpha_{\text{th}}$, the batch is deferred and valid samples are accumulated into subsequent batches, preventing high-variance updates. Unlike GRPO's per-response length normalization $\frac{1}{|y_i|}$, GPG normalizes by the total token count $\frac{1}{\sum_{i=1}^{G}|y_i|}$ across all responses. The objective follows Eq. 2.1.1. The policy gradient is given in Eq. 5.1.3.1.

$$\nabla_\theta J_{\text{GPG}}(\theta) = \mathbb{E}_{x \sim \mathcal{D}, \{y_i\}_{i=1}^{G} \sim \pi_{\theta_{\text{old}}}(\cdot|x)} \left[ \frac{1}{\sum_{i=1}^{G}|y_i|} \sum_{i=1}^{G} \sum_{t=1}^{|y_i|} \alpha_{\text{GPG}} A_{i,t} \, \nabla_\theta \log \pi_\theta(y_i^t|x, y_i^{<t}) \right] \quad (5.1.3.1)$$

The gradient coefficient is $\text{GC}_{\text{GPG}}(x, y_i, t) = \alpha_{\text{GPG}} A_{i,t}$, where $\alpha_{\text{GPG}}$ is a batch-level rescaling factor outside GC.

## 5.2 Advantage Shaping

★ **Outcome Reward** — *When* a verifiable scalar correctness score on the complete response is available. *How:* subtract a baseline (group mean or learned critic) from the terminal reward to form the advantage.
*e.g. GRPO, PPO (§5.2.1)*

**Length Penalty** — *When* a correctness reward exists but completions are verbose or format-inefficient. *How:* augment the outcome reward with a length-/format-aware term that compresses reasoning while preserving accuracy.
*e.g. LCPO (LCPO-Exact, LCPO-Max), GRPO-λ, GRPO-LEAD, Ada-GRPO, DAPO, Lite PPO (§5.2.2)*

**Self-Certainty Reward** — *When* verifiable labels are absent or unreliable, or confidence should weight updates. *How:* derive the signal from the model's own distribution (entropy, self-certainty, semantic entropy).
*e.g. Entropy Minimization (EM-FT, EM-RL, EM-INF), INTUITOR, SEED-GRPO, EMPO (§5.2.3)*

**Process Reward** — *When* terminal outcome rewards are too sparse for credit assignment. *How:* supply dense step-level process rewards (derived implicitly from outcome labels, or from a process reward model) as intermediate per-step feedback.
*e.g. PRIME, PURE (§5.2.4)*

**Baseline Estimation** — *When* variance reduction should target the subtracted baseline itself. *How:* replace the group mean with learned critics, variance-minimizing optimal baselines, or Bayesian value trackers.
*e.g. VC-PPO, VAPO, ORZ, OPO, SPO (§5.2.5)*

**Partition Function Estimation** — *When* the objective is built on the closed-form KL-optimal policy $\pi^* \propto \pi_{\text{ref}} \exp(r/\beta)$, whose normalizing partition function $Z(x)$ is intractable. *How:* approximate $Z(x)$ (estimate it from the group-mean reward, or learn a $Z_\phi$ network) or cancel it exactly via group normalization, yielding a tractable squared-error reward-matching loss.
*e.g. MDPO, FlowRL, GIFT, GVPO (§5.2.6)*

**Advantage Stability** — *When* homogeneous group rewards collapse the group-relative advantage toward zero, stalling gradients. *How:* add an advantage-momentum term against the frozen reference to sustain non-vanishing signal.
*e.g. AAPO (§5.2.7)*

**Hybrid SFT with RL** — *When* sequential SFT→RL cold start over-shifts the policy distribution (catastrophic forgetting) before RL begins. *How:* fuse the SFT and RL objectives in a single stage, augmenting rollout groups with demonstrations and balancing the two via entropy-aware weighting.
*e.g. SRFT, HPT (§5.2.8)*

**Advantage Shaping**

What signal should the advantage be shaped from, beyond a plain outcome reward?

Figure 13: **Advantage shaping.** Selecting how the per-response advantage is reshaped in RLVR—by terminal correctness, a length/format term, intrinsic model confidence, dense step-level process rewards, an improved baseline estimator, a partition-function objective, a homogeneous-group stability term, or SFT–RL fusion—according to the available reward and credit-assignment needs (§5.2; ★ default).

While the previous subsection addressed the IS ratio that multiplies the advantage, this subsection surveys methods that reshape the advantage signal $A_i$ itself across eight design axes: 1. outcome rewards, 2. length-aware reward shaping, 3. self-certainty rewards, 4. process rewards, 5. baseline estimation, 6. partition function approximation, 7. advantage stability under homogeneous rewards, and 8. hybrid SFT–RL objectives (Figure 13). Methods whose primary contribution spans multiple subsections are cross-referenced.

### 5.2.1 Outcome Reward

Standard outcome reward models assign a scalar correctness score to each complete response, forming the basic advantage signal used by GRPO (Shao et al., 2024), PPO (Schulman et al., 2017b; Ouyang et al., 2022),

and most other methods in this survey. In these works, the reward can also be regarded as token level, where all the previous rewards are zero except the last token which will be the outcome reward. After deriving the reward, the advantage is derived by subtracting a baseline (e.g., the group mean in GRPO, a learned critic in PPO) from the outcome score.

### 5.2.2 Length Penalty

The following works augment the outcome reward with a length-aware component to discourage verbose reasoning without sacrificing correctness. Apart from the following works, DAPO Yu et al. (2025) in Section 5.4.2 and Lite PPO Liu et al. (2026b) in Section 5.4.2 also include penalty for overlong responses.

**Length-Controlled Policy Optimization (LCPO)**   Reasoning models generate CoT outputs of uncontrollable length, making it infeasible to allocate test-time compute to a user-specified budget while maintaining accuracy. LCPO (Aggarwal & Welleck, 2025) (App. Tables 4 and 9) directly addresses this by a length-aware reward $r(x, y_i, n^*)$ that incorporates a user-specified token budget $n^*$ appended to each prompt $x$. The underlying training objective otherwise follows the GRPO formulation (Eq. (2.5.1.1)) with $G$ generations, substituting this length-aware reward into the advantage computation in place of the standard $r(x, y_i)$. The **exact-length** variant (LCPO-Exact) symmetrically penalizes deviation from $n^*$ via Eq. (5.2.2.1)

$$r(y, y^*, n^*) = \mathbb{I}(y = y^*) - \alpha_{\text{len}} |n^* - |y|| \tag{5.2.2.1}$$

where $\mathbb{I}(\cdot)$ is the correctness indicator, $|y|$ is the generated length, and $\alpha_{\text{len}}$ is the correctness-length trade-off. The **maximum-length** variant (LCPO-Max) applies a soft upper-bound constraint via Eq. (5.2.2.2), penalizing only over-budget outputs

$$r(y, y^*, n^*) = \mathbb{I}(y = y^*)\text{clip}(\alpha_{\text{len}} (n^* - |y|) + \delta_{\text{offset}}, \, 0, \, 1) \tag{5.2.2.2}$$

where $\delta_{\text{offset}}$ ensures slightly over-budget correct answers remain preferable to incorrect ones. LCPO-Exact enforces precise length matching; LCPO-Max allows the model to use fewer tokens when the problem does not require the full budget. The advantage is computed via group normalization with $\mu(x)$ and $\sigma(x)$ as in Eq. (2.5.1.4). The gradient coefficient is $\text{GC}_{\text{LCPO}}(x, y_i, t) = c_{i,t} \rho_{i,t} A_{i,t} + \beta\left(\frac{\pi_{\text{ref}}(y_i^t|x, y_i^{<t})}{\pi_\theta(y_i^t|x, y_i^{<t})} - 1\right)$, identical to the GRPO gradient coefficient (Eq. (2.5.1.2)).

**GRPO-$\lambda$**   While length-penalty reward functions can mitigate overthinking in GRPO, they tend to cause premature training collapse: as completion length decreases, model accuracy abruptly collapses, often early in training. GRPO-$\lambda$ (Dai et al., 2025a) (App. Tables 4 and 9) addresses this instability by dynamically adjusting the reward strategy based on the per-group correctness ratio. For each batch, query-completion groups are ranked by their correctness ratio. The top-$\lambda$ fraction (those with sufficiently high correctness ratio, indicating mature reasoning capability) receive an efficiency-prioritized reward with a length penalty (Eq. (5.2.2.3))

$$r(x, y_i) = \begin{cases} 1 - \alpha_{\text{len}} \cdot \sigma\left(\frac{|y_i| - \mu_{\text{correct}}}{\sigma_{\text{correct}}}\right) & \text{if } \mathbb{I}(y_i = y^*) = 1 \\ 0 & \text{otherwise} \end{cases} \tag{5.2.2.3}$$

where $\mu_{\text{correct}}$ and $\sigma_{\text{correct}}$ are the mean and standard deviation of completion lengths over correct responses within the group, and $\alpha_{\text{len}}$ controls the penalty strength. The remaining groups fall back to the standard accuracy-prioritized outcome reward $r(x, y_i) = \mathbb{I}(y_i = y^*)$, prioritizing reasoning capability over efficiency. The advantage $A_i$ for each sample is computed via group normalization as $A_i = \frac{r_i - \mu(r_i)}{\sigma(r_i)}$ and broadcast uniformly to all corresponding response tokens before the standard GRPO parameter update.

**GRPO-LEAD**   GRPO's binary rewards yield no signal when all group responses agree, tolerate speculative guessing via zero reward for incorrect answers, and over-optimize easy problems. To solve these problems, GRPO-LEAD (Zhang & Zuo, 2025) (App. Tables 4 and 9) modifies the reward $r(x, y_i)$ and applies difficulty-aware advantage reweighting. The reward in Eq. (5.2.2.4) uses the standardized length deviation $z_i = \frac{|y_i| - \mu_\ell}{\sigma_\ell + \epsilon}$

of correct responses ($\mu_\ell, \sigma_\ell$ are the mean and standard deviation of completion lengths within the group) to down-weight verbose solutions, and assigns $-1$ to incorrect ones

$$r(x, y_i) = \begin{cases} \exp(-\alpha_{\text{len}} z_i) & \text{if } y_i = y^* \\ -1 & \text{if } y_i \neq y^* \end{cases} \tag{5.2.2.4}$$

where $\alpha_{\text{len}} > 0$ controls length penalization strength. The group-normalized advantage $A_i = \frac{r(x,y_i)-\mu(x)}{\sigma(x)}$ (Eq. (2.5.1.4)), where $\mu(x)$ and $\sigma(x)$ are the mean and standard deviation of rewards within the group, is then reweighted by a difficulty-aware logistic function $w(x)$ in Eq. (5.2.2.5)

$$w(x) = C + \frac{B - C}{1 + \exp[k(x - x_0)]}, \qquad \hat{A}'_i = A_i \begin{cases} w(\mu(x)) & \text{if } A_i > 0 \\ w(1 - \mu(x)) & \text{if } A_i \leq 0 \end{cases} \tag{5.2.2.5}$$

where $B$, $C$, $k$, and $x_0$ are hyperparameters controlling the sensitivity of the reweighting to problem difficulty. Correct responses on hard problems ($\mu(x) \approx 0 \Rightarrow w(\mu(x)) \approx B$) and incorrect responses on easy problems ($\mu(x) \approx 1 \Rightarrow w(1-\mu(x)) \approx B$) both receive amplified updates. The gradient coefficient is $\text{GC}_{\text{GRPO-LEAD}}(x, y_i, t) = c_{i,t}\, \rho_{i,t}\, \hat{A}'_i$, identical to the GRPO gradient coefficient (Eq. (2.5.1.2)) but with the reweighted advantage $\hat{A}'_i$ replacing $A_i$ and without the KL regularization term.

**Adaptive GRPO (Ada-GRPO)**  Ada-GRPO (Wu et al., 2025a) (App. Tables 4 and 9) addresses the *format collapse* problem in standard GRPO, where GRPO's accuracy-only objective creates a self-reinforcing cycle: the highest-accuracy format, typically Long CoT, is increasingly reinforced, suppressing more efficient formats (Direct Answer, Short CoT) regardless of task difficulty. It introduces a *dynamic reward-scaling strategy* that incentivizes less-used reasoning formats via a two-stage framework: SFT to teach four reasoning formats (Direct Answer, Short CoT, Code, Long CoT), then RL with the scaled reward. Given the binary correctness reward $r(x, y_i) = \mathbb{I}(y_i = y^*)$, Ada-GRPO scales the reward as Eq. 5.2.2.6

$$r'(x, y_i) = \alpha_i(t) \cdot r(x, y_i), \qquad \alpha_i(t) = 1 + \frac{1}{2}\left(\frac{G}{F(y_i)} - 1\right)\left(1 + \cos\left(\frac{\pi\, t}{T}\right)\right), \tag{5.2.2.6}$$

where $F(y_i)$ counts how many responses in the group share the same reasoning format as $y_i$ (identified via format-specific special tokens, e.g., ``), and $\frac{t}{T}$ is the fractional training progress. So $\alpha_i(t) \in [1, \frac{G}{F(y_i)}]$: at $t = 0$ each format receives its full inverse-frequency boost $\frac{G}{F(y_i)}$; as $t \to T$ the cosine schedule decays $\alpha_i(t) \to 1$, eliminating the format bias once training stabilises. The scaled rewards $r'(x, y_i)$ replace $r(x, y_i)$ in the group-normalized advantage (Eq. 2.5.1.4), i.e., $A(x, y_i) = \frac{r'(x,y_i)-\mu'(x)}{\sigma'(x)}$ where $\mu'(x)$ and $\sigma'(x)$ are the mean and standard deviation of the scaled rewards $\{r'(x, y_1), \ldots, r'(x, y_G)\}$. The model is then optimized with the GRPO objective (Eq. 2.5.1.1). The gradient coefficient is $\text{GC}_{\text{Ada-GRPO}}(x, y_i, t) = c_{i,t}\, \rho_{i,t}\, A_{i,t} + \beta\left(\frac{\pi_{\text{ref}}(y^t|x,y^{<t})}{\pi_\theta(y^t|x,y^{<t})} - 1\right)$, identical to the GRPO gradient coefficient (Eq. 2.5.1.2).

### 5.2.3  Self-Certainty Reward

When labeled outcome rewards are unavailable or unreliable, self-certainty rewards derive the training signal directly from the model's own output distribution, enabling label-free or unsupervised RL. The following approaches differ in how certainty is measured.

**Unreasonable Effectiveness of Entropy Minimization in LLM Reasoning**  Unlike prior RLVR methods that rely on labeled outcome rewards or verified answers as training signal, this work (Agarwal et al., 2025) (App. Tables 5 and 10) is motivated by the observation that instruction-tuned LLMs (i.e., after SFT but before RL post-training) already possess underappreciated reasoning capabilities that can be elicited by simply minimizing output entropy, i.e., concentrating probability mass on the model's most confident outputs without any labeled data. Entropy minimization (EM) achieves this through three methods, each targeting a distinct post-training stage: unsupervised finetuning (EM-FT), RL with entropy-based rewards (EM-RL), and inference-time logit optimization (EM-INF).

**EM-FT** (unsupervised finetuning) directly minimizes the token-level entropy of the policy on self-generated outputs in Eq. 5.2.3.1

$$J_{\text{EM-FT}}(\theta) = -\mathbb{E}_{x \sim \mathcal{D}, \, y \sim \pi_\theta(\cdot|x)} \left[ \frac{1}{|y|} \sum_{t=1}^{|y|} H\big(\pi_\theta(\cdot \mid x, y^{<t})\big) \right] \tag{5.2.3.1}$$

where $H(\pi_\theta(\cdot \mid x, y^{<t})) = -\sum_{j \in \mathcal{V}} \pi_\theta(j \mid x, y^{<t}) \log \pi_\theta(j \mid x, y^{<t})$ is the Shannon entropy over the vocabulary $\mathcal{V}$.

**EM-RL** uses REINFORCE with $G$ generations per prompt. Two alternative (not combined) entropy-based rewards $r(x, y)$ are defined in Eq. 5.2.3.2.

$$r_{\text{seq}}(x, y) = \sum_{t=1}^{|y|} \log \pi_\theta(y^t \mid x, y^{<t}), \qquad r_{\text{tok}}(x, y) = -\sum_{t=1}^{|y|} H\big(\pi_\theta(\cdot \mid x, y^{<t})\big) \tag{5.2.3.2}$$

The gradient coefficient is $\text{GC}_{\text{EM-RL}}(x, y, t) = A_{i,t} + \beta\Big(\frac{\pi_{\text{ref}}(y^t|x,y^{<t})}{\pi_\theta(y^t|x,y^{<t})} - 1\Big)$, where $r \in \{r_{\text{seq}}, r_{\text{tok}}\}$, $A_i = r(x, y_i) - b(x, y_i)$ and $b(x, y_i) = \frac{1}{G-1} \sum_{j \neq i} r(x, y_j)$ is the LOO baseline.

**EM-INF** reduces entropy at inference time without updating model parameters $\theta$. At each decoding step, the model produces a logit vector $z_t \in \mathbb{R}^{|\mathcal{V}|}$ via a standard forward pass and $z_t$ is then treated as a free variable and updated for a few gradient-descent steps (5–15 in practice) to minimize $H(\text{softmax}(z_t))$ down to a floor $\delta_{\text{floor}}$, while $\theta$ remains frozen. This is equivalent to optimizing a standalone softmax layer with $|\mathcal{V}|$ parameters, so the overhead is negligible compared to the model's forward pass. After logit optimization, sampling-based decoding selects the next token as usual. Unlike temperature scaling which preserves the logit ordering, logit optimization can reorder non-top logits, potentially improving reasoning chains in high-uncertainty settings. The key limitation is that EM is most effective when model confidence correlates with correctness, while it is less suited when the pretrained model lacks the target reasoning behaviors or when the task diverges significantly from the pretraining distribution.

**INTUITOR**  Current RLVR methods require domain-specific verifiers and gold-standard solutions, limiting their applicability to verifiable domains. INTUITOR (Zhao et al., 2025b) (App. Tables 5 and 10) replaces the outcome reward $r(x, y)$ with a **self-certainty** reward $r_{\text{sc}}(x, y)$: an intrinsic confidence measure termed **Reinforcement Learning from Internal Feedback (RLIF)**. Self-certainty is defined as the average KL divergence between a uniform distribution $U$ over the vocabulary $\mathcal{V}$ and the model's next-token distribution in Eq. 5.2.3.3.

$$r_{\text{sc}}(x, y) = \frac{1}{|y|} \sum_{t=1}^{|y|} \text{KL}\big(U \,\|\, \pi_\theta(\cdot|x, y^{<t})\big) = -\frac{1}{|y||\mathcal{V}|} \sum_{t=1}^{|y|} \sum_{j=1}^{|\mathcal{V}|} \log\big(|\mathcal{V}|\pi_\theta(j|x, y^{<t})\big) \tag{5.2.3.3}$$

This score $r_{\text{sc}}(x, y)$ replaces $r(x, y)$ in the GRPO group-normalized advantage (Eq. (2.5.1.4)). The gradient coefficient is $\text{GC}_{\text{INTUITOR}}(x, y_i, t) = c_{i,t}\rho_{i,t}A_{i,t} + \beta\Big(\frac{\pi_{\text{ref}}(y_i^t|x,y_i^{<t})}{\pi_\theta(y_i^t|x,y_i^{<t})} - 1\Big)$, identical to the GRPO gradient coefficient (Eq. (2.5.1.2)). A key design choice is using *online* self-certainty from the evolving policy $\pi_\theta$ rather than a frozen model, which prevents reward hacking.

**Semantic Entropy EnhanceD GRPO (SEED-GRPO)**  GRPO treats all training prompts equally during updates, ignoring differences in model confidence. When an LLM produces diverse responses to the same prompt, it often signals uncertainty or limited reasoning ability and applying large updates in such cases can amplify noise in the reward signal and harm learning. Unlike standard GRPO, SEED-GRPO (Chen et al., 2025) (App. Tables 5 and 10) introduces prompt-level uncertainty-aware advantage modulation: it computes the *semantic entropy* of the response group and applies a monotonically decreasing scaling function to reduce the advantage magnitude for high-entropy (high-uncertainty) prompts, while also removing std-normalization $A_i = r(x, y_i) - \mu(x)$, length normalization, and KL divergence. Given $G$ responses clustered into $K$ semantic

equivalence classes $\{C_1, \ldots, C_K\}$, the semantic entropy is approximated via Monte Carlo in Eq. 5.2.3.4.

$$\text{SE}(x) \approx -\frac{1}{K} \sum_{k=1}^{K} \log p(C_k|x), \quad p(C_k|x) = \sum_{y_i \in C_k} \pi_{\theta_{\text{old}}}(y_i|x) \tag{5.2.3.4}$$

The uncertainty-aware advantage is defined in Eq. 5.2.3.5

$$\tilde{A}_i = A_i \cdot f\left(\alpha_s \frac{\text{SE}(x)}{\text{SE}_{\text{max}}(x)}\right) \tag{5.2.3.5}$$

where $\text{SE}_{\text{max}}(x) = \log G$ is the maximum entropy (when every response falls into a distinct cluster), $\alpha_s$ controls sensitivity, and $f$ is a monotonically decreasing scaling function (linear by default). The gradient coefficient is $\text{GC}_{\text{SEED-GRPO}}(x, y_i, t) = c_i \, \rho_i \, \tilde{A}_i$, identical to the GRPO gradient coefficient (Eq. (2.5.1.2)) but without KL regularization and with semantic entropy modulation of the advantage.

**Entropy-Minimized Policy Optimization (EMPO)**   Unlike SEED-GRPO which still needs outcome rewards $r(x, y_i)$, EMPO (Zhang et al., 2025d) (App. Tables 5 and 10) replaces them entirely with **semantic entropy** as the sole reward signal: a lower semantic entropy indicates that the model's outputs cluster into fewer, more consistent meanings, which correlates with higher accuracy. Given $x \sim \mathcal{D}$ and $\{y_i\}_{i=1}^{G} \sim \pi_{\theta_{\text{old}}}(\cdot|x)$, EMPO clusters the responses into $K$ meaning clusters $\{C_1, \ldots, C_K\}$ via semantic equivalence (regular expressions for math tasks; a DeBERTa-v3-large entailment model for free-form question answering). The semantic entropy over the meaning distribution is defined (Zhang et al., 2025d) in Eq. (5.2.3.6)

$$H = -\sum_{k=1}^{K} p(C_k \mid x) \log p(C_k \mid x), \quad p(C_k \mid x) \approx \frac{|C_k|}{G} \tag{5.2.3.6}$$

where $|C_k|$ denotes the number of responses in cluster $C_k$. Each response $y_i$ receives a reward $r(x, y_i)$ equal to the probability of its assigned meaning cluster (Eq. (5.2.3.7)), with all responses in the same cluster $C_k$ receiving identical rewards regardless of surface-level differences, directly incentivizing convergence toward larger, more dominant clusters and thus minimizing $H$.

$$r(x, y_i) = p(C_k \mid x) \approx \frac{|C_k|}{G} \text{ where } y_i \in C_k \tag{5.2.3.7}$$

This reward $r(x, y_i)$ replaces the external reward in the GRPO group-normalized advantage (Eq. (2.5.1.4)): $A_i = \frac{r(x, y_i) - \mu(x)}{\sigma(x)}$. To prevent reward hacking, EMPO applies dual entropy thresholds $H_{\text{low}}$ and $H_{\text{high}}$: prompts with $H < H_{\text{low}}$ are excluded to preserve diversity and reduce the risk of overfitting to trivial predictions, while prompts with $H > H_{\text{high}}$ are excluded because all responses are too scattered to provide reliable signal. The final EMPO objective follows the GRPO formulation (Eq. (2.5.1.1)), subject to $H_{\text{low}} < H < H_{\text{high}}$. The gradient coefficient is $\text{GC}_{\text{EMPO}}(x, y_i, t) = \mathbb{I}[H_{\text{low}} < H < H_{\text{high}}]c_{i,t} \, \rho_{i,t} \, A_{i,t} + \beta\left(\frac{\pi_{\text{ref}}(y_i^t|x, y_i^{<t})}{\pi_\theta(y_i^t|x, y_i^{<t})} - 1\right)$ with $\rho = 1$.

### 5.2.4   Process Reward

Dense process rewards provide intermediate feedback at the step level, enabling richer credit assignment than sparse outcome rewards at the end of the response.

**Process Reinforcement through IMplicit rEwards (PRIME)**   While dense process rewards provide token-level credit assignment far more informative than sparse outcome rewards, collecting step-level annotations at scale is prohibitively expensive. This motivates PRIME's implicit approach (Cui et al., 2025a) (App. Tables 5 and 10) that derives process rewards online from outcome labels alone. It maintains the *LLM policy* $\pi_\theta$ and an *implicit PRM* $\pi_\phi$. At each training iteration, $\pi_\phi$ performs a forward pass over the current batch of rollouts to produce the token-level implicit process reward defined in Eq. (5.2.4.1). The response-level implicit reward is obtained by summing over all tokens: $r^o(x, y) = \sum_{t=1}^{|y|} r^p(y^t|x, y^{<t})$.

$$r^p(y^t|x, y^{<t}) = \beta \log \frac{\pi_\phi(y^t|x, y^{<t})}{\pi_{\text{ref}}(y^t|x, y^{<t})} \tag{5.2.4.1}$$

The outcome reward $r(x, y)$ is a rule-based verifier defined in Eq. (5.2.4.2) for math and code respectively:

$$r_{\text{math}}(x, y) = \begin{cases} 1 & \text{if matched} \\ 0 & \text{otherwise} \end{cases} \qquad r_{\text{code}}(x, y) = \frac{\sum \#\text{passes}}{\sum \#\text{test cases}} \qquad (5.2.4.2)$$

Once rollouts are graded by the outcome verifier, PRIME applies an *accuracy filter* to retain only prompts of medium difficulty, i.e., prompts for which the $G$ rollouts contain a mix of correct and incorrect responses. Prompts where all rollouts are correct (too easy) or all are wrong (too hard) are discarded, yielding the filtered set $\mathcal{T}$. With $\mathcal{T}$ so defined, $\pi_\phi$ is updated online on the same batch via binary cross-entropy loss against the outcome labels (Eq. (5.2.4.3)), keeping it calibrated to the current policy distribution and mitigating reward hacking from distribution shift.

$$\mathcal{L}_{\text{CE}}(\phi) = -\mathbb{E}_{(x,y,r(x,y))\sim\mathcal{T}}[r(x, y) \cdot \log \sigma(r^o(x, y)) + (1 - r(x, y)) \cdot \log(1 - \sigma(r^o(x, y)))] \qquad (5.2.4.3)$$

Given $G$ rollouts $\{y_1, \ldots, y_G\}$ sampled from $\pi_{\theta_{\text{old}}}(\cdot|x)$ for each prompt $x \in \mathcal{T}$, the advantage combines the dense process returns from $\pi_\phi$ with sparse outcome rewards through a leave-one-out (LOO) baseline with no standard deviation normalization, as shown in Eq. (5.2.4.4):

$$A_{i,t} = \underbrace{\sum_{s=t}^{|y_i|} \gamma^{s-t} \left[ r^p(y_i^s|x, y_i^{<s}) - \frac{1}{G-1} \sum_{j\neq i} \mu_\phi^p(y_j|x) \right]}_{\text{LOO with implicit process rewards}} + \underbrace{r^o(x, y_i) - \frac{1}{G-1} \sum_{j\neq i} r^o(x, y_j)}_{\text{LOO with outcome rewards}} \qquad (5.2.4.4)$$

where $\mu_\phi^p(y_j|x) = \frac{1}{|y_j|} \sum_{s=1}^{|y_j|} r^p(y_j^s|x, y_j^{<s})$ is the mean token-level process reward over response $y_j$. The gradient coefficient is $\text{GC}_{\text{PRIME}}(x, y_i, t) = c_{i,t}\, \rho_{i,t}\, A_{i,t}$, identical to the GRPO gradient coefficient (Eq. (2.5.1.2)) but without KL regularization.

**Process sUpervised Reinforcement lEarning (PURE)**  The standard approach in process-reward RL sums discounted future rewards to compute each step's value, but because PRM rewards are imperfect, this accumulation amplifies estimation errors and enables the model to exploit steps that receive spuriously high rewards. PURE (Cheng et al., 2025b) (App. Tables 5 and 10) replaces this sum-form with a *min-form credit assignment* so that only the worst reasoning step determines the response value. For an $n$-step response, let $r_t^p$ denote the process reward assigned by the PRM to step $t \in \{1, \ldots, n\}$, and let $t_w = \arg\min_{1\leq t\leq n} r_t^p$ be the index of the worst step. Steps up to and including the worst step $t_w$ receive the minimum of all process rewards as return, and steps after $t_w$ contribute nothing. To implement this without changing the return computation logic, PURE transforms the raw process rewards via a temperature-controlled softmax that concentrates weight on the lowest-reward step (Eq. 5.2.4.5)

$$r_i^{p*} = \frac{\exp\left(-\frac{r_i^p}{T}\right)}{\sum_{j=1}^n \exp\left(-\frac{r_j^p}{T}\right)} r_i^p \qquad (5.2.4.5)$$

where $T$ is the transform temperature, and $r_i^{p*}$ denotes the transformed process reward (as distinguished from the raw PRM reward $r_i^p$). For advantage estimation, PURE uses RLOO combining outcome verifiable rewards $r^o(x, y_i)$ and token-level transformed process rewards $r_{i,j}^{p*}$. Given $G$ responses $\{y_1, \ldots, y_G\}$ per prompt $x$ with maximum generation length $N$ (a hyperparameter, e.g., $N = 8192$), the advantage for response $y_i$ at token $t$ is defined in Eq. 5.2.4.6.

$$A_{i,t} = \underbrace{r_i^o - \frac{1}{G-1} \sum_{k\neq i} r_k^o}_{\text{RLOO (outcome)}} + \underbrace{\sum_{j=t}^N \gamma^{j-t} r_{i,j}^{p*} - \frac{\sum_{k\neq i} \sum_{l=1}^N \sum_{j=l}^N \gamma^{j-l} r_{k,j}^{p*}}{(G-1)N}}_{\text{RLOO (process)}} \qquad (5.2.4.6)$$

The process-reward baseline is normalized by the fixed maximum generation length $N$ rather than the actual response length $|y_k|$ to avoid biasing the model toward shorter responses. The RLOO advantage

uses a leave-one-out mean without standard-deviation normalization (i.e., advantage normalization is 1). The gradient coefficient is $\text{GC}_{\text{PURE}}(x, y_i, t) = \rho_{i,t} A_{i,t} + \beta \left( \frac{\pi_{\text{ref}}(y_i^t | x, y_i^{<t})}{\pi_\theta(y_i^t | x, y_i^{<t})} - 1 \right)$, following the GRPO gradient coefficient (Eq. (2.5.1.2)) but without clipping and with the RLOO process-reward advantage replacing the group-normalized advantage.

### 5.2.5 Baseline Estimation

The baseline subtracted from the reward reduces gradient variance without introducing bias. While GRPO (Shao et al., 2024) uses the group mean and PPO (Schulman et al., 2017b; Ouyang et al., 2022) a learned critic, this subsubsection surveys different baseline estimation from value function estimation, variance minimization and Bayesian based methods.

**Value-Calibrated PPO (VC-PPO)** VC-PPO (Yuan et al., 2025) (App. Tables 5 and 10) addresses PPO's failure in long-CoT tasks where output length collapses due to value initialization bias through two techniques: *value pretraining* and *Decoupled GAE*. Value pretraining initializes $V_\phi$ by offline regression on Monte Carlo returns from a fixed SFT policy, eliminating the bias that arises when the value model is initialized from a reward model trained only on terminal tokens. Decoupled GAE assigns asymmetric $\lambda$ values: $\lambda_{\text{critic}} = 1$ for the value target (reducing to Monte Carlo returns for unbiased reward propagation in long sequences) and $\lambda_{\text{actor}} = 0.95$ for the actor (reducing advantage variance). The value target with Decoupled GAE is given in Eq. 5.2.5.1

$$V^{\text{target}}(x, y^{<t}) = \begin{cases} \sum_{l=0}^{T-t-1} \lambda^l \, \delta_{t+l} + V(x, y^{<t}) & \text{if } \lambda < 1.0 \\ \sum_{l=0}^{T-t-1} r_{t+l} & \text{if } \lambda = 1.0 \end{cases} \tag{5.2.5.1}$$

where $\delta_{t+l} = r_{t+l} + \gamma V(x, y^{<t+l+1}) - V(x, y^{<t+l}) = r_{t+l} + V(x, y^{<t+l+1}) - V(x, y^{<t+l})$ is the TD error with $\gamma = 1$ (Eq. 2.4.1.2). The value model is trained with the standard PPO value function MSE loss. The gradient coefficient is $\text{GC}_{\text{VC-PPO}}(x, y, t) = c_t \rho_t A_t$, identical to the PPO gradient coefficient (Eq. 2.4.2.7) but without KL regularization in the reward signal.

**Value-Augmented Policy Optimization (VAPO)** While VC-PPO addresses PPO's length-collapse failure, three further challenges remain in long-CoT RLVR: the fixed $\lambda_{\text{policy}}$ cannot adapt to heterogeneous response lengths, symmetric clipping suppresses exploration of low-probability tokens, and per-sequence loss normalization biases gradient updates toward shorter sequences. VAPO (Yue et al., 2025) (App. Tables 5 and 10) introduces four additional modifications over VC-PPO.

*Length-adaptive GAE* replaces VC-PPO's fixed $\lambda_{\text{policy}} = 0.95$ with a response-length-dependent value in Eq. 5.2.5.2

$$\lambda_{\text{policy}} = 1 - \frac{1}{\alpha \, |y|} \tag{5.2.5.2}$$

where $\alpha$ controls the bias–variance tradeoff. *Clip-higher* decouples the clipping range into $\varepsilon_{\text{low}} < \varepsilon_{\text{high}}$, leaving more room for low-probability tokens to increase and mitigating entropy collapse. *Token-level loss* replaces the per-sequence average $\frac{1}{|y_i|}$ with a global token average $\frac{1}{\sum_{i=1}^{G} |y_i|}$, giving longer sequences proportionally more weight. A *positive-example NLL loss* on the subset $\mathcal{T} = \{y_i \mid r(x, y_i) = 1\}$ of correct responses reinforces successful reasoning patterns. The NLL loss and combined objective are in Eq. 5.2.5.3.

$$J_{\text{NLL}}(\theta) = \frac{1}{\sum_{y_i \in \mathcal{T}} |y_i|} \sum_{y_i \in \mathcal{T}} \sum_{t=1}^{|y_i|} \log \pi_\theta(y_i^t \mid x, y_i^{<t}) \qquad J(\theta) = J_{\text{VAPO}}(\theta) + \alpha_{\text{NLL}} J_{\text{NLL}}(\theta) \tag{5.2.5.3}$$

The gradient coefficient is $\text{GC}_{\text{VAPO}}(x, y, t) = c_t \rho_t A_t^{\text{GAE}(\gamma, \lambda_{\text{policy}})} + \alpha_{\text{NLL}} \mathbb{I}(y \in \mathcal{T})$, where the first term is the PPO gradient coefficient (Eq. 2.4.2.7) with length-adaptive $\lambda_{\text{policy}}$ and asymmetric clipping $(\varepsilon_{\text{low}}, \varepsilon_{\text{high}})$, and

the second term is the positive-example NLL contribution with weight $\alpha_{\text{NLL}}$ for correct responses ($y \in \mathcal{T}$). Unlike VC-PPO ($\beta = 0$), VAPO uses KL divergence (folded into per-token rewards as in standard RLHF PPO).

**Open-Reasoner-Zero (ORZ)**  Unlike prior work that uses GRPO without a dedicated value network, ORZ (Hu et al., 2025b) (App. Tables 5 and 10) is the first open-source large-scale Reasoner-Zero training framework, adopting vanilla PPO with a learned critic $V_\phi$ and batch-level advantage normalization in place of GRPO entirely. GRPO's group mean assigns a single scalar baseline to every token in a response and cannot identify which specific tokens are problematic, whereas ORZ's token-level critic assigns lower expected returns to states with repetitive patterns, making those token advantages strongly negative and actively discouraging degenerate generation. For each prompt $x \sim \mathcal{D}$, $G$ different responses $y_i \sim \pi_{\theta_{\text{old}}}(\cdot|x)$ are generated, and the baseline is estimated via a trained value function $V_\phi(x, y_i^{<t})$. Setting both GAE parameters to unity ($\gamma = 1$, $\lambda = 1$) with a terminal-only binary outcome reward $r(x, y_i) \in \{0, 1\}$ ($r_{i,t} = 0$ for $t < |y_i|$, $r_{i,|y_i|} = r(x, y_i)$), the GAE formula (Eq. 2.4.1.2) simplifies via telescoping.

$$A_{i,t}^{\text{GAE}(1,1)} = \sum_{k=0}^{|y_i|-t-1} \delta_{i,t+k} = \underbrace{\sum_{k=0}^{|y_i|-t-1} r_{i,t+k}}_{= r(x,y_i)} + \underbrace{V_\phi(x, y_i)}_{=0} - V_\phi(x, y_i^{<t}) = r(x, y_i) - V_\phi(x, y_i^{<t}) \qquad (5.2.5.4)$$

Batch-level advantage normalization is applied, i.e., $\hat{A}_{i,t} = \frac{A_{i,t}^{\text{GAE}(1,1)} - \mu_{\text{batch}}}{\sigma_{\text{batch}}}$ where $\mu_{\text{batch}}$ and $\sigma_{\text{batch}}$ are the batch mean and standard deviation respectively. The gradient coefficient is $\text{GC}_{\text{ORZ}}(x, y, t) = c_t \rho_t \hat{A}_{i,t}$, identical to the PPO gradient coefficient (Eq. 2.4.2.7).

**Optimal Policy Optimization (OPO)**  GRPO's loose on-policy setting causes large policy shifts, entropy collapse, and reduced output diversity, motivating OPO (Hao et al., 2025) (App. Tables 5 and 10) to enforce strict exact on-policy training and to replace the group-mean baseline with a theoretically optimal variance-minimizing alternative. It makes two key changes: (1) it enforces *exact on-policy training* by eliminating the importance ratio $\rho_{i,t}$ and clipping entirely, updating the policy only on freshly sampled responses; and (2) it replaces the group-mean baseline with an *optimal variance-minimizing baseline*. The theoretical optimal baseline minimizes the variance of the policy gradient estimator with respect to $b$ and is given in Eq. 5.2.5.5. Under the assumption that token-level gradients are approximately orthogonal with identically distributed norms ($\|\nabla_\theta \log \pi_\theta(y|x)\|^2 \propto |y|$), the baseline simplifies to a length-weighted reward average over the $G$ sampled responses.

$$b^* = \frac{\mathbb{E}_{y \sim \pi_\theta(\cdot|x)}\left[\|\nabla_\theta \log \pi_\theta(y|x)\|^2 \, r(x, y)\right]}{\mathbb{E}_{y \sim \pi_\theta(\cdot|x)}[\|\nabla_\theta \log \pi_\theta(y|x)\|^2]} \approx \frac{\sum_{i=1}^{G} |y_i| r(x, y_i)}{\sum_{i=1}^{G} |y_i|} \qquad (5.2.5.5)$$

The advantage is $A_i = r(x, y_i) - b^*(x)$ without group standard deviation normalization. Note the gradient does not include per-response length normalization $\frac{1}{|y_i|}$ (unlike GRPO Eq. (2.5.1.1)) and the length effect is instead absorbed by the optimal baseline $b^*$. The policy gradient is in Eq. 5.2.5.6.

$$\nabla_\theta J_{\text{OPO}}(\theta) = \mathbb{E}_{x \sim \mathcal{D}, \{y_i\}_{i=1}^{G} \sim \pi_\theta(\cdot|x)} \left[ \frac{1}{G} \sum_{i=1}^{G} \sum_{t=1}^{|y_i|} A_i \, \nabla_\theta \log \pi_\theta(y_i^t \mid x, y_i^{<t}) \right] \qquad (5.2.5.6)$$

The gradient coefficient is $\text{GC}_{\text{OPO}}(x, y_i, t) = A_i$.

**Single-stream Policy Optimization (SPO)**  GRPO's group-based design suffers from frequent degenerate groups and imposes synchronization barriers in distributed training that stall throughput in agentic settings with variable-length responses, motivating SPO (Xu & Ding, 2026) (App. Tables 5 and 10) to abandon the group paradigm entirely. SPO generates a single response ($G = 1$) per prompt $x \sim \mathcal{D}$, $y \sim \pi_{\theta_{\text{old}}}(\cdot|x)$, and estimates the value function via a persistent Bayesian value tracker. For the input binary reward

$r(x, y) \in \{0, 1\}$, SPO maintains a Beta distribution to track the success probability, whose parameters and value estimate are updated as Eq. 5.2.5.7

$$a(x) \leftarrow \kappa(x)\, a(x) + r(x, y), \quad b(x) \leftarrow \kappa(x)\, b(x) + (1 - r(x, y)), \quad \hat{v}(x) = \frac{a(x)}{a(x) + b(x)} \tag{5.2.5.7}$$

where $\kappa(x) = 2^{-D(x)/D_{\text{half}}}$ discounts past observations by the KL divergence $D(x)$ between the current policy and the policy that last acted on $x$, with half-life $D_{\text{half}}$. The advantage $A(x, y)$ is computed using the pre-update baseline $\hat{v}(x)$ firstly. Next, the advantage is normalized globally across the batch $\mathcal{B}$ (batch-level, not per-group as in GRPO) to derive $\hat{A}(x, y)$, as shown in Eq. 5.2.5.8

$$\hat{A}(x, y) = \frac{A(x, y) - \mu_{\mathcal{B}}}{\sigma_{\mathcal{B}}} \qquad A(x, y) = r(x, y) - \hat{v}(x) \tag{5.2.5.8}$$

where $\mu_{\mathcal{B}}$ and $\sigma_{\mathcal{B}}$ are the mean and standard deviation of the raw advantages $\{r(x, y) - \hat{v}(x)\}_{x \in \mathcal{B}}$ across the batch. The objective follows the PPO-Clip formulation (Eq. (2.4.2.3)) with asymmetric clipping (Clip-Higher). The gradient coefficient is $\text{GC}_{\text{SPO}}(x, y, t) = c_t\, \rho_t\, \hat{A}(x, y)$. It further enables prioritized sampling with weights $w(x) \propto \sqrt{\hat{v}(x)(1 - \hat{v}(x))} + \epsilon$, focusing computation on prompts near $\hat{v}(x) \approx 0.5$.

### 5.2.6 Partition Function Estimation

The intractable partition function $Z(x)$ in the closed-form optimal policy, i.e., $\pi^*(y|x) \propto \pi_{\text{ref}}(y|x) \exp\left(\frac{r(x,y)}{\beta}\right)$ must be approximated or eliminated for tractable training. Other than the following methods, GVPO (Zhang et al., 2025b) in Section 4.1.3 cancel $Z(x)$ to minimize the variance.

**Mirror Descent Policy Optimization (MDPO)** Standard RL methods fix a static reference policy throughout training, causing the KL anchor to grow stale as the policy evolves. MDPO (Kimi Team, 2025) (App. Tables 5 and 10) applies mirror descent: each iteration solves a KL-regularized subproblem (Eq. (2.1.1) with $\pi_{\text{ref}}$ replaced by $\pi_{\theta_{\text{old}}}$), then advances the KL center to the current policy. A length reward is added to the outcome reward, given in Eq. 5.2.6.1

$$\hat{r}(x, y_i) = r(x, y_i) + w \cdot \text{r}_{\text{len}}(y_i), \quad \text{r}_{\text{len}}(y_i) = \begin{cases} \lambda & \text{if correct} \\ \min(0, \lambda) & \text{if incorrect} \end{cases} \tag{5.2.6.1}$$

where $\lambda = 0.5 - \frac{|y_i| - \min_{1 \le k \le G}|y_k|}{\max_{1 \le k \le G}|y_k| - \min_{1 \le k \le G}|y_k|}$. The KL-constrained problem admits a closed-form optimal policy $\pi_\theta^*(y|x) \propto \pi_{\theta_{\text{old}}}(y|x) \exp\left(\frac{r(x,y)}{\beta}\right)$ and MDPO fits $\pi_\theta(y|x)$ to $\pi_\theta^*(y|x)$ in squared error. The partition function $\beta \log Z(x)$ is estimated from the group mean reward $\mu(x)$. The resulting objective $J(\theta) = -L(\theta)$ is given in Eq. 5.2.6.2.

$$J(\theta) = \mathbb{E}_{x \sim \mathcal{D},\, y \sim \pi_{\theta_{\text{old}}}(\cdot|x)} \left[ -\left( \hat{r}(x, y) - \mu(x) - \beta \log \frac{\pi_\theta(y|x)}{\pi_{\theta_{\text{old}}}(y|x)} \right)^2 \right] \tag{5.2.6.2}$$

The gradient coefficient is $\text{GC}_{\text{MDPO}}(x, y_i) = 2\beta \left[ \hat{r}(x, y_i) - \mu(x) - \beta \log \frac{\pi_\theta(y_i|x)}{\pi_{\theta_{\text{old}}}(y_i|x)} \right]$. After each iteration, the reference policy is updated to the current policy and the optimizer state is reset. Lastly, the training process utilizes curriculum sampling (easy-to-hard) and prioritized sampling, where prompts with lower success rates receive more iterations.

**Flow Reinforcement Learning (FlowRL)** Reward-maximizing RL methods such as PPO and GRPO tend to overfit to dominant reward signals, causing mode collapse that reduces diversity among reasoning paths and limits generalization. Instead of the clipped surrogate objective, FlowRL (Zhu et al., 2025) (App. Tables 5 and 10) shifts to reward distribution matching by introducing a learnable partition function $Z_\phi(x)$ and minimizing $\mathcal{D}_{\text{KL}}(\pi_\theta(\cdot|x) \| \pi^*(\cdot|x))$ where $\pi^*(y|x) = \frac{\exp(r(x,y)/\beta) \cdot \pi_{\text{ref}}(y|x)}{Z_\phi(x)}$, which is equivalent to the trajectory balance objective in Eq. 5.2.6.3.

$$J_{\text{FlowRL}}(\theta, \phi) = \mathbb{E}_{x \sim \mathcal{D},\, \{y_i\}_{i=1}^G \sim \pi_{\theta_{\text{old}}}(\cdot|x)} \left[ -\rho \cdot \left( \log Z_\phi(x) + \frac{1}{|y|} \log \frac{\pi_\theta(y|x)}{\pi_{\text{ref}}(y|x)} - \beta\, r(x, y) \right)^2 \right] \tag{5.2.6.3}$$

where $\rho = \text{clip}\left(\frac{\pi_\theta(y|x)}{\pi_{\theta_{\text{old}}}(y|x)}, 1-\varepsilon, 1+\varepsilon\right)^{\text{detach}}$ is a sequence-level, gradient-detached importance weight, $\hat{r}(x,y) = \frac{r(x,y)-\mu(x)}{\sigma(x)}$ is the group-normalized outcome reward, and $\delta(x,y;\theta,\phi) = \log Z_\phi(x) + \frac{1}{|y|}\log\frac{\pi_\theta(y|x)}{\pi_{\text{ref}}(y|x)} - \beta\, r(x,y)$ is the flow-balance residual. The objective $J_{\text{FlowRL}}(\theta,\phi)$ is minimized *jointly* with respect to **two learnable components**: (1) the **policy network** $\pi_\theta$, whose parameters receive gradients through the log-probability term $\frac{1}{|y|}\log\pi_\theta(y|x)$; and (2) the **partition function network** $Z_\phi$, implemented as a 3-layer MLP, whose parameters receive gradients through the $\log Z_\phi(x)$ term. At each training step, both $\theta$ and $\phi$ are updated simultaneously to drive the flow-balance residual $\delta(x,y;\theta,\phi)$ toward zero. The gradient coefficient is $\text{GC}_{\text{FlowRL}}(x,y) = -2\rho\,\delta(x,y;\theta,\phi)$, with $\rho$ detached so that the importance-sampling correction does not itself contribute to the gradient signal and $|y|$ is utilized for length normalization.

**Group Implicit Fine Tuning (GIFT)**  Unlike GRPO which maximizes cumulative normalized rewards through a non-convex clipped surrogate requiring careful tuning of clipping hyperparameters and prone to overfitting, GIFT (Wang, 2025) (App. Tables 5 and 10) replaces this objective with a convex MSE loss that aligns normalized implicit rewards (from DPO's reward formulation) to normalized explicit rewards (from UNA's reward-alignment principle), while retaining GRPO's on-policy group sampling with $G$ generations for stable exploration. DPO's implicit reward $r_\theta(x,y) = \beta\log\frac{\pi_\theta(y|x)}{\pi_{\text{ref}}(y|x)} + \beta\log Z(x)$ contains an intractable partition function $\log Z(x)$. GIFT addresses it through GRPO-style group normalization, in which $\beta$ and $\log Z(x)$ cancel out exactly when the group mean is subtracted and the result is divided by the standard deviation. For each prompt $x$, a group of $G$ responses $\{y_1, \ldots, y_G\}$ is sampled from $\pi_\theta(\cdot|x)$. The implicit reward is computed in Eq. 5.2.6.4.

$$\hat{r}_\theta(x,y_i) = \log\frac{\pi_\theta(y|x)}{\pi_{\text{ref}}(y|x)} \tag{5.2.6.4}$$

Both the explicit reward $r(x,y_i)$ and implicit reward $\hat{r}_\theta(x,y_i)$ are normalized to obtain advantage across the group using group mean and group standard deviation in Eq. 5.2.6.5. For the explicit reward $r(x,y_i)$, the group mean and standard deviation are $\mu(x)$ and $\sigma(x)$. For the implicit reward $\hat{r}_\theta(x,y_i)$, the group mean and standard deviation are $\hat{\mu}(x)$ and $\hat{\sigma}(x)$.

$$A(x,y_i) = \frac{r(x,y_i)-\mu(x)}{\sigma(x)}, \qquad \hat{A}_\theta(x,y_i) = \frac{\hat{r}_\theta(x,y_i)-\hat{\mu}(x)}{\hat{\sigma}(x)} \tag{5.2.6.5}$$

Following UNA's reward-alignment principle, the objective is $J_{\text{GIFT}}(\theta) = -L_{\text{GIFT}}(\theta)$, where the loss minimizes the MSE between normalized implicit and explicit rewards in Eq. 5.2.6.6.

$$J_{\text{GIFT}}(\theta) = \mathbb{E}_{x\sim\mathcal{D},\,\{y_i\}_{i=1}^G\sim\pi_\theta(\cdot|x)}\left[-\left(A(x,y)-\hat{A}_\theta(x,y)\right)^2\right] \tag{5.2.6.6}$$

In practice, both the *response-level* implicit reward $\hat{r}_\theta(x,y_i) = \sum_{t=1}^{|y_i|}\log\frac{\pi_\theta\left(y_i^t|x,y_i^{<t}\right)}{\pi_{\text{ref}}\left(y_i^t|x,y_i^{<t}\right)}$ and the *token-level* implicit reward $\hat{r}_\theta(x,y_i) = \frac{1}{|y_i|}\sum_{t=1}^{|y_i|}\log\frac{\pi_\theta\left(y_i^t|x,y_i^{<t}\right)}{\pi_{\text{ref}}\left(y_i^t|x,y_i^{<t}\right)}$ have been empirically evaluated. The response-level formulation (without $\frac{1}{|y_i|}$ length normalization) is adopted as it consistently outperforms the token-level variant. The gradient coefficient is $\text{GC}_{\text{GIFT}}(x,y_i,t) = \frac{2\left(A(x,y)-\hat{A}_\theta(x,y)\right)}{\text{sg}(\hat{\sigma}_\theta)} - \left(A(x,y)-\hat{A}_\theta(x,y)\right)^2$, which drives the policy to match normalized implicit rewards to normalized explicit rewards.

### 5.2.7  Advantage Stability

When group rewards are homogeneous, the group-relative advantage collapses toward zero and stalls gradient updates. AAPO (Xiong et al., 2025a) directly addresses this through an advantage momentum term.

**Advantage-Augmented Policy Optimization (AAPO)**  AAPO (Xiong et al., 2025a) (App. Tables 6 and 11) addresses the vanishing gradient problem that arises when all group responses receive similar rewards, causing the group-relative advantage to collapse toward zero and stalling training. The key innovation is an *advantage momentum* term, i.e., the clipped reward gap between the current policy's response and a response

51

from the frozen reference model $\pi_{\text{ref}}$, which prevents vanishing gradients when the group-relative advantage approaches zero. The augmented advantage is given in Eq. 5.2.7.1

$$A_{i,t} = \underbrace{\frac{r(x,y_i) - \mu(x)}{\sigma(x)}}_{A_{\text{GRPO}}(x,y_i)} + \text{clip}\left(\underbrace{r(x,y_i) - r(x,\tilde{y}_i)}_{\text{advantage momentum}}, \; \delta_L, \; \delta_H\right) \tag{5.2.7.1}$$

where $\tilde{y}_i \sim \pi_{\text{ref}}(\cdot|x)$ is a reference-model response and $[\delta_L, \delta_H]$ bound the momentum for stability. Because $\pi_{\text{ref}}$ is frozen while $\pi_\theta$ improves, the momentum term grows over training, ensuring a non-vanishing gradient signal even when the group-relative component vanishes. Like GPG, AAPO normalizes by the total token count $\frac{1}{\sum_{i=1}^{G}|y_i|}$ across all responses (group length norm) rather than GRPO's per-response $\frac{1}{|y_i|}$. In addition, AAPO drops the importance ratio, clipping, and KL penalty, directly optimizing the cross-entropy loss weighted by the augmented advantage $A_{i,t}$. The gradient coefficient is $\text{GC}_{\text{AAPO}}(x,y_i,t) = A_{i,t}$.

### 5.2.8   Hybrid SFT with RL

SRFT (Fu et al., 2025) unifies SFT and RL in a single-stage objective by augmenting on-policy rollout groups with human demonstrations and applying entropy-aware weighting to balance the two objectives without manual scheduling or a separate SFT cold-start phase. HPT (Lv et al., 2025) in Section 2.6 has also proposed to combine SFT with RL properly based on average reward obtained.

**Supervised Reinforcement Fine-Tuning (SRFT)**   SRFT (Fu et al., 2025) (App. Tables 6 and 11) is motivated by the observation that SFT and RL exert complementary but asymmetric effects on policy distributions: SFT induces coarse-grained global shifts while RL performs fine-grained selective updates, and that sequential SFT $\rightarrow$ RL risks over-shifting the distribution before RL begins, prompting a single-stage unification with entropy-aware weighting. It augments the on-policy rollout group with demonstrations to form $G_{\text{aug}} = G_{\text{roll}} \cup G_{\text{demo}}$, and computes the advantage via group normalization over $G_{\text{aug}}$: $A_i = \frac{r(x,y_i) - \mu(G_{\text{aug}})}{\sigma(G_{\text{aug}})}$, where $\mu(G_{\text{aug}})$ and $\sigma(G_{\text{aug}})$ are the mean and standard deviation of outcome rewards $r(x,y_i)$ across all samples in $G_{\text{aug}}$. The demo RL component sets $\pi_\beta = 1$ and omits clipping. The on-policy self-exploration uses binary reward $r(x,y) \in \{+1, -1\}$ in a REINFORCE objective. The total objective $J_{\text{SRFT}}(\theta) = -\mathcal{L}_{\text{SRFT}}(\theta)$ is Eq. 5.2.8.1

$$J_{\text{SRFT}}(\theta) = w_{\text{SFT}} \, \mathbb{E}_{(x,y)\sim\mathcal{D}_{\text{demo}}}\left[\frac{1}{|y|}\sum_{t=1}^{|y|}\log\pi_\theta(y^t|x,y^{<t})\right] + \mathbb{E}_{(x,y_k)\sim\mathcal{D}_{\text{demo}}}\left[\frac{1}{|y_k|}\sum_{t=1}^{|y_k|}\rho_{k,t}A_k\right] \tag{5.2.8.1}$$
$$+ \mathbb{E}_{x\sim\mathcal{D}, \, y\sim\pi_\theta(\cdot|x)}[w_{\text{RL}} \cdot \mathbb{I}[r(x,y)\!=\!+1] - \mathbb{I}[r(x,y)\!=\!-1]]$$

where $\rho_{k,t} = \frac{\pi_\theta(y_k^t|x,y_k^{<t})}{\pi_\beta(y_k^t|x,y_k^{<t})} = \pi_\theta(y_k^t|x,y_k^{<t})$ since $\pi_\beta = 1$, $\text{sg}(\cdot)$ denotes stop-gradient, and the two entropy-aware weights have opposite dependencies: $w_{\text{SFT}} = 0.5 \cdot \text{sg}(e^{-H(\pi_\theta)})$ suppresses SFT when entropy is high, while $w_{\text{RL}} = 0.1 \cdot \text{sg}(e^{+H(\pi_\theta)})$ amplifies positive reinforcement at high entropy. Since the demo SFT and demo RL losses apply to the same samples, their gradient coefficients add. For demonstration samples $(x,y_k) \sim \mathcal{D}_{\text{demo}}$ (with $\frac{1}{|y_k|}$ normalization): $\text{GC}_{\text{SRFT}}^{\text{demo}}(x,y_k,t) = w_{\text{SFT}} + \rho_{k,t}A_k$, where the first term comes from MLE and the second from the GRPO surrogate (similar to Eq. 2.5.1.2 but without the clipping indicator and KL term). For on-policy rollout samples $y \sim \pi_\theta(\cdot|x)$ (without $\frac{1}{|y|}$ normalization): $\text{GC}_{\text{SRFT}}^{\text{self}}(x,y,t) = w_{\text{RL}} \cdot \mathbb{I}[r(x,y)\!=\!+1] - \mathbb{I}[r(x,y)\!=\!-1]$, a response-level coefficient derived via the REINFORCE score function.

## 5.3 Advantage Normalization

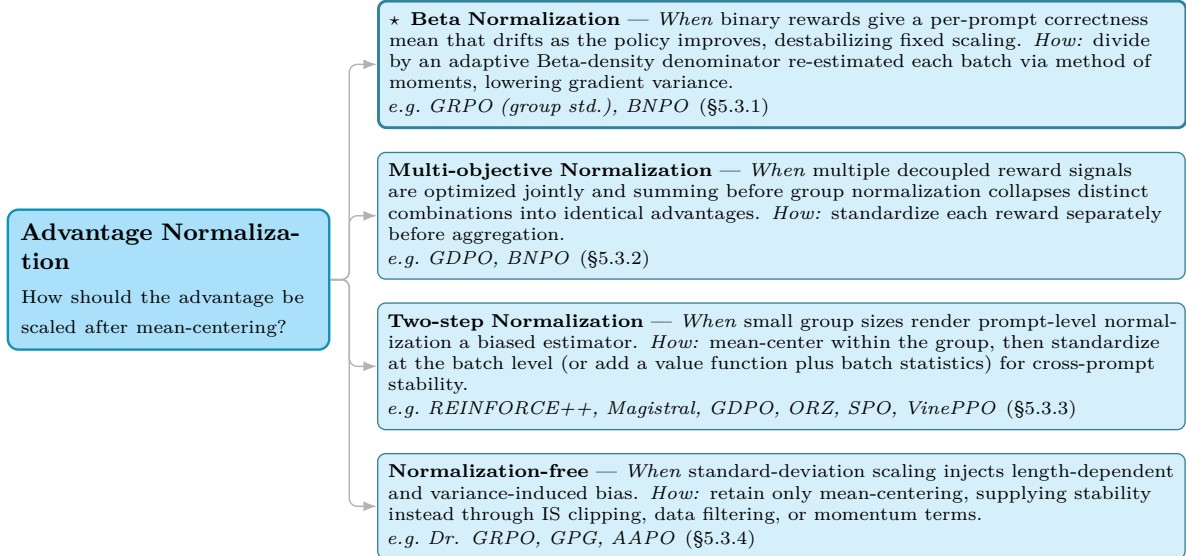

⋆ **Beta Normalization** — *When* binary rewards give a per-prompt correctness mean that drifts as the policy improves, destabilizing fixed scaling. *How:* divide by an adaptive Beta-density denominator re-estimated each batch via method of moments, lowering gradient variance.
*e.g. GRPO (group std.), BNPO* (§5.3.1)

**Multi-objective Normalization** — *When* multiple decoupled reward signals are optimized jointly and summing before group normalization collapses distinct combinations into identical advantages. *How:* standardize each reward separately before aggregation.
*e.g. GDPO, BNPO* (§5.3.2)

**Advantage Normalization**

How should the advantage be scaled after mean-centering?

**Two-step Normalization** — *When* small group sizes render prompt-level normalization a biased estimator. *How:* mean-center within the group, then standardize at the batch level (or add a value function plus batch statistics) for cross-prompt stability.
*e.g. REINFORCE++, Magistral, GDPO, ORZ, SPO, VinePPO* (§5.3.3)

**Normalization-free** — *When* standard-deviation scaling injects length-dependent and variance-induced bias. *How:* retain only mean-centering, supplying stability instead through IS clipping, data filtering, or momentum terms.
*e.g. Dr. GRPO, GPG, AAPO* (§5.3.4)

Figure 14: **Advantage normalization.** How the mean-centered group advantage is scaled—an adaptive Beta-density denominator, per-reward standardization before aggregation, group-then-batch two-step standardization, or no scaling—to curb gradient variance and bias under binary, multi-objective, or small-group rewards (§5.3; ⋆ default).

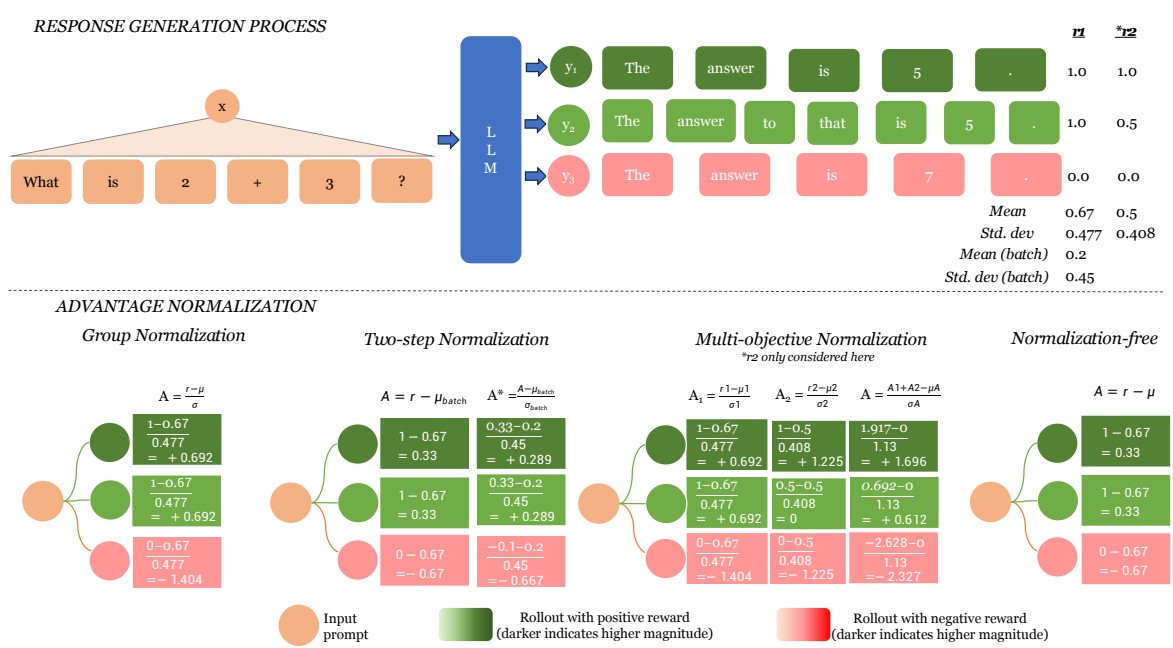

Figure 15: Advantage Normalization: Group normalization, Two-step normalization, Multi-objective normalization and Normalization-free.

Normalizing the advantage before multiplying by the IS ratio controls the effective learning rate and gradient variance (Figure 14). This subsection surveys four strategies, also depicted in Figure 15. Group normalization

subtracts the group mean and divides by the group standard deviation to standardize rewards within each prompt's rollouts, using an adaptive Beta-based scheme that tracks the evolving reward distribution (Xiao et al., 2025). Beta Normalization is a special case of group normalization where the standard deviation is replaced by a Beta distribution. Multi-objective normalization standardizes each reward signal separately, sums them, and re-standardizes the combined score to balance different objectives in decoupled per-reward settings (Xiao et al., 2025; Liu et al., 2026a). Two-step normalization first removes the group mean from the raw reward, then standardizes the result across the entire batch for cross-prompt stability (Hu et al., 2025a). Normalization-free simply subtracts the group mean without any scaling, preserving the raw reward differences (Liu et al., 2025d). Methods that additionally drop normalization as part of a broader regularization-free design are discussed with DAPO (Yu et al., 2025) in Section 5.4.2 and Lite PPO (Liu et al., 2026b) in Section 5.4.2.

### 5.3.1 Beta Normalization

BNPO (Xiao et al., 2025) replaces GRPO's (Shao et al., 2024) static group standard deviation denominator with an adaptive Beta density whose parameters are re-estimated each batch via method of moments, yielding lower-variance gradients as the correctness distribution shifts during training.

**Beta-Normalized Policy Optimization (BNPO)** The GRPO's fixed group normalization strategy by the group standard deviation $\sigma(x)$ under binary rewards $r(x, y) \in \{0, 1\}$, is equivalent to using Beta parameters $(\alpha, \beta) = \left(\frac{3}{2}, \frac{3}{2}\right)$ regardless of training stage. However, the distribution of the per-prompt $\mu(x) = \frac{1}{G} \sum_{i=1}^{G} r(x, y_i)$ shifts as $\pi_\theta$ improves, suggesting that the normalization itself should adapt. BNPO (Xiao et al., 2025) (App. Tables 6 and 11) addresses this by dynamically normalizing advantages using a Beta distribution whose parameters are re-estimated each batch via method of moments. BNPO replaces the fixed denominator with an adaptive Beta density $f(\mu(x); \alpha, \beta)$ in Eq. (5.3.1.1).

$$A_{\alpha,\beta}(x, y) = \frac{r(x, y) - \mu(x)}{f_N(\mu(x); \alpha, \beta)} \tag{5.3.1.1}$$

The parameters $(a, b)$ of the underlying Beta distribution $f_D(\mu(x); a, b)$ are estimated via method of moments over $\{\mu(x_i)\}$ in the current batch, and the normalizing parameters $(\alpha, \beta)$ are set to the variance-minimizing values as shown in Eq. (5.3.1.2).

$$
\begin{aligned}
a &= \left(\frac{\mathbb{E}_x[\mu(x)] \left(1 - \mathbb{E}_x[\mu(x)]\right)}{\mathrm{Var}_x[\mu(x)]} - 1\right) \mathbb{E}_x[\mu(x)], & \alpha &= 1 + \frac{a}{3} \\
b &= \left(\frac{\mathbb{E}_x[\mu(x)] \left(1 - \mathbb{E}_x[\mu(x)]\right)}{\mathrm{Var}_x[\mu(x)]} - 1\right) (1 - \mathbb{E}_x[\mu(x)]), & \beta &= 1 + \frac{b}{3}
\end{aligned}
\tag{5.3.1.2}
$$

Setting $(\alpha, \beta) = (1, 1)$ recovers REINFORCE with baseline ($f = 1$); $(\alpha, \beta) = \left(\frac{3}{2}, \frac{3}{2}\right)$ recovers GRPO since $f\left(\mu; \frac{3}{2}, \frac{3}{2}\right) \propto \sqrt{\mu(1-\mu)} = \sigma(x)$ for Bernoulli rewards. BNPO dynamically adjusts $(\alpha, \beta)$ instead, yielding lower-variance gradients throughout training. The gradient coefficient is $\mathrm{GC}_{\mathrm{BNPO}}(x, y_i, t) = c_{i,t}\, \rho_{i,t}\, A_{\alpha,\beta}(x, y_i)$, identical to the GRPO gradient coefficient (Eq. (2.5.1.2)) but without the KL regularization term. For $N$ binary reward functions $r_1(x, y), \ldots, r_n(x, y)$ (e.g. accuracy and format rewards), BNPO normalizes each reward separately before averaging in Eq. (5.3.1.3)

$$A(x, y) = \frac{1}{N} \sum_{n=1}^{N} \frac{r_n(x, y) - \mu_n(x)}{f(\mu_n(x); \alpha_n, \beta_n)} \tag{5.3.1.3}$$

where $r_n$ denotes the $n$-th reward function and $\mu_n(x)$ its per-prompt $\mu$, avoiding reward collapse from summing rewards before normalization.

### 5.3.2 Multi-objective Normalization

GDPO (Liu et al., 2026a) addresses training with multiple reward functions, where naively summing rewards prior to group normalization can lead to reward collapse, erasing signal differences between responses. BNPO, as discussed in Section 5.3.1, also addresses the integration of multiple reward signals.

**Group reward-Decoupled Normalization Policy Optimization (GDPO)**   When extending GRPO to $n$ different reward functions $r_1(x, y), \ldots, r_n(x, y)$, summing all rewards and applying group-relative normalization (Eq. (2.5.1.4)) to the aggregate, distinct reward combinations collapse into identical advantage values (reward collapse). GDPO (Liu et al., 2026a) (App. Tables 6 and 11) resolves this by performing decoupled group-wise normalization, i.e., normalizing each reward independently before aggregation. Each per-reward advantage is computed in Eq. (5.3.2.1):

$$A_n(x, y_i) = \frac{r_n(x, y_i) - \mu_n(x)}{\sigma_n(x)} \tag{5.3.2.1}$$

where $\mu_n(x)$ and $\sigma_n(x)$ are the mean and standard deviation of $\{r_n(x, y_i)\}_{i=1}^{G}$ across the $G$ responses for prompt $x$. The combined advantage sums the per-reward advantages in Eq. (5.3.2.2).

$$A_{\text{sum}}(x, y_i) = \sum_{n=1}^{N} A_n(x, y_i) \tag{5.3.2.2}$$

Lastly, a batch-level re-normalization ensures the magnitude does not scale with $n$ in Eq. (5.3.2.3) where $\mu_{\mathcal{B}}$ and $\sigma_{\mathcal{B}}$ are the mean and standard deviation of $\{A_{\text{sum}}(x, y)\}$ across all responses in the batch $\mathcal{B}$.

$$\hat{A}_{\text{sum}}(x, y_i) = \frac{A_{\text{sum}}(x, y_i) - \mu_{\mathcal{B}}}{\sigma_{\mathcal{B}}} \tag{5.3.2.3}$$

The gradient coefficient is $\text{GC}_{\text{GDPO}}(x, y_i, t) = c_{i,t}\, \rho_{i,t}\, \hat{A}_{\text{sum}}(x, y_i)$, identical to the GRPO gradient coefficient (Eq. 2.5.1.2) but with the two-step normalized advantage $\hat{A}_{\text{sum}}$ and without KL divergence.

### 5.3.3 Two-step Normalization

In two-step normalization, the advantage is derived through two-step: either 1. group mean, group standard deviation, batch mean and batch standard deviation in Magistral (Mistral-AI et al., 2025) in Section 3.2.1, GDPO (Liu et al., 2026a) in Section 5.3.2 and REINFORCE++ (Hu et al., 2025a) or 2. value function and batch mean and batch standard deviation in ORZ (Hu et al., 2025b) in Section 5.2.5, SPO (Xu & Ding, 2026) in Section 5.2.5 and VinePPO (Kazemnejad et al., 2025) in Section 4.2.2.

**REINFORCE++**   Unlike GRPO and RLOO, which use prompt-level (local) normalization, a theoretically biased estimator that causes training instability and overfitting when group sizes are small. REINFORCE++ (Hu et al., 2025a) (App. Tables 6 and 11) addresses these issues by combining local group normalization with global batch-level advantage normalization, retaining the PPO clipped surrogate (Eq. (2.4.2.3)). REINFORCE++ proposes two variants: the base REINFORCE++ ($G \geq 1$) follows PPO, and REINFORCE++$_{w/\text{Baseline}}$ ($G > 1$) is built on GRPO with $G$ generations per prompt. In the base **REINFORCE++** variant ($G \geq 1$), the raw advantage folds a $k_1$-style KL penalty into the outcome reward $r(x, y_i)$, and is then normalized across the entire batch as shown in Eq. 5.3.3.1

$$A_{i,t} = r(x, y_i) - \beta \sum_{s=t}^{|y_i|} \log \frac{\pi_{\theta_{\text{old}}}(y_i^s \mid x, y_i^{<s})}{\pi_{\text{ref}}(y_i^s \mid x, y_i^{<s})}, \quad \hat{A}_{i,t}^{\text{norm}} = \frac{A_{i,t} - \mu_{\text{batch}}}{\sigma_{\text{batch}}} \tag{5.3.3.1}$$

where $\mu_{\text{batch}}$ and $\sigma_{\text{batch}}$ are computed over all token-level advantages $\{A(x, y_i, t)\}$ in the batch. For the REINFORCE++$_{w/\text{Baseline}}$ variant ($G > 1$), the advantage is computed in two steps: first subtracting the group $\mu_{\text{group}}$ reward for reshaping, then normalizing by batch-level mean $\mu_{\text{batch}}$ and standard deviation $\sigma_{\text{batch}}$, as shown in Eq. 5.3.3.2.

$$A'_i = r(x, y_i) - \mu_{\text{group}}, \quad \hat{A}_i^{\text{norm}} = \frac{A'_i - \mu_{\text{batch}}}{\sigma_{\text{batch}}} \tag{5.3.3.2}$$

Both variants follow the GRPO objective (Eq. (2.5.1.1)) but replace GRPO's $k_3$ KL estimator with the $k_2$ estimator $D_{\text{KL}}^{(k_2)}(x, y_i, t) = \frac{1}{2}\left(\log\frac{\pi_\theta(y_i^t|x,y_i^{<t})}{\pi_{\text{ref}}(y_i^t|x,y_i^{<t})}\right)^2$ as a separate loss term; the only difference is how $\hat{A}^{\text{norm}}$ is computed (Eq. (5.3.3.1) vs. Eq. (5.3.3.2)). The gradient coefficient is $\text{GC}_{\text{RF++}}(x, y_i, t) = c_{i,t}\,\rho_{i,t}\,A_{i,t}^{\text{norm}} + \lambda\log\frac{\pi_{\text{ref}}(y_i^t|x,y_i^{<t})}{\pi_\theta(y_i^t|x,y_i^{<t})}$, which replaces the GRPO gradient coefficient's $k_3$ KL term $\beta\left(\frac{\pi_{\text{ref}}}{\pi_\theta} - 1\right)$ (Eq. (2.5.1.2)) with the $k_2$ KL term $\lambda\log\frac{\pi_{\text{ref}}(y_i^t|x,y_i^{<t})}{\pi_\theta(y_i^t|x,y_i^{<t})}$.

### 5.3.4 Normalization-free

Some methods drop advantage normalization entirely. Dr. GRPO (Liu et al., 2025d) in Section 5.4.3 removes both std-normalization and per-response length normalization, retaining only mean-centering to eliminate length-dependent and variance-induced biases. GPG (Chu et al., 2026) in Section 5.1.3 and AAPO (Xiong et al., 2025a) in Section 5.2.7 similarly forgo std-normalization, relying on IS clipping, data filtering, or momentum terms for stability.

### 5.4 Length Normalization

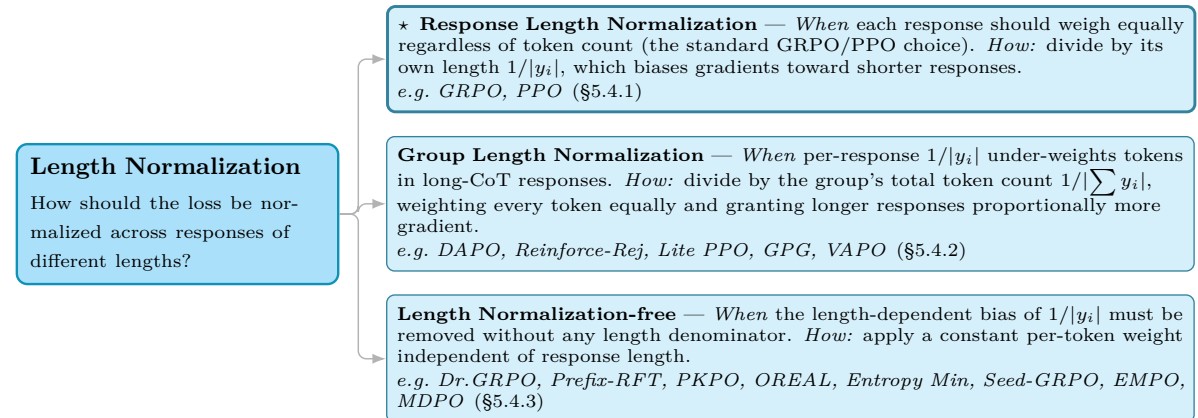

Figure 16: **Length normalization.** The denominator that aggregates per-token gradients across responses of differing length—per-response length, group-response length, or a constant per-token weight—and its resulting bias toward shorter or longer generations (§5.4; ⋆ default).

Length normalization determines how per-token gradient contributions are aggregated across responses of varying lengths (Figure 16). Figure 17 illustrates three common strategies. **Response length normalization** divides by each response's own length $\frac{1}{|y_i|}$ (e.g., 1/5, 1/7), treating every response equally but potentially biasing training toward shorter outputs (Shao et al., 2024; Schulman et al., 2017b; Ouyang et al., 2022). **Group length normalization** divides by the total length of all responses $\frac{1}{\sum_i |y_i|}$ (e.g., 1/17 for all tokens), giving longer responses proportionally more gradient weight (Yu et al., 2025; Xiong et al., 2025b; Liu et al., 2026b). **Length normalization-free** methods apply no scaling (weight = 1 per token), (e.g., 1 for all tokens), letting every token contribute equally regardless of response length (Liu et al., 2025d). This choice also interacts with several methods discussed later, including VAPO (Yue et al., 2025) (Section 5.2.5), MAGIC (He et al., 2025) (Section 5.5.2), and PURE (Cheng et al., 2025a) (Section 5.2.4).

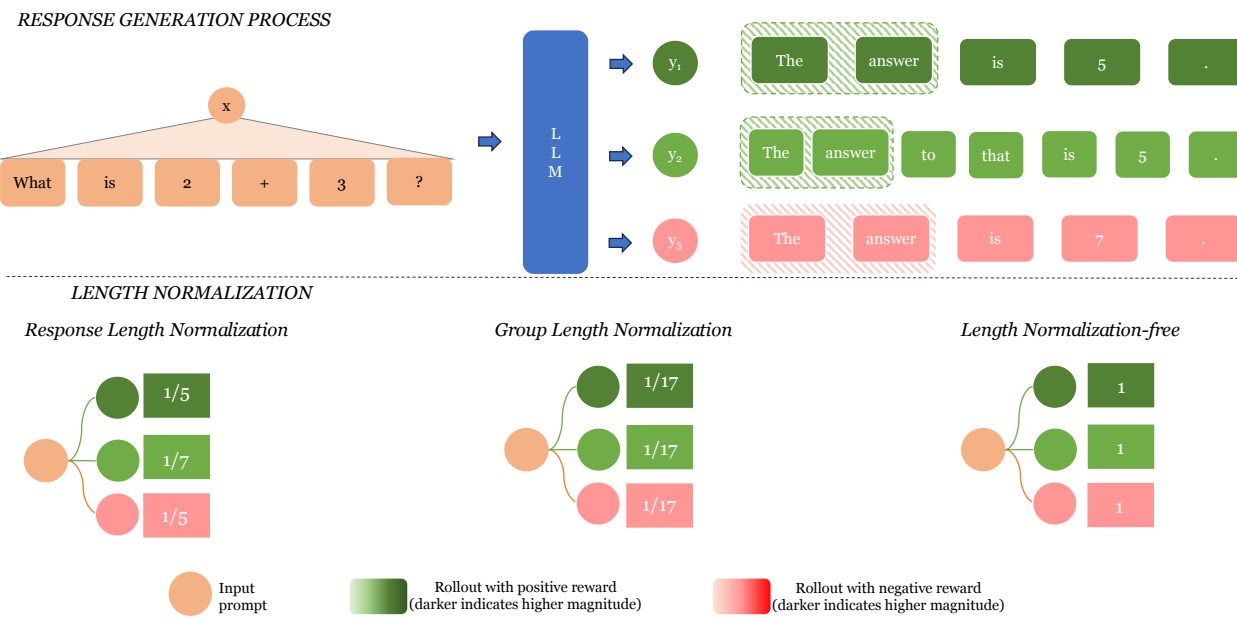

Figure 17: Length Normalization: Response length normalization, group-length normalization and length normalization-free.

### 5.4.1 Response Length Normalization

The standard per-response normalization $\frac{1}{|y_i|}$, used in GRPO (Shao et al., 2024) and PPO (Schulman et al., 2017b; Ouyang et al., 2022), ensures that each response contributes equally to the gradient regardless of token count. While this prevents long responses from dominating updates, it can inadvertently bias the model toward shorter responses when combined with per-group advantage normalization, motivating the group-level and normalization-free alternatives discussed in the subsections below.

### 5.4.2 Group Length Normalization

Group length normalization $\frac{1}{\sum_{i=1}^{G}|y_i|}$ replaces per-response normalization, giving every token equal contribution and longer responses proportionally more gradient weight (Yu et al., 2025; Xiong et al., 2025b; Liu et al., 2026b). In addition, GPG (Chu et al., 2026) in Section 5.1.3 and VAPO (Yue et al., 2025) in Section 5.2.5 also use group-total token normalization.

**Dynamic sAmpling Policy Optimization (DAPO)** Applying naive GRPO to large-scale long-CoT RL yields only 30% on AIME 2024 (vs. DeepSeek-R1-Zero's 47%), with entropy collapse, reward noise, and training instability as the root causes. DAPO (Yu et al., 2025) (App. Tables 6 and 11) diagnoses these four failure modes and open-sources a complete, reproducible fix. DAPO introduces four modifications to address entropy collapse, reward noise, and training instability in long-CoT RL. (1). **Clip-Higher** decouples the clipping bounds into $\varepsilon_{\text{low}}$ and $\varepsilon_{\text{high}}$ (with $\varepsilon_{\text{high}} > \varepsilon_{\text{low}}$), widening the upper bound to encourage exploration. (2). **Dynamic Sampling** filters out prompts where all $G$ outputs receive identical rewards (i.e., requiring $0 < |\{y_i : r(x, y_i) = 1\}| < G$). (3). **Token-level loss** replaces the per-response $\frac{1}{|y_i|}$ normalization with $\frac{1}{\sum_{i=1}^{G}|y_i|}$ over all tokens in the group. (4). **Overlong Reward Shaping** adds a soft length penalty $r_{\text{length}}(x, y)$ (Eq. 5.4.2.1) to the outcome reward $r(x, y)$, where $L_{\text{max}}$ is the hard generation cap and $L_{\text{cache}}$ is the width of a buffer zone before it. The DAPO objective is Eq. 5.4.2.2.

$$r_{\text{length}}(x, y) = \begin{cases} 0, & |y| \leq L_{\text{max}} - L_{\text{cache}}, \\ \frac{(L_{\text{max}} - L_{\text{cache}}) - |y|}{L_{\text{cache}}}, & L_{\text{max}} - L_{\text{cache}} < |y| \leq L_{\text{max}}, \\ -1, & |y| > L_{\text{max}}. \end{cases} \tag{5.4.2.1}$$

$$J_{\text{DAPO}}(\theta) = \mathbb{E}_{x \sim \mathcal{D}, \; \{y_i\}_{i=1}^G \sim \pi_{\theta_{\text{old}}}(\cdot|x)}$$

$$\left[ \frac{1}{\sum_{i=1}^G |y_i|} \sum_{i=1}^G \sum_{t=1}^{|y_i|} \min(\rho_{i,t} A_{i,t}, \; \text{clip}(\rho_{i,t}, 1 - \varepsilon_{\text{low}}, 1 + \varepsilon_{\text{high}}) A_{i,t}) \right] \quad (5.4.2.2)$$

$$\text{s.t.} \quad 0 < |\{y_i : r(x, y_i) = 1\}| < G$$

where $A_{i,t} = A_{\text{GRPO}(x,y_i)} = \frac{r(x,y_i) - \mu(x)}{\sigma(x)}$ (Eq. 2.5.1.4) and $\rho_{i,t} = \frac{\pi_\theta(y_i^t|x,y_i^{<t})}{\pi_{\theta_{\text{old}}}(y_i^t|x,y_i^{<t})}$. The gradient coefficient is $\text{GC}_{\text{DAPO}}(x, y_i, t) = c_{i,t} \, \rho_{i,t} \, A_{i,t}$, identical to the GRPO gradient coefficient (Eq. 2.5.1.2) but without the KL regularization term.

**Reinforce with Rejection (Reinforce-Rej)** The authors argue that GRPO's performance advantage over vanilla REINFORCE stems not from reward normalization but from implicitly discarding prompts where all sampled responses are incorrect. Motivated by this, Reinforce-Rej (Xiong et al., 2025b) (App. Tables 6 and 11) makes this filtering explicit. Reinforce-Rej shares several design choices with DAPO (Yu et al., 2025): the prompt-level mixed-correctness filter (restricting training to prompts where $-1 < \mu(x) = \frac{1}{n} \sum_{i=1}^n r(x, y_i) < 1$), asymmetric clipping, and the removal of KL divergence or entropy regularization.

The key distinction from DAPO lies in the **advantage formulation**: Reinforce-Rej applies the binary reward $r(x, y_i) \in \{-1, +1\}$ *directly* as the advantage, entirely foregoing the group mean and standard deviation normalization used in DAPO (and GRPO). This is motivated by the finding that reward normalization contributes minimally to performance, while prompt-level filtering is the dominant factor. The gradient coefficient is $\text{GC}_{\text{Reinforce-Rej}}(x, y_i, t) = \mathcal{I}(-1 < \mu(x) < 1) c_{i,t} \, \rho_{i,t} \, r(x, y_i)$.

**Lite PPO** Motivated by growing confusion over conflicting RL technique recommendations, Lite PPO (Liu et al., 2026b) (App. Tables 6 and 11) identifies a minimal two-technique recipe that unlocks learning capacity in critic-free policies using a vanilla PPO loss, outperforming technique-heavy methods such as DAPO while adding only two targeted changes to the GRPO baseline. First, it uses *mixed advantage normalization*: the $\mu_{\text{group}}$ is computed at the group level (per prompt) while the standard deviation ($\sigma_{\text{batch}}$) is computed at the batch level (across all $N \times G$ responses), preventing the small-denominator instability that arises when group rewards are highly concentrated as shown in Eq. (2.5.1.4).

$$A_{\text{Lite}}(x, y_i) = \frac{r(x, y_i) - \mu_{\text{group}}(x)}{\sigma_{\text{batch}}} \quad (5.4.2.3)$$

Second, it adopts *token-level loss aggregation*: the per-response $\frac{1}{|y_i|}$ normalization in GRPO is replaced by $\frac{1}{\sum_{i=1}^G |y_i|}$ over all tokens, so every token contributes equally regardless of sequence length. The gradient coefficient is $\text{GC}_{\text{LitePPO}}(x, y_i, t) = c_{i,t} \, \rho_{i,t} \, A_{\text{Lite}}(x, y_i)$, identical to the GRPO gradient coefficient (Eq. (2.5.1.2)) but without the KL regularization term.

### 5.4.3 Length Normalization-free

Dr. GRPO (Liu et al., 2025d) removes length normalization entirely, arguing that $\frac{1}{|y_i|}$ introduces a length-dependent bias that systematically underweights tokens in longer responses. Besides that, length normalization-free can be also found in Prefix-RFT in Section 4.2.1, PKPO (Walder & Karkhanis, 2025) in Section 4.3.4, OREAL (Lyu et al., 2025) in Section 5.1.1, Entropy Min (Agarwal et al., 2025) in Section 5.2.3, Seed-GRPO (Chen et al., 2025) in Section 5.2.3, EMPO (Zhang et al., 2025d) in Section 5.2.3, and MDPO (Kimi Team, 2025) in Section 5.2.6.

**GRPO Done Right (Dr.GRPO)** Standard GRPO introduces optimization biases via per-response length normalization $\frac{1}{|y_i|}$ and advantage standard normalization that artificially inflate response lengths and miscalibrate gradient magnitudes across question difficulties. Dr. GRPO (Liu et al., 2025d) (App. Tables 6 and 11) removes three sources of bias from standard GRPO: (i) the per-response length normalization $\frac{1}{|y_i|}$, (ii) the standard normalization $\sigma(x)$ in the advantage, and (iii) the KL divergence. The unbiased advantage

retains only mean-centering as defined in Eq. (5.4.3.1), where $\mu(x)$ is the group reward mean and $r(x, y_i)$ is the outcome reward.

$$A_i = r(x, y_i) - \mu(x) \tag{5.4.3.1}$$

The resulting objective is given in Eq. (5.4.3.2).

$$J_{\text{Dr.GRPO}}(\theta) = \mathbb{E}_{x \sim \mathcal{D}, \ \{y_i\}_{i=1}^{G} \sim \pi_{\theta_{\text{old}}}(\cdot|x)} \left[ \frac{1}{G} \sum_{i=1}^{G} \sum_{t=1}^{|y_i|} \min(\rho_{i,t} A_i, \ \text{clip}(\rho_{i,t}, 1-\varepsilon, 1+\varepsilon) A_i) \right] \tag{5.4.3.2}$$

The gradient coefficient is $\text{GC}_{\text{Dr.GRPO}}(x, y_i, t) = c_{i,t} \, \rho_{i,t} \, A_i$.

## 5.5 Regularization

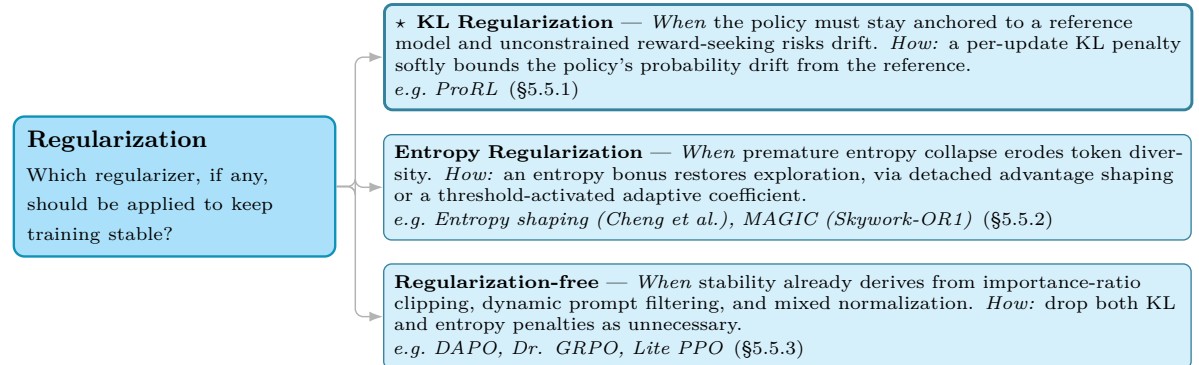

Figure 18: **Regularization.** Which explicit regularizer constrains the RLVR update—a KL penalty anchoring the policy to a reference, an entropy bonus countering premature entropy collapse, or none when clipping and prompt filtering already supply stability (§5.5; ⋆ default).

Explicit regularization prevents policy collapse or divergence from reference behaviors (Figure 18). KL regularization anchors the policy to a reference model and entropy regularization maintains output diversity. Methods that drop all regularization rely on IS clipping, dynamic sampling, and mixed normalization for stability are discussed in Regularization-free.

### 5.5.1 KL Regularization

KL divergence from a reference policy provides a soft anchor preventing excessive policy drift per update. Most papers include KL divergence to constrain the difference from the reference model.

**Prolonged Reinforcement Learning (ProRL)** Standard RLVR training stalls after a few hundred steps because the cumulative KL from a fixed reference policy eventually dominates the objective and suppresses useful gradient signal. To solve this problem, ProRL (Liu et al., 2025a) (App. Tables 6 and 11) introduces *reference policy resets*, which periodically hard-resetting $\pi_{\text{ref}}$ to a recent snapshot of $\pi_\theta$ along with the optimizer state to prevent the cumulative KL from dominating and stalling policy updates. In addition, it adopts DAPO's *decoupled clipping* ($\varepsilon_{\text{high}} > \varepsilon_{\text{low}}$), where the larger upper bound allows greater probability increases for unlikely tokens (*clip-higher*), combined with *dynamic sampling* that filters prompts where all $G$ group responses are uniformly correct or incorrect (zero advantage). Lately, it retains the KL penalty $\beta \, D_{\text{KL}}^{(i,t)}$ in Eq. (2.5.1.1). The gradient coefficient is $\text{GC}_{\text{ProRL}}(x, y_i, t) = c_{i,t} \, \rho_{i,t} \, A_i + \beta \left( \frac{\pi_{\text{ref}}(y_i^t|x,y_i^{<t})}{\pi_\theta(y_i^t|x,y_i^{<t})} - 1 \right)$, identical to the GRPO gradient coefficient (Eq. (2.5.1.2)) with asymmetric clipping bounds.

### 5.5.2 Entropy Regularization

Entropy regularization adds a bonus rewarding diverse token distributions to prevent premature entropy collapse. The following two papers differ in how the bonus adapts: entropy shaping (Cheng et al., 2025a) uses a detached advantage-scaling term with a self-regulating negative feedback loop, while MAGIC (Skywork-OR1) (He et al., 2025) activates the bonus only when entropy drops below a target threshold via a fixed step-size rule.

**Reasoning with Exploration: An Entropy Perspective**  The authors observe that the policy's token-level entropy is consistently higher at positions that matter most for reasoning: **pivotal tokens** that serve as logical connectors (e.g. *because*, *therefore*), **reflective actions** where the model self-verifies or self-corrects, and **rare behaviors** that go beyond the base model's typical strategies (Cheng et al., 2025a) (App. Tables 6 and 11). Since entropy naturally signals these exploratory reasoning moments, they propose shaping the advantage with an entropy-based term. Let $H_t = -\sum_{v \in \mathcal{V}} \pi_\theta(v \mid x, y^{<t}) \log \pi_\theta(v \mid x, y^{<t})$ be the per-token policy entropy. The shaped advantage that replaces $A_{i,t}$ in the GRPO gradient (Eq. (2.5.1.2)) is defined in Eq. (5.5.2.1), where $\alpha > 0$ scales the entropy contribution and the min clips it so that $\phi(H_t) \leq \frac{|A_t|}{2}$, preventing the term from dominating or reversing the sign of $A_t$.

$$A_t^{\text{shaped}} = A_t + \phi(H_t), \quad \phi(H_t) = \min\left(\alpha \cdot H_t^{\text{detach}}, \frac{|A_t|}{2}\right) \tag{5.5.2.1}$$

Because $H_t^{\text{detach}}$ is detached from the computational graph ($\nabla_\theta H_t^{\text{detach}} = 0$), the shaped advantage adjusts update magnitudes without altering gradient flow. In addition, it adopts asymmetric clipping ($\varepsilon_{\text{low}} = 0.2$, $\varepsilon_{\text{high}} = 0.28$) and token-level loss averaging across the batch, i.e., $\frac{1}{\sum_{i=1}^{G} |y_i|}$ rather than per-response $\frac{1}{|y_i|}$ normalization from DAPO.

The gradient coefficient is $\text{GC}(x, y_i, t) = c_{i,t} \cdot \rho_{i,t} \cdot A_t^{\text{shaped}}$, identical to the GRPO gradient coefficient (Eq. (2.5.1.2)) but with $A_t^{\text{shaped}}$ replacing $A_{i,t}$ and without the KL regularization term. The method is also self-regulating via a negative feedback loop: high $H_t$ increases $\phi(H_t)$, producing a stronger update that sharpens the distribution at that position, which lowers $H_t$ and automatically shrinks the entropy bonus in subsequent iterations to avoid over-encouragement without manual scheduling.

**Multi-stage Adaptive entropy scheduling for GRPO In Convergence (MAGIC)**  While prior RLVR methods apply RL to base models with fixed entropy regularization, MAGIC (He et al., 2025) (App. Tables 6 and 11) targets long CoT SFT models with adaptive threshold-based entropy control. In addition, MAGIC modifies the GRPO objective (Eq. (2.5.1.1)) by replacing per-response length normalization $\frac{1}{|y_i|}$ with batch-level token averaging $\frac{1}{\mathcal{N}_k}$ where $\mathcal{N}_k = \sum_{i \in \mathcal{T}_k} \sum_{j=1}^{G} |y_{i,j}|$ is the total token count, removing KL divergence, filtering out zero-advantage prompt groups ($\mathcal{T}_k := \{i \in [N] : \exists j \in [G], A_{i,j} \neq 0\}$). The objective is given in Eq. (5.5.2.2)

$$J_{\text{MAGIC}}(\theta) = \mathbb{E}_{x \sim \mathcal{D}, \{y_j\}_{j=1}^{G} \sim \pi_{\theta_{\text{old}}}(\cdot|x)} \left[ \frac{1}{\mathcal{N}_k} \sum_{i \in \mathcal{T}_k} \sum_{j=1}^{G} \sum_{t=1}^{|y_{i,j}|} \right.$$

$$\left. \left( \min\left(\rho_{i,j}^t A_{i,j}, \text{clip}(\rho_{i,j}^t, 1-\varepsilon, 1+\varepsilon) A_{i,j}\right) + \alpha_k H\left(\pi_\theta(\cdot \mid x_i, y_{i,j}^{<t})\right) \right) \right] \tag{5.5.2.2}$$

where $\rho_{i,j}^t = \frac{\pi_\theta(y_{i,j}^t \mid x_i, y_{i,j}^{<t})}{\pi_{\theta_{\text{old}}}(y_{i,j}^t \mid x_i, y_{i,j}^{<t})}$, $A_{i,j} = \frac{r(x_i, y_{i,j}) - \mu(x_i)}{\sigma(x_i)}$ is the group-normalized advantage (Eq. (2.5.1.4)), and $H(\cdot)$ is the next-token entropy. The entropy coefficient $\alpha_k$ adapts at each step $k$ via Eq. (5.5.2.3):

$$\alpha_k = c_k \cdot \mathbb{I}\{e_k \leq e_{\text{tgt}}\}, \quad c_{k+1} = \begin{cases} c_k + \Delta & \text{if } e_k < e_{\text{tgt}} \\ c_k - \Delta & \text{if } e_k > e_{\text{tgt}} \end{cases} \quad c_0 = 0 \tag{5.5.2.3}$$

where $e_k$ is the current entropy, $e_{\text{tgt}}$ the target lower bound, $\Delta$ a fixed step-size hyperparameter, and the bonus activates only when $e_k$ drops below $e_{\text{tgt}}$. The gradient coefficient is $\text{GC}_{\text{MAGIC}}(x, y_{i,j}, t) = c_{i,j}^t \, \rho_{i,j}^t \, A_{i,j} - \alpha_k \, \log \pi_\theta \big( y_{i,j}^t \mid x_i, y_{i,j}^{<t} \big)$ where $c_{i,j}^t \in \{0, 1\}$ is the PPO clipping indicator that zeros out any token whose importance ratio $\rho_{i,j}^t$ falls outside $[1-\varepsilon, \, 1+\varepsilon]$ which is distinct from the adaptive entropy schedule scalar $c_k$ in Eq. (5.5.2.3).

### 5.5.3 Regularization-free

Methods including DAPO (Yu et al., 2025) in Section 5.4.2, Dr. GRPO (Liu et al., 2025d) in Section 5.4.3, and Lite PPO (Liu et al., 2026b) in Section 5.4.2 remove both KL divergence and entropy regularization, relying on IS ratio clipping, dynamic prompt filtering, and mixed normalization for stability.

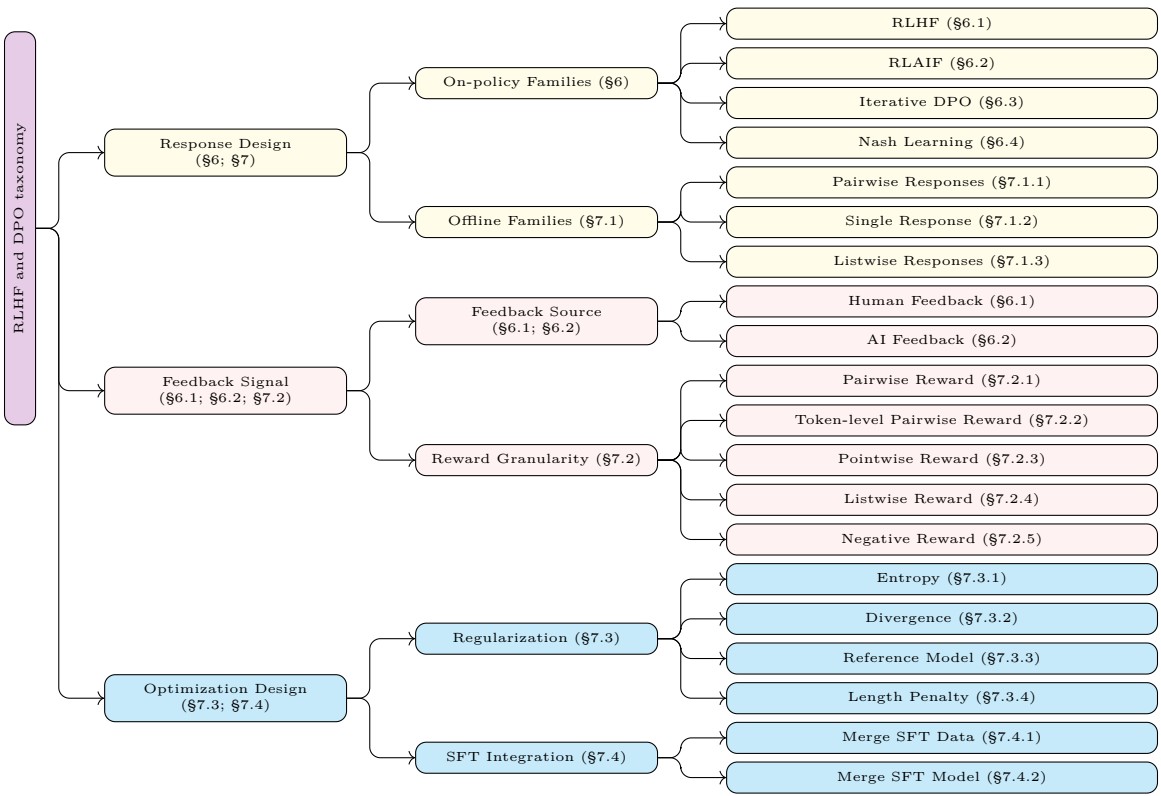

Figure 19: Comprehensive RLHF and DPO taxonomy.

# 6 RLHF and Iterative DPO: On-Policy Methods

**Response Sampling**

Should responses be generated online from the current policy, or drawn from a fixed dataset?

⋆ **On-policy** — *When* policy-data mismatch and reward-model staleness must be removed during training. *How:* regenerate responses from the current policy each step and score them with a reward model, LLM-judge, or preference oracle. *e.g. RLHF (PPO), RLAIF (Constitutional AI, Distilled/Direct RLAIF, Rule-Based Rewards), Iterative DPO (RSO, Self-Rewarding LMs, CRINGE, Meta-Rewarding), Nash learning (NLHF/Nash-MD, SPPO, DNO)* (§6)

**Offline** — *When* reward modeling and online rollouts are to be avoided and a fixed preference corpus already exists. *How:* optimize the pre-collected preferences directly, accepting distribution mismatch from unknown mixed behavior policies. *e.g. DPO, SLiC-HF, DPOP, sDPO, beta-DPO, IPO, RPO, GPO, KTO, DRO, UNA, RRHF, PRO, LiPO, TDPO, D2O, NPO, CPO, f-DPO, SimPO, R-DPO, ORPO, UFT, PAFT* (§7)

Figure 20: **Response sampling paradigm.** Whether preference-optimization responses are continuously regenerated from the current policy and scored online, or drawn once from a fixed, pre-collected preference corpus (⋆ default).

RLHF and RLVR differ primarily in how their reward signals are constructed (Figure 20). In RLHF, a reward model is trained to capture human preferences via the Bradley-Terry model: given a set of prompts, the framework iteratively generates responses, scores them with the learned reward model, and updates the LLM policy via reinforcement learning. Because the training distribution is continuously refreshed through this generation loop, RLHF is inherently on-policy. RLAIF follows the same on-policy paradigm but replaces the learned BT reward model with an LLM-as-a-judge, sidestepping the need for explicit human annotation.

DPO, by contrast, is an offline method: it optimizes the policy directly on a fixed, pre-collected preference dataset, bypassing the need for an explicit reward model or online generation. Iterative DPO extends this by closing the loop: at each iteration, the current policy generates multiple responses, a preference signal is used to identify desired and undesired outputs from among them, and the model is fine-tuned on the resulting pairs. This makes iterative DPO an on-policy approach that progressively refines the training distribution. Nash learning-based methods take a fundamentally different stance by replacing the pointwise reward model with a preference model grounded in game theory. A comprehensive summary of the RLHF and DPO algorithms, including offline DPO, is provided in Algorithm 2, and a detailed taxonomy of these methods is illustrated in Figure 19.

---

**Algorithm 2** RLHF and DPO: Key Design Choices

---

     **Unified Policy Gradient:** same form as Algorithm 1
1: **Initialise:** $\pi_\theta$, $\pi_{\text{ref}}$, $\pi_{\text{old}} \leftarrow \pi_\theta$ *(on-policy)* | fixed dataset $\mathcal{D}_{\text{off}}$ *(offline)*
2: **for** $iter = 1, 2, \ldots, N$ **do**
3:     **Prompt Sampling:** sample $x \sim \mathcal{D}$
4:     **Response Generation**
        **6** *On-policy*: generate $y \sim \pi_\theta(\cdot \mid x)$ from current policy
        **7** *Offline*: responses from fixed $\mathcal{D}_{\text{off}}$, not generated on-policy
           **7.1** *Response type:* pairwise | single | listwise
5:     **Reward**
        **6** *On-policy:* RLHF | RLAIF | Iterative DPO | Nash Learning
        **7.2** *Offline:* pairwise | token-level | pointwise | listwise | negative
6:     **for** $t = 1, \ldots, |y|$ **do**
7:         **Gradient Coefficient**
           **7.3** *Regularisation:* entropy | divergence | ref model | length penalty
8:         **Accumulate:** $g \mathrel{+}= \text{GC}(x, y, t) \cdot \nabla_\theta \log \pi_\theta(y^t \mid x, y^{<t})$
9:     $\theta \leftarrow \theta + \eta\, g$;   $\pi_{\text{old}} \leftarrow \pi_\theta$ *(on-policy only)*;
    **7.4** *Merge SFT (offline):* merge SFT data | merge SFT model | none
10: **return** $\pi_\theta$

---

## 6.1 RLHF

| Reward Model / Feedback Source |
|---|
| What should supply the reward signal that scores responses? |

⋆ **Human Feedback** — *When* human pairwise preference rankings are obtainable. *How:* fit a Bradley-Terry pointwise scalar reward from them, then score responses for KL-regularized PPO optimization.
*e.g. RLHF-OpenAI, RLHF-Anthropic (BT reward model with PPO) (§6.1)*

**AI Feedback** — *When* human harmlessness/preference annotation is prohibitively costly (and reward-model staleness under policy drift should be avoided). *How:* an LLM supplies the reward directly, either by grading enumerated binary safety propositions individually (rule-based) and summing them into a weighted feature reward, or by emitting a holistic pairwise or pointwise (1–10) judgment, optionally online.
*e.g. RLAIF-OpenAI Rule-Based Reward (RBR) (§6.2); Constitutional AI (RLAIF-Anthropic), RLAIF-Google (Distilled RLAIF, Direct RLAIF) (§6.2)*

Figure 21: **Reward model / feedback source.** What supplies the scoring signal in on-policy RLHF—a Bradley–Terry reward fitted to human pairwise preferences versus an LLM (grading enumerated safety propositions rule-based, or judging holistically) (§6; ⋆ default).

In RLHF, for each prompt, one response is generated by LLM (Figure 21). Then, a BT reward model is trained based on pairwise preference datasets. Then, the prompt and response will be sent to BT reward model for reward. The reward will then be utilized to optimize LLM through PPO. More details on RLHF can be found in Section 2.4.3.

## 6.2 RLAIF

In RLAIF, the trained BT reward model is replaced with LLM-as-a-Judge. For each prompt and response, they are combined through a system prompt and sent to LLM for evaluation.

**RLAIF-Anthropic**   Constitutional AI (Bai et al., 2022b) (App. Tables 12 and 14) was developed to address two problems with an earlier training method pursued by Anthropic (Bai et al., 2022a): the high cost of hiring people to label harmful content, and the tendency of the resulting models to simply refuse sensitive questions instead of responding in a helpful, nuanced way. Unlike RLHF-Anthropic, which requires human crowd-worker annotations for both helpfulness and harmlessness labels, Constitutional AI replaces all harmlessness feedback with model-generated signals guided by a written "constitution" of ∼16 natural-language harmlessness principles. Like RLHF-Anthropic, the RLAIF stage retains the same PPO-based RL pipeline and continues to use human feedback for helpfulness. The training proceeds in two stages.

In the *Supervised Learning (SL)* stage, a helpful RLHF model generates initial responses to red-team prompts. Each response is then iteratively critiqued and revised guided by a constitutional principle randomly sampled from the constitution; optional CoT prompting (Wei et al., 2023) improves critique quality. The revised responses, together with human helpfulness samples, are used for SFT to produce the SL-CAI model. The authors found that increasing the number of revisions progressively improved harmlessness scores while slightly reducing helpfulness, and that the critique-revision approach outperformed direct revision alone.

In the *RLAIF* stage, harmlessness preference labels are generated by an independent feedback model. Given a response pair generated by SL-CAI and a principle drawn uniformly from the constitutional set, the model is queried via a multiple-choice prompt to identify the less harmful response. The normalized log-probabilities over the answer choices are then used as soft preference targets for training. A preference model (PM) is then trained on a mixture of these AI harmlessness labels and human helpfulness labels, and PPO fine-tunes the SL-CAI policy against this PM. The resulting RL-CAI models are "harmless but non-evasive", i.e., engaging with sensitive queries and explaining their objections rather than refusing outright.

This study demonstrated the feasibility of self-supervised AI alignment by utilizing AI to collect preference data for harmlessness, replacing costly human annotation for that dimension while maintaining helpfulness via human feedback.

**RLAIF-Google** Unlike RLAIF by Anthropic (Bai et al., 2022b), who applied AI feedback only to harmlessness, the authors (Lee et al., 2024) (App. Tables 12 and 14) extended RLAIF to summarization and helpfulness as well. The paper proposes two RLAIF strategies that differ in how the AI signal is used to train the policy. *Distilled RLAIF* follows the canonical RLHF pipeline: AI-generated pairwise preferences train a RM, which then guides policy optimization via REINFORCE. The process is structurally identical to human-feedback RLHF, with AI labels substituting for human labels. *Direct RLAIF* (d-RLAIF), the key novel contribution, bypasses RM training entirely: the LLM is prompted to score each response on a scale of 1–10, and this pointwise score is used directly as the RL reward signal. d-RLAIF addresses the "staleness" issue of Distilled RLAIF, where the RM trained on generations from the initial policy becomes increasingly out-of-distribution as the policy improves during training.

Both strategies share the same AI feedback collection framework. A structured prompt is constructed from four components: 1. Preamble, 2. Few-shot exemplars (optional), 3. Sample to annotate, and 4. Ending. A two-step CoT procedure generates the AI preference. Firstly, the full prompt elicits a rationale from the LLM, and then the LLM's response is appended with a completion cue (e.g., "Preferred Summary=") and re-submitted to the LLM, which generates a preference token ("1" or "2"). The log-probabilities of these two tokens are extracted and converted via softmax to a soft preference distribution (e.g., 0.6 vs. 0.4). To mitigate positional bias, each pair is evaluated in both candidate orderings and the scores are averaged.

During the evaluation process, three key metrics were employed: 1. AI-labeler alignment: the degree of agreement between AI and human labelers, 2. win rate: the likelihood of a response being selected by human labelers when compared between two candidates, and 3. harmless rate: the percentage of responses deemed harmless by human evaluators. They observed that the RLHF policy sometimes hallucinated when the RLAIF policy did not, and RLAIF sometimes produced less coherent summaries as compared to RLHF. Three main conclusions were drawn. Firstly, RLAIF achieved comparable performance to RLHF in summarization and helpful dialogue generation tasks, but outperformed RLHF in the harmless task. Secondly, RLAIF demonstrated the ability to enhance a SFT policy even when the LLM labeler was of the same size as the policy. Lastly, Direct RLAIF surpassed Distilled RLAIF in terms of alignment.

**RLAIF-OpenAI** While Anthropic and Google distill AI preferences into reward models, OpenAI's RLAIF (Mu et al., 2024) (App. Tables 12 and 14) aligns safety by decomposing policies into fine-grained, LLM-graded propositions. This creates a Rule-Based Reward (RBR) integrated directly into PPO training, requiring minimal human calibration. To solve this problem, the authors divided the task of rating responses by LLM into specific rules that explicitly describe the desired and undesired behaviors and used the behaviors on individual tasks to cover complex evaluation behaviors. This could simplify the task of AI evaluation and allow fine grained control of model responses. Based on the prompt, the authors proposed a content policy and a behavior policy. The content policy defined precisely what content in a prompt is considered an unsafe request, and the behavior policy referred to how LLM should handle various kinds of unsafe requests defined in the content policy. In the case of applying LLM as a chat model, the content policy included erotic content, hate speech, criminal advice and self-harm, while the behavior policy contained Hard Refusal, Soft Refusal and Comply as the three response types.

Based on the content and behavior policies, an auxiliary safety rule-based reward function (RBR) was built for RL training. To begin with, the authors discovered that AI was better at classifying individual tasks rather than multilayered tasks like holistic rating. Thus, they proposed **propositions**, which are binary statements about responses given a prompt, and **rules**, which determine the ranking of responses for a given response type. Based on each individual proposition, a feature like $\phi_i(x, y)$ for the $i$-th feature could be obtained using classification-prompts for each proposition and a grader LLM. Eventually, the total reward as a weighted linear combination of all features was shown in Eq. 6.2.1 where $N$ features were considered in total.

$$r_{\mathrm{rbr}}(x, y, w) = \sum_{i=1}^{N} w_i \phi_i(x, y) \tag{6.2.1}$$

In the paper, the authors utilized 20 features for Hard-Refusal, 23 features for Soft-Refusal, and 18 features for Comply. Synthetic data $D_{\text{RBR}} = \{(x, y_1, y_2, \ldots, y_G)\}$ where the ranked $G$ responses $y_1 > y_2 > \ldots > y_G$ were utilized for fitting the weights in Eq. 6.2.1. The obtained reward from RBR was combined with the original reward model, i.e. $r_{\text{RM}}$ to play the role of total reward, i.e., $r_{\text{tot}}$ through $r_{\text{tot}} = r_{\text{RM}} + r_{\text{rbr}}$. Lastly, the weights were fitted by minimizing the hinge loss in Eq. 6.2.2 over all pairwise comparisons extracted from the ranked completions in $D_{\text{RBR}}$, where $|D_{\text{RBR}}|$ refers to the total number of prompts in the dataset.

$$L(w) = \frac{1}{|D_{\text{RBR}}|} \sum_{(x, y_w, y_l) \in D_{\text{RBR}}} \left( \max \left( 0, 1 + r_{\text{tot}}(x, y_l, w) - r_{\text{tot}}(x, y_w, w) \right) \right) \tag{6.2.2}$$

A comparison between the original help-only reward model, i.e., $r_{\text{RM}}$ and the final total reward, i.e., $r_{\text{tot}}$ was conducted. Results showed that $r_{\text{RM}}$ was tough to separate disallowed, perfect refusal and bad refusal. However, for $r_{\text{tot}}$, the rewards of different categories of prompts could be separated. Similar patterns were observed for error rate when comparing $r_{\text{RM}}$ with $r_{\text{tot}}$.

## 6.3   Iterative DPO

While DPO is traditionally used as an offline optimization technique, iterative DPO adapts it into an on-policy framework by generating candidate responses from the current policy for a given set of prompts and then labeling them as preferred or dispreferred. The model is subsequently updated using these pairwise preference signals. Consequently, iterative DPO is best understood as an on-policy method.

**Rejection Sampling Optimization (RSO)**   DPO optimizes a language model on offline preference pairs, but they are limited to human-collected data from unknown mixed policies, and it introduces a distribution mismatch that degrades alignment quality. RSO (Liu et al., 2024) (App. Tables 12 and 14) mitigates the policy–data distribution mismatch inherent in offline preference optimization methods by using rejection sampling to approximate samples from the estimated optimal policy.

The procedure operates as follows: (1) sample $y \sim \pi_{\text{sft}}(y|x)$ and $u \sim U[0,1]$; (2) compute $M = \min\{m \mid m\pi_{\text{sft}}(y|x) \geq \pi^\star(y|x)\}$; (3) accept $y$ if $u < \frac{\pi^\star(y|x)}{M\pi_{\text{sft}}(y|x)}$; otherwise reject; and (4) repeat until sufficient accepted samples are collected, ensuring proximity to the target policy. Since $M$ is intractable, RSO approximates the density ratio using a reward model, yielding the acceptance probability in Eq. 6.3.1

$$P_{\text{accept}}(x, y) = \frac{\pi^\star(y|x)}{M\pi_{\text{sft}}(y|x)} = \exp\left( \frac{r_\phi(x, y) - r_{\max}}{\beta} \right) \tag{6.3.1}$$

where $r_{\max}$ denotes the maximum reward among current candidates and $\beta$ controls selectiveness. As $\beta \to \infty$, all samples are accepted, while $\beta \to 0$ retains only the highest-reward response.

**Self-Rewarding Language Models**   A key limitation of DPO is the high cost of collecting new human preference data. Iterative (online) DPO addresses this by using a single LLM to jointly perform instruction following and self-instruction creation, rather than separating these capabilities into distinct models (Yuan et al., 2024) (App. Tables 12 and 14). In the *Self-Instruction Creation* phase, $G$ candidate responses are generated for a given prompt, and the same LLM acts as its own reward model via LLM-as-a-Judge prompting to evaluate each response. Rather than scoring five separate dimensions, the judge assigns a single additive score from 0 to 5, awarding one point for each of five quality criteria met: relevance, coverage, usefulness, clarity, and expertise, preceded by a brief chain-of-thought justification. The highest- and lowest-scoring responses form a preference pair where pairs with equal scores are discarded. During *Instruction Following* training, DPO is applied to these self-generated preference pairs, allowing the model to improve both its instruction-following and reward-modeling abilities simultaneously.

**ContRastive Iterative Negative GEneration (CRINGE)**   The CRINGE loss (Adolphs et al., 2023) handles positive and negative responses separately. Positive responses $(x, y_w)$ are processed like SFT, while negative responses $(x, y_l)$ contrast each negative token $y_l^t$ against a positive token. Let $s_{\theta,t}$ represent the

model output score for token $t$. Xu et al. (2024b) select top-k scores $\{s_{\theta,t}[1], ..., s_{\theta,t}[k]\}$ excluding $s_{\theta,t}[y_l^t]$, then sample via $s_{\theta,t}^* \sim \text{Softmax}(s_{\theta,t}[1], \ldots, s_{\theta,t}[k])$. The binary CRINGE loss is Eq. 6.3.2 and extending CRINGE to preference feedback (Xu et al., 2024c) (App. Tables 12 and 14) yields the pairwise loss in Eq. 6.3.3.

$$J_{Bin}(\pi_\theta) = \log P_\theta([x, y_w]) + \alpha \left[ \sum_{t=1}^{T} \log \left( \frac{\exp(s_{\theta,t}^*)}{\exp(s_{\theta,t}^*) + \exp(s_{\theta,t}[y_l^t])} \right) \right] \tag{6.3.2}$$

$$J_{Pair}(\pi_\theta) = g_\theta(x, y_w, y_l) J_{Bin}(\pi_\theta) \tag{6.3.3}$$

The gate function $g_\theta(x, y_w, y_l) = \sigma \left( \frac{b - (\log P_\theta(y_w|x) - \log P_\theta(y_l|x))}{\tau} \right)$ controls the loss: approaching zero when $y_w$ is much better than $y_l$, and one otherwise. Parameters $b$ and $\tau$ control the margin and smoothness respectively.

**Meta-Rewarding Language Models**  Self-Rewarding LMs improve the LLM policy through iterative DPO, but overlook the judge: a stagnant judge saturates feedback quality and causes reward hacking. To address this, the authors introduce Meta-Rewarding, a self-improvement framework that upgrades both the actor and the judge simultaneously through iterative DPO (Wu et al., 2025b) (App. Tables 12 and 14). The key insight is to introduce a third role: the meta-judge, which evaluates the quality of the judge's own evaluations, providing a feedback signal to improve judging ability. The process of assigning rewards to evaluations is termed "meta-rewarding." Length bias, a known failure mode in reward models where judges tend to favor verbose responses, is also explicitly mitigated. In this framework, the LLM simultaneously serves three roles: (1) actor (generating responses), (2) judge (evaluating responses), and (3) meta-judge (evaluating the quality of the judges). All three roles are performed by a single model, preserving the self-improving, human-supervision-free nature of the pipeline.

## 6.4  Nash Learning based Methods

Standard RLHF and DPO methods reduce preferences to a scalar reward via the Bradley-Terry model, which is brittle to non-transitive annotations and optimizes absolute scores rather than comparative win rates. Nash learning-based methods address this by framing alignment as a two-player zero-sum preference game where policies are directly compared via win probabilities rather than scalar rewards.

**Nash Learning from Human Feedback (NLHF)**  In RLHF, the BT model is used to learn a scalar reward function, which is then optimized via RL. A core limitation of this pipeline is that the objective effectively optimizes reward score rather than win probability against other policies. In addition, non-transitivity (e.g., $y_1 > y_2$, $y_2 > y_3$, but $y_3 > y_1$) can be amplified by annotator disagreement, and inaccurate preference ordering can misguide the policy toward a narrow set of responses and reduce diversity (Bertrand et al., 2023). Nash learning addresses these issues by defining the objective directly in preference space without a reward surrogate or reward model and by using the Nash equilibrium as the solution concept instead of reward maximization (Munos et al., 2024) (App. Tables 12 and 14). The preference probability between two policies $\pi_\theta(y|x)$ and $\pi_\theta'(y|x)$ is defined as Eq. 6.4.1 where this preference model does not depend on $\theta$.

$$P_\theta(\pi_\theta(y|x) > \pi_\theta'(y|x)) = \mathbb{E}_{x \sim \mathcal{D}, y \sim \pi_\theta(\cdot|x), y' \sim \pi_\theta'(\cdot|x)} \left[ P(y > y'|x) \right] \tag{6.4.1}$$

This formulation directly compares policies, eliminating the need for the BT model. The optimal policy is obtained by the Nash equilibrium in Eq. 6.4.2.

$$\pi_\theta^*(y|x) = \arg\max_{\pi_\theta} \min_{\pi_\theta'} P_\theta(\pi_\theta(y|x) > \pi_\theta'(y|x)) \tag{6.4.2}$$

For LLM alignment, a KL constraint to a reference model is introduced, yielding Eq. 6.4.3 to ensure that the distance from the aligned model to the initial model remains limited.

$$P_{\theta,\beta}(\pi_\theta(y|x) > \pi'_\theta(y|x)) = P_\theta(\pi_\theta(y|x) > \pi'_\theta(y|x)) - \beta D_{\text{KL}}(\pi_\theta(\cdot|x)||\pi_{\text{ref}}(\cdot|x)) \\ + \beta D_{\text{KL}}(\pi'_\theta(\cdot|x)||\pi_{\text{ref}}(\cdot|x)) \tag{6.4.3}$$

Building on the refined preference model, the authors introduced the Nash-Mirror Descent (Nash-MD) algorithm. This algorithm uses a regularized policy (Eq. 6.4.4) and a policy update step (Eq. 6.4.5), where $\alpha_t$ denotes the learning rate at step $t$. In Eq. 6.4.4, $\pi^t(y|x)$ is used rather than $\pi_\theta^t(y|x)$ because after the $t$-th optimization, the policy remains fixed and is no longer subject to further optimization.

$$\pi_{\text{mix}}^t(y|x) = \frac{\pi^t(y|x)^{1-\alpha_t\beta}\pi_{\text{ref}}(y|x)^{\alpha_t\beta}}{\sum_{y'}\pi^t(y'|x)^{1-\alpha_t\beta}\pi_{\text{ref}}(y'|x)^{\alpha_t\beta}} \tag{6.4.4}$$

$$\pi_\theta^{t+1}(y|x) = \arg\max_{\pi_\theta}\left[\alpha_t P_{\theta,\beta}(\pi_\theta(y|x) > \pi_{\text{mix}}^t(y|x)) - D_{\text{KL}}(\pi_\theta(\cdot|x)||\pi_{\text{mix}}^t(\cdot|x))\right] \tag{6.4.5}$$

The method is proven to converge by maintaining last-iterate policies.

**Self-Play Preference Learning (SPPO)**  Nash-MD's Mirror Descent requires maintaining a geometric mixture policy $\pi_{\text{mix}}^t$ through two-timescale updates (Eq. 6.4.4), introducing significant implementation complexity for large LLMs. SPPO (Wu et al., 2025c) (App. Tables 12 and 14) reinterprets RLHF as a two-player zero-sum game with a single agent representing both players and a multiplicative-weights self-play rule where the policy competes directly against its own previous iterate. The agent samples multiple trajectories evaluated by humans or models, using win rate as reward. This avoids explicit scalar reward regression (e.g., Bradley–Terry-style reward modeling) but still relies on a preference oracle (human judgements or a learned preference model, e.g., PairRM), and it naturally handles noisy and intransitive preferences. For LLMs, by leveraging the symmetry of the preference function (Freund & Schapire, 1999), the iterative/online policy update is derived as Eq. 6.4.6

$$\pi_\theta^{t+1}(y|x) = \frac{\pi^t(y|x)e^{\left(\frac{1}{\beta}P_\theta(y>\pi^t|x)\right)}}{Z_{\pi^t}(x)} \tag{6.4.6}$$

where $Z_{\pi^t}(x) = \sum_y \pi^t(y|x)e^{\left(\frac{1}{\beta}P_\theta(y>\pi^t|x)\right)}$ is the normalizing partition function and $P(\mathbf{y} > \pi|x) = E_{y'\sim\pi(\cdot|x)}[P(y>y'|x)]$ is the expected probability that $y$ is preferred over a random response $y'$ drawn from policy $\pi^t$. By reformulating Eq. 6.4.6, the authors derived $\log\left(\frac{\pi_\theta^{t+1}(y|x)}{\pi^t(y|x)}\right) = \frac{1}{\beta}P_\theta(y>\pi^t|x) - \log Z_{\pi^t}(x)$. A MSE loss is then adopted as the practical objective to update the policy, as shown in Eq. 6.4.7.

$$\pi_\theta^{t+1} = \arg\min_{\pi_\theta}\mathbb{E}_{x\sim X, y\sim\pi^t(y|x)}\left\{\left[\log\left(\frac{\pi_\theta(y|x)}{\pi^t(y|x)}\right) - \left(\frac{1}{\beta}P_\theta(y>\pi^t|x) - \log Z_{\pi^t}(x)\right)\right]^2\right\} \tag{6.4.7}$$

The estimations of $P_\theta(y > \pi^t|x)$ and $\log Z_{\pi^t}(x)$ are conducted through sampling and averaging. The authors opt to sample $G$ responses $y_1, y_2, \ldots, y_G \sim \pi^t$ for each prompt $x$, and represent the empirical distribution as $\hat{\pi}^t$. Consequently, $P_\theta(y > \hat{\pi}^t|x) = \frac{1}{G}\sum_{i=1}^G P_\theta(y > y_i|x)$ and $Z_{\hat{\pi}^t}(x) = \mathbb{E}_{y\sim\pi^t(y|x)}\left[e^{\left(\frac{1}{\beta}P_\theta(y>\pi^t|x)\right)}\right]$.

**Direct Nash Optimization (DNO)**  Previous Nash learning algorithms used purely on-policy methods requiring unstable two-timescale updates (e.g., $\pi_{\text{mix}}^t(y|x)$ and $\pi_\theta^{t+1}(y|x)$ in (Munos et al., 2024)). DNO addressed this with batched on-policy learning and single-timescale updates, improving sampling efficiency (Rosset et al., 2024) (App. Tables 12 and 14). Rather than directly seeking a Nash equilibrium via $\pi_\theta^t(y|x) \to \pi_\theta^\star(y|x)$, DNO views the update as regressing the policy-induced internal reward $r_{\theta,t}(x,y)$ toward a preference-based reward $r_t(x,y) = \mathbb{E}_{y'\sim\pi_t}[P(y>y' \mid x)]$ defined by pairwise preferences.

The scaled DNO algorithm proceeds as follows: (1) construct dataset $D_t = \{(x, y_{\text{gold}})\}$ where $x \sim \mathcal{D}$ and $y_{\text{gold}}$ are teacher responses sampled as $y_{\text{gold}} \sim \pi_{\text{gold}}(y|x)$; (2) sample $G$ outputs per prompt: $\{y_1^t, \ldots, y_G^t\} \sim \pi^t(y|x)$

which is the fixed policy of $\pi_\theta^t(y|x)$ after the training process; (3) rank responses $\{y_1^t, \ldots, y_G^t, y_{\text{gold}}\}$; (4) filter preference pairs and only pairs $(y_w^t, y_l^t)$ whose ranking gap exceeds a threshold are retained as $\mathcal{D}_{t+1}$. Lastly, $\pi_{t+1}^\theta$ is obtained via a single-timescale contrastive learning step (Eq. 6.4.8), where $\pi^t$ serves simultaneously as the reference policy.

$$\pi_\theta^{t+1} = \arg\max_{\pi_\theta} \mathbb{E}_{(x, y_w^t, y_l^t) \sim D_{t+1}} \log \left[ \sigma \left( \beta \log \left( \frac{\pi_\theta(y_w^t|x)}{\pi^t(y_w^t|x)} \right) - \beta \log \left( \frac{\pi_\theta(y_l^t|x)}{\pi^t(y_l^t|x)} \right) \right) \right] \tag{6.4.8}$$

The developed algorithm closely resembles iterative/online DPO, leading the authors to assert that such an approach could approximate the Nash equilibrium for general preferences.

# 7 DPO: Offline Methods

This section surveys representative offline-based algorithms, with specific focus on DPO-based variants. We organize them along the structure of the training signal including response format (pairwise, pointwise, or listwise); the form of the reward signal (pairwise, token-level, pointwise, listwise, or negative); regularization strategy; and how SFT is integrated with preference learning.

## 7.1 Response Generation

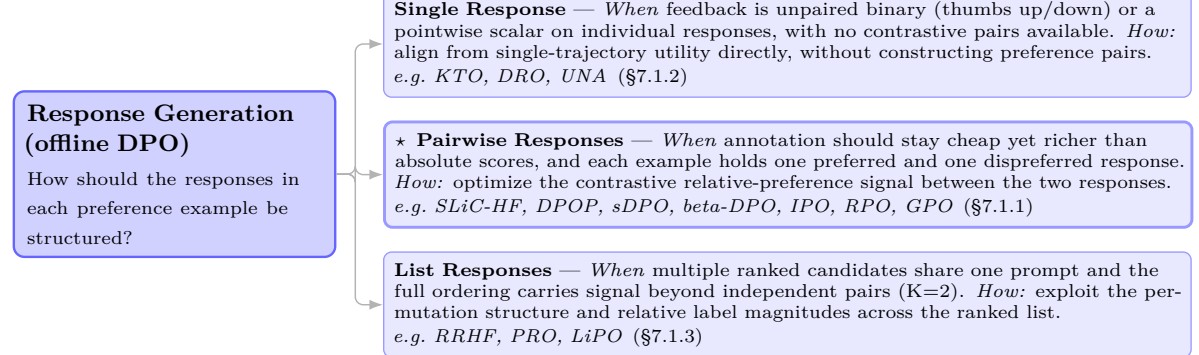

Figure 22: **Response generation (offline DPO).** How the responses underlying each offline training example are structured—a single scored response, a contrastive preferred/dispreferred pair, or a ranked list—trading annotation ease against signal richness (§7.1; ⋆ default).

A fundamental design axis in offline-based algorithms is how the response data are structured (Figure 22). Methods differ in whether they operate on pairwise comparisons between a preferred and a dispreferred response, single responses with binary or scalar feedback, or listwise rankings across multiple candidates.

### 7.1.1 Pairwise Responses

The dominant paradigm in offline preference optimization is the pairwise formulation, in which each training example consists of a prompt $x$, a preferred response $y_w$, and a dispreferred response $y_l$. This contrastive structure directly encodes relative human judgment and forms the foundation of DPO and its many variants. Pairwise data has become the standard not by accident: it is cognitively easier for annotators to rank two responses than to assign absolute scores, it yields richer and more reliable preference signal, and it arises naturally in practical settings such as A/B testing.

**Sequence Likelihood Calibration with Human Feedback (SLiC-HF)** PPO-based RLHF incurs high complexity and memory overhead through its online rollout loop, simultaneous multi-model requirements (policy, value, reward, and SFT), and costly fresh data collection for each new model. SLiC-HF (Zhao

et al., 2023) (App. Tables 12 and 14) addresses these bottlenecks by replacing the online RL loop with a simple offline calibration objective that directly reuses off-policy human preference data collected for reward models. It uses a max-margin ranking objective plus a SFT-anchoring regularizer in Eq. 7.1.1.1 to simplify the PPO-based RLHF pipeline, and its objective gradient is shown in Eq. 7.1.1.2.

$$J_{\text{SLiC-HF}}(\pi_\theta) = \mathbb{E}_{(x,y_w,y_l)\sim\mathcal{D}} \left[ -\max\left(0, \delta_m - \log\pi_\theta(y_w|x) + \log\pi_\theta(y_l|x)\right) + \lambda_{\text{sft}} \log\pi_\theta(y_{\text{sft}}|x) \right] \quad (7.1.1.1)$$

$$\begin{aligned} \nabla_\theta J_{\text{SLiC-HF}}(\pi_\theta) = \mathbb{E}_{(x,y_w,y_l)\sim\mathcal{D}}[\mathbb{I}(m>0)\left(\nabla_\theta \log\pi_\theta(y_w|x) - \nabla_\theta \log\pi_\theta(y_l|x)\right) \\ + \lambda_{\text{sft}}\nabla_\theta \log\pi_\theta(y_{\text{sft}}|x)] \end{aligned} \quad (7.1.1.2)$$

where $m = \delta_m - \log\pi_\theta(y_w|x) + \log\pi_\theta(y_l|x)$. The gradient coefficients are $\text{GC}^w_{\text{SLiC-HF}} = \mathbb{I}(m>0)$, $\text{GC}^l_{\text{SLiC-HF}} = -\mathbb{I}(m>0)$, and $\text{GC}^{\text{sft}}_{\text{SLiC-HF}} = \lambda_{\text{sft}}$. Here, $\delta_m$ acts as a margin separating desired from undesired responses, while the regularization term $\lambda_{\text{sft}} \log\pi_\theta(y_{\text{sft}} \mid x)$ encourages the learned policy to remain close to the initial SFT model.

The authors proposed two variants: SLiC-HF-direct and SLiC-HF-sample-rank. SLiC-HF-direct directly uses human preference data to define the preferred response $y_w$ and the dispreferred response $y_l$. In contrast, SLiC-HF-sample-rank first generates multiple responses from the SFT model and then employs a ranking or reward model to select $y_w$ and $y_l$. By drawing training samples from the SFT model's own output distribution, the sample-rank variant yields more stable learning and was found to converge more robustly.

**DPO-Positive (DPOP)**    Unlike DPO, which only enforces a relative preference margin between preferred and dispreferred responses, DPOP (Pal et al., 2024) (App. Tables 12 and 14) additionally penalizes any drop in the preferred response's absolute likelihood below the reference model, directly fixing the failure mode where DPO degrades preferred-response quality on near-identical response pairs. To be more specific, DPOP modifies DPO to avoid pathological updates where the likelihood of preferred outputs also decreases. This phenomenon is theoretically proven and is especially severe when response pairs have small edit distances (e.g., "2+2=4" vs. "2+2=5"). To expose this limitation, they constructed modified ARC (Clark et al., 2018), HellaSwag (Zellers et al., 2019), and Metamath (Yu et al., 2024) datasets enriched with small-edit-distance pairs, and proposed DPOP, defined in Eq. 7.1.1.3, where the added hinge term $\lambda_{\text{pos}} \max\left(0, \log\left(\frac{\pi_{\text{ref}}(y_w|x)}{\pi_\theta(y_w|x)}\right)\right)$ prevents the preferred response from becoming less likely than under the reference.

$$\begin{aligned} J_{\text{DPOP}}(\pi_\theta) = \mathbb{E}_{(x,y_w,y_l)\sim\mathcal{D}} \Bigg\{ &\log\left[\sigma\left(\beta\log\frac{\pi_\theta(y_w|x)}{\pi_{\text{ref}}(y_w|x)} - \beta\log\frac{\pi_\theta(y_l|x)}{\pi_{\text{ref}}(y_l|x)}\right)\right] \\ &-\lambda_{\text{pos}}\max\left(0, \log\frac{\pi_{\text{ref}}(y_w|x)}{\pi_\theta(y_w|x)}\right) \Bigg\} \end{aligned} \quad (7.1.1.3)$$

The gradient follows the unified form (Eq. 2.1.1.1), where the GC for positive response is $\text{GC}^w_{\text{DPOP}} = \beta\sigma\left(\beta\log\frac{\pi_\theta(y_l|x)}{\pi_{\text{ref}}(y_l|x)} - \beta\log\frac{\pi_\theta(y_w|x)}{\pi_{\text{ref}}(y_w|x)}\right) + \lambda_{\text{pos}}\mathbb{I}(\pi_{\text{ref}}(y_w|x) > \pi_\theta(y_w|x))$ and the GC for negative response is $\text{GC}^l_{\text{DPOP}} = -\beta\sigma\left(\beta\log\frac{\pi_\theta(y_l|x)}{\pi_{\text{ref}}(y_l|x)} - \beta\log\frac{\pi_\theta(y_w|x)}{\pi_{\text{ref}}(y_w|x)}\right)$. The first term in $\text{GC}^w$ matches the standard DPO coefficient, while the indicator term activates when the preferred response likelihood drops below the reference, providing an additional upward push.

**Stepwise DPO (sDPO)**    Standard DPO fixes the SFT model as the reference throughout training, but since the reference model acts as an alignment lower bound, a weakly aligned reference limits how well the target model can ultimately be optimized. sDPO extends DPO by iteratively updating the reference model to provide a progressively stronger lower bound for optimization (Kim et al., 2025) (App. Tables 12 and 14). Preference data are partitioned into stages, and DPO is applied sequentially; the partially aligned model from each stage is reused as the reference model for the next stage. This procedure improved performance over standard DPO on multiple-choice benchmarks and yielded a monotonic increase in $r_{\text{ref}}(x, y_w, y_l) = \log\frac{\pi_{\text{ref}}(y_w|x)}{\pi_{\text{ref}}(y_l|x)}$, while also reducing the initial optimization loss when initializing from the updated reference.

$\beta$**-DPO**  DPO is highly sensitive to the choice of the trade-off parameter $\beta$, particularly under varying pairwise data quality. $\beta$-DPO (Wu et al., 2024) (App. Tables 12 and 14) systematically investigates the joint influence of $\beta$ and preference data quality. The authors observe that low-gap pairs where the chosen and rejected responses exhibit small reward discrepancies typically benefit from smaller $\beta$ values, enabling more assertive policy updates. In contrast, high-gap pairs, characterized by large reward discrepancies, require larger $\beta$ values to avoid overfitting and overly aggressive updates. To address the limitations of a static $\beta$, the paper proposes batch-level dynamic $\beta_{\text{batch}}$ calibration. Specifically, $\beta_{\text{batch}}$ is adjusted based on the average reward discrepancy within each batch in Eq. 7.1.1.4.

$$\beta_{\text{batch}} = [1 + \alpha_\beta(\mathbb{E}_{i\sim\text{batch}}[M_i] - M_0)]\beta_0 \tag{7.1.1.4}$$

Here, $M_i = \beta_0 \log\left(\frac{\pi_\theta(y_w^{(i)}|x^{(i)})}{\pi_{\text{ref}}(y_w^{(i)}|x^{(i)})}\right) - \beta_0 \log\left(\frac{\pi_\theta(y_l^{(i)}|x^{(i)})}{\pi_{\text{ref}}(y_l^{(i)}|x^{(i)})}\right)$ is the individual reward discrepancy for triplet $i$, measuring the log-probability gap between the winning and losing responses under the implicit DPO reward model. The threshold $M_0$ is not fixed but is dynamically maintained as a momentum-based running mean of $M_i$ across batches, i.e., $M_0 \leftarrow mM_0 + (1-m)\mathbb{E}_{i\sim\text{batch}}[M_i]$ with momentum $m \in [0,1)$. Then, $\beta$-guided data filtering reduces the influence of outliers by assigning each triplet a sampling probability proportional to a Gaussian centered at $M_0$, so samples with reward discrepancies far from the mean are selected less frequently.

**Identity Preference Optimization (IPO)**  Two key assumptions underlying RLHF are identified: (i) pairwise preferences are substituted with pointwise rewards, and (ii) a reward model trained on such rewards is assumed to generalize to out-of-distribution samples (Azar et al., 2024) (App. Tables 12 and 14). While DPO avoids the second approximation by directly optimizing the policy from preference data, it still relies on the first through the BT formulation. As a result, DPO continues to depend on a pointwise-reward assumption and can lose effective KL regularization under deterministic preferences. The authors showed that substituting pairwise preferences with pointwise rewards can lead to instability when preferences are deterministic or nearly deterministic, i.e., $P(y_w > y_l) = 1$. In this regime, $r_\theta(x, y_w) - r_\theta(x, y_l) = \beta\left[\log\left(\frac{\pi_\theta(y_w|x)}{\pi_{\text{ref}}(y_w|x)}\right) - \log\left(\frac{\pi_\theta(y_l|x)}{\pi_{\text{ref}}(y_l|x)}\right)\right] \to +\infty$ which effectively weakens the KL regularization imposed by $\beta$ and can cause overfitting to the preference dataset. To address this issue, the authors proposed IPO, which directly optimizes preference probabilities while retaining KL regularization to a reference policy, as shown in Eq. 7.1.1.5.

$$\pi^*(y|x) = \arg\max_{\pi_\theta} \mathbb{E}_{x\sim\mathcal{D}}\left[\mathbb{E}_{y\sim\pi_\theta(y|x),\, y'\sim\pi'_\theta(y|x)}P(y > y'|x) - \beta D_{\text{KL}}(\pi_\theta(\cdot|x)\|\pi_{\text{ref}}(\cdot|x))\right] \tag{7.1.1.5}$$

In this formulation, $\pi'_\theta(y|x)$ denotes a fixed sampling or behavior policy (typically the data-collection policy) used to draw comparison responses, and it is not optimized during training. From this objective, the authors derived a squared-loss formulation that can be optimized directly from preference data, avoiding both reward modeling and reinforcement learning, as shown in Eq. 7.1.1.6. The gradient follows the unified form (Eq. 2.1.1.1) with coefficients $\text{GC}_{\text{IPO}}^w = -2\left(\log\frac{\pi_\theta(y_w|x)}{\pi_{\text{ref}}(y_w|x)} - \log\frac{\pi_\theta(y_l|x)}{\pi_{\text{ref}}(y_l|x)} - \frac{1}{2\beta}\right)$ and $\text{GC}_{\text{IPO}}^l = 2\left(\log\frac{\pi_\theta(y_w|x)}{\pi_{\text{ref}}(y_w|x)} - \log\frac{\pi_\theta(y_l|x)}{\pi_{\text{ref}}(y_l|x)} - \frac{1}{2\beta}\right)$.

$$J_{\text{IPO}}(\pi_\theta) = \mathbb{E}_{(x,y_w,y_l)\sim\mathcal{D}}\left\{-\left[\log\left(\frac{\pi_\theta(y_w|x)}{\pi_{\text{ref}}(y_w|x)}\right) - \log\left(\frac{\pi_\theta(y_l|x)}{\pi_{\text{ref}}(y_l|x)}\right) - \frac{1}{2\beta}\right]^2\right\} \tag{7.1.1.6}$$

This objective constrains the gap between the log-likelihood ratios of preferred and dispreferred responses, ensuring that the learned policy remains close to the reference model even under deterministic preferences.

**Reward-aware Preference Optimization (RPO)**  Prior work on DPO ignores the reward scores between different preference pairs. RPO (Sun et al., 2025) (App. Tables 12 and 14) was proposed to exploit this

information. The objective function is shown in Eq. 7.1.1.7 to minimize the gap between the implicit and explicit reward differences where $g(x,y) = \sigma(y) \log\left(\frac{\sigma(y)}{\sigma(x)}\right) + (1 - \sigma(y)) \log\left(\frac{1-\sigma(y)}{1-\sigma(x)}\right)$.

$$J_{\mathrm{RPO}}(\pi_\theta) = -\mathbb{E}_{(x,y_w,y_l)\sim\mathcal{D}}\, g\bigg(\beta \log\left(\frac{\pi_\theta(y_w|x)}{\pi_{\mathrm{ref}}(y_w|x)}\right) - \beta \log\left(\frac{\pi_\theta(y_l|x)}{\pi_{\mathrm{ref}}(y_l|x)}\right),$$
$$\eta(r_\phi(x,y_w) - r_\phi(x,y_l))\bigg) \tag{7.1.1.7}$$

The gradient coefficients are $\mathrm{GC}^w_{\mathrm{RPO}} = -\mathrm{GC}^l_{\mathrm{RPO}} = \beta(\sigma(c) - \sigma(z_\theta))$, where the implicit reward difference $z_\theta = \beta\left(\log\left(\frac{\pi_\theta(y_w|x)}{\pi_{\mathrm{ref}}(y_w|x)}\right) - \log\left(\frac{\pi_\theta(y_l|x)}{\pi_{\mathrm{ref}}(y_l|x)}\right)\right)$ and the explicit reward difference $c = \eta(r_\phi(x,y_w) - r_\phi(x,y_l))$.

**Generalized Preference Optimization (GPO)**    Unlike DPO, IPO, and SLiC-HF, which each fix a specific loss function independently, GPO (Tang et al., 2024) (App. Tables 12 and 14) unifies them under a single family parameterized by a convex function $f$, enabling principled comparisons of existing algorithms and a systematic study of how the choice of $f$ governs the implicit offline regularization. GPO applied a Taylor expansion around 0 to the loss, as shown in Eq. 7.1.1.8, where $r_\theta(x,y_w,y_l) = \beta \log\left(\frac{\pi_\theta(y_w|x)}{\pi_{\mathrm{ref}}(y_w|x)}\right) - \beta \log\left(\frac{\pi_\theta(y_l|x)}{\pi_{\mathrm{ref}}(y_l|x)}\right)$.

$$J_{\mathrm{GPO}}(\pi_\theta) = \mathbb{E}_{(x,y_w,y_l)\sim\mathcal{D}}[-f(r_\theta(x,y_w,y_l))] \tag{7.1.1.8}$$
$$\approx -f(0) - f'(0)\mathbb{E}_{(x,y_w,y_l)\sim\mathcal{D}}\left[r_\theta(x,y_w,y_l)\right] - \frac{f''(0)}{2}\mathbb{E}_{(x,y_w,y_l)\sim\mathcal{D}}\left[(r_\theta(x,y_w,y_l))^2\right]$$

The general gradient coefficients are $\mathrm{GC}^w_{\mathrm{GPO}} = -\beta f'(r_\theta)$ and $\mathrm{GC}^l_{\mathrm{GPO}} = \beta f'(r_\theta)$. For DPO, $f(r) = -\log\sigma(r)$ yields $f'(r) = -\sigma(-r)$, so $\mathrm{GC}^w = \beta\sigma(-r_\theta(x,y_w,y_l)))$, recovering the standard DPO gradient coefficient. $-f'(0)\mathbb{E}_{(x,y_w,y_l)\sim\mathcal{D}}[r_\theta(x,y_w,y_l)]$ was termed *preference optimization*: it focuses on maximizing the difference between desired and undesired responses, playing a role analogous to a reward signal. $-\frac{f''(0)}{2}\mathbb{E}_{(x,y_w,y_l)\sim\mathcal{D}}[(r_\theta(x,y_w,y_l))^2]$ was termed *offline regularization*: its goal lies in minimizing the difference between the current policy and the reference policy, analogous to a KL divergence penalty.

### 7.1.2   Single Response

Most RLHF/PPO methods follow a single-response paradigm: a response is sampled from the policy given a prompt, a reward is derived, advantages are computed, and the LLM is optimized via PPO. In the offline setting, binary feedback (e.g., thumbs up/down) is often more practical to collect than pairwise rankings. KTO (Ethayarajh et al., 2024) and DRO (Richemond et al., 2024) investigate how to align policies from such single-trajectory signals, bypassing the need for contrastive preference pairs. More recently, UNA (Wang et al., 2026) leverages pointwise reward signals to bridge offline preference learning and online PPO, across pairwise, binary and pointwise feedback.

**Kahneman-Tversky Optimization (KTO)**    Unlike DPO, which maximizes pairwise preference likelihood via the BT model, KTO (Ethayarajh et al., 2024) (App. Tables 12 and 14) directly optimizes LLM from cheap unpaired binary (desirable/undesirable) feedback without requiring ranked response pairs. Motivated by Kahneman and Tversky's prospect theory (Tversky & Kahneman, 1992), KTO adopts a human-aware value function that captures loss aversion. When applied to LLM alignment, the prospect-theoretic value is instantiated via a sigmoid utility over rewards $z = r_\theta(x,y) = \beta \log\left(\frac{\pi_\theta(y|x)}{\pi_{\mathrm{ref}}(y|x)}\right)$, yielding the unified form in Eq. 7.1.2.1.

$$v(z) = \begin{cases} (z - z_0)^\alpha & \text{if } z \geq z_0 \;\Rightarrow\; \lambda_D\sigma(r_\theta(x,y_w) - z_0) & \text{for } (x,y_w) \sim \mathcal{D} \\ -\lambda(z_0 - z)^\alpha & \text{if } z < z_0 \;\Rightarrow\; \lambda_U\sigma(z_0 - r_\theta(x,y_l)) & \text{for } (x,y_l) \sim \mathcal{D} \end{cases} \tag{7.1.2.1}$$

Here, $\lambda_y \in \{\lambda_D, \lambda_U\}$ refers to the weight/sensitivity for desirable and undesired examples, and $z_0$ denotes the reference point in Eq. 7.1.2.2. The KTO objective is then derived in Eq. 7.1.2.3.

$$z_0 = \beta\, \mathbb{E}_{x\sim\mathcal{D}}[D_{\mathrm{KL}}(\pi_\theta(\cdot|x)\|\pi_{\mathrm{ref}}(\cdot|x))] = \max\left(0, \frac{1}{m}\sum_{i\neq j}\beta\log\frac{\pi_\theta(y_j|x_i)}{\pi_{\mathrm{ref}}(y_j|x_i)}\right) \quad (7.1.2.2)$$

$$J_{\mathrm{KTO}}(\pi_\theta) = \mathbb{E}_{(x,y)\sim\mathcal{D}}[v_\theta(x,y) - \lambda_y] \quad (7.1.2.3)$$

Treating $z_0$ as a constant (computed as a batch statistic), the gradient follows the unified form where the gradient coefficient depends on desirability: $\mathrm{GC}^w_{\mathrm{KTO}} = \lambda_D\,\beta\,\sigma(r_\theta(x,y_w) - z_0)(1 - \sigma(r_\theta(x,y_w) - z_0))$ and $\mathrm{GC}^l_{\mathrm{KTO}} = -\lambda_U\,\beta\,\sigma(z_0 - r_\theta(x,y_l))(1 - \sigma(z_0 - r_\theta(x,y_l)))$.

**Direct Reward Optimization (DRO)**  While KTO uses prospect-theoretic utility assumptions to handle binary feedback, DRO (Richemond et al., 2024) (App. Tables 12 and 14) derives its alignment objective from first principles by directly leveraging the KL-regularized RLHF optimality condition, jointly optimizing both the policy and a learned value function without relying on any utility model. DRO reformulated the policy-reward relationship in Eq. 7.1.2.4 where $V(x) = \beta\log(Z(x))$.

$$r(x,y) - V(x) = \beta\log\left(\frac{\pi_\theta(y|x)}{\pi_{\mathrm{ref}}(y|x)}\right) \quad (7.1.2.4)$$

The resulting DRO objective minimizes mean squared error, as in Eq. 7.1.2.5.

$$J_{\mathrm{DRO}}(\pi_\theta, V_\phi) = \mathbb{E}_{(x,y)\sim\mathcal{D}}\left[-\frac{1}{2}\left(r(x,y) - V_\phi(x) - \beta\log\left(\frac{\pi_\theta(y|x)}{\pi_{\mathrm{ref}}(y|x)}\right)\right)^2\right] \quad (7.1.2.5)$$

Since estimating $V(x)$ is challenging, DRO-V approximates it with a neural network $V_\phi$, jointly optimizing policy and value networks. The policy gradient resembles standard policy gradient with value baseline plus regularization and the value update is similar to TD learning with a KL divergence term in Eq. 7.1.2.6 and Eq. 7.1.2.7 with $\mathrm{GC}_{\mathrm{DRO}} = \beta\left(r(x,y) - V_\phi - \beta\log\frac{\pi_\theta(y|x)}{\pi_{\mathrm{ref}}(y|x)}\right)$.

$$\nabla_\theta J(\pi_\theta, V_\phi) = \mathbb{E}_{(x,y)\sim\mathcal{D}}\left[\beta\left(r(x,y) - V_\phi - \beta\log\frac{\pi_\theta(y|x)}{\pi_{\mathrm{ref}}(y|x)}\right)\nabla_\theta\log\pi_\theta(y|x)\right] \quad (7.1.2.6)$$

$$\nabla_\phi J(\pi_\theta, V_\phi) = -\mathbb{E}_{(x,y)\sim\mathcal{D}}\left\{\left[V_\phi - r(x,y) + \beta\log\left(\frac{\pi_\theta(y|x)}{\pi_{\mathrm{ref}}(y|x)}\right)\right]\nabla_\phi V_\phi\right\} \quad (7.1.2.7)$$

Key implementation details include separate policy/value networks, rescaling the policy gradient by $1/\beta$, and using multiple value outputs per batch.

**UNified Alignment (UNA)**  RLHF/PPO is computationally expensive and unstable, DPO is restricted to pairwise preference data, and KTO handles only binary feedback, leaving scalar reward signals from reward models and LLMs unexploited. UNA (Wang et al., 2026) (App. Tables 13 and 15) bridges this gap by providing a unified framework that accommodates all three feedback types through a generalized implicit reward function. Starting from the same objective of RLHF and DPO, UNA proves that the optimal policy satisfies Eq. 7.1.2.8 via the log-sum inequality rather than DPO's partition-function argument

$$r_\theta(x,y) = \beta\log\frac{\pi_\theta(y|x)}{\pi_{\mathrm{ref}}(y|x)} \quad (7.1.2.8)$$

which eliminates the intractable partition function $Z(x)$ present in the DPO derivation (Eq. 2.4.4.1). With this implicit reward, UNA reframes all alignment as *minimizing the gap between implicit and explicit rewards*. While UNA supports any discrepancy measure $g$ (e.g., MSE, BCE), we present the MSE instantiation in

Eq. 7.1.2.9 where $r_\phi$ is an explicit reward from human labels, a reward model, or an LLM evaluator. The gradient follows the unified form (Eq. 2.1.1.1) with coefficient $\text{GC}_{\text{UNA}} = -2\beta \left( r_\theta(x, y) - r_\phi(x, y) \right)$.

$$J_{\text{UNA}} = -\mathbb{E}_{(x,y)\sim\mathcal{D}} \left[ r_\theta(x, y) - r_\phi(x, y) \right]^2 \tag{7.1.2.9}$$

For pairwise data, UNA is mathematically equivalent to DPO. For binary feedback (thumbs up/down treated as scores 1/0), it improves over KTO. For scalar scores from reward models or LLMs, it enables reward distillation, outperforming both DPO and KTO.

### 7.1.3 List Responses

While previous PPO/DPO studies focused on pairwise preferences or binary response, the following methods have explored direct listwise preference optimization for LLMs.

**Rank Responses to align language models with Human Feedback (RRHF)** RLHF training required multiple components, including a policy, value (or value head), reward, and reference model, leading to high memory costs. To reduce this overhead, the authors proposed RRHF, which incorporated alignment directly into SFT while achieving comparable performance (Yuan et al., 2023b) (App. Tables 13 and 15). RRHF sampled multiple responses from different models and ranked them with the training model's own length-normalized log probabilities, training the model to match rankings from reward models or human annotators, as shown in Eq. 7.1.3.1.

$$J_{\text{RRHF}}(\pi_\theta) = \mathbb{E}_{(x,y,\phi)\sim\mathcal{D}} \left[ -\sum_{\phi_i < \phi_j} \max \left( 0, \frac{\log \pi_\theta(y_i|x)}{|y_i|} - \frac{\log \pi_\theta(y_j|x)}{|y_j|} \right) + \log \pi_\theta(y_{i'}|x) \right] \tag{7.1.3.1}$$

$-\sum_{\phi_i < \phi_j} \max \left( 0, \frac{\log \pi_\theta(y_i|x)}{|y_i|} - \frac{\log \pi_\theta(y_j|x)}{|y_j|} \right)$ penalizes rank-order violations between the model's length-normalized log probabilities and human reward rankings $\phi_i(x, y_i)$, $\phi_j(x, y_j)$. $i'$ denotes the optimal response from the multiple candidates, and $\log \pi_\theta(y_{i'}|x)$ is the SFT term for instruction following. Compared to PPO, RRHF requires neither a reference model nor a value model, and can dispense with the reward model entirely when rankings are provided directly by human annotators.

**Preference Ranking Optimization (PRO)** While RRHF's hinge loss assigns a binary gradient signal to each rank violation regardless of the preference gap, PRO (Song et al., 2024) (App. Tables 13 and 15) integrates alignment directly into SFT via an InfoNCE-based objective with a dynamic temperature that scales each contrast proportionally to the reward gap, enabling gap-aware optimization over listwise preference rankings of arbitrary length. Suppose there is one prompt $x$ and $K$ responses $y_1, y_2, \ldots, y_K$, which are ranked based on the preference scores, i.e., $y_1 > y_2 > \ldots > y_K$. This can be decomposed into $K$ sub-tasks. The first sub-task takes $y_1$ as the positive sample while $y_2, \ldots, y_K$ are negative samples. In the second sub-task, $y_1$ is dropped, $y_2$ becomes the positive sample, and $y_3, \ldots, y_K$ remain negative. This process continues for $K-1$ sub-tasks. Based on these $K-1$ sub-tasks and InfoNCE, the alignment objective is formulated as shown in Eq. 7.1.3.2

$$J_{\text{align}}(\pi_\theta) = \mathbb{E}_{(x,y)\sim\mathcal{D}} \left[ \log \left( \prod_{k=1}^{K-1} \frac{\exp \left( \frac{r_\theta(x, y_k)}{T_k^k} \right)}{\sum_{i=k}^{K} \exp \left( \frac{r_\theta(x, y_i)}{T_i^k} \right)} \right) \right] \tag{7.1.3.2}$$

where $T_i^k = \frac{1}{r_\theta(x, y_k) - r_\theta(x, y_i)}$ measures the distances between two responses, and $T_k^k = \min_{i>k} T_i^k$ measures the minimum distance between the positive response $y_k$ and all negative responses $y_{k+1}, \ldots, y_K$ for the $k$-th task. Lastly, the overall PRO objective merges SFT and alignment: $J_{\text{PRO}}(\pi_\theta) = J_{\text{SFT}}(\pi_\theta) + \alpha J_{\text{align}}(\pi_\theta)$.

**Listwise Preference Optimization (LiPO)**   Pairwise methods like DPO treat every pair from a ranked list independently and discard permutation structure beyond $K{=}2$, while even existing listwise methods such as $\text{DPO}_{\text{PL}}$ and PRO optimize rank-ordering alone without exploiting actual score magnitudes. LiPO (Liu et al., 2025c) (App. Tables 13 and 15) draws inspiration from Learning-to-Rank methodologies (Liu et al., 2009) to handle listwise data directly. The authors highlight two main advantages of using listwise preferences: (1) evaluating all candidates under the same prompt systematically enhances policy learning, and (2) leveraging the relative label values between responses improves alignment. The LiPO loss function is defined in Eq. 7.1.3.3

$$J_{\text{lambda-loss}}(\pi_\theta) = \mathbb{E}_{(x,y,\phi)\sim\mathcal{D}} \left[ \sum_{\phi_i > \phi_j} \Delta_{i,j} \log\left(1 + e^{-(s_i - s_j)}\right) \right] \tag{7.1.3.3}$$

where $\Delta_{i,j} = |G_i - G_j| \left| \frac{1}{D(\tau(i))} - \frac{1}{D(\tau(j))} \right|$ is the Lambda weight, and $G_i$ is the gain of response $y_i$, defined as $G_i = 2^{r_\phi(x,y_i)} - 1$ with $\phi_i(x,y_i)$ denoting human-labelled scores. $D$ is a rank discount function with $D(\tau(s_i)) = \log(1 + \tau(s_i))$, where $\tau(s_i)$ is the rank position of $y_i$ in the ranking permutation induced by $s$. Thus, LiPO is a listwise method even though its loss can be written in terms of pairs. $s$ refers to the scores of each response as shown $s(\pi_\theta) = \{s_1, \ldots, s_K\} = \left\{ \beta \log\left(\frac{\pi_\theta(y_1|x)}{\pi_{\text{ref}}(y_1|x)}\right), \ldots, \beta \log\left(\frac{\pi_\theta(y_K|x)}{\pi_{\text{ref}}(y_K|x)}\right) \right\}$. The authors also evaluated alternative loss functions: $L_{\text{list\_mle}}$, $L_{\text{pair\_logistic}}$, $L_{\text{pair\_hinge}}$, $L_{\text{point\_mse}}$, $L_{\text{point\_sigmoid}}$, and $L_{\text{softmax}}$. Experiments yielded the ranking: $L_{\text{lambda-loss}} > (L_{\text{list\_mle}} \approx L_{\text{pair\_logistic}} \approx L_{\text{pair\_hinge}}) > L_{\text{softmax}} > L_{\text{point\_sigmoid}} > L_{\text{point\_mse}}$.

## 7.2   Reward

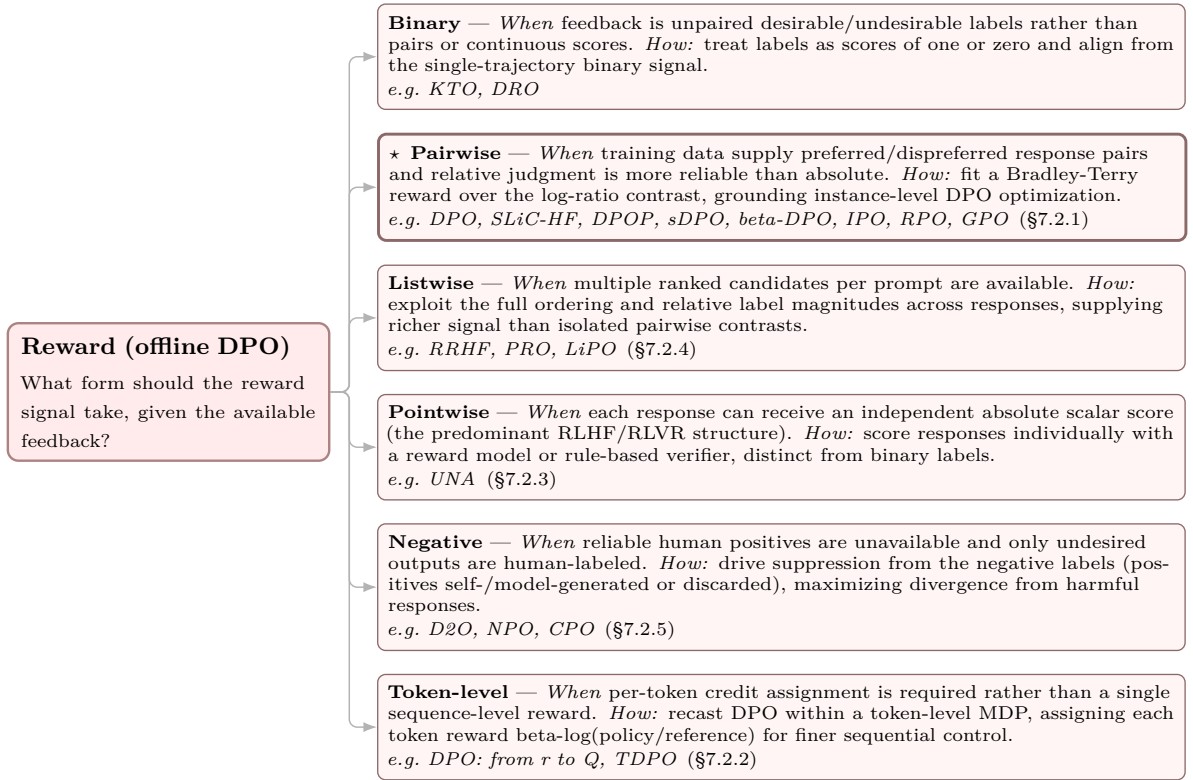

**Reward (offline DPO).**

**What form should the reward signal take, given the available feedback?**

**Binary** — *When* feedback is unpaired desirable/undesirable labels rather than pairs or continuous scores. *How:* treat labels as scores of one or zero and align from the single-trajectory binary signal.
*e.g. KTO, DRO*

★ **Pairwise** — *When* training data supply preferred/dispreferred response pairs and relative judgment is more reliable than absolute. *How:* fit a Bradley-Terry reward over the log-ratio contrast, grounding instance-level DPO optimization.
*e.g. DPO, SLiC-HF, DPOP, sDPO, beta-DPO, IPO, RPO, GPO* (§7.2.1)

**Listwise** — *When* multiple ranked candidates per prompt are available. *How:* exploit the full ordering and relative label magnitudes across responses, supplying richer signal than isolated pairwise contrasts.
*e.g. RRHF, PRO, LiPO* (§7.2.4)

**Pointwise** — *When* each response can receive an independent absolute scalar score (the predominant RLHF/RLVR structure). *How:* score responses individually with a reward model or rule-based verifier, distinct from binary labels.
*e.g. UNA* (§7.2.3)

**Negative** — *When* reliable human positives are unavailable and only undesired outputs are human-labeled. *How:* drive suppression from the negative labels (positives self-/model-generated or discarded), maximizing divergence from harmful responses.
*e.g. D2O, NPO, CPO* (§7.2.5)

**Token-level** — *When* per-token credit assignment is required rather than a single sequence-level reward. *How:* recast DPO within a token-level MDP, assigning each token reward beta-log(policy/reference) for finer sequential control.
*e.g. DPO: from r to Q, TDPO* (§7.2.2)

Figure 23: **Reward (offline DPO).** The granularity and structure of the offline DPO reward signal—binary, pairwise, listwise, pointwise, negative-only, or token-level—matched to the available feedback and credit-assignment requirements (§7.2; ★ default).

Reward design is central to preference-based alignment, as the reward structure directly shapes what behaviors a policy learns to reinforce or suppress (Figure 23). This section categorizes methods by reward granularity: pairwise rewards compare responses directly, with extensions to token-level MDPs; pointwise rewards assign absolute scalar scores and underpin most RLHF pipelines; listwise rewards rank multiple responses simultaneously; and negative rewards focus solely on suppressing undesired outputs. Figure 24 provides a concrete example contrasting different response types and their corresponding reward signals.

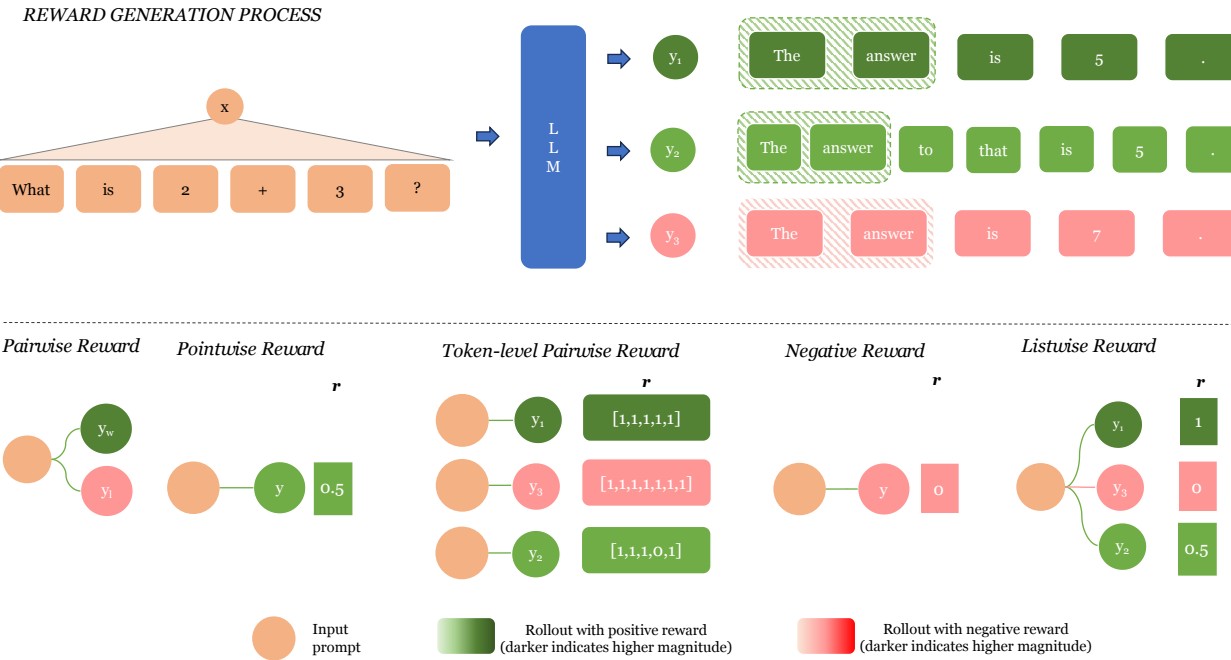

Figure 24: RLHF and DPO Response and Reward generation: Pairwise Reward, Pointwise Reward, Token-level Pairwise Reward, Negative Reward and Listwise Reward.

### 7.2.1 Pairwise Reward

The BT reward model trained from pairwise preference data underlies DPO (Rafailov et al., 2023) and its many variants surveyed in Section 7.1.1.

### 7.2.2 Token-level Pairwise Reward

Originally derived in a contextual bandit setting, DPO assigns reward to a response as a whole, leaving token-level credit assignment implicit. This line of work reinterprets DPO within a token-level MDP framework, making that credit assignment explicit and tractable.

**DPO: from r to Q**  Although DPO solves the same KL-regularized objective as RLHF, it was derived in a contextual bandit setting, i.e., treating the full response as a single arm, while classical RLHF explicitly optimizes over token-level MDPs. This mismatch leaves open whether DPO can perform token-level credit assignment or be extended to sequential multi-step tasks. In this work, it is demonstrated that DPO was capable of performing token-level credit assignment (Rafailov et al., 2024) (App. Tables 13 and 15). The token-level MDP is defined as $M = (S, A, f, r, \rho_0)$, where $S$ is the state space, $A$ is the action space, $f(s|a)$ describes the state transition given an action, $r$ is the reward function, and $\rho_0$ is the initial state distribution. The token-level MDP is formulated within the maximum entropy RL framework, as illustrated in Eq. 7.2.2.1.

$$\pi^*(y|x) = \arg\max_{\pi_\theta} E_{a_t \sim \pi_\theta(a_t|s_t)} \left\{ \sum_{t=0}^{T} [r_\theta(s_t, a_t) + \beta \log(\pi_{\text{ref}}(a_t|s_t))] + \beta H(\pi_\theta) \middle| s_0 \sim \rho(s_0) \right\} \quad (7.2.2.1)$$

Under maximum entropy RL, the relationship between the optimal Q-function $Q_\theta(s_t, a_t)$ and the optimal value function $V_\theta(s_t)$ is given in Eq. 7.2.2.2.

$$Q_\theta(s_t, a_t) - V_\theta(s_t) = \beta \log(\pi_\theta(a_t|s_t)) \tag{7.2.2.2}$$

The Bellman equation is given in Eq. 7.2.2.3.

$$Q_\theta(s_t, a_t) = r_\theta(s_t, a_t) + \beta \log(\pi_{\mathrm{ref}}(a_t|s_t)) + V_\theta(s_{t+1}) \tag{7.2.2.3}$$

Substituting $Q_\theta(s_t, a_t)$ from Eq. 7.2.2.3 into Eq. 7.2.2.2 yields $r_\theta(s_t, a_t) = V_\theta(s_t) - V_\theta(s_{t+1}) + \beta \log\left(\frac{\pi_\theta(a_t|s_t)}{\pi_{\mathrm{ref}}(a_t|s_t)}\right)$.
Summing both sides over $t$ and using $V_\theta(s_T) = 0$, the cumulative reward can be re-expressed as in Eq. 7.2.2.4.

$$\sum_{t=0}^{T-1} r_\theta(s_t, a_t) = V_\theta(s_0) + \sum_{t=0}^{T-1} \beta \log\left(\frac{\pi_\theta(a_t|s_t)}{\pi_{\mathrm{ref}}(a_t|s_t)}\right) \tag{7.2.2.4}$$

The term $V_\theta(s_0)$ cancels when substituted into the BT model, as shown in Eq. 7.2.2.5, where $y_w$ contains $N$ tokens and $y_l$ contains $M$ tokens.

$$P_\theta(y_w > y_l) = \sigma\left[\sum_{t=0}^{N-1} \beta \log\left(\frac{\pi_\theta(a_t^w|s_t^w)}{\pi_{\mathrm{ref}}(a_t^w|s_t^w)}\right) - \sum_{t=0}^{M-1} \beta \log\left(\frac{\pi_\theta(a_t^l|s_t^l)}{\pi_{\mathrm{ref}}(a_t^l|s_t^l)}\right)\right] \tag{7.2.2.5}$$

As a result, the bandit formulation was extended to a token-level MDP, where each token generation received a reward $\beta \log\left(\frac{\pi_\theta(a_t|s_t)}{\pi_{\mathrm{ref}}(a_t|s_t)}\right)$.

**Token-level DPO (TDPO)** DPO regularizes the policy at the response level, but LLM generation is inherently sequential and auto-regressive. TDPO (Zeng et al., 2024) (App. Tables 13 and 15) exploits this structure by introducing forward KL (sequential KL divergence) constraints at the token level rather than at the sentence level, enabling finer-grained diversity control. In addition, the reward discount is set to one and the token-wise reward is defined as $r_{\theta,t} = r_\theta([x, y^{<t}], y^t)$. The Q-value, value function, and advantage function are defined accordingly, with the total reward $r_\theta(x, y) = \sum_{t=1}^T r_\theta([x, y^{<t}], y^t)$. The TDPO objective is then formulated in Eq. 7.2.2.6.

$$\pi^*(y|x) = \arg\max_{\pi_\theta} \mathbb{E}_{x,y^{<t}\sim\mathcal{D}} \left[\mathbb{E}_{y^t\sim\pi_\theta(y^t|[x,y^{<t}])} A^{\pi_{\mathrm{ref}}}([x, y^{<t}], y^t)\right. \tag{7.2.2.6}$$
$$\left. - \beta D_{\mathrm{KL}}\left(\pi_\theta(\cdot|[x, y^{<t}])||\pi_{\mathrm{ref}}(\cdot|[x, y^{<t}])\right)\right]$$

From this objective, the relationship between the Q-value and the optimal policy is derived in Eq. 7.2.2.7.

$$Q^{\pi_{\mathrm{ref}}}([x, y^{<t}], y^t) = \beta \log\left(\frac{\pi_\theta(y^t|[x, y^{<t}])}{\pi_{\mathrm{ref}}(y^t|[x, y^{<t}])}\right) + \beta \log\left(Z([x, y^{<t}])\right) \tag{7.2.2.7}$$

However, since $Z([x, y_w^{<t}]) \neq Z([x, y_l^{<t}])$, the normalization terms cannot be canceled as in DPO. To resolve this, the authors introduced a sequential KL divergence defined in Eq. 7.2.2.8.

$$D_{\mathrm{SeqKL}}(x, y; \pi_1||\pi_2) = \sum_{t=1}^T D_{\mathrm{KL}}(\pi_1(\cdot|[x, y^{<t}])||\pi_2(\cdot|[x, y^{<t}])) \tag{7.2.2.8}$$

With sequential KL divergence, the normalization terms cancel under the BT model, yielding Eq. 7.2.2.9.

$$P_\theta(y_w > y_l|x) = \sigma(r_\theta(x, y_w) - r_\theta(x, y_l)) = \sigma(u_\theta(x, y_w, y_l) - \delta_\theta(x, y_w, y_l)) \tag{7.2.2.9}$$

Here, $u_\theta$ captures the log-probability ratio between the policy and reference model, while $\delta_\theta$ accounts for the difference in sequential forward KL divergence between preferred and dispreferred responses. The resulting preference probability is optimized using cross-entropy loss, and stopping the gradient on $y_w$ is further proposed to improve performance.

### 7.2.3  Pointwise Reward

Pointwise reward methods assign an absolute scalar score to each individual response rather than comparing pairs. This is the predominant reward structure in RLHF and RLVR pipelines, where a trained reward model or rule-based verifier scores each response independently. For offline policy learning, KTO, DRO and UNA utilize pointwise rewards and they can be found in Section 7.1.2. In particular, UNA utilizes pointwise reward, while KTO and DRO utilize binary rewards.

### 7.2.4  List Reward

Listwise reward methods generalize pairwise comparisons by assigning ranked scores to multiple responses simultaneously, providing a richer training signal. As with the listwise response methods described above in Section 7.1.3, these approaches can more fully exploit the relative ordering of candidate responses under the same prompt.

### 7.2.5  Negative Reward

Recent studies show that modern LLMs often surpass human performance in tasks like translation and summarization. Consequently, model-generated outputs can serve as preferred responses, while undesired outputs are leveraged for alignment to suppress harmful behavior.

**Distributional Dispreference Optimization ($D^2O$)**  Unlike DPO, which optimizes at the instance level over paired positive and negative responses, Negating Negatives proposes $D^2O$ (Duan et al., 2024) (App. Tables 13 and 15), which operates at the distribution level using *only* human-labeled negative samples, replacing noisy human positives with on-policy self-generated responses as anchors. They therefore proposed to discard human-labeled positive samples, generate new positive responses with an LLM, and train on these LLM-generated positive responses paired with human-labeled negative responses. The $D^2O$ objective is defined in Eq. 7.2.5.1 where $y_i \sim \pi_{\text{ref}}(y|x)$ are $G$ sampled responses. The reference models $\pi_{\text{ref}}^+$ and $\pi_{\text{ref}}^-$ represent more aligned models (previous iteration) and less aligned models (initial) respectively. The objective maximizes divergence from harmful responses, effectively suppressing undesirable behaviors. The gradient coefficient for the on-policy response $y_i$ is $\text{GC}_{D^2O}^w = \frac{\gamma}{G} \sigma\left(\alpha \log \frac{\pi_\theta(y_l|x)}{\pi_{\text{ref}}^+(y_l|x)} - \frac{\gamma}{G} \sum_{j=1}^G \log \frac{\pi_\theta(y_j|x)}{\pi_{\text{ref}}^-(y_j|x)}\right)$, and for the undesired response $y_l$ it is $\text{GC}_{D^2O}^l = -\alpha \, \sigma\left(\alpha \log \frac{\pi_\theta(y_l|x)}{\pi_{\text{ref}}^+(y_l|x)} - \frac{\gamma}{G} \sum_{j=1}^G \log \frac{\pi_\theta(y_j|x)}{\pi_{\text{ref}}^-(y_j|x)}\right)$.

$$J_{D^2O}(\pi_\theta) = \mathbb{E}_{(x,y_l)\sim\mathcal{D}} \left\{ \log\left[\sigma\left(\frac{\gamma}{G}\sum_{i=1}^G \log \frac{\pi_\theta(y_i|x)}{\pi_{\text{ref}}^-(y_i|x)} - \alpha \log \frac{\pi_\theta(y_l|x)}{\pi_{\text{ref}}^+(y_l|x)}\right)\right] \right\} \tag{7.2.5.1}$$

**Negative Preference Optimization (NPO)**  Gradient ascent (GA) can suppress undesired responses but often degrades overall performance (Maini et al., 2024). To mitigate this, NPO adapts DPO by retaining only the negative component (Zhang et al., 2024) (App. Tables 13 and 15), discarding the positive response $y_w$ entirely. NPO significantly slows catastrophic collapse. The NPO objective is defined in Eq. 7.2.5.2, where $y_l \sim \mathcal{D}_{\text{FG}}$ denotes the human-labeled negative (forget) sample.

$$J_{\text{NPO}}(\pi_\theta) = \mathbb{E}_{(x, y_l)\sim\mathcal{D}} \left[\frac{2}{\beta} \log \sigma\left(-\beta \log \frac{\pi_\theta(y_l|x)}{\pi_{\text{ref}}(y_l|x)}\right)\right] \tag{7.2.5.2}$$

77

The gradient coefficient for the undesired response $y_l$ is $\text{GC}_{\text{NPO}}^l = -2\,\sigma\left(\beta \log \frac{\pi_\theta(y_l|x)}{\pi_{\text{ref}}(y_l|x)}\right)$.

**Contrastive Preference Optimization (CPO)**  CPO was proposed to improve machine translation (MT) performance in moderately-sized LLMs (Xu et al., 2024b) (App. Tables 13 and 15). To construct higher-quality supervision, the authors generated translations using GPT-4 (OpenAI et al., 2024) and ALMA-13B-LoRA (Xu et al., 2024a). These outputs, together with the human gold reference $y_{\text{ref}}$, form a triplet that is scored by reference-free evaluators where the highest-scoring translation is labeled as desired ($y_w$) and the lowest-scoring as undesired ($y_l$), while the intermediate-scoring translation is discarded. The resulting dataset enabled training with the CPO objective in Eq. 7.2.5.3, where $y_w$ may originate from any of the three candidates (model-generated or human reference). The gradient coefficients are $\text{GC}_{\text{CPO}}^w = \beta\,\sigma\left(\beta \log \frac{\pi_\theta(y_l|x)}{\pi_\theta(y_w|x)}\right) + 1$ and $\text{GC}_{\text{CPO}}^l = -\beta\,\sigma\left(\beta \log \frac{\pi_\theta(y_l|x)}{\pi_\theta(y_w|x)}\right)$. In particular, the reference model terms cancel, eliminating the need for an explicit $\pi_{\text{ref}}$ by assuming a uniform prior, i.e., $\pi_{\text{ref}} \sim \mathcal{U}$. Lastly, a behavior cloning term is added to stay close to the preferred data distribution.

$$J_{\text{CPO}}(\pi_\theta) = \mathbb{E}_{(x,y_w,y_l)\sim\mathcal{D}}\{[\log\left(\sigma(\beta \log \pi_\theta(y_w|x) - \beta \log \pi_\theta(y_l|x))\right)] + [\log \pi_\theta(y_w|x)]\} \qquad (7.2.5.3)$$

## 7.3  Regularization

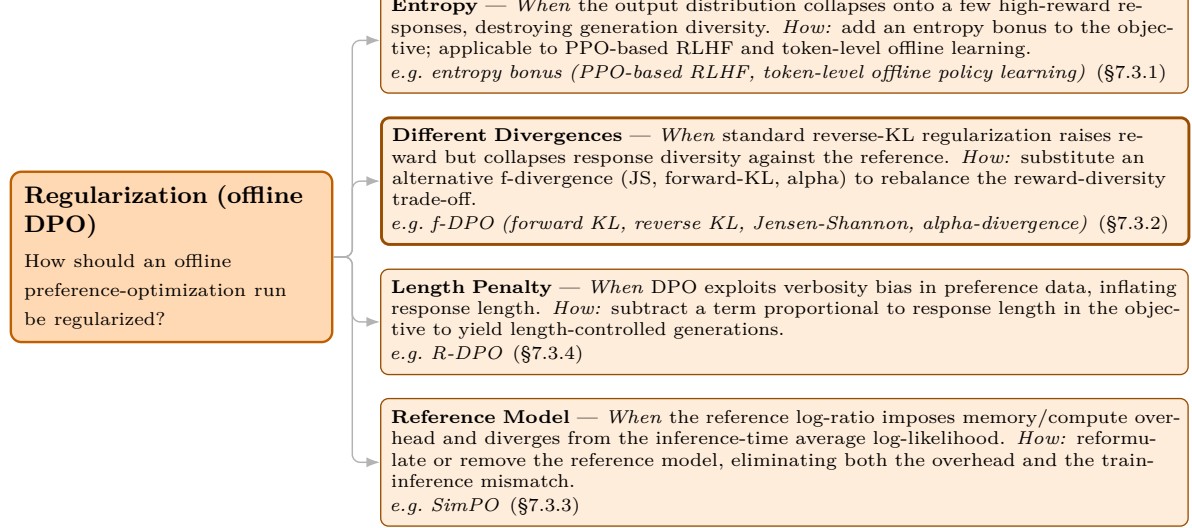

Figure 25: **Regularization (offline DPO).** How an offline preference-optimization policy is regularized—an entropy bonus, an alternative $f$-divergence rebalancing reward against diversity, a length penalty against verbosity bias, or reformulating/removing the reference model (§7.3).

Regularization is central to offline alignment (Figure 25): without constraints, policy optimization tends to overfit to the training preferences, leading to reward hacking or degenerate outputs. This section covers methods that address regularization via entropy bonuses, alternative divergence measures, adjustments to the reference model, and explicit length penalties.

### 7.3.1  Entropy

Entropy regularization encourages the policy to maintain diversity in its output distribution, preventing collapse onto a narrow set of high-reward responses. This technique is commonly employed in PPO-based RLHF and can also be incorporated into offline policy learning in Section 7.2.2.

### 7.3.2 Divergence

Rather than relying solely on the reverse KL divergence, several approaches substitute alternative divergences to strike a more favorable balance between maximizing reward and constraining the policy to remain close to a reference model.

$f$-**DPO** Previous studies used KL divergence to minimize discrepancy between the policy and pretrained model, but found that while reward increased during alignment, response diversity decreased (Wiher et al., 2022). This degradation was attributed to the KL term, prompting exploration of alternative $f$-divergences (Wang et al., 2024) (App. Tables 13 and 15) such as forward KL, reverse KL, Jensen-Shannon (JS), and $\alpha$-divergence, with the general $f$-divergence shown in Eq. 7.3.2.1.

$$D_f(p, q) = \mathbb{E}_{x \sim q(x)} \left[ f\left(\frac{p(x)}{q(x)}\right) \right] \tag{7.3.2.1}$$

In this context, $f$ represents various divergence functions. In the traditional RL framework, the reverse KL divergence, defined as $f(x) = x \log x$, is typically employed. The authors tested the $\alpha$-divergence, given by $f(x) = \frac{x^{1-\alpha} - (1-\alpha)x - \alpha}{\alpha(\alpha-1)}$, along with the forward KL divergence, $f(x) = -\log x$, and the Jensen-Shannon (JS) divergence, $f(x) = x \log x - (x + 1) \log\left(\frac{x+1}{2}\right)$. These divergences are considered within the framework of the constrained objective function as illustrated in Eq. 7.3.2.2.

$$\pi_\theta^*(y|x) = \arg\max_{\pi_\theta} \mathbb{E}_{(x,y) \sim \mathcal{D}} \left[ r_\theta(x, y) - \beta f\left(\frac{\pi_\theta(y|x)}{\pi_{\text{ref}}(y|x)}\right) \right]$$
$$\text{s.t.} \quad \sum_y \pi_\theta(y|x) = 1 \quad \forall x \tag{7.3.2.2}$$
$$\pi_\theta(y|x) \geq 0 \quad \forall x$$

Using the Lagrange method, the authors transformed the constraints into the objective function. In particular, the equality constraint $\sum_y \pi_\theta(y|x) = 1$ is enforced via the multiplier $\lambda$, while the non-negativity constraint $\pi_\theta(y|x) \geq 0$ is handled through Karush–Kuhn–Tucker (KKT) complementary slackness on the dual variables $\alpha(y) \geq 0$, yielding the transformed RL objective in Eq. 7.3.2.3.

$$J(\pi_\theta, \lambda, \alpha) = \mathbb{E}_{(x,y) \sim \mathcal{D}} \left[ r_\theta(x, y) - \beta f\left(\frac{\pi_\theta(y|x)}{\pi_{\text{ref}}(y|x)}\right) - \lambda \left( \sum_y \pi_\theta(y|x) - 1 \right) + \sum_y \alpha(y) \pi_\theta(y|x) \right] \tag{7.3.2.3}$$

Based on the new objective function, the optimal policy can be expressed as Eq. 7.3.2.4.

$$\pi_\theta(y|x) = \pi_{\text{ref}}(y|x)(f')^{-1} \left( \frac{r_\theta(y|x) - \lambda + \alpha(y)}{\beta} \right) \tag{7.3.2.4}$$

Under additional conditions, i.e., (1) $\pi_{\text{ref}}(y|x) > 0$ and (2) $f'$ being invertible with $0 \notin \text{dom}(f')$, the reward function for a specific divergence $f$ can be reformulated as Eq. 7.3.2.5.

$$r_\theta(x, y) = \beta f'\left(\frac{\pi_\theta(y|x)}{\pi_{\text{ref}}(y|x)}\right) + C \tag{7.3.2.5}$$

Integrating this reward model into the BT model enabled deriving the probability of desired over undesired responses for the objective function. Experiments revealed a reward-diversity trade-off: RKL and JSD achieved high rewards, while FKL and $\alpha$ divergence showed better entropy with lower rewards. In particular, JSD matched RKL's rewards with higher diversity, suggesting its potential for future alignment research.

### 7.3.3 Reference Model

The reference model constrains the aligned policy from deviating too far from the reference model. Some methods re-examine or eliminate this reference model dependency, either by modifying how the reference enters the objective or by removing it entirely through alternative reward formulations.

**Simple Preference Optimization (SimPO)** DPO's implicit reward relies on the log-ratio to a reference model, which both adds memory and compute overhead and is misaligned with the average log-likelihood metric used during generation. SimPO (Meng et al., 2024) (App. Tables 13 and 15) eliminates both issues by adopting a reference-free, length-normalized reward that directly aligns training with inference. Firstly, they introduce length normalization ($\frac{\alpha}{|y|} \log \pi_\theta(y|x)$) to avoid length bias. Next they introduce a reward-margin $\gamma$ to separate preferred and dis-preferred responses in Eq. 7.3.3.1. $|y_w|$ and $|y_l|$ are response lengths, $\alpha$ scales the reward difference, and the reference model can be removed. The gradient follows the unified form (Eq. 2.1.1.1) with the built-in $\frac{1}{|y|}$ normalization, i.e., $\text{GC}_{\text{SimPO}}^w = \alpha \, \sigma\left(\frac{\alpha}{|y_l|} \log \pi_\theta(y_l|x) - \frac{\alpha}{|y_w|} \log \pi_\theta(y_w|x) + \gamma\right)$ and $\text{GC}_{\text{SimPO}}^l = -\alpha \, \sigma\left(\frac{\alpha}{|y_l|} \log \pi_\theta(y_l|x) - \frac{\alpha}{|y_w|} \log \pi_\theta(y_w|x) + \gamma\right)$.

$$J_{\text{SimPO}}(\pi_\theta) = \mathbb{E}_{(x,y_w,y_l)\sim\mathcal{D}} \left( \log \left( \sigma \left( \frac{\alpha}{|y_w|} \log \pi_\theta(y_w|x) - \frac{\alpha}{|y_l|} \log \pi_\theta(y_l|x) - \gamma \right) \right) \right) \tag{7.3.3.1}$$

### 7.3.4 Length Penalty

SimPO (Section 7.3.3) has tried to solve the overlong response generation by normalizing over the response length $|y|$. Other than that, R-DPO focuses on generating length-controlled responses by subtracting the response length $|y|$.

**Regularized DPO (R-DPO)** R-DPO (Park et al., 2024) (App. Tables 13 and 15) addresses DPO's tendency to exploit preference data biases, particularly verbosity. It incorporates output length directly into the RL objective in Eq. 7.3.4.1 where the term $\alpha|y|$ penalizes response length and $\alpha$ controls its significance.

$$\pi^*(y|x) = \arg\max_{\pi_\theta} \mathbb{E}_{x\sim\mathcal{D}} \left\{ \mathbb{E}_{y\sim\pi_\theta(y|x)} \left[ r_\theta(x,y) - \alpha|y| \right] - \beta D_{\text{KL}}(\pi_\theta(\cdot|x)\|\pi_{\text{ref}}(\cdot|x)) \right\} \tag{7.3.4.1}$$

The corresponding optimal internal reward function becomes Eq. 7.3.4.2, where $Z(x) = \sum_y \pi_{\text{ref}}(y|x) \, e^{\frac{1}{\beta}(r_\theta(x,y)-\alpha|y|)}$. The resulting R-DPO objective is then Eq. 7.3.4.3.

$$r_\theta^{\text{R-DPO}}(x,y) = \beta \log \left( \frac{\pi_\theta(y|x)}{\pi_{\text{ref}}(y|x)} \right) + \beta \log Z(x) - \alpha|y| \tag{7.3.4.2}$$

$$J_{\text{R-DPO}}(\pi_\theta) = \mathbb{E}_{(x,y_w,y_l)\sim\mathcal{D}} \left( \log \left( \sigma \left( \beta \log \left( \frac{\pi_\theta(y_w|x)}{\pi_{\text{ref}}(y_w|x)} \right) - \beta \log \left( \frac{\pi_\theta(y_l|x)}{\pi_{\text{ref}}(y_l|x)} \right) - (\alpha|y_w|-\alpha|y_l|) \right) \right) \right) \tag{7.3.4.3}$$

The gradient coefficient is $\text{GC}_{\text{R-DPO}}^w = \beta \, \sigma(r_\theta(x,y_l) - r_\theta(x,y_w) + \alpha(|y_w|-|y_l|))$ for desired response and $\text{GC}_{\text{R-DPO}}^l = -\beta \, \sigma(r_\theta(x,y_l) - r_\theta(x,y_w) + \alpha(|y_w|-|y_l|))$ for undesired responses. The length penalty $\alpha(|y_w|-|y_l|)$ shifts the sigmoid argument, increasing the gradient magnitude when the preferred response is longer and reducing it when shorter.

### 7.4 Merge SFT

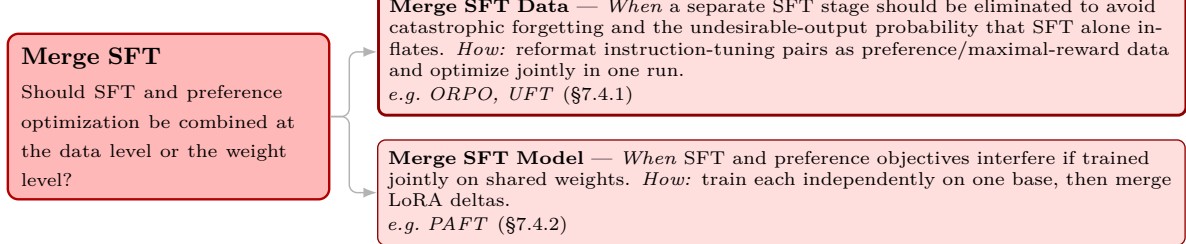

Figure 26: **Merge SFT.** At what stage SFT and preference optimization are unified to avoid catastrophic forgetting—jointly at the data/objective level, or by merging independently trained weights (§7.4).

A distinct line of research investigates how to combine SFT with preference optimization (Figure 26). Rather than running SFT and alignment as sequential stages, which can cause catastrophic forgetting, these methods either merge the two training datasets into a unified objective or combine separately trained model weights.

### 7.4.1 Merge SFT Data

One approach to unifying SFT and alignment is to reformat instruction-tuning data as preference data and optimize both jointly in a single training run. This eliminates the separate SFT stage and allows the model to simultaneously develop instruction-following ability and preference alignment.

**Odds Ratio Preference Optimization (ORPO)** The authors observed that SFT on desirable data also increased undesirable data probability, since such data were grammatically correct and similar to desired outputs. While PPO and DPO addressed this through separate alignment stages, ORPO (Hong et al., 2024) (App. Tables 13 and 15) combined these processes. The authors defined the odds ratio $OR_\theta(x, y_w, y_l)$ in Eq. 7.4.1.1, where $odds_\theta(y|x) = \frac{\pi_\theta(y|x)}{1-\pi_\theta(y|x)}$ quantifies the relative likelihood of producing $y_w$ over $y_l$.

$$OR_\theta(x, y_w, y_l) = \frac{odds_\theta(y_w|x)}{odds_\theta(y_l|x)} = \frac{\pi_\theta(y_w|x)\left(1 - \pi_\theta(y_l|x)\right)}{\pi_\theta(y_l|x)\left(1 - \pi_\theta(y_w|x)\right)} \tag{7.4.1.1}$$

The ORPO objective function is shown in Eq. 7.4.1.2 where $\lambda$ balances between SFT and $OR_\theta(x, y_w, y_l)$.

$$J_{\text{ORPO}} = \mathbb{E}_{(x,y_w,y_l)\sim\mathcal{D}}\left[\log\left(\pi_\theta(y_w|x)\right) + \lambda\log\left(\sigma\left(\log OR_\theta(x, y_w, y_l)\right)\right)\right] \tag{7.4.1.2}$$

**Unified Fine-Tuning (UFT)** UFT (Wang et al., 2025b) (App. Tables 13 and 15) takes a direct approach to merging SFT and alignment: it *combines the two stages into a single training run* by converting SFT data into alignment-compatible format using UNA's generalized implicit reward in Eq. 7.1.2.8. The key observation is that high-quality instruction-tuning pairs $(x, y)$ can be treated as score-based alignment data with maximal reward $r_\phi(x, y) = r_{\max}$. Once reformatted, instruction-tuning and alignment data are mixed and trained jointly with the UNA MSE objective in Eq. 7.4.1.3 and $\text{GC}_{\text{UFT}} = -2\beta\left(\beta\log\frac{\pi_\theta(y|x)}{\pi_{\text{ref}}(y|x)} - r_\phi(x, y)\right)$.

$$J_{\text{UFT}}(\pi_\theta) = \mathbb{E}_{(x,y)\sim\mathcal{D}_{\text{mix}}}\left[-\left(r_\phi(x, y) - \beta\log\frac{\pi_\theta(y|x)}{\pi_{\text{ref}}(y|x)}\right)^2\right] \tag{7.4.1.3}$$

Here, $\mathcal{D}_{\text{mix}}$ combines instruction-tuning data (with $r_\phi = r_{\max}$) and alignment data (with scores from human annotators, reward models, or LLMs).

### 7.4.2  Merge SFT Model

An alternative strategy is to train the SFT and alignment models separately and then merge the resulting model weights. By keeping the two objectives independent during training, this approach avoids interference and can preserve the distinct benefits of each stage.

**PArallel training for LLM Fine-Tuning (PAFT)**   Sequential SFT and alignment training often cause catastrophic forgetting of task-specific capabilities acquired during SFT: a phenomenon known as the *alignment tax* (Ouyang et al., 2022). To address this, PAFT (Pentyala et al., 2024) (App. Tables 13 and 15) performs SFT and DPO in parallel on the same pretrained model, then merges the resulting adapters. The LoRA-based $\delta$ models are defined as $\pi_\delta^{\mathrm{SFT}}(y|x) = \pi_\theta^{\mathrm{SFT}}(y|x) - \pi_\theta^{\mathrm{pre}}(y|x)$ and $\pi_\delta^{\mathrm{DPO}}(y|x) = \pi_\theta^{\mathrm{DPO}}(y|x) - \pi_\theta^{\mathrm{pre}}(y|x)$. The paper observes that DPO produces naturally sparse delta parameters while SFT does not, causing parameter interference during merging. To promote sparsity in the SFT adapter, an $\ell_1$ regularization term is added to the SFT loss in Eq. 7.4.2.1.

$$\mathcal{L}_{\mathrm{SFT_{sparse}}} = \mathcal{L}_{\mathrm{SFT}} + \lambda\|\delta_{\mathrm{sft}}\|_1 \tag{7.4.2.1}$$

The final merged model is then obtained by combining both sparse delta parameters with the pretrained model through Eq. 7.4.2.2, using merging strategies such as TIES (Yadav et al., 2023).

$$\pi_\theta^{\mathrm{merge}}(y|x) = f\left(\pi_\theta^{\mathrm{pre}}(y|x),\, \pi_\delta^{\mathrm{DPO}}(y|x),\, \pi_\delta^{\mathrm{SFT}+\ell_1}(y|x)\right) \tag{7.4.2.2}$$

## 8  Future Directions

Future directions are organized along the seven clusters that structure the training pipeline: the theoretical **foundations** of the gradient coefficient, the quality of **prompts**, the design of **responses**, the reliability of **feedback** and evaluation, the faithfulness of **reward** models, the efficiency of the learning **algorithm**, and **extensions** to new modalities, languages, and model analysis.

### 8.1  Foundations: Gradient Coefficient Theory and Convergence

Without a clear understanding of gradient coefficient behavior and convergence, practitioners tune hyperparameters by trial and error with no guarantee of correctness.

**Principled Gradient Coefficient Design**   The gradient coefficient $\mathrm{GC}(x, y, t)$ in Eq. 2.1.1.1 has five design axes that current work treats independently: (i) the **IS ratio** $\rho_t$, which controls how much weight is given to responses generated by a previous model version, from symmetric clipping (GRPO) to asymmetric stop-gradient truncation (CISPO) to full removal (GPG); (ii) the **advantage estimator** $A$, which measures how much better a response is compared to a baseline, from a learned value function via GAE (PPO) to a group mean baseline (GRPO) to entropy-weighted token filtering (Beyond 80/20); (iii) **advantage normalization**, which rescales advantage values to keep training stable, from group std (GRPO) to batch std (REINFORCE++) to none (Dr. GRPO (Liu et al., 2025d)); (iv) **response length normalization**, $\frac{1}{|y_i|}$ versus **group length normalization** $\frac{1}{\sum_i |y_i|}$ (DAPO, Magistral); and (v) **regularization**, which prevents the model from drifting too far from its starting point, from a KL penalty toward $\pi_{\mathrm{ref}}$ (Eq. 2.1.1) to an entropy bonus (Qwen3) to none (CISPO, GPG). No existing work studies how these axes interact and what combination of the five axes is optimal under a fixed compute budget.

**Convergence Guarantees**   Can convergence guarantees be established for GRPO given its group baseline that keeps shifting during training, the large LLM action space, and the fact that the reward $r(x, y)$ depends on the entire response rather than individual steps? If not, how can practitioners tell whether training has truly stalled or is only temporarily plateauing due to exploration?

**Reference Model Design and Update**   The reference model $\pi_{\mathrm{ref}}$ determines the best policy the model can reach $\pi^*(y|x) \propto \pi_{\mathrm{ref}}(y|x)\exp(r(x, y)/\beta)$ in Eq. 2.4.4.1 and sets the target for the KL penalty in Eq. 2.1.1, yet all current methods either fix it for the entire training run or update it heuristically (Wu et al., 2024).

When should $\pi_{\text{ref}}$ be updated during training, and how does the choice between a fixed and a moving reference affect the stability of the KL penalty and the quality of the final policy?

## 8.2 Prompt: Curriculum Design and Self-Play

Prompt quality determines the diversity and difficulty of the training experiences the policy encounters.

**Multi-Dimensional Curriculum Design**   Can a method that jointly schedules both domain and difficulty, i.e., treating each domain-difficulty combination as a separate option to explore, outperform single-axis difficulty curricula? Should domain coverage and difficulty be co-scheduled, or should the policy fully cover one domain before moving to the next?

**Self-Play Prompt Bias**   In single-agent self-play where proposer and solver share the same parameters, the proposer tends to generate tasks it already knows how to solve, which narrows the training distribution and introduces a bias that limits prompt diversity, since the model only trains on problems it can already handle. How can this bias be detected and corrected? For tasks that cannot be checked automatically (e.g., causal reasoning, scientific hypothesis generation), how can the validity of self-generated prompts be guaranteed without human annotation?

## 8.3 Response: Sampling, Diversity, and Length

Response-level design determines how rollouts are generated, how diverse the response distribution stays during training, and how reasoning length is controlled.

**Self-Play Response Verification and Transfer**   When the solver responds to a self-proposed task, there is no external oracle to confirm whether the reasoning chain is correct. What stopping criteria prevent the solver from being rewarded for incorrect reasoning that it produces with high confidence? Do the response strategies learned through self-play transfer to held-out real-world benchmarks, or does the self-play distribution drift away from genuine task requirements over time?

**Response Diversity and Off-Policy Reuse**   As RL training progresses, the policy tends to converge toward a narrow set of high-reward response patterns, reducing the variety of outputs it generates. How can response diversity be maintained without sacrificing reward quality? How many times can responses generated by an old model be reused across gradient steps before the IS ratio $\rho_t$ in Eq. 2.1.1.1 becomes too outdated to produce reliable gradient updates, and what threshold should trigger new rollout generation?

**Reasoning Length and Efficiency**   Can the appropriate number of tokens for a given prompt be estimated before generation, so that the model can dynamically adjust its output length based on task difficulty? How can a gradient update rule that explicitly accounts for response length optimize for both conciseness and correctness without distorting the advantage signal?

## 8.4 Feedback: Supervision Quality and Evaluation

**Feedback** is the training signal, i.e., human labels, AI scores, or a mix of both that tells the policy which responses to reinforce during training. **Evaluation** is the check performed after training that measures whether the trained policy has actually improved. Although both operate on response quality, they fail in different ways. Feedback is collected during training: annotators and AI judges tend to prefer longer or better-formatted responses regardless of content, collapsing helpfulness, factual accuracy, safety, and style into one score that the policy can learn to game without genuinely improving. Evaluation is conducted after training: benchmark scores can rise while real utility falls, because a fixed benchmark does not adapt to the new failure modes that a more capable policy introduces.

**Data Quality and Hybrid Supervision**   Pairwise comparisons and binary ratings mix helpfulness, factual accuracy, safety, and style into a single signal. In addition, verbosity and formatting alone can inflate

preference scores independent of content quality. In hybrid human-AI supervision, what criteria should govern the allocation of scarce human labels versus AI feedback, and how should the judge be designed to resist shortcuts based on surface features such as length and formatting bias? As the policy improves during training, the judge's own training distribution becomes stale; how should this gradual loss of reliability in the judge's scores be detected and corrected to keep the feedback signal reliable throughout training?

**Evaluation Standardization and Principled Stopping**   Evaluation measures whether training produced a genuinely better model, but papers report incompatible metrics like win rates, benchmark accuracies, and RLHF scores, making cross-method comparison unreliable. Iterative training also risks optimizing for the evaluation metric rather than genuine quality improvement: the policy improves on the evaluation metric without improving in deployment, because the metric does not capture all the ways a response can fail. What minimal set of evaluation dimensions like truthfulness, toxicity, over-refusal, instruction-following, calibration, jailbreak robustness is necessary and sufficient to detect the distinct failure modes of different alignment methods, and how should static benchmarks and adversarial tests be combined to cover failure modes that neither captures alone?

## 8.5   Reward: Models, Granularity, and Safety

Reward model design sets the performance ceiling of the whole pipeline.

**Verifiable Rewards Beyond Math and Code**   All current RLVR systems rely on domains where correctness can be checked automatically by an automated tool such as a compiler or symbolic verifier, which is the main bottleneck preventing RLVR from scaling beyond math and code. How can reliable verifiable reward signals be constructed for tasks such as factual question answering, scientific reasoning, and long-form writing, where no such automated checker exists?

**Reward Granularity and Process Supervision**   ORMs give a single reward for the full response; PRMs (Lightman et al., 2023) give step-level supervision; and token-level credit methods (Beyond 80/20, Clip-Cov) assign weights at individual token positions. Do step-level or token-level rewards raise the performance ceiling, or do they only speed up convergence to the same endpoint as ORMs? Across this ORM → PRM → token-level spectrum, what granularity is optimal under a fixed compute budget, and should the granularity follow a training-stage schedule, i.e., starting with coarse whole-response rewards for stable early training and gradually shifting to finer step- or token-level rewards as the policy matures, and with what criterion triggering the transition?

## 8.6   Algorithm: Optimization and Pipeline Integration

Algorithmic questions concern how the policy update is formulated and how the SFT-to-RL pipeline is connected.

**Offline-Online Convergence and Advanced Objectives**   Under what conditions does iterative offline DPO converge to the same policy as online RLVR? Do common training failure modes such as reward hacking (the model exploits loopholes in the reward), verbosity bias (the model produces unnecessarily long responses), and diversity collapse (the model stops exploring different response styles) become worse or better for Nash (Munos et al., 2024) and listwise (Liu et al., 2025c) methods as scale increases beyond 70B, relative to PPO and DPO?

**SFT-RL Pipeline Integration**   Can joint SFT-RL objectives with a gradually adjusted balance between SFT and RL losses, as partially shown by HPT (Lv et al., 2025) prevent the model from forgetting previously learned knowledge while keeping the exploration needed for effective RL?

### 8.7 Extensions: Frontiers Beyond Current Scope

The final cluster covers settings where the current pipeline does not directly apply: multi-modal and agentic tasks, continual and cross-lingual training, and understanding why certain behaviors emerge during RL.

**Multimodal and Agentic RLVR**   How should multi-turn GRPO be extended beyond single-response rollouts to handle tool calls, changing observations, and sparse rewards in complex tasks that require a long sequence of actions? What verifiable reward designs are robust to the agent learning to exploit weaknesses in the training environment?

**Continual and Cross-Lingual Alignment**   Do RLVR reasoning gains transfer across languages in a positive direction (English RL improves multilingual math performance), a negative direction (English training introduces biases into non-English output), or does this depend on model scale and pretraining language balance?

**Mechanistic Interpretability of RL Training**   During RLVR, models develop self-verification and reflection behaviors such as the "aha moment" in DeepSeek-R1-Zero, but it is not known which parts of the network produce them. Which attention heads and weight components are most responsible for these behaviors and change most during RL training, and can this knowledge be used to freeze the rest and reduce training cost?

## 9   Conclusion

This survey organizes LLM post-training from MLE through SFT, actor-critic RLHF and RLVR, and offline and iterative DPO under a single gradient coefficient framework. Every method reviewed is recovered by specifying a data source $\mathcal{D}$ including prompts and responses, a gradient coefficient $GC(x, y, t)$ in Eq. 2.1.1.1. Within RLVR, the survey organizes existing methods along three axes. The first is **prompt sampling** (Section 3), which covers both prompt generation: spanning human-annotated and synthetically generated data and prompt selection, including static curricula, adaptive difficulty curricula, and reward-based filtering. The second is **response sampling** (Section 4), which is divided into response generation and response selection. Response generation encompasses on-policy rollouts, off-policy approaches via replay buffers and knowledge distillation, asynchronous multi-node training, prefix-conditioned rollouts, and tree-structured rollouts. Response selection covers positive/negative filtering, reward-based filtering, advantage-based filtering, and bootstrapped selection. The third is **gradient coefficient design** (Section 5), which the survey breaks down into five sub-axes: the importance sampling ratio (spanning token-level and sequence-level clipping through to IS-free objectives), advantage shaping (covering outcome rewards, length penalties, self-certainty rewards, process rewards, baseline and partition function estimation, advantage stability, and hybrid SFT–RL objectives), advantage normalization (beta, multi-objective, two-step, and normalization-free variants), length normalization (response-level, group-level, and normalization-free), and regularization (KL penalty, entropy bonus, or none). For **on-policy methods with unverifiable rewards** (Section 6), the survey further discusses RLHF, RLAIF, iterative DPO, and Nash learning-based methods. For **offline DPO** (Section 7), the survey covers four axes: response generation (pairwise, single, and list responses), reward (pairwise, token-level pairwise, pointwise, list, and negative reward), regularization (entropy, divergence, reference model, and length penalty), and merging with SFT, showing how each variant differs in data structure and gradient coefficient form. Seven open problems remain along the training pipeline: 1. foundations, 2. prompt, 3. response, 4. feedback, 5. reward, 6. algorithm and 7. extensions.

## 10   Limitations

**Comparability of empirical evidence**   The surveyed papers differ in base models, training data, reward sources, compute budgets, benchmarks, and reporting protocols. Our comparisons therefore focus on algorithmic design choices rather than controlled empirical rankings. We summarize recurring patterns, such as normalization, reward granularity, online versus offline optimization, and PPO/GRPO/DPO trade-offs, while avoiding strong causal claims across heterogeneous settings.

**Scope of framework and settings**   The gradient-coefficient view is most direct for rollout-based methods; offline DPO-style objectives are included under the same notation when useful, but are organized separately around preference construction, reward parameterization, regularization, and SFT integration. Concretely, the standard DPO objective is a coupled contrastive loss over response pairs: it continuously increases the likelihood difference between the preferred response and the dispreferred one through their ratio, cancelling the intractable partition function $Z(x)$.

Beyond the framework boundary, the gradient-coefficient lens also abstracts away two practical concerns that matter in deployment: reward hacking (the framework does not model how reward models can be exploited under over-optimization, which interacts with gradient coefficient magnitude and the strength of KL regularization) and implementation cost (PPO requires four models in memory simultaneously: policy, reference, reward, and value whereas GRPO, RLOO, and offline DPO do not, a constraint invisible in the unified objective).

Lastly, we focus on single-turn, text-only post-training, leaving multimodal, agentic and multi-turn RL, continual and cross-lingual alignment, pretraining-scale RL, systems issues, and non-language applications mainly to future work.

## List of Symbols

| Symbol | Definition |
|---|---|
| **Policies and Models** | |
| $\pi_\theta$ | The policy model (i.e. the LLM being aligned), parameterized by $\theta$. |
| $\pi_{\theta_{\mathrm{old}}}$ | The old (rollout) policy: a frozen snapshot of $\pi_\theta$ used to sample training responses. Refreshed after each update step in GRPO, or after several steps in PPO. |
| $\pi_{\mathrm{ref}}$ | The frozen reference policy, typically the SFT checkpoint. Serves as the KL anchor in RLHF, DPO, and GRPO objectives. |
| $\pi_{\mathrm{sft}}$ | The SFT policy, which is the initialisation point for RL-based post-training. Responses are sampled from $\pi_{\mathrm{sft}}$ in offline methods such as RFT. |
| $\pi_b$ | Behaviour / data-collection policy that originally generated off-policy responses (e.g. in RePO, TOPR, LUFFY, GVPO). Distinct from $\pi_{\theta_{\mathrm{old}}}$: $\pi_b$ may be an entirely frozen external model rather than a stale snapshot of the current policy. |
| $\pi^*$ | The theoretically optimal policy: the closed-form maximiser of the KL-regularised reward objective (used in DPO and IPO derivations). |
| $\pi_{\mathrm{mix}}^t$ | Geometric mixture policy used in Nash-Mirror Descent (Nash-MD). |
| $\pi_\theta'$ | Fixed sampling / comparison policy used in IPO and Nash learning to draw contrast responses $y'$; not optimised during training. |
| $\pi_{\mathrm{ref}}^+,\ \pi_{\mathrm{ref}}^-$ | More-aligned (previous-iteration) and less-aligned (initial) reference models used in D$^2$O to anchor positive and negative gradients respectively. |
| $\theta$ | Trainable parameters of the policy model. |
| $\phi$ | Parameters of an auxiliary model, specifically the reward model $r_\phi(x,y)$ or the value network $V_\phi(x)$ in DRO-V. |
| **Prompts and Responses** | |
| $x$ | Input prompt / query given to the LLM. |
| $y$ | Model response / output sequence generated for prompt $x$. |
| $y^t$ | The token generated at position $t$ within response $y$; equivalently written $a_t$ in the MDP notation (where $a_t = y^t$, $s_t = (x, y^{<t})$). |

*(continued on next page)*

| Symbol | Definition |
|---|---|
| $y^{<t}$ | All tokens generated before position $t$, i.e. the prefix $(y^1, \ldots, y^{t-1})$. |
| $y^*$ | The ground-truth / correct reference response, used in the correctness indicator $\mathbb{I}(y = y^*)$. |
| $y_w$ | The preferred ("winning") response in a pairwise preference sample $(x, y_w, y_l)$. |
| $y_l$ | The dispreferred ("losing") response in a pairwise preference sample. |
| $y_{\text{sft}}$ | A demonstration response drawn from the SFT dataset or model, used as a supervised anchor in methods such as SLiC-HF that combine a ranking objective with an SFT regularisation term. |
| $y^{\text{pre}}$ | Expert prefix: a fixed prefix of length $L$ from a reference trace, prepended to on-policy rollouts to densify rewards in BREAD and Prefix-RFT. |
| $|y|$ | Length of response $y$ in tokens. |
| $G$ | Number of response rollouts sampled per prompt in GRPO-family methods. |
| $\mu$ | The mean reward across the $G$ sampled responses for a given prompt in GRPO-family methods. |
| $\sigma$ | The standard deviation of the rewards across the $G$ sampled responses for a given prompt in GRPO-family methods. |
| **Datasets and Distributions** | |
| $\mathcal{D}$ | Training data / prompt distribution; the source from which $x \sim \mathcal{D}$ is drawn during on-policy RL. |
| $\mathcal{D}_{\text{pre}}$ | Pretraining corpus; used in the MLE pretraining objective. |
| $\mathcal{D}_{\text{sft}}$ | Supervised fine-tuning dataset of $(x, y)$ demonstration pairs. |
| $\mathcal{D}_{\text{RBR}}$ | Synthetic dataset $\{(x, y_1, y_2, \ldots, y_K)\}$ of ranked completions used to fit the Rule-Based Reward (RBR) weights in RLAIF-OpenAI (Eq. 6.2.2). |
| $\mathcal{B}$ | Replay buffer storing previously generated off-policy responses together with their log-probabilities under the behaviour policy $\pi_b$; used in RePO to augment on-policy rollouts without additional generation cost. |
| **Rewards and Objectives** | |
| $r(x, y)$ | General scalar reward assigned to response $y$ given prompt $x$. |
| $r_\phi(x, y)$ | Explicit reward model with learned parameters $\phi$, trained from human preference data via the Bradley–Terry objective. |
| $r_\theta(x, y)$ | Implicit reward expressed through the policy ratio: $\beta \log \frac{\pi_\theta(y|x)}{\pi_{\text{ref}}(y|x)}$. |
| $r_t$ | Per-step reward in the MDP formulation; equals 0 for all non-terminal tokens under an Outcome Reward Model (ORM), and equals $r(x, y)$ only at the terminal step $t = T$. |
| $r_\theta^p(y^t|x, y^{<t})$ | Implicit process reward at token $t$ from the co-trained PRM in PRIME: $\beta \log \frac{\pi_\theta(y^t|x, y^{<t})}{\pi_{\text{ref}}(y^t|x, y^{<t})}$. |
| $r_\phi^o(x, y)$ | Outcome (verifiable) reward, e.g. binary correctness for math or pass-rate for code (PRIME, PURE). |
| $r_{\text{rbr}}, r_{\text{RM}}, r_{\text{tot}}$ | Rule-based reward, reward-model score, and total combined reward in RLAIF-OpenAI (Eq. 6.2.1). |
| $r_{\text{sc}}(x, y)$ | Self-certainty reward (INTUITOR): average KL divergence between the uniform vocabulary distribution and the policy's next-token distribution, measuring intrinsic model confidence without external labels. |
| $r_{\text{solve}}, r_{\text{propose}}$ | Solver and proposer rewards in Absolute Zero: $r_{\text{solve}}$ indicates task correctness while $r_{\text{propose}}$ rewards tasks of moderate difficulty (Eq. 3.1.2.1). |

*(continued from previous page)*

| Symbol | Definition |
|--------|------------|
| $J(\cdot)$ | General policy objective function to be *maximized.* |
| $L(\cdot)$ | Loss function to be *minimized,* used for reward model training. |
| $Z(x)$ | Partition function / normalization constant depending only on the prompt: $Z(x) = \sum_y \pi_{\text{ref}}(y|x)e^{r_\theta(x,y)/\beta}$. Intractable to compute directly; cancelled in the DPO derivation by taking reward *differences.* |
| $\text{GC}(x,y,t)$ | Gradient coefficient: the scalar multiplying $\nabla_\theta \log \pi_\theta(y^t|x,y^{<t})$ in the unified policy gradient (Eq. 2.1.1.1). Different post-training methods differ *only* in their choice of GC. |
| **MDP Notation** | |
| $s_t$ | MDP state at step $t$; in the LLM instantiation $s_t = (x, y^{<t})$. |
| $a_t$ | MDP action at step $t$; in the LLM instantiation $a_t = y^t$. |
| $\tau$ | Full trajectory $\tau = (s_1, a_1, s_2, a_2, \ldots, s_T, a_T)$ in general RL. |
| $T$ | Total number of tokens (episode horizon). |
| $G_t$ | Discounted return from step $t$: $G_t = \sum_{t'=t}^{T} \gamma^{t'-t} r_{t'}$. With $\gamma = 1$ and ORM, $G_t = r(x,y)$ for all $t$. |
| $G_t^{\min}$ | Min-form return in PURE: equals the minimum process reward $r_{t_w}^p$ for all steps up to and including the worst step $t_w$, and zero thereafter. |
| $\gamma$ | Discount factor ($\gamma \in [0,1]$; set to 1 for LLM text generation). |
| $S_{\text{terminal}}$ | Set of step-ending token positions in a response $y$, used by the Process Reward Model (PRM) to determine which positions receive nonzero reward (Eq. 2.3.3). |
| **Value Functions and Advantage Estimation** | |
| $V^\pi(s_t)$ | State-value function: expected return from state $s_t$ under policy $\pi$. |
| $Q^\pi(s_t, a_t)$ | Action-value function: expected return after taking action $a_t$ in state $s_t$ under policy $\pi$. |
| $A^\pi(s_t, a_t)$ | Advantage function: $Q^\pi(s_t, a_t) - V^\pi(s_t)$, measuring how much better action $a_t$ is relative to the average action in state $s_t$. |
| $V_\phi$ | Learned critic (value network) with parameters $\phi$, used in PPO and PPO-based methods (VC-PPO, ORZ, VAPO). |
| $\hat{V}_t^{\text{target}}$ | Value regression target for the critic: either the MC return $G_t$ (unbiased) or the TD target $r_t + \gamma V_{\phi^-}(s_{t+1})$ (lower variance). |
| $\hat{V}_{\text{MC}}(x, y^{<t})$ | Monte Carlo value estimate at token position $t$: the average outcome reward over $G$ continuations sampled from $\pi_\theta(\cdot|x, y^{<t})$, used in VinePPO as a drop-in replacement for the learned critic. |
| $A_{\text{MC},t}$ | MC-based token-level advantage in VinePPO: $\hat{V}_{\text{MC}}(x, y^{<t+1}) - \hat{V}_{\text{MC}}(x, y^{<t})$, exploiting the zero intermediate-reward structure of language generation. |
| $\delta_t$ | One-step TD residual: $\delta_t = r_t + \gamma V_\phi(s_{t+1}) - V_\phi(s_t)$; the building block of GAE. |
| $\lambda$ | GAE interpolation parameter: $\lambda = 0$ gives the one-step TD advantage; $\lambda = 1$ gives the full MC advantage. |
| $A_t^{\text{GAE}(\gamma,\lambda)}$ | Generalised Advantage Estimation: $\sum_{l=0}^{T-t} (\gamma\lambda)^l \delta_{t+l}$ (Eq. 2.4.1.2). |
| $A_{\text{GRPO}}(x, y_i)$ | GRPO group-normalized advantage. |
| $A_{\text{RLOO}}(x, y_i)$ | REINFORCE leave-one-out advantage. |

*(continued on next page)*

*(continued from previous page)*

| Symbol | Definition |
|---|---|
| $\mathrm{SE}(x)$ | Semantic entropy of the response group for prompt $x$ (SEED-GRPO): approximated by clustering $G$ responses into semantic equivalence classes and computing entropy over their pooled probabilities; used to scale down advantages for high-uncertainty prompts. |

**Importance Sampling and Clipping**

| Symbol | Definition |
|---|---|
| $\rho_{i,t}$ | Per-token importance sampling (IS) ratio for response $i$ at token $t$: $\rho_{i,t} = \frac{\pi_\theta(y_i^t\vert x,y_i^{<t})}{\pi_{\theta_{\mathrm{old}}}(y_i^t\vert x,y_i^{<t})}$. Corrects for distribution shift when reusing off-policy rollouts. |
| $\rho_i$ | Sequence-level IS ratio for response $i$, aggregating token-level ratios into a single per-response weight. In GSPO it is the length-normalised geometric mean $\rho_i = \left(\frac{\pi_\theta(y_i\vert x)}{\pi_{\theta_{\mathrm{old}}}(y_i\vert x)}\right)^{1/\vert y_i\vert}$; in GEPO it is the group-expectation-normalised ratio $\rho_i^{\mathrm{GEPO}}$ (Eq. 4.1.4.2). |
| $\hat{\rho}_{i,t}$ | Clipped IS ratio with stop-gradient applied (CISPO): $\hat{\rho}_{i,t} = \mathrm{clip}(\rho_{i,t}, 1 - \varepsilon_{\mathrm{low}}, 1 + \varepsilon_{\mathrm{high}})$, treated as a constant during differentiation so the gradient never zeroes out from clipping. |
| $c_t$, $c_{i,t}$ | PPO/GRPO clipping indicator: equals 0 when $\rho_t$ exceeds the trust region $[1 - \varepsilon, 1 + \varepsilon]$ in the direction favored by the advantage (zeroing the gradient), and 1 otherwise (Eq. 2.4.2.5). |
| $c_i$ | Sequence-level clipping indicator (GSPO): same binary form as $c_{i,t}$ but computed from the sequence-level ratio $\rho_i$ and advantage $A_i$, clipping the entire response at once rather than per token. |
| $\varepsilon$ | Symmetric PPO/GRPO clipping range; the IS ratio is clipped to $[1 - \varepsilon, 1 + \varepsilon]$. |
| $\varepsilon_{\mathrm{low}}$, $\varepsilon_{\mathrm{high}}$ | Asymmetric clipping bounds used by DAPO, OLMo 3, Magistral, VAPO, and CISPO ($\varepsilon_{\mathrm{high}} > \varepsilon_{\mathrm{low}}$ permits larger updates on high-advantage tokens). |
| $\varepsilon_{\mathrm{KL}}$ | Maximum allowable KL divergence per update step in TRPO's hard trust-region constraint (Eq. 2.4.2.1); approximated by clipping in PPO. |

**KL Divergence, Entropy, and Regularisation**

| Symbol | Definition |
|---|---|
| $D_{\mathrm{KL}}(\pi\Vert\pi')$ | KL divergence from distribution $\pi$ to $\pi'$. The reverse KL $D_{\mathrm{KL}}(\pi_\theta\Vert\pi_{\mathrm{ref}})$ is zero-forcing and prevents reward hacking. |
| $D_{\mathrm{SeqKL}}(x,y;\pi_1\Vert\pi_2)$ | Sequential (token-level cumulative) KL divergence used in TDPO: $\sum_{t=1}^{T} D_{\mathrm{KL}}(\pi_1(\cdot\vert x,y^{<t})\Vert\pi_2(\cdot\vert x,y^{<t}))$, enabling cancellation of per-step partition functions in the BT model (Eq. 7.2.2.8). |
| $D_{\mathrm{KL}}^{(i,t)}$ | Schulman unbiased KL estimator for KL$(\pi_\theta\Vert\pi_{\mathrm{ref}})$ at token $(i,t)$: $\frac{\pi_{\mathrm{ref}}}{\pi_\theta} - \log\frac{\pi_{\mathrm{ref}}}{\pi_\theta} - 1 \geq 0$ (used in the GRPO objective, Eq. 2.5.1.1). |
| $\beta$ | KL regularisation coefficient controlling deviation from $\pi_{\mathrm{ref}}$ (Eq. 2.1.1); also scales the implicit reward in DPO. |
| $H(\pi_\theta(\cdot\vert x,y^{<t}))$ | Shannon entropy of the policy's next-token distribution at context $(x,y^{<t})$: $-\sum_{j\in\mathcal{V}} \pi_\theta(j\vert\cdot)\log\pi_\theta(j\vert\cdot)$. |
| $\mathcal{V}$ | Vocabulary of the language model. |

**Preference Learning**

| Symbol | Definition |
|---|---|
| $\sigma(\cdot)$ | Sigmoid (logistic) function: $\sigma(z) = 1/(1+e^{-z})$, used in the Bradley–Terry pairwise preference model $P(y_w > y_l\vert x) = \sigma(r_\phi(x,y_w) - r_\phi(x,y_l))$. |
| $P(y_w > y_l\vert x)$ | Pairwise preference probability under the Bradley–Terry model: probability that response $y_w$ is preferred over $y_l$ given prompt $x$. |
| $z_0$ | KTO reference point: $\beta\,\mathbb{E}_{x\sim D}[D_{\mathrm{KL}}(\pi_\theta(\cdot\vert x)\Vert\pi_{\mathrm{ref}}(\cdot\vert x))]$ (Eq. 7.1.2.2). |

*(continued from previous page)*

| Symbol | Definition |
|---|---|
| $z_\theta$ | Implicit pairwise reward difference under the current policy (RPO, $\beta$-DPO): $z_\theta = \beta\left[\log\frac{\pi_\theta(y_w|x)}{\pi_{\text{ref}}(y_w|x)} - \log\frac{\pi_\theta(y_l|x)}{\pi_{\text{ref}}(y_l|x)}\right]$; compared against an explicit reward difference to calibrate the gradient coefficient. |
| $u_\theta$, $\delta_\theta$ | TDPO decomposition of the response-level reward difference: $u_\theta$ collects per-token log-ratios and $\delta_\theta$ captures the difference in sequential forward KL divergences between preferred and dispreferred responses; together they replace the intractable partition-function terms (Eq. 7.2.2.9). |
| $\lambda_D$, $\lambda_U$ | KTO loss weights for desirable and undesirable examples, respectively. |
| $\delta_m$ | Margin hyperparameter in SLiC-HF's max-margin ranking objective: the minimum required log-probability gap between the preferred and dispreferred responses before the ranking loss activates (Eq. 7.1.1.1). |
| $\lambda_{\text{sft}}$ | SFT regularisation weight in SLiC-HF and related methods: scales the supervised term that anchors the policy to demonstration responses, preventing drift from the initial SFT model. |
| $\lambda_{\text{pos}}$ | DPOP penalty weight: scales the hinge term that prevents the preferred response $y_w$ from becoming less likely than under $\pi_{\text{ref}}$, directly addressing the likelihood-degradation failure mode of DPO (Eq. 7.1.1.3). |
| $\tau(\cdot)$ | Rank position function: $\tau(s_i)$ is the rank of response $y_i$ in the permutation induced by scores $s$ (used in LiPO's rank discount). |
| $D(\cdot)$ | Rank discount function: $D(\tau(s_i)) = \log(1 + \tau(s_i))$ (LiPO). |
| $\Delta_{i,j}$ | Lambda weight in LiPO combining gain and rank-discount differences: $\|G_i - G_j\|\|\frac{1}{D(\tau(i))} - \frac{1}{D(\tau(j))}\|$. |
| $\phi_i(x,y)$ | (i) Binary proposition feature in RLAIF-OpenAI's Rule-Based Reward (RBR): a binary classification output for the $i$-th proposition given prompt $x$ and response $y$. (ii) Human-labelled reward / ranking score used in RRHF and LiPO to define the preference ordering over multiple candidate responses. |
| $w_i$ | Weight for feature $\phi_i$ in the RBR reward: $r_{\text{rbr}}(x,y,w) = \sum_{i=1}^{N} w_i\phi_i(x,y)$. |

**Nash and Self-Play Methods**

| Symbol | Definition |
|---|---|
| $P(\pi_\theta \succ \pi_\theta')$ | Policy-level preference probability in Nash learning: expected probability that a response from $\pi_\theta$ is preferred over one from $\pi_\theta'$, averaged over prompts $x \sim \mathcal{D}$ (Eq. 6.4.1). Replaces the scalar BT reward and supports non-transitive preferences. |
| $\alpha_t$ | Learning rate at iteration $t$ in Nash-MD; controls the interpolation weight of the regularised mixture policy $\pi_{\text{mix}}^t$ between the current iterate $\pi^t$ and the reference $\pi_{\text{ref}}$. |

**Adaptive Curriculum and Prompt Selection**

| Symbol | Definition |
|---|---|
| $d$ | Prompt difficulty score in AdaRFT: $d = 100 \times (1 - \bar{r}_{\text{ref}}(x,y))$, where $\bar{r}_{\text{ref}}$ is the average reward of a reference LLM on the prompt; higher $d$ indicates harder prompts. |
| $d_T$ | Global target difficulty in AdaRFT: updated online based on the current policy's batch-average reward to keep the training success rate near $p^*$ (Eq. 3.2.2.1). |
| $p^*$ | Target reward threshold in AdaRFT: the desired batch-average success rate (motivated as $p^* \approx 0.5$ for binary rewards to maximise gradient signal). |

**Hyperparameters**

| Symbol | Definition |
|---|---|
| $\alpha$ | A multipurpose scaling hyperparameter; its role changes per method. |

*(continued on next page)*

*(continued from previous page)*

| Symbol | Definition |
|---|---|
| $\alpha_{\text{len}}$ | Length penalty coefficient: scales the length deviation term in length-aware reward functions (LCPO, GRPO-$\lambda$, GRPO-LEAD) to trade off response correctness against verbosity. |
| $n^*$ | User-specified token budget appended to each prompt in LCPO; the length-aware reward penalises deviations from $n^*$ to enforce inference-time compute control. |
| $\kappa$ | Policy-shaping constant in LUFFY: $f(u) = \frac{u}{u+\kappa}$ (with $\kappa = 0.1$); amplifies gradients for low-probability tokens from off-policy traces to prevent entropy collapse. |
| $\lambda_{\text{ent}}$ | Entropy bonus weight in GFPO: scales the entropy regularisation term $\lambda_{\text{ent}} H(\pi_\theta)$ added to the GRPO objective to prevent premature policy collapse (Eq. 4.3.2.2). |
| $\eta_{\text{neg}}$ | Negative-sample penalty coefficient in OREAL: balances the contribution of the token-weighted negative term relative to the positive BC term (Eq. 5.1.1.8). |
| $\beta_0$ | Initial (base) KL coefficient in $\beta$-DPO; serves as the starting value before batch-level dynamic calibration adjusts it to $\beta_{\text{batch}}$. |
| $\beta_{\text{batch}}$ | Batch-level dynamic KL coefficient in $\beta$-DPO: adjusted per batch based on the average reward discrepancy $M_i$ relative to a running threshold $M_0$ (Eq. 7.1.1.4). |
| $\alpha_\beta$ | Sensitivity scaling factor in $\beta$-DPO: controls how aggressively $\beta_{\text{batch}}$ responds to deviations of the average reward discrepancy $M_i$ from the threshold $M_0$. |

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

## A   RLVR - detailed characterization of all papers

| Paper | IS | Clipping | Reward | Baseline | Adv Norm | Length Norm | Partition Fn | KL | Entropy |
|---|---|---|---|---|---|---|---|---|---|
| The Art of Scaling RL Khatri et al. (2026) (§2.7) | token | asymmetry | outcome | group mean | batch std | group length norm | no | no | no |
| Absolute Zero Zhao et al. (2025a) (§3.1.2) | token | symmetry | outcome | group mean | batch std | length norm | no | no | yes |
| Qwen 3 Yang et al. (2025) (§3.2.1) | token | symmetry | outcome | group mean | group std | length norm | no | yes | yes |
| OLMo 3 Olmo Team et al. (2025) (§3.2.1) | token | asymmetry | outcome | group mean | 1 | group length norm | no | no | no |
| Magistral Mistral-AI et al. (2025) (§3.2.1) | token | asymmetry | outcome | group mean, batch mean | batch std | group length norm | no | no | no |
| AdaRFT Shi et al. (2025) (§3.2.2) | token | symmetry | outcome | value function | 1 | length norm | no | yes | yes |
| SRPO Zhang et al. (2025e) (§3.2.3) | token | symmetry | outcome | group mean | group std | length norm | no | no | no |

Table 2: RLVR Methods: Prompt Source (7 papers). Methods innovating on training data sourcing, curriculum design, and prompt selection.

| Paper | IS | Clipping | Reward | Baseline | Adv Norm | Length Norm | Partition Fn | KL | Entropy |
|---|---|---|---|---|---|---|---|---|---|
| RePO Li et al. (2025a) (§4.1.2) | token | symmetry | outcome | group mean | group std | length norm | no | no | no |
| TOPR Roux et al. (2025) (§4.1.2) | sequence | asymmetry | outcome | no | 1 | length norm | no | no | no |
| LUFFY Yan et al. (2025) (§4.1.3) | token | symmetry | outcome | group mean | 1 | group length norm | no | no | yes |
| GVPO Zhang et al. (2025b) (§4.1.3) | no | no | outcome | group mean | 1 | length norm | cancel | yes | no |
| GEPO Zhang et al. (2025a) (§4.1.4) | group | symmetry | outcome | group mean | 1 | length norm | no | yes | no |
| BREAD Zhang et al. (2025f) (§4.2.1) | token | symmetry | outcome | group mean | group std | length norm | no | yes | no |
| Prefix-RFT Huang et al. (2025) (§4.2.1) | token | symmetry | outcome | group mean | 1 | no | no | no | no |
| TreePO Li et al. (2025b) (§4.2.2) | token | asymmetry | outcome | different sub-group means | same sub-group std | group length norm | no | no | no |
| VinePPO Kazemnejad et al. (2025) (§4.2.2) | token | symmetry | outcome | value function, batch mean | batch std | length norm | no | yes | no |
| S-GRPO Dai et al. (2025b) (§4.2.2) | token | symmetry | outcome | group mean | 1 | length norm | no | no | no |
| GFPO Shrivastava et al. (2026) (§4.3.2) | token | symmetry | outcome | group mean | group std | group length norm | no | yes | yes |
| RFT Yuan et al. (2023a) (§2.2.2) | no | no | outcome | no | 1 | length norm | no | no | no |
| PODS Xu et al. (2025) (§4.3.3) | token | symmetry | outcome | group mean | group std | length norm | no | no | no |
| CPPO Lin et al. (2025) (§4.3.3) | token | symmetry | outcome | group mean | group std | length norm | no | yes | no |
| PKPO Walder & Karkhanis (2025) (§4.3.4) | no | no | outcome | group mean | 1 | no | no | no | no |

Table 3: RLVR Methods: Response Design (15 papers). Methods innovating on response generation and selection strategies.

| Paper | IS | Clipping | Reward | Baseline | Adv Norm | Length Norm | Partition Fn | KL | Entropy |
|---|---|---|---|---|---|---|---|---|---|
| PPO for LLM Schulman et al. (2017b); Ouyang et. al. (2022) (§2.4.2) | token | symmetry | outcome | value function | 1 | length norm | no | yes | no |
| GRPO Shao et al. (2024) (§2.5.1) | token | symmetry | outcome | group mean | group std | length norm | no | yes | no |
| CISPO MiniMax-M1 Team (2025) (§5.1.1) | token | asymmetry | outcome | group mean | group std | group length norm | no | no | no |
| SAPO Gao et al. (2025) (§5.1.1) | token | asymmetry | outcome | group mean | group std | length norm | no | no | no |
| Beyond 80/20 Wang et al. (2025a) (§5.1.1) | token | asymmetry | outcome | group mean | group std | group length norm | no | no | no |
| Clip-Cov & KL-Cov Cui et al. (2025b) (§5.1.1) | token | symmetry (Clip-Cov) no (KL-Cov) | outcome | group mean | group std | length norm | no | no (Clip-Cov) yes (KL-Cov) | no |
| OREAL Lyu et al. (2025) (§5.1.1) | no | no | outcome | group mean | 1 | no | no | yes | no |
| GSPO Zheng et al. (2025) (§5.1.2) | sequence | symmetry | outcome | group mean | group std | length norm | no | no | no |
| GMPO Zhao et al. (2026) (§5.1.2) | token | symmetry | outcome | group mean | group std | length norm | no | no | no |
| GPG Chu et al. (2026) (§5.1.3) | no | no | outcome | group mean | 1 | group length norm | no | no | no |
| LCPO Aggarwal & Welleck (2025) (§5.2.2) | token | symmetry | outcome, length penalty | group mean | group std | length norm | no | yes | no |
| GRPO-λ Dai et al. (2025a) (§5.2.2) | token | symmetry | outcome, length penalty | group mean | group std | length norm | no | yes | no |
| GRPO-LEAD Zhang & Zuo (2025) (§5.2.2) | token | symmetry | outcome, length penalty | group mean | group std | length norm | no | no | no |
| Ada-GRPO Wu et al. (2025a) (§5.2.2) | token | symmetry | outcome | group mean | group std | group length norm | no | yes | no |

Table 4: RLVR Methods: Gradient Coefficient Part 1 (14 papers). Methods innovating on importance ratios and advantage computation.

| Paper | IS | Clipping | Reward | Baseline | Adv Norm | Length Norm | Partition Fn | KL | Entropy |
|---|---|---|---|---|---|---|---|---|---|
| Entropy Min Agarwal et al. (2025) (§5.2.3) | no | no | self-certainty | group mean | 1 | no | no | yes | yes |
| INTUITOR Zhao et al. (2025b) (§5.2.3) | token | symmetry | self-certainty | group mean | group std | length norm | no | yes | no |
| Seed-GRPO Chen et al. (2025) (§5.2.3) | sequence | symmetry | outcome | group mean | 1 | no | no | no | yes |
| EMPO Zhang et al. (2025d) (§5.2.3) | token | symmetry | self-certainty | group mean | group std | no | no | yes | no |
| PRIME Cui et al. (2025a) (§5.2.4) | token | symmetry | process | group mean | 1 | length norm | no | no | no |
| PURE Cheng et al. (2025b) (§5.2.4) | token | no | process | group mean | 1 | length norm | no | yes | no |
| VC-PPO Yuan et al. (2025) (§5.2.5) | token | symmetry | outcome | value function | 1 | length norm | no | no | no |
| VAPO Yue et al. (2025) (§5.2.5) | token | asymmetry | outcome | value function | 1 | group length norm | no | yes | no |
| ORZ Hu et al. (2025b) (§5.2.5) | token | symmetry | outcome | value function, batch mean | batch std | length norm | no | no | no |
| OPO Hao et al. (2025) (§5.2.5) | no | no | outcome | variance minimization | 1 | no | no | no | no |
| SPO Xu & Ding (2026) (§5.2.5) | token | symmetry | outcome | value function, batch mean | batch std | length norm | no | no | no |
| MDPO Kimi Team (2025) (§5.2.6) | no | no | outcome | group mean | 1 | no | estimated | no | no |
| FlowRL Zhu et al. (2025) (§5.2.6) | sequence | symmetry | outcome | group mean | group std | length norm | estimated | yes | no |
| GIFT Wang (2025) (§5.2.6) | no | no | outcome | group mean | group std | no | cancel | yes | no |

Table 5: RLVR Methods: Gradient Coefficient Part 2 (14 papers). Methods innovating on reward design, value baselines, and partition functions.

| Paper | IS | Clipping | Reward | Baseline | Adv Norm | Length Norm | Partition Fn | KL | Entropy |
|---|---|---|---|---|---|---|---|---|---|
| AAPO Xiong et al. (2025a) (§5.2.7) | no | no | outcome | group mean | group std | group length norm | no | no | no |
| SRFT Fu et al. (2025) (§5.2.8) | token | no | outcome | group mean | group std | length norm | no | no | yes |
| HPT Lv et al. (2025) (§2.6) | token | symmetry | outcome | group mean | group std | length norm | no | no | no |
| BNPO Xiao et al. (2025) (§5.3.1) | token | symmetry | outcome | group mean | Beta distribution | length norm | no | no | no |
| GDPO Liu et al. (2026a) (§5.3.2) | token | symmetry | outcome | group mean, batch mean | batch mean, batch std | length norm | no | yes | no |
| REINFORCE++ Hu et al. (2025a) (§5.3.3) | token | symmetry | outcome | group mean, batch mean | batch std | length norm | no | yes | no |
| DAPO Yu et al. (2025) (§5.4.2) | token | asymmetry | outcome, length penalty | group mean | group std | group length norm | no | no | no |
| Reinforce-Rej Xiong et al. (2025b) (§5.4.2) | token | asymmetry | outcome | no | 1 | length norm | no | no | no |
| Lite PPO Liu et al. (2026b) (§5.4.2) | token | symmetry | outcome | group mean | batch std | group length norm | no | no | no |
| Dr.GRPO Liu et al. (2025d) (§5.4.3) | token | symmetry | outcome | group mean | 1 | no | no | no | no |
| ProRL Liu et al. (2025a) (§5.5.1) | token | asymmetry | outcome | group mean | group std | length norm | no | yes | no |
| Entropy Explor Cheng et al. (2025a) (§5.5.2) | token | asymmetry | outcome | value function, group mean | group std | group length norm | no | no | yes |
| MAGIC He et al. (2025) (§5.5.2) | token | symmetry | outcome | group mean | group std | group length norm | no | no | yes |

Table 6: RLVR Methods: Gradient Coefficient Part 3 (13 papers). Methods innovating on advantage normalization, length scaling, and regularization.

| Paper | Improve Over (Methodology) | Base Model | Fine-tuning Data | Evaluation Benchmarks | Compare With (Methodology) |
|---|---|---|---|---|---|
| The Art of Scaling RL Khatri et al. (2026) (§2.7) | GRPO | 8B dense, Llama-4 Scout (17B×16 MoE) | Polaris-53K (math), Deepcoder (math+code runs) | AIME-24, LiveCodeBench (Jan–Jun 2025) | DeepSeek (GRPO), Qwen2.5 (DAPO), Magistral, MiniMax |
| Absolute Zero Zhao et al. (2025a) (§3.1.2) | REINFORCE++ | Qwen2.5-7B, Qwen2.5-7B-Coder, Llama-3.1-8B | Self-play (zero human-curated data) | HumanEval+, MBPP, LiveCodeBench, AIME 2024, AIME 2025, AMC, MATH-500, Minerva, OlympiadBench | GRPO, REINFORCE++, SFT |
| Qwen 3 Yang et al. (2025) (§3.2.1) | GRPO | Qwen3-235B-A22B-Base, Qwen3-32B-Base | ~4K query-verifier pairs (math, code, STEM) | AIME 2024/25, MATH-500, GPQA Diamond, ZebraLogic, LiveCodeBench v5, Codeforces | SFT |
| OLMo 3 Olmo Team et al. (2025) (§3.2.1) | GRPO | OLMo 3 Base 7B, OLMo 3 Base 32B | Dolci Think RL (math, code, IF, chat) | MATH, AIME 2024, AIME 2025, OMEGA, MMLU, GPQA, IFEval, IFBench, HumanEval+, LiveCodeBench v3, ZebraLogic, BBH | SFT, DPO, RLVR |
| Magistral Mistral-AI et al. (2025) (§3.2.1) | GRPO | Mistral Small 3 Instruct, Mistral Medium 3 Instruct | 38K curated math, 35K code problems | AIME 2024, AIME 2025, MATH-500, GPQA Diamond, LiveCodeBench (v5, v6), Aider Polyglot, Humanity's Last Exam | SFT, GRPO |
| AdaRFT Shi et al. (2025) (§3.2.2) | PPO | Qwen2.5-Math-1.5B, Qwen2.5-7B | DeepScaleR | MATH-500, GSM8K, OlympiadBench, Minerva, AMC, AIME 2024 | PPO, GRPO, REINFORCE++ |
| SRPO Zhang et al. (2025e) (§3.2.3) | GRPO | Qwen2.5-32B-Base | Curated Math + Code | AIME 2024, LiveCodeBench | GRPO |

Table 7: RLVR Methods: Prompt Source (7 papers). Methods innovating on training data sourcing, curriculum design, and prompt selection.

| Paper | Improve Over (Methodology) | Base Model | Fine-tuning Data | Evaluation Benchmarks | Compare With (Methodology) |
|---|---|---|---|---|---|
| RePO Li et al. (2025a) (§4.1.2) | GRPO | Qwen2.5-Math-1.5B/7B (-Instruct), Qwen3-1.7B | DeepMath-103K | GSM8K, MATH-500, Minerva, OlympiadBench, AIME 2024, AIME 2025, AMC, MMLU-Pro, ARC-C, GPQA Diamond, BBH, IFEval | GRPO, Dr.GRPO |
| TOPR Roux et al. (2025) (§4.1.2) | REINFORCE | Llama 3 8B Instruct, DeepSeek-R1 8B | GSM8K, MATH | GSM8K, MATH-500 | REINFORCE, Off-policy REINFORCE, Truncated IS, DPO, PPO |
| LUFFY Yan et al. (2025) (§4.1.3) | GRPO | Qwen2.5-Math-7B, Qwen2.5-Math-1.5B, Qwen2.5-7B-Instruct, LLaMA-3.1-8B | OpenR1-Math-220K | AIME 2024/25, AMC, MATH-500, Minerva, OlympiadBench, ARC-C, GPQA, MMLU-Pro | GRPO, SFT |
| GVPO Zhang et al. (2025b) (§4.1.3) | GRPO, ReMax, REINFORCE++ | Qwen2.5-Math-7B, Qwen2.5-Math-1.5B, Llama-3.1-8B-Instruct | MATH | AIME 2024, AMC, MATH-500, Minerva, OlympiadBench | GRPO, Dr.GRPO, ReMax, REINFORCE++ |
| GEPO Zhang et al. (2025a) (§4.1.4) | GRPO, GSPO | Qwen3-1.7B, Qwen3-8B | MATH (Level 3-5) | AMC, AIME 2024, AIME 2025, MATH-500 | BNPO, Dr.GRPO, GRPO, GSPO, TOPR, CISPO |
| BREAD Zhang et al. (2025f) (§4.2.1) | GRPO | Qwen2.5-1.5B-Instruct, Qwen2.5-3B-Instruct | MATH, NuminaMath-CoT | MATH, NuminaMath-CoT, GPQA | GRPO, SFT, SFT+GRPO |
| Prefix-RFT Huang et al. (2025) (§4.2.1) | SFT, GRPO | Qwen2.5-Math-7B, Qwen2.5-Math-1.5B, LLaMA-3.1-8B | OpenR1-Math-220K | AIME 2024/25, AMC, MATH-500, Minerva, OlympiadBench, ARC-C, GPQA Diamond, MMLU-Pro | SFT, GRPO, SFT+GRPO, LUFFY, ReLIFT |
| TreePO Li et al. (2025b) (§4.2.2) | GRPO, DAPO | Qwen2.5-7B, Qwen2.5-7B-Instruct, Qwen2.5-Math-7B-Instruct | MATH (Level 3-5), DeepScaleR | AIME 2024, AMC, MATH-500, Minerva, OlympiadBench | GRPO |
| VinePPO Kazemnejad et al. (2025) (§4.2.2) | PPO | DeepSeekMath 7B, RhoMath 1.1B | GSM8K, MATH | GSM8K, MATH | PPO, GRPO, RLOO, RestEM, DPO+ |
| S-GRPO Dai et al. (2025b) (§4.2.2) | GRPO | Qwen3-8B, Qwen3-14B, DeepSeek-R1-Distill-Qwen-7B, DeepSeek-R1-Distill-Qwen-14B | DeepMath-103K | GSM8K, AIME 2024, AMC, MATH-500, GPQA Diamond | GRPO, DEER, ConCISE, RL+Length Penalty, ShortBetter |
| GFPO Shrivastava et al. (2026) (§4.3.2) | GRPO | Phi-4-reasoning | 72K math problems | AIME 2024, AIME 2025, GPQA, Omni-MATH, LiveCodeBench | SFT, GRPO |
| RFT Yuan et al. (2023a) (§2.2.2) | SFT | LLaMA-7B, LLaMA-13B, LLaMA2-7B, LLaMA2-13B | GSM8K | GSM8K | SFT |
| PODS Xu et al. (2025) (§4.3.3) | GRPO | Qwen2.5-3B-Instruct, Qwen2.5-7B-Instruct, Llama3.2-3B-Instruct | GSM8K, MATH | GSM8K, MATH | GRPO, GRPO-GA |
| CPPO Lin et al. (2025) (§4.3.3) | GRPO | Qwen2.5-1.5B-Instruct, Qwen2.5-7B-Instruct | GSM8K, MATH | GSM8K, MATH, AMC, AIME 2024 | GRPO, DAPO, Dr.GRPO |
| PKPO Walder & Karkhanis (2025) (§4.3.4) | REINFORCE, RLOO | Gemma 2 2B, Gemma 2 9B, Llama 3.1 8B | MATH, MBPP, ARC-AGI-1 | MATH, HumanEval+, ARC-AGI-1 | RLOO |

Table 8: RLVR Methods: Response Design (15 papers). Methods innovating on response generation and selection strategies.

| Paper | Improve Over (Methodology) | Base Model | Fine-tuning Data | Evaluation Benchmarks | Compare With (Methodology) |
|---|---|---|---|---|---|
| PPO for LLM Schulman et al. (2017b); Ouyang et al. (2022) (§2.4.2) | SFT | GPT-3 (1.3B/6B/175B) | Labeler demonstrations + human-labeled comparisons | TruthfulQA, RealToxicityPrompts, Winogender, CrowS-Pairs, API human eval | SFT |
| GRPO Shao et al. (2024) (§2.5.1) | PPO | DeepSeekMath-7B-Instruct | GSM8K, MATH | GSM8K, MATH, MGSM-zh, CMATH | PPO, DPO, RFT, SFT |
| CISPO MiniMax-M1 Team (2025) (§5.1.1) | GRPO, DAPO | MiniMax-Text-01 (456B) | 50K math, SynLogic (53K logic), 30K code, SWE-bench-derived SE, 25K general domain | MATH-500, AIME 2024, AIME 2025, LiveCodeBench, FullStackBench, GPQA Diamond, HLE, ZebraLogic, MMLU-Pro, SWE-bench Verified, OpenAI-MRCR, LongBench-v2, TAU-bench, SimpleQA, MultiChallenge | GRPO, DAPO |
| SAPO Gao et al. (2025) (§5.1.1) | GSPO, GRPO | Qwen3-30B-A3B-Base, Qwen3-VL-30B-A3B | Math, Code, Logic | AIME 2025, HMMT25, BeyondAIME, LiveCodeBench, ZebraLogic, MathVision | GSPO, GRPO |
| Beyond 80/20 Wang et al. (2025a) (§5.1.1) | DAPO | Qwen3-8B, Qwen3-14B, Qwen3-32B | DAPO-Math-17K | AIME 2024, AIME 2025, AMC, MATH-500, Minerva, OlympiadBench | DAPO |
| Clip-Cov & KL-Cov Cui et al. (2025b) (§5.1.1) | GRPO | Qwen2.5-7B, Qwen2.5-32B | DAPO-Math-17K | AIME 2024/25, AMC, MATH-500, Omni-MATH, OlympiadBench, Minerva | GRPO, Clip-higher |
| OREAL Lyu et al. (2025) (§5.1.1) | REINFORCE | Qwen2.5-7B, Qwen2.5-32B, DeepSeek-R1-Distill-Qwen-7B | NuminaMath, MATH, AMC/AIME competitions | MATH-500, AIME 2024, AIME 2025, LiveMathBench, OlympiadBench | REINFORCE, GRPO, PRIME |
| GSPO Zheng et al. (2025) (§5.1.2) | GRPO | Qwen3-30B-A3B-Base | Math + Code | AIME 2024, LiveCodeBench, Codeforces | GRPO |
| GMPO Zhao et al. (2026) (§5.1.2) | GRPO | Qwen2.5-Math-1.5B, Qwen2.5-Math-7B, DeepSeek-R1-Distill-Qwen-7B, Qwen3-32B, Qwen2.5-VL-Instruct-7B | MATH, DeepScaleR, CountDown, Geometry3K | AIME 2024, AMC, MATH-500, Minerva, OlympiadBench, Geometry3K | GRPO, Dr.GRPO |
| GPG Chu et al. (2026) (§5.1.3) | GRPO | DeepSeek-R1-Distill-Qwen-1.5B, Qwen2.5-Math-7B, Qwen2.5-VL-3B-Instruct, Qwen2-VL-2B | open-s1, open-rs, MATH-lighteval, DAPO-Math-17K, SAT dataset, GEOQA training set, Flower102, Pets37, FGVC-Aircraft, Cars196, LISA training set | AIME 2024, MATH-500, AMC, Minerva, OlympiadBench, CV-Bench, GEOQA, Flower102, Pets37, FGVC-Aircraft, Cars196, LISA | GRPO, Dr.GRPO, DAPO, SFT |
| LCPO Aggarwal & Welleck (2025) (§5.2.2) | GRPO | DeepScaleR-1.5B-Preview | DeepScaleR-Preview-Dataset | AIME 2025, MATH, AMC, OlympiadBench, GPQA, LSAT, MMLU | S1 (budget forcing) |
| GRPO-λ Dai et al. (2025a) (§5.2.2) | GRPO | Qwen3-8B | DeepMath-103K | AIME 2024, AMC, MATH-500, Minerva, OlympiadBench | GRPO |
| GRPO-LEAD Zhang & Zuo (2025) (§5.2.2) | GRPO | DeepSeek-R1-Distill-Qwen-7B, DeepSeek-R1-Distill-Qwen-14B | DeepScaleR | AIME 2024, AIME 2025, LiveCodeBench | GRPO |
| Ada-GRPO Wu et al. (2025a) (§5.2.2) | GRPO | Qwen2.5-Base-3B/7B/14B | AQuA-Rat (SFT), CSQA, GSM8K, MATH (RL) | CSQA, OBQA, SVAMP, GSM8K, MATH, AIME 2025, BBH | SFT, GRPO |

Table 9: RLVR Methods: Gradient Coefficient Part 1 (14 papers). Methods innovating on importance ratios and advantage computation.

| Paper | Improve Over (Methodology) | Base Model | Fine-tuning Data | Evaluation Benchmarks | Compare With (Methodology) |
|---|---|---|---|---|---|
| Entropy Min Agarwal et al. (2025) (§5.2.3) | REINFORCE | Qwen2.5-Math-7B, Eurus-2-7B-SFT, Llama-3.1-8B-Instruct | NuminaMath + Eurus-2 coding (unlabeled prompts, no answer labels) | MATH-500, AMC, AIME 2024, Minerva, OlympiadBench, LeetCode, LiveCodeBench v2, SciCode, UGPhysics | SFT, RLOO, GRPO, SC-RL |
| INTUITOR Zhao et al. (2025b) (§5.2.3) | GRPO | Qwen2.5-1.5B, Qwen2.5-3B, Qwen2.5-7B, Qwen2.5-14B, Llama3.2-3B-Instruct, OLMo-2-1124-7B-SFT, Qwen3-14B | MATH, Codeforces | MATH-500, GSM8K, LiveCodeBench, CRUXEval-O, MMLU-Pro, AlpacaEval | GRPO, GRPO-PV |
| Seed-GRPO Chen et al. (2025) (§5.2.3) | GRPO | Qwen2.5-Math-1.5B, Qwen2.5-Math-7B, DeepSeek-R1-Distill-Qwen-7B | MATH (Level 3-5) | AIME 2024, AMC, MATH-500, Minerva, OlympiadBench | Dr.GRPO, GRPO, DAPO, GPG, SRPO, RAFT++ |
| EMPO Zhang et al. (2025d) (§5.2.3) | GRPO | Qwen2.5-Math-1.5B-Base, Qwen2.5-Math-7B-Base, Qwen2.5-3B-Instruct, Qwen2.5-7B-Instruct | NuminaMath-CoT (20K math), TriviaQA (10K), TruthfulQA (500) | Minerva Math, MATH, AMC23, OlympiadBench, AIME 2024, TriviaQA, TruthfulQA | GRPO, ODPO, SFT |
| PRIME Cui et al. (2025a) (§5.2.4) | RLOO | Qwen2.5-Math-7B-Base | MathInstruct, NuminaMath, Code-Feedback | AIME 2024, AMC, MATH-500, Minerva, OlympiadBench, LeetCode, LiveCodeBench | RLOO, REINFORCE, GRPO, PPO, SFT, VinePPO |
| PURE Cheng et al. (2025b) (§5.2.4) | RLOO | Qwen2.5-7B, Qwen2.5-Math-7B, Qwen2.5-Math-1.5B | MATH | AIME 2024, AMC, MATH-500, Minerva, OlympiadBench | RLOO, GRPO, REINFORCE++, DPO |
| VC-PPO Yuan et al. (2025) (§5.2.5) | PPO | Qwen2.5-32B-Base | Past AIME problems + synthetic hard math | AIME 2024, GPQA, Codeforces | PPO, GRPO |
| VAPO Yue et al. (2025) (§5.2.5) | PPO | Qwen2.5-32B-Base | DAPO-Math-17K | AIME 2024 | PPO, DAPO, GRPO |
| ORZ Hu et al. (2025b) (§5.2.5) | GRPO | Qwen2.5-0.5B-Base, Qwen2.5-1.5B-Base, Qwen2.5-7B-Base, Qwen2.5-32B-Base | AIME, MATH, NuminaMath, Tulu3 MATH, OpenR1-Math-220K, AoPS forum + synthesized tasks | AIME 2024, AIME 2025, MATH-500, GPQA Diamond, MMLU, MMLU-Pro | GRPO, DAPO |
| OPO Hao et al. (2025) (§5.2.5) | GRPO | DeepSeek-R1-Distill-Qwen-7B | Skywork-OR1-RL-Data | MATH-500, AIME 2024, AIME 2025 | GRPO, SFT |
| SPO Xu & Ding (2026) (§5.2.5) | GRPO | Qwen3-8B | DAPO (English subset) | AIME 2024, AIME 2025, BeyondAIME, BRUNO25, HMMT25 | GRPO |
| MDPO Kimi Team (2025) (§5.2.6) | REINFORCE | Kimi k1.5 Base (proprietary MM) | Proprietary RL prompts (math, code, vision) | AIME 2024, MATH-500, HumanEval-Mul, LiveCodeBench, Codeforces, MathVista, MMMU, MathVision, MMLU, IFEval, CLUEWSC, C-Eval | ReST, DPO |
| FlowRL Zhu et al. (2025) (§5.2.6) | REINFORCE++, PPO, GRPO | Qwen2.5-7B-Base, Qwen2.5-32B-Base, DeepSeek-R1-Distill-Qwen-7B | DAPO-Math-17K, DeepCoder | MATH-500, AMC, AIME 2024/25, Minerva, OlympiadBench, LiveCodeBench, Codeforces, HumanEval+ | REINFORCE++, PPO, GRPO |
| GIFT Wang (2025) (§5.2.6) | GRPO | DeepSeek-LLM-7B-Chat, Qwen2.5-7B-Instruct, Qwen2.5-32B-Base, Qwen3-32B-Base | GSM8K, MATH, DAPO-Math-17K, Infinity | GSM8K, MATH, AIME, TruthfulQA, BBQ, MBPP, ARC-C, Winogender, GPQA, MUSR, AlpacaEval, Arena-Hard | GRPO, DPO, UNA, PPO |

Table 10: RLVR Methods: Gradient Coefficient Part 2 (14 papers). Methods innovating on reward design, value baselines, and partition functions.

| Paper | Improve Over (Methodology) | Base Model | Fine-tuning Data | Evaluation Benchmarks | Compare With (Methodology) |
|---|---|---|---|---|---|
| AAPO Xiong et al. (2025a) (§5.2.7) | GRPO, GPG | DeepSeek-R1-Distill-Qwen-1.5B, Qwen2.5-Math-7B, Llama-3.2-1B-Instruct, Llama-3.2-3B-Instruct | open-rs, simplelr_qwen_level3to5 | AIME 2024, MATH-500, AMC, Minerva, OlympiadBench | GRPO, GPG, PRIME |
| SRFT Fu et al. (2025) (§5.2.8) | SFT, GRPO | Qwen2.5-Math-7B | OpenR1-Math-46k-8192 | AIME 2024, AMC, MATH-500, Minerva, OlympiadBench, ARC-C, GPQA Diamond, MMLU-Pro | SFT, GRPO, PPO, LUFFY |
| HPT Lv et al. (2025) (§2.6) | SFT, GRPO | Qwen2.5-Math-7B, Qwen2.5-Math-1.5B, LLaMA-3.1-8B | Math problems with solution trajectories | AIME 2024, AIME 2025, AMC, MATH-500, Minerva, OlympiadBench, GPQA Diamond, ARC-C | SFT, GRPO, LUFFY, SRFT, SFT→GRPO |
| BNPO Xiao et al. (2025) (§5.3.1) | GRPO, REINFORCE, ReMax, REINFORCE++ | Qwen2.5-Math-1.5B, Qwen2.5-Math-7B | MATH | AIME 2024, AIME 2025, AMC, MATH-500 | GRPO, ReMax, REINFORCE++, REINFORCE |
| GDPO Liu et al. (2026a) (§5.3.2) | GRPO | Qwen2.5-1.5B-Instruct, Qwen2.5-3B-Instruct, DeepSeek-R1-Distill-Qwen-1.5B, DeepSeek-R1-Distill-Qwen-7B, Qwen3-4B-Instruct | DeepScaleR-Preview-Dataset (math), ToolACE, Hammar, xLAM (tool calling), Eurus-2-RL (coding) | AIME 2024, AMC, MATH, Minerva, OlympiadBench, BFCL-v3, APPS, CodeContests, Codeforces, TACO | GRPO |
| REINFORCE++ Hu et al. (2025a) (§5.3.3) | REINFORCE | Llama-3-8B-SFT, Qwen2.5-Math-7B-Base, Qwen2.5-7B-Base | 20K diverse prompts (RLHF), ORZ, DAPO, MATH | AIME 2024, AIME 2025, MATH-500, AMC, Chat-Arena-Hard, K&K, HMMT, CMIMC | PPO, GRPO, RLOO, ReMax |
| DAPO Yu et al. (2025) (§5.4.2) | GRPO | Qwen2.5-32B-Base | DAPO-Math-17K | AIME 2024 | GRPO |
| Reinforce-Rej Xiong et al. (2025b) (§5.4.2) | REINFORCE | Qwen2.5-Math-7B-Base, LLaMA-3.2-3B-Instruct | NuminaMath | MATH-500, Minerva Math, OlympiadBench | RAFT, RAFT++, Iterative DPO, REINFORCE, GRPO, PPO |
| Lite PPO Liu et al. (2026b) (§5.4.2) | GRPO, DAPO | Qwen3-4B-Base, Qwen3-8B-Base | SimpleRL-Zoo-Data, DeepMath-103K | MATH-500, OlympiadBench, AMC, Minerva, AIME 2024, AIME 2025 | GRPO, DAPO |
| Dr. GRPO Liu et al. (2025d) (§5.4.3) | GRPO | Qwen2.5-Math-7B, Qwen2.5-Math-1.5B, Llama-3.2-3B | MATH | AIME 2024, AMC, MATH-500, Minerva, OlympiadBench | GRPO, PRIME |
| ProRL Liu et al. (2025a) (§5.5.1) | GRPO, DAPO | DeepSeek-R1-Distill-Qwen-1.5B | DeepScaleR, Eurus-2-RL, SCP-116K, Reasoning Gym, Llama-Nemotron | AIME 2024/25, AMC, MATH, Minerva, OlympiadBench, APPS, CodeContests, Codeforces, TACO, LiveCodeBench, HumanEval+, GPQA Diamond, IFEval, Reasoning Gym | GRPO |
| Entropy Explor Cheng et al. (2025a) (§5.5.2) | PPO, GRPO | Qwen2.5-Base-7B, Qwen2.5-Math-Base-7B | DAPO-Math-17K | AIME 2024/25, AMC, MATH-500 | PPO, GRPO, PRIME, GPG |
| MAGIC He et al. (2025) (§5.5.2) | GRPO | DeepSeek-R1-Distill-Qwen-7B, DeepSeek-R1-Distill-Qwen-32B | Skywork-OR1-RL-Data | AIME 2024, AIME 2025, LiveCodeBench | GRPO |

Table 11: RLVR Methods: Gradient Coefficient Part 3 (13 papers). Methods innovating on advantage normalization, length scaling, and regularization.

# B   RLHF - detailed characterization of all papers

| Paper | Response | Feedback Type | Reward | Training Paradigm | Entropy Reg. | Divergence Reg. | Length Penalty | Ref. Model | Merge w/ SFT |
|---|---|---|---|---|---|---|---|---|---|
| RLAIF-Anthropic (Bai et al., 2022b) (§6.2) | pairwise | HF/AF | pairwise | on-policy | no | RKL | no | yes | no |
| RLAIF-Google (Lee et al., 2024) (§6.2) | pairwise | AF | pairwise | offline | no | RKL | no | yes | no |
| RLAIF-OpenAI (Mu et al., 2024) (§6.2) | list | HF/AF | pointwise | on-policy | no | RKL | no | yes | no |
| RSO (Liu et al., 2024) (§6.3) | pairwise | HF | pairwise | on-policy | no | RKL | no | yes | no |
| Self-Rewarding LMs (Yuan et al., 2024) (§6.3) | pairwise | AF | pointwise | on-policy | no | RKL | no | yes | no |
| CRINGE (Xu et al., 2024c) (§6.3) | pairwise | HF | pairwise | on-policy | no | none | no | no | yes |
| Meta-Rewarding LMs (Wu et al., 2025b) (§6.3) | pairwise | AF | pairwise | on-policy | no | RKL | no | yes | no |
| NLHF (Munos et al., 2024) (§6.4) | pairwise | HF | pairwise | on-policy | no | RKL | no | yes | no |
| SPPO (Wu et al., 2025c) (§6.4) | pairwise | AF | pairwise | on-policy | no | RKL | no | yes | no |
| DNO (Rosset et al., 2024) (§6.4) | pairwise | AF | pairwise | on-policy | no | RKL | no | yes | no |
| DPO (Rafailov et al., 2023) (§2.4.4) | pairwise | HF | pairwise | offline | no | RKL | no | yes | no |
| SLiC-HF (Zhao et al., 2023) (§7.1.1) | pairwise | HF | pairwise | offline | no | none | no | no | yes |
| DPOP (Smaug) (Pal et al., 2024) (§7.1.1) | pairwise | HF | pairwise | offline | no | RKL | no | yes | no |
| sDPO (Kim et al., 2025) (§7.1.1) | pairwise | HF | pairwise | on-policy | no | RKL | no | yes | no |
| $\beta$-DPO (Wu et al., 2024) (§7.1.1) | pairwise | HF | pairwise | offline | no | RKL | no | yes | no |
| IPO (Azar et al., 2024) (§7.1.1) | pairwise | HF | pairwise | offline | no | RKL | no | yes | no |
| RPO (Sun et al., 2025) (§7.1.1) | pairwise | AF | pointwise | on-policy | no | RKL | no | yes | no |
| GPO (Tang et al., 2024) (§7.1.1) | pairwise | HF | pairwise | offline | no | RKL | no | yes | no |
| KTO (Ethayarajh et al., 2024) (§7.1.2) | single | HF | binary | offline | no | RKL | no | yes | no |
| DRO (Richemond et al., 2024) (§7.1.2) | single | HF | binary | offline | no | RKL | no | yes | no |

Table 12: RLHF methods: methodology summary, papers 1–20 (DPO baseline through Nash/self-play variants).

| Paper | Response | Feedback Type | Reward | Training Paradigm | Entropy Reg. | Divergence Reg. | Length Penalty | Ref. Model | Merge w/ SFT |
|---|---|---|---|---|---|---|---|---|---|
| UNA (Wang et al., 2026) (§7.1.2) | single | HF/AF | pointwise | offline | no | RKL | no | yes | no |
| RRHF (Yuan et al., 2023b) (§7.1.3) | list | HF/AF | list | offline | no | none | yes | no | yes |
| PRO (Song et al., 2024) (§7.1.3) | list | HF | list | offline | no | none | no | no | yes |
| LiPO (Liu et al., 2025c) (§7.1.3) | list | HF | list | offline | no | RKL | no | yes | no |
| DPO-r-to-Q (Rafailov et al., 2024) (§7.2.2) | pairwise | HF | token | offline | yes | RKL | no | yes | no |
| TDPO (Zeng et al., 2024) (§7.2.2) | pairwise | HF | token | offline | no | FKL | no | yes | no |
| D²O (Duan et al., 2024) (§7.2.5) | single | HF | negative | on-policy | no | Jeffrey Divergence | no | yes | no |
| NPO (Zhang et al., 2024) (§7.2.5) | single | HF | negative | offline | no | RKL | no | yes | no |
| CPO (Xu et al., 2024b) (§7.2.5) | pairwise | AF | pairwise | offline | no | none | no | no | yes |
| f-DPO (Wang et al., 2024) (§7.3.2) | pairwise | HF | pairwise | offline | no | others | no | yes | no |
| SimPO (Meng et al., 2024) (§7.3.3) | pairwise | HF | pairwise | offline | no | none | yes | no | no |
| R-DPO (Park et al., 2024) (§7.3.4) | pairwise | HF | pairwise | offline | no | RKL | yes | yes | no |
| ORPO (Hong et al., 2024) (§7.4.1) | pairwise | HF | pairwise | offline | no | none | no | no | yes |
| UFT (Wang et al., 2025b) (§7.4.1) | single | HF/AF | pointwise | offline | no | RKL | no | yes | yes |
| PAFT (Pentyala et al., 2024) (§7.4.2) | pairwise | HF | pairwise | offline | no | RKL | no | yes | yes |

Table 13: RLHF methods: methodology summary, papers 21–35 (token-level reward through SFT-merge variants).

| Paper | Improve Over (Methodology) | Base Model | Fine-tuning Data | Evaluation Benchmarks | Compare With (Methodology) |
|---|---|---|---|---|---|
| RLAIF-Anthropic (Bai et al., 2022b) (§6.2) | RLHF-Anthropic | Anthropic LM series (1B–52B; 52B for main results) | Red-team prompts (Ganguli et al. 2022) + human helpfulness data | Helpfulness Elo, Harmlessness Elo (crowdworker), HHH binary eval | Helpful RLHF, HH RLHF |
| RLAIF-Google (Lee et al., 2024) (§6.2) | RLAIF-Anthropic | PaLM 2 XS (policy) | Reddit TL;DR (summary), OpenAI Human Preferences (helpful), Anthropic HH (harmless) | Win rate (human eval), AI-labeled alignment, harmless rate | RLHF, SFT |
| RLAIF-OpenAI (Xu et al., 2024) (§6.2) | RLHF/PPO (helpful-only) | Internal OpenAI LLMs (Large→GPT-4, Medium, Small, XSmall) | Safety-relevant RL prompts ($P_s$; 6.7K), Gold set (518 human-labeled), $D_{\text{HHH}}$ (6.7K×4 synthetic) | XSTest (overrefusal rate), WildChat (safety human eval), MMLU, HellaSwag, GPQA, Lambada | Helpful-only PPO, Human-PPO (safety data) |
| RSO (Liu et al., 2024) (§6.3) | SLiC, DPO | T5-large (770M), T5-XXL (11B) | HH-RLHF, Reddit TL;DR | Proxy/Gold reward win rate, AutoSxS (PaLM 2-L), human eval | SLiC, DPO, RAFT, ReST |
| Self-Rewarding LMs (Yuan et al., 2024) (§6.3) | Offline DPO | LLaMA-2-70B | Self-generated preference pairs, IFEval-style prompts, LLM-as-judge scoring | AlpacaEval 2.0, pairwise human eval | DPO (M1), SFT, GPT-4 |
| CRINGE (Xu et al., 2024c) (§6.3) | Binary CRINGE | OPT-1.3B, GPT2Medium, LLaMA-7B | Blended Skill Talk, ConvAI2, AlpacaFarm PREF | AlpacaFarm win rate, Repeat@3-gram, F1 | PPO, DPO, Binary CRINGE, Hard Margin CRINGE, SFT, Best-of-N, Iterative DPO |
| Meta-Rewarding LMs (Wu et al., 2025b) (§6.3) | Self-Rewarding LMs | LLaMA-3-8B-Instruct | Self-generated + meta-judged preference pairs | AlpacaEval 2.0, MT-Bench, Arena-Hard | Self-Rewarding LMs (+LC), SPPO, GPT-4 |
| NLHF (Munos et al., 2024) (§6.4) | RLHF/PPO | T5X (S/L/XL) (text summarization) | Reddit TL;DR summaries, human preference annotations | Pairwise win rate (PaLM 2 Large judge) AlpacaEval 2.0, | SFT, RLHF, Self-Play, Best-Response |
| SPPO (Wu et al., 2025c) (§6.4) | Nash RLHF | Mistral-7B-Instruct-v0.2, Llama-3-8B-Instruct | UltraFeedback (AI-labeled via PairRM) | MT-Bench, Arena-Hard, AlpacaEval 2.0, Open LLM Leaderboard | DPO, IPO, SFT |
| DNO (Rosset et al., 2024) (§6.4) | Nash RLHF, DPO | Orca-2-7B, Mistral-7B | UltraFeedback prompts + GPT-4-Turbo teacher responses | GPT-4 win rate, MT-Bench | DPO, SPIN, SFT, Self-Rewarding LM |
| DPO (Rafailov et al., 2023) (§2.4.4) | RLHF/PPO | Pythia-2.8B, LLaMA-7B, GPT-J-6B | Anthropic HH, TL;DR (Reddit) | TL;DR win rate, human preference eval | PPO, SFT, SLiC |
| SLiC-HF (Zhao et al., 2023) (§7.1.1) | PPO/RLHF | T5-Large (770M), T5-XXL (11B) | TL;DR (Reddit summarization) | Human win rate, ranker win rate, ROUGE-1/2/L | PPO/RLHF, SFT |
| DPOP (Smaug) (Pal et al., 2024) (§7.1.1) | DPO | Mistral-7B, LLaMA-2-7B, Yi-34B (Smaug-34B), Qwen-72B (Smaug-72B) | MetaMath DPO, UltraFeedback, Orca-DPO-Pairs | MT-Bench, Open LLM Leaderboard | DPO, IPO, SLiC-HF |
| sDPO (Kim et al., 2025) (§7.1.1) | DPO | SOLAR-10.7B, Mistral-7B | OpenOrca, UltraFeedback Cleaned (multi-stage) | Open LLM Leaderboard (ARC, HellaSwag, MMLU, TruthfulQA), MT-Bench, EQ Bench | DPO, SFT |
| β-DPO (Wu et al., 2024) (§7.1.1) | DPO | Pythia-410M/1.4B/2.8B, Mistral-7B, Llama3-8B | HH-RLHF, TL;DR, UltraChat-200k | GPT-4 win rate (HH, TL;DR), AlpacaEval 2 (LC & WR) | DPO, IPO, KTO, SPPO |
| IPO (Azar et al., 2024) (§7.1.1) | DPO | Synthetic datasets | Anthropic HH (pairwise subset) lmsys-1M | Controlled preference experiments, empirical stability analysis | DPO |
| RPO (Sun et al., 2025) (§7.1.1) | DPO | LLaMA-3-8B, LLaMA-3-70B | (120k prompts, AI-labeled by Nemotron-4-340B-RM) | AvgReward & Win-Rate (lmsys test, AlpacaEval) | DPO, SimPO, KTO, RLOO, SFT |
| GPO (Tang et al., 2024) (§7.1.1) | DPO, IPO, SLiC | Unspecified LM (summarization task, internal setup) | TL;DR (Stiennon et al. 2020) | Win rate vs. $\pi_{\text{ref}}$ (PaLM-2 judge), KL divergence | DPO, IPO, SLiC (GPO variants) |
| KTO (Ethayarajh et al., 2024) (§7.1.2) | DPO | Pythia-1.4B–12B, LLaMA-7B/13B/30B, Mistral-7B | UltraFeedback (binary labels) | GPT-4 win rate, GSM8K, BBH, MMLU | DPO, offline PPO, SLiC, CSFT, SFT |
| DRO (Richemond et al., 2024) (§7.1.2) | KTO | T5-L (770M), T5-XL (3B) | UltraFeedback (binary feedback) | Side-by-side win rate (PaLM2 judge) | KTO, SFT |

Table 14: RLHF methods: experimental summary, papers 1–20. (DPO baseline through Nash/self-play variants).

| Paper | Improve Over (Methodology) | Base Model | Fine-tuning Data | Evaluation Benchmarks | Compare With (Methodology) |
|---|---|---|---|---|---|
| UNA (Wang et al., 2026) (§7.1.2) | DPO, KTO | Mistral-7B-v0.1, Mistral-7B-v0.1-Inst | HelpSteer2 | AlpacaEval 2.0, MT-Bench, Open LLM Leaderboard | DPO, KTO, PPO |
| RRHF (Yuan et al., 2023b) (§7.1.3) | PPO/RLHF | LLaMA-7B, Alpaca-7B | Anthropic HH, Stanford Alpaca (Wombat) | reward model score, PPL, human preference eval | PPO, SFT, Best-of-n |
| PRO (Song et al., 2024) (§7.1.3) | PPO/RRHF | LLaMA-7B | HH-RLHF | BLEU, reward model score, human preference eval | PPO, RRHF, SFT |
| LiPO (Liu et al., 2025c) (§7.1.3) | DPO | T5-large (770M), T5-XXL (11B) | Reddit TL;DR, Anthropic HH (ranked lists) | Reward model win rate, AutoSxS, Human SxS | DPO, SLiC, PRO, list-MLE |
| DPO-r-to-Q (Rafailov et al., 2024) (§7.2.2) | DPO (bandit) | Pythia 2.8B | TL;DR (Reddit) | GPT-4 win rate (TL;DR), token-level reward analysis, beam search win rate | DPO, SFT |
| TDPO (Zeng et al., 2024) (§7.2.2) | DPO | GPT-2 Large, Pythia-2.8B | IMDb, HH-RLHF | Output diversity metrics, MT-Bench (GPT-4 win rate) | DPO, f-DPO, SFT, PPO |
| D²O (Duan et al., 2024) (§7.2.5) | DPO | Alpaca-7B, Phi-3-mini, Qwen2-1.5B | PKU-SafeRLHF | Harmlessness, Helpfulness, GR1, GR2, Win Rate vs. Alpaca, MMLU | DPO, IPO, SLiC-HF, SimPO, GA, SFT |
| NPO (Zhang et al., 2024) (§7.2.5) | Gradient Ascent | Llama-2-7B-chat | TOFU (fictitious persona data) | TOFU Forget Quality, Model Utility | Gradient Ascent, DPO, KTO |
| CPO (Xu et al., 2024b) (§7.2.5) | SFT + RLHF for MT | ALMA-7B, ALMA-13B (LLaMA-based MT) | WMT translation pairs + AI-labeled preferences (GPT-4, ALMA-13B-LoRA) | COMET, BLEURT, WMT test sets (En→De, En↔Zh) | ALMA, supervised MT, DPO |
| f-DPO (Wang et al., 2024b) (§7.3.2) | DPO (RKL only) | GPT-2-large, Pythia-2.8B | HH-RLHF, IMDB-sentiment dataset | Reward-diversity trade-off, MT-bench | DPO, PPO |
| SimPO (Meng et al., 2024) (§7.3.3) | DPO | Mistral-7B, LLaMA-3-8B, Gemma-2-9B | UltraFeedback Cleaned | AlpacaEval 2.0 (LC win rate), MT-Bench, Arena-Hard | RRHF, SLiC-HF, DPO, IPO, CPO, KTO, ORPO, R-DPO |
| R-DPO (Park et al., 2024) (§7.3.4) | DPO | Pythia-2.8B | Anthropic HH, TL;DR summarization | AlpacaEval (length-controlled), TL;DR win rate | DPO, SFT |
| ORPO (Hong et al., 2024) (§7.4.1) | DPO + SFT (sequential) | OPT-125M–1.3B, Phi-2 (2.7B), LLaMA-2-7B, Mistral-7B | Anthropic HH, UltraFeedback | AlpacaEval 1.0/2.0, MT-Bench, IFEval | SFT, DPO, PPO |
| UFT (Wang et al., 2025b) (§7.4.1) | SFT + DPO (sequential) | Mistral-7B, Qwen-32B | UltraChat (SFT), HelpSteer2 (alignment) | AlpacaEval 2.0, MT-Bench, Open LLM Leaderboard | SFT, DPO, KTO |
| PAFT (Pentyala et al., 2024) (§7.4.2) | SFT + DPO (sequential) | Mistral-7B, Llama-3-8B | UltraChat (SFT), UltraFeedback (DPO) | AlpacaEval, Open LLM Leaderboard | SFT, DPO, KTO |

Table 15: RLHF methods: experimental summary, papers 21–35. (token-level reward through SFT-merge variants).

