# OpenReview forum: "Reinforcement Learning for LLM Post-Training: A Survey"
_TMLR — Under review for TMLR_

### Review · Reviewer_W6Kw · 2026-06-16

**Summary Of Contributions:**

This article surveys reinforcement learning methods for LLM post-training and aims to provide a unified framework for summarizing and comparing existing approaches, including RLHF, PPO, DPO, GRPO, RLVR, RLAIF, and iterative DPO. The central idea is that these methods can all be written in a unified policy-gradient form, and their main differences lie in how they define the gradient coefficient. This coefficient may incorporate different factors such as reward, advantage, importance ratios, KL regularization, and entropy regularization.

Under this framework, pretraining and supervised fine-tuning can be viewed as special cases where the gradient coefficient is always equal to 1, meaning that all tokens are reinforced equally. More advanced post-training methods instead modify the gradient coefficient using reward or preference signals. The article organizes these methods into several major directions, including RLHF / PPO, offline preference optimization methods such as DPO, and verifiable-reward-based methods such as RLVR / GRPO. It also highlights several open problems in the field, including the optimal design of gradient coefficients, convergence analysis, prompt and response sampling strategies, and the quality and safety of reward signals.

**Audience:**

Yes

**Audience Explanation:**

Researchers interested in LLM alignment and post-training would find this survey useful because it organizes RLHF, DPO, GRPO, and RLVR under a clear unified framework for technical comparison. It also serves as a helpful reference and guide to many existing methods in this area.

**Broader Impact Concerns:**

I do not see any concerns on the ethical implications of the work.

**Claims And Evidence:**

Yes

**Claims Explanation:**

Yes. This paper is a survey, and most of the methods discussed are existing approaches that, based on my current knowledge, are accurately described. The main contribution of the paper is not to introduce a new algorithm, but to use a unified mathematical model and notation to organize these methods and compare them from a common perspective. In particular, the paper frames different post-training methods through their gradient coefficients, which provides a useful way to understand their similarities and differences. Although I have not checked every formula in full detail, the derivations and the underlying intuition appear to be reasonable and consistent with the literature.

**Requested Changes:**

Given that this paper is a detailed survey, I do not think there are any major issues that must be addressed to support acceptance. The following changes would further strengthen the paper:
1. The paper covers many methods, so adding a table or figure that compares RLHF/PPO, DPO, RLVR/GRPO, RLAIF, and iterative DPO along the main design axes would make the survey easier to follow.
2. The paper should include more discussion of the limitations of the unified framework. For example, the gradient-coefficient view is useful, but it does not fully capture practical issues such as optimization stability, data quality, reward hacking, and implementation cost.

---

> ### Author Response · Authors · 2026-07-02
> **Rebuttal for the Reviewer**
>
> Q1: The paper covers many methods, so adding a table or figure that compares RLHF/PPO, DPO, RLVR/GRPO, RLAIF, and iterative DPO along the main design axes would make the survey easier to follow.
>
> A1: Thank you for the reviewer’s feedback. The comparison of all the methods can be found in appendix A and B from 9 different dimensions: 1. importance sampling, 2. clipping, 3. reward, 4. baseline, 5. advantage normalization, 6. length normalization, 7. KL divergence, 8. entropy, and 9. partition function.
>
> We have also added Figure 3 to plot this more clear: “Overview of the nine per-paper analysis axes used to characterize each method—importance sampling, clipping, reward, baseline, advantage normalization, length normalization, KL divergence, entropy, and partition function—and the six visualization categories: pretraining/SFT, RLHF/PPO, RLAIF, RLVR/GRPO, DPO (offline), and DPO (on-policy).”
>
> Q2: The paper should include more discussion of the limitations of the unified framework. For example, the gradient-coefficient view is useful, but it does not fully capture practical issues such as optimization stability, data quality, reward hacking, and implementation cost.
>
> A2: Thank you for the reviewer’s feedback. We have added a limitation section.
>
> Modification on the paper:
> Limitations
>
> Comparability of empirical evidence The surveyed papers differ in base models, training data, reward sources, compute budgets, benchmarks, and reporting protocols. Our comparisons therefore focus on algorithmic design choices rather than controlled empirical rankings. We summarize recurring patterns, such as normalization, reward granularity, online versus offline optimization, and PPO/GRPO/DPO trade-offs, while avoiding strong causal claims across heterogeneous settings.
>
> Scope of framework and settings The gradient-coefficient view is most direct for rollout-based methods; offline DPO-style objectives are included under the same notation when useful, but are organized separately around preference construction, reward parameterization, regularization, and SFT integration. Concretely, the standard DPO objective is a coupled contrastive loss over response pairs: it continuously increases the likelihood difference between the preferred response and the dispreferred one through their ratio, cancelling the intractable partition function Z(x).
>
> Beyond the framework boundary, the gradient-coefficient lens also abstracts away two practical concerns that matter in deployment: reward hacking (the framework does not model how reward models can be exploited under over-optimization, which interacts with gradient coefficient magnitude and the strength of KL regularization) and implementation cost (PPO requires four models in memory simultaneously: policy, reference, reward, and value whereas GRPO, RLOO, and offline DPO do not, a constraint invisible in the unified objective).
>
> Lastly, we focus on single-turn, text-only post-training, leaving multimodal, agentic and multi-turn RL, continual and cross-lingual alignment, pretraining-scale RL, systems issues, and non-language applications mainly to future work.

---

### Review · Reviewer_KDTa · 2026-06-16

**Summary Of Contributions:**

This paper surveys reinforcement learning for LLM post-training, organized around a single unifying lens: a token-level policy-gradient estimator in which each method is recovered by specifying a data source, a gradient coefficient that sets the per-token reinforcement, and a stabilization mechanism. From this view, it derives MLE, SFT, REINFORCE, actor-critic, TRPO, PPO, GRPO, and DPO, then decomposes the RLVR literature along prompt sampling, response sampling, and gradient-coefficient design, with separate treatment of RLHF and offline DPO, and an appendix characterizing roughly seventy methods in consistent tables. The main strengths are careful self-contained derivations, standardized notation that makes disparate methods directly comparable, and unusually current coverage.

**Audience:**

Yes

**Audience Explanation:**

Clearly yes. RL-based post-training is a highly active area, and existing surveys lean qualitative rather than algorithmic. A technically grounded reference that expresses many methods in one notation and isolates the precise design choice distinguishing each is useful to both newcomers and practitioners, and the currency of the coverage adds further value.

**Claims And Evidence:**

Yes

**Claims Explanation:**

For the most part, yes. The central claim, that surveyed methods share a common policy-gradient form with an identifiable gradient coefficient, is backed by derivations that are correct where I checked them, and the coverage claim is well supported by the tables. Two claims overstate the evidence and should be tightened: the abstract presents the unified framework as the authors' derivation when the body attributes it to prior work, and the "empirical results" tables record base models, data, and benchmarks rather than the reported numbers, so they compare experimental design rather than outcomes.

**Requested Changes:**

1. Reword the overstated claims to match the evidence: attribute the unified framework to prior work in the abstract and contributions
2. Add a short paragraph delimiting where the single-estimator unification holds versus where it is an organizing abstraction, especially for offline DPO, whose coupled, off-policy form sits uneasily with the on-policy per-token estimator.
3. Add an explicit scope-and-limitations subsection stating what is deferred (e.g., agentic/multi-turn and multimodal RLVR), plus a compact practitioner-facing decision guide distilled from the tables.
4. It would be better if the author could improve navigability and polish: ensure methods in the tables map to the discussed subsections.

---

> ### Author Response · Authors · 2026-07-02
> **Rebuttal for the Reviewer**
>
> Q1: Reword the overstated claims to match the evidence: attribute the unified framework to prior work in the abstract and contributions
>
> A1: Thank you for the reviewer’s suggestion. We agree that the distinction between the prior framework and our contribution should be made more explicit. We have revised the abstract, contributions, paper organization, and Section 2 to consistently state that the unified policy-gradient framework is adopted from and extended based on Shao et al. (2024).
> Specifically, the unified policy-gradient formulation originates from Shao et al. (2024). Our contribution is to extend this analytical lens across the broader landscape of LLM post-training methods—including pretraining, SFT, RLHF, RLVR, DPO-based alignment, and recent variants—and to use the resulting gradient-coefficient perspective as a consistent framework for comparing algorithms and their design choices throughout the survey.
> Modification in the paper:
> 1. Abstract: We adopt and extend the unified policy-gradient framework of Shao et al. (2024) as an organizing lens spanning pretraining, SFT, RLHF, RLVR, and more recent techniques.
> 2. Contribution: For each algorithm, we derive the policy-gradient objective and identify the gradient coefficient that encapsulates its core design decisions, thereby expressing it within the unified analytical lens of Shao et al. (2024) that we adopt and extend throughout the survey.
> 3. Contribution: We adopt and extend the unified policy gradient framework by Shao et al. (2024) that includes PPO-based RLHF, RLVR, and DPO-based alignment.
> 4. Paper Organization: Section 2 adopts and extends a unified post-training framework (UPT) by Shao et al. (2024), deriving pretraining through MLE, REINFORCE, actor-critic, TRPO, PPO, DPO, and GRPO for RLHF and RLVR.
> 5. Evolution of Language Model Training Paradigms: In this section, we adopt and extend the unified post-training framework based on a single policy gradient estimator by Shao et al. (2024).
>
>
> Q2: Add a short paragraph delimiting where the single-estimator unification holds versus where it is an organizing abstraction, especially for offline DPO, whose coupled, off-policy form sits uneasily with the on-policy per-token estimator.
>
> A2: Thank you for the reviewer's feedback. We agree that offline DPO differs structurally from on-policy rollout methods, and we have clarified this distinction in the revised manuscript. However, we respectfully believe that these differences do not invalidate the single-estimator perspective. Rather, they correspond to changes in the sampling procedure and gradient coefficient while preserving the underlying policy-gradient form.
>
> To better reflect these relationships, we have added Figure 3, which provides an overview of the nine analysis axes (importance sampling, clipping, reward, baseline, advantage normalization, length normalization, KL divergence, entropy, and partition function) and the six method categories (pretraining/SFT, RLHF/PPO, RLAIF, RLVR/GRPO, and offline/on-policy variants of DPO). A more detailed discussion of how DPO fits within the unified gradient formulation is provided in Section 2.4.4.
>
> Q3: Add an explicit scope-and-limitations subsection stating what is deferred (e.g., agentic/multi-turn and multimodal RLVR), plus a compact practitioner-facing decision guide distilled from the tables.
>
> A3: Thank you for the reviewer’s feedback. We have added a limitation section (section 10) and a subsection for practitioner-facing decision guides (subsection 2.8) to solve these problems respectively.
>
> Q4: It would be better if the author could improve navigability and polish: ensure methods in the tables map to the discussed subsections.
>
> A4: Thank you for the reviewer’s feedback. We have added 1. “Navigation click” for the table in the appendix to go back to the corresponding sections and 2. “Click” on each page to go back to table content.

---

### Review · Reviewer_FDwF · 2026-06-23

**Summary Of Contributions:**

This paper surveys reinforcement-learning-based post-training methods for LLMs, including RLHF, RLVR, PPO/GRPO-style methods, and DPO/iterative-DPO variants. Its main contribution is to organize these methods under a unified policy-gradient view, where different algorithms are compared through their data source, response sampling strategy, and gradient coefficient. The paper also provides standardized notation and detailed appendix tables comparing implementation details across recent methods.

Overall, the paper is timely and potentially useful as a technical reference for researchers working on LLM post-training. Its strongest aspect is that it tries to go beyond a high-level narrative survey and instead provides a mathematically unified notation for comparing many recent methods.

However, I have several concerns about the paper’s novelty as a survey contribution, its positioning relative to prior work, and the amount of critical synthesis beyond taxonomy and formula rewriting.

- The main unifying idea needs clearer attribution and positioning. The paper presents the unified policy-gradient / gradient-coefficient framework as a central contribution, but this view appears to be largely inherited from prior work, especially Shao et al. (2024), which the paper itself cites. The authors should clearly separate what is adopted from prior work, what is re-derived or re-notated, and what is genuinely new in this survey.

- I would suggest that the paper should better differentiate itself from existing surveys. It's 2026 now and there are already many recent surveys on RLHF, RLVR, LLM post-training, etc. The paper should more explicitly explain why another survey is needed now, what gap it fills relative to these works, and what unique perspective or organization it provides.

- The coverage is broad, but many sections read like method-by-method summaries. I would expect more synthesis on questions such as which design choices consistently matter, when group normalization or length normalization helps or hurts, what the practical trade-offs are between PPO, GRPO, RFT, iterative DPO, and offline DPO, and which empirical findings are robust versus benchmark- or model-specific.

**Audience:**

Yes

**Audience Explanation:**

The paper is relevant to TMLR’s audience because RL-based LLM post-training is an active and important topic.

**Broader Impact Concerns:**

I do not see major broader-impact concerns requiring rejection. The paper is a survey and does not introduce a new training method or dataset.

**Claims And Evidence:**

Yes

**Claims Explanation:**

The paper’s main descriptive claims are mostly supported: it does provide a broad technical survey of RLHF/RLVR/DPO-style post-training methods, derives many objectives under a common notation, and includes detailed comparison tables. These support the claim that the paper can serve as a technical reference.

However, some claims about novelty and comprehensiveness need more careful qualification. The unified gradient-coefficient framework appears closely related to prior work, particularly Shao et al. (2024), so the paper should more explicitly distinguish what is inherited from prior work and what is newly contributed by this survey. Also, because the field is moving quickly, the paper should state its paper-selection criteria, time cutoff, and inclusion/exclusion rules. Without this, claims of comprehensive coverage are hard to fully assess.

**Requested Changes:**

- I would encourage the authors to clarify the novelty of the unified policy-gradient / gradient-coefficient framework. The paper should explicitly separate what is adopted from prior work, especially Shao et al. (2024), from what is newly contributed by this survey.

- Also please add a clear survey methodology. The paper should describe how the surveyed papers were collected and selected, including search sources, time range, keywords, inclusion/exclusion criteria, and filtering process.

- Add more visualization could improve readability of the survey paper. The paper currently relies heavily on text, equations, and tables. For a survey, additional visualizations such as a method timeline, taxonomy map, pipeline comparison, or algorithm-design matrix would make the paper easier to navigate.

- The paper should explain what the taxonomy teaches us: which design choices seem robust, which are empirically unclear, and which comparisons are not reliable due to different models, datasets, reward functions, or benchmarks.

---

> ### Author Response · Authors · 2026-07-02
> **Rebuttal for the Reviewer**
>
> Q1: I would encourage the authors to clarify the novelty of the unified policy-gradient / gradient-coefficient framework. The paper should explicitly separate what is adopted from prior work, especially Shao et al. (2024), from what is newly contributed by this survey.
>
> A1: Thank you for the reviewer’s suggestion. We agree that the distinction between the prior framework and our contribution should be made more explicit. We have revised the abstract, contributions, paper organization, and Section 2 to consistently state that the unified policy-gradient framework is adopted from and extended based on Shao et al. (2024).
> Specifically, the unified policy-gradient formulation originates from Shao et al. (2024). Our contribution is to extend this analytical lens across the broader landscape of LLM post-training methods—including pretraining, SFT, RLHF, RLVR, DPO-based alignment, and recent variants—and to use the resulting gradient-coefficient perspective as a consistent framework for comparing algorithms and their design choices throughout the survey.
>
> Modification in the paper:
> 1. Abstract: We adopt and extend the unified policy-gradient framework of Shao et al. (2024) as an organizing lens spanning pretraining, SFT, RLHF, RLVR, and more recent techniques.
> 2. Contribution: For each algorithm, we derive the policy-gradient objective and identify the gradient coefficient that encapsulates its core design decisions, thereby expressing it within the unified analytical lens of Shao et al. (2024) that we adopt and extend throughout the survey.
> 3. Contribution: We adopt and extend the unified policy gradient framework by Shao et al. (2024) that includes PPO-based RLHF, RLVR, and DPO-based alignment.
> 4. Paper Organization: Section 2 adopts and extends a unified post-training framework (UPT) by Shao et al. (2024), deriving pretraining through MLE, REINFORCE, actor-critic, TRPO, PPO, DPO, and GRPO for RLHF and RLVR.
> 5. Evolution of Language Model Training Paradigms: In this section, we adopt and extend the unified post-training framework based on a single policy gradient estimator by Shao et al. (2024).
>
> Q2: Also please add a clear survey methodology. The paper should describe how the surveyed papers were collected and selected, including search sources, time range, keywords, inclusion/exclusion criteria, and filtering process.
>
> A2: Thank you for the reviewer’s suggestion. We have added §1.5 "Survey Scope and Literature Selection" to discuss 1. Scope and criteria, 2. Sources and time range, 3. Search procedure and 4. Paper Timeline.
>
> Q3: Add more visualization could improve readability of the survey paper. The paper currently relies heavily on text, equations, and tables. For a survey, additional visualizations would make the paper easier to navigate.
>
> A3: We have added the following figures:
> Figure 1: Timeline of representative RL-based LLM post-training methods
> Figure 3: Overview of the nine per-paper analysis axes used to characterize each method and the six visualization categories.
> Figure 4: Method-selection decision tree for LLM post-training.
> Figure 6-7, 9-14, 16, 18, 20-23, 25-26: New subfigures for the main takeaway for each subsection.
>
> Q4: The paper should explain what the taxonomy teaches us: which design choices seem robust, which are empirically unclear, and which comparisons are not reliable due to different models, datasets, reward functions, or benchmarks.
>
> A4: Thank you for the reviewer’s suggestion. We have expanded Section 2.8 to emphasize what the taxonomy teaches.
>
> Specifically, we added a practical method-selection procedure (Figure 4) that explains when different post-training strategies are appropriate, progressing from prompt engineering and SFT to offline DPO, RLVR, and RLHF/RLAIF/iterative DPO based on the need for exploration and the source of the reward signal. We also emphasize that these methods are complementary and are often combined in practice rather than being mutually exclusive.
>
> We further clarify that the taxonomy is intended to explain the assumptions and trade-offs behind different design choices, not to universally rank methods, since many reported results are not directly comparable due to differences in base models, datasets, reward functions, training budgets, and evaluation benchmarks. Readers are then guided to the detailed analyses in Sections 3–7, where methods are systematically compared using the same nine analytical dimensions.
>
> Q5: I would suggest that the paper should better differentiate itself from existing surveys.
>
> A5: Thank you for the suggestion. We have revised Subsection 1.4 to clarify the contribution and comparison with other review papers. The revised text now emphasizes that our survey provides a unified, mechanics-focused treatment of LLM post-training objectives, covering SFT, PPO-based RLHF, DPO, RLVR, and GRPO under a common policy-gradient framework with standardized notation and fine-grained decomposition of algorithmic design choices.